# CARL: Preserving Causal Structure in Representation Learning

**Yulong Li[1,*], Xiwei Liu[1,*], Feilong Tang[3], Zhixiang Lu[2], Ming Hu[3], Yichen Li[1], Haochen Xue[1]**
Peixin Guo[2], Jionglong Su[2], Yutong Xie[1], Eran Segal[1,†], Imran Razzak[1,†]

[1]Mohamed bin Zayed University of Artificial Intelligence
[2]Xi'an Jiaotong-Liverpool University
[3]Monash University

[*]Equal contribution
[†]Corresponding authors

## ABSTRACT

Cross-modal representation learning is fundamental for extracting structured information from multimodal data to enable semantic understanding and reasoning. However, current methods optimize statistical objectives without explicit causal constraints, where nonlinear mappings can introduce spurious dependencies or eliminate critical mediators, leading to representation-induced structural drift that undermines the reliability of causal inference. Therefore, establishing theoretical guarantees for causal invariance in cross-modal representation learning remains a foundational challenge. To this end, we propose Causal Alignment and Representation Learning (CARL), which explicitly embeds causal structure preservation constraints into cross-modal alignment objectives. Specifically, CARL introduces a multi-consistency loss architecture that jointly optimizes conditional independence preservation and information bottleneck regularization to balance cross-modal compression with critical variable retention, ensuring low-density modalities are not masked by high-density reconstruction demands. We further incorporate monotonic alignment consistency loss to establish correspondence between semantic similarity and representation distance through Spearman correlation, and Markov boundary preservation loss to maintain identifiability conditions including backdoor, frontdoor, and instrumental variable criteria in the shared representation space. In synthetic experiments with known causal ground truth, CARL achieves state-of-the-art performance in preserving conditional independence patterns and maintaining causal query identifiability under structural uncertainty. Real-world validation on Human Phenotype Project data reveals that CARL successfully preserves causal structures between fundus vascular representations and cardiovascular events, demonstrating its capacity for reliable cross-modal causal inference in complex biomedical applications.

## 1 INTRODUCTION

Representation learning is fundamental to modern deep learning Bengio et al. (2013), yet prevalent methods risk distorting the very structure they aim to capture Arjovsky et al. (2019). By optimizing purely statistical criteria, objectives based on reconstruction Goodfellow et al. (2014); Kingma & Welling (2013), contrastive learning Radford et al. (2021), or large-scale pretraining Dosovitskiy et al. (2021) can inadvertently corrupt the underlying causal graph of the data. These nonlinear mappings may introduce spurious dependencies or eliminate critical mediators Geirhos et al. (2020), a failure we term representation-induced structural drift. This drift critically undermines the promise of causal machine learning: reliable interventional estimation Shalit et al. (2017), counterfactual reasoning Zuo et al. (2023), and out-of-distribution generalization Liu et al. (2021a). For instance, in the Human Phenotype Project (HPP)[1], such drift could obscure the true causal link between retinal

---

[1]https://knowledgebase.pheno.ai/

vascular features and cardiovascular events amidst confounders like age and blood pressure Shapira et al. (2024). To alleviate this, we propose the Causal Structure Preservation (CSP) principle, which requires that representations preserve key conditional independences, Markov boundary information, and the identifiability of causal effects. Establishing theoretical guarantees for such causal invariance in cross-modal learning is a foundational problem we address herein.

A critical gap exists between the optimization objectives of current cross-modal representation learning methods and the preservation of causal invariance. Sun et al. (2025). For instance, classical approaches like DCCA Andrew et al. (2013) focus on maximizing correlation without structural verification mechanisms, failing to guarantee Markov boundary preservation and identifiability consistency. More contemporary paradigms, while powerful, inherit similar blindness to causality. Contrastive models like CLIP Radford et al. (2021) learn cross-modal alignment via an InfoNCE loss but fail to enforce a monotonic relationship between semantic similarity and representation distance, a property crucial for structural consistency. Pivot-based alignment paradigms like ALIGN Jia et al. (2021) and ImageBind Girdhar et al. (2023) optimize geometric properties of representations rather than the explicit structural criteria required for causal inference. Consequently, none of these prominent approaches can guarantee that their learned representations faithfully maintain key causal properties like conditional independence, Markov boundaries, or identifiability conditions.

This abovementioned gap exposes three fundamental challenges that hinder reliable cross-modal causal inference: *i):* Cross-modal Information Bottleneck (CIB): asymmetric compression from information density differences across modalities causes key variables from low-density modalities to be masked by reconstruction demands of high-density modalities, weakening identifiability of causal queries. *ii):* Modal Alignment Consistency (MAC): after independent encoding, shared space distances lack monotone correspondence with semantic similarity, preventing distance-based retrieval from guaranteeing semantic consistency. *iii):* Cross-modal Identifiability Consistency (CIC): identifiability conditions including backdoor, frontdoor, and instrumental variables lack establishment criteria in shared representations Pearl (2009). The core challenge, therefore, is to establish verifiable, structure-preserving mechanisms that are robust to such modal heterogeneity.

To this end, we introduce Causal Alignment and Representation Learning (CARL), a framework that establishes verifiable structure-preserving mechanisms while handling modal heterogeneity by embedding causal invariance constraints into cross-modal alignment objectives. CARL implements our novel CSP principle by co-optimizing three synergistic losses: a *conditional independence loss* to preserve the causal graph; a *Markov boundary retention loss*, implemented via conditional mutual information estimation, to ensure key variables from low-density modalities are not masked by high-density ones; and a *monotonic alignment consistency loss* guaranteeing semantic consistency by maximizing the Spearman correlation between semantic differences and representation distances. This joint optimization ensures crucial identifiability conditions, including backdoor, frontdoor, and instrumental variable criteria, are preserved in the shared representation space. Validation on HPP demonstrates CARL maintains causal structure among the representation of retinal fundus, sleep, anthropometrics and cardiovascular. Our main contributions are:

❶ We formalize the Causal Structure Preservation principle and indentify three core theoretical challenges in causally-aware cross-modal representation learning (CIB, MAC, and CIC), establishing a new theoretical foundation for causal invariance guarantees in cross-modal scenarios.

❷ We propose CARL, a framework that jointly optimizes three structure-preserving losses: conditional independence, Markov boundary retention, and monotonic alignment consistency, achieving verifiable causal structure preservation and cross-modal alignment under heterogeneity.

❸ Empirical validation on HPP demonstrates that CARL preserves the causal relationships between retinal vascular features and cardiovascular events established in existing research, validating the framework's effectiveness in maintaining cross-modal causal structure.

## 2 PROBLEM FORMULATION AND CAUSAL SETUP

We formalize the causal structure preservation problem by establishing the causal model and cross-modal setting (Section 2.1), encoder configurations (Section 2.2), and the CSP principle (Section 2.3), then instantiate these concepts in a biomedical application (Section 2.4).

## 2.1 Causal Model and Cross-Modal Setting

We formalize the underlying causal mechanism with a Directed Acyclic Graph (DAG) $\mathcal{G} = (V, E)$, where $V = \{T, Y^*, M, X\}$ consists of a treatment $T \in \mathcal{T}$, a set of potential mediators $M = \{M_1, \ldots, M_k\}$, a true outcome $Y^* \in \mathcal{Y}$, and a set of covariates $X \in \mathcal{X}$. These variables may be continuous, discrete, or mixed. The edges $E$ represent direct causal relationships, where the parent set of any node $V_i \in V$ is denoted by $\mathrm{Pa}_\mathcal{G}(V_i)$. We assume the true graph $\mathcal{G}^*$ satisfies the Causal Markov and Faithfulness assumptions Pearl (2009), enabling conditional independencies to be inferred from the graph. Appendix A.4 details these assumptions and Causal Sufficiency.

In the cross-modal setting, we observe tabular data $(T, M, X)$ alongside image modalities: $I^M$ for the mediator and $I^Y$ as a proxy for the true outcome, $Y^*$. For instance, in biomedical phenotyping (Section 2.4), $T$ may represent blood pressure, $M$ captures vascular features observed both as measurements and fundus images, and $Y^*$ denotes cardiovascular outcomes. The observed outcome $Y$ equals $Y^*$ when directly available; otherwise, $Y = \phi(I^Y)$ serves as a feature-extracted proxy. CARL supports three configurations based on this: IM (using $I^M$), IY (using $I^Y$), and DUAL (both).

We aim to learn encoder family $\mathcal{E} = \{E_T, E_M, E_{I^M}, E_{I^Y}\}$ that map these heterogeneous inputs to a shared representation space $\mathcal{Z} \subset \mathbb{R}^d$, yielding representation $Z_T = E_T(T)$, $Z_M \in \{E_M(M), E_{I^M}(I^M)\}$, and $Z_Y \in \{E_Y(Y), E_{I^Y}(I^Y)\}$ Our core objective, structure preservation, dictates that if the original variables satisfy $T \perp Y^* \mid M$, the learned representations must adhere to the constraint $MI(Z_T; Z_Y \mid Z_M) \leq \zeta^*$, where $MI(\cdot; \cdot \mid \cdot)$ is the conditional mutual information and $\zeta^*$ is the minimal approximation error for the encoder class.

**Assumption 1 (Causal Markov and Faithfulness):** The data generating process is assumed to follow a DAG $\mathcal{G}$ that satisfies the Causal Markov and Faithfulness assumptions. We further assume that causal effects are identifiable, which requires that either a set of covariates $X$ blocks all backdoor paths, or the frontdoor/IV criteria are met (Appendix A.2), and that both positivity (overlap) and consistency (SUTVA) hold (Appendix A.3).

**Assumption 2 (Separable Measurement and Encoders):** Images modalities are assumed to follow a separable measurement structure $I^M = g_M(a_M, b_{\text{style}}; \eta_M)$, $I^Y = g_Y(a_Y, b_{\text{style}}; \eta_Y)$, where $a_M, a_Y$ represent semantic content variables (the causal signal we aim to preserve), $b_{\text{style}}$ captures style factors, and $\eta_M, \eta_Y$ denote measurement noise. We assume $(\eta_M, \eta_Y) \perp (T, M, Y^*, X)$ and $b_{\text{style}} \perp (T, Y^*) \mid (M, X)$ (Appendix A.6.1). All encoders mapping to the shared space $\mathcal{Z}$ are assumed to be Lipschitz-bounded for stability (Appendix A.8.2).

With the causal model and structural assumptions in place, we now describe the encoder architectures that map heterogeneous inputs to a shared representation space while respecting causal constraints. We consider three configurations corresponding to different image modality availability.

## 2.2 Cross-Modal Encoding Configurations

Building upon the causal assumptions defined previously, our framework learns an encoder family $\mathcal{E}$ to map heterogeneous inputs into a unified representation space $\mathcal{Z} \subset \mathbb{R}^d$. To support variable tracing and ensure stability, all encoders are deterministic, Lipschitz-constrained mappings, with tabular inputs processed by multilayer perceptrons and images by convolutional architectures. The framework flexibly supports three configurations based on the availability of image data:

**Scenario 1. (IM):** image $I^M$ represents the mediator, where encoder $E_{I^M} : I^M \to \mathcal{Z}_{I^M}$ and tabular encoder $E_M : M \to \mathcal{Z}_M$ learn shared representations through alignment losses. The constraint ensures no spurious path $T \to I^M \to Y^*$ is introduced.

**Scenario 2. (IY):** image $I^Y$ serves as outcome proxy. Encoder $E_{I^Y}$ generates $\mathcal{Z}_{I^Y} = E_{I^Y}(I^Y) \in \mathcal{Z}$. An observable outcome $Y = \phi(I^Y) \in \mathbb{R}^k$ is extracted, assuming a monotonic and calibratable relationship with the true outcome $Y^*$. The calibration function $h : \mathcal{Z} \to \mathbb{R}^k$ then map the representation back to a prediction $\hat{Y} = h(\mathcal{Z}_{I^Y})$, optimized to align with the observed value $Y$.

**Scenario 3. (DUAL):** both image modalities are used simultaneously, but conditioning on $I^M$ and $I^Y$ jointly is avoided in loss computation to prevent collider bias.

## 2.3 THE CAUSAL STRUCTURE PRESERVATION PRINCIPLE

Having defined the causal graph $\mathcal{G}$ and encoder configurations in the previous subsections, we now formalize what it means for learned representations to preserve causal structure. We capture this through the Causal Structure Preservation (CSP) principle.

**Definition 2.1 ($\varepsilon$-Causal Structure Preservation).** A representation learning system $(\mathcal{E}, \mathcal{Z})$ satisfies $\varepsilon$-CSP if and only if there exists $\varepsilon \geq 0$ such that the following three conditions hold: **(i)** *Conditional Independence Transfer*: For any conditional independence $T \perp Y^* \mid M$ implied by the original causal graph, the corresponding representations must satisfy $MI(Z_T; Z_Y \mid Z_M) \leq \varepsilon$; **(ii)** *Markov Boundary Preservation*: $MI(Z_M; Z_Y) \geq MI(M; Y^*) - \varepsilon$; **(iii)** *Monotonic Alignment Consistency*: In the IY configuration, the Spearman correlation $\rho_S$ between semantic differences $\Delta a_{ij} = |a_i - a_j|$ and representation distances $\Delta z_{ij} = \|z_i - z_j\|_2$ must be lower-bounded: $\rho_S(\Delta a, \Delta z) \geq c_0 - \varepsilon$, where $c_0 \in (0, 1]$ is a constant determined by the order-preserving margin assumption. These three conditions jointly ensure causal structure preservation: condition (i) prevents spurious dependencies by enforcing conditional independencies from the original graph; condition (ii) prevents information loss by requiring the mediator to retain predictive power; condition (iii) ensures geometric consistency between semantic similarity and representation distance. Together, they maintain the identifiability of causal effects in the representation space.

These three conditions jointly ensure causal structure preservation: condition *i)* prevents spurious direct pathways by enforcing conditional independence constraints from the original causal graph; condition *ii)* prevents trivial satisfaction of *i)* through information loss by requiring the mediator to retain predictive information about the outcome; and condition *iii)* ensures geometric consistency by aligning semantic similarity with representation distance. Together, they guarantee that causal relationships are neither distorted by spurious shortcuts nor weakened by information bottlenecks.

## 2.4 MOTIVATING APPLICATION: HUMAN PHENOTYPE PROJECT

Having formalized the framework, we now present our motivating application. We conduct our real-world evaluation on the HPP, a large-scale, longitudinal study that collects deep phenotypic profiles from a diverse population, offering a holistic perspective on the factors influencing health and disease Kohn et al. (2025). As previewed in Figure 1, the dataset is distinctly multi-modal, containing a wide variety of participant information categorized into tabular records, time series, and images, covering multiple physiological systems. Specifically, the HPP encompasses health information spanning over 30 distinct modalities, including but not limited to gut microbiome, body composition, genetics, and blood tests. This rich, multi-modal structure, with its inherent complexity and potential for confounding, makes the HPP an ideal and challenging testbed for evaluating CARL's ability to preserve causal structures in a real-world biomedical application.

This rich multi-modal structure presents a critical methodological challenge: standard representation learning methods that optimize purely statistical objectives may inadvertently distort the underlying causal relationships. For instance, a naive encoder could learn to predict cardiovascular events directly from retinal images by exploiting spurious correlations, thereby bypassing the true mediating pathway through vascular features and blood pressure. Such representation-induced structural drift would corrupt our ability to perform valid causal inference and estimate intervention effects. This motivates our focus on causal structure preservation in cross-modal representation learning.

## 3 METHODOLOGY

Our proposed CARL framework provides a unified approach to preserving causal structure within cross-modal representation learning. CARL learns an encoder family $\mathcal{E} = \{E_T, E_M, E_{I^M}, E_{I^Y}\}$ which can be extended with optional encoders $E_Y, E_X$ for outcome encoding and confounder control. This is designed to map heterogeneous inputs, spanning tabular variables $(T, M, Y^*, X)$ and image observations $I \in \{I^M, I^Y\}$ to a shared representation space while preserving causal relationships. The core of our framework is a multi-objective optimization strategy guided by three synergistic losses: conditional independencies preservation $\mathcal{L}_{CI}$, Markov boundary retention $\mathcal{L}_{MBR}$, and monotonic alignment consistency $\mathcal{L}_{MAC}$ (Section 3.1). The learned representations enable causal discovery via the PC algorithm with variable tracing back to original inputs (Section 3.2). We establish theoretical properties proving these losses suffice for causal structure preservation (Section 3.3).

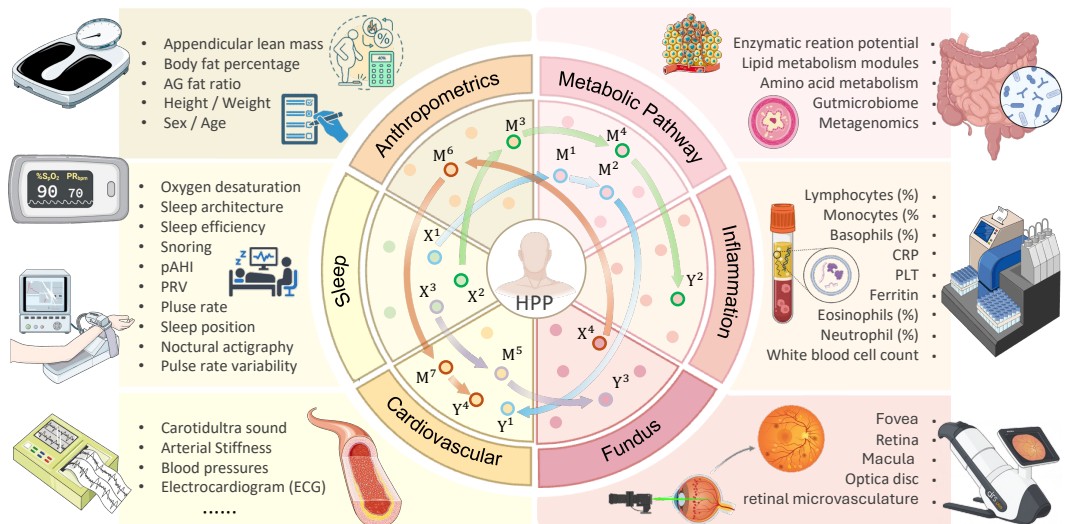

Figure 1: **Overview of the multi-modal Human Phenotype Project (HPP) dataset**. This diagram illustrates the six key body-level domains central to our study. The indexed variable series $\{X^i, Y^i, M^i\}$ represent the complex web of causal pathways. For clarity, this figure selectively illustrates a subset of these causal relationship, focusing on significant pathways that are either discovered in prior work or newly validated by our model. Refer Appendix H for detailed description.

## 3.1 STRUCTURE-PRESERVING LOSSES

We implement the CSP principle through three complementary loss functions. These losses are defined using a training target $y$ (representing the outcome $Y$ or its proxy $\phi(I^Y)$) and an auxiliary outcome embedding $\psi_Y : \mathbb{R}^k \to \mathbb{R}^d$ for InfoNCE computation, while encoder $E_Y$ produces $Z_Y = E_Y(Y)$ for downstream tasks. This joint optimization can yield causally coherent representations.

**Conditional independence preservation.** To enforce $Z_T \perp Y^* \mid Z_M$, we minimize the Conditional Mutual Information (CMI), $MI(Z_T; Y \mid Z_M)$. Following the variational approach for CMI estimation (Poole et al., 2019), we estimate $MI(Z_T; Y \mid Z_M)$ using the negative log-likelihood difference between two independently parameterized neural predictors, $q_\theta$ and $q_\phi$:

$$L_{CI} = \mathbb{E}[-\log q_\phi(y \mid z_m)] - \mathbb{E}[-\log q_\theta(y \mid z_t, z_m)] \tag{1}$$

Here, $q_\theta$ approximates the joint conditional distribution $p(y \mid z_t, z_m)$ while $q_\phi$ learns $p(y \mid z_m)$ with $z_t$ and $z_m$ being the treatment and mediator representations. For continuous $y \in \mathbb{R}^k$, both heads output diagonal-Gaussian parameters ($\mu, \sigma^2 \in \mathbb{R}^k$); for discrete $y$, they output class probabilities.

**Markov boundary retention.** This component prevents the trivial satisfaction of conditional independence that can occur from discarding mediator information:

$$L_{MBR} = -\text{InfoNCE}(z_m, \psi_Y(y)) \tag{2}$$

where $\psi_Y : \mathbb{R}^k \to \mathbb{R}^d$ is a deterministic outcome embedding. Following van den Oord et al. (2019), the InfoNCE objective provides a lower bound on the mutual information $MI(Z_M; \psi_Y(Y))$ (hence also regularizing $MI(Z_M; Y)$) and prevents the collapse of the mediator representation. When outcome embeddings are unavailable, an energy-style regularizer $\lambda_{en}\|Z_M\|^2$ can be used instead. In the DUAL configuration, the two image modalities never jointly appear within the same conditioning set of a loss term.

**Monotonic alignment consistency.** When ground-truth semantic labels $\{a_i\}_{i=1}^n \in \mathbb{R}$ are available that quantify the underlying semantic content of each sample, we enforce monotonic correspondence between semantic ordering and representation distance, this loss maximizes their Spearman's rank correlation Spearman (1904) ($\rho_S$):

$$\mathcal{L}_{MAC} = -\rho_S(\text{soft\_rank}(\Delta a), \text{soft\_rank}(\Delta z)) \tag{3}$$

where $\Delta a_{ij} = |a_i - a_j|$ denotes semantic amplitude differences with $a \in \mathbb{R}$ as semantic labels, $\Delta z_{ij} = \|z_i - z_j\|_2$ denotes L2 representation distances, and soft\_rank (Blondel et al., 2020) provides

a differentiable rank approximation. The weight $w_{MAC}$ is set to 0 if semantic ordering $\{a_i\}_{i=1}^n$ is not provided. This constraint ensures that semantically similar samples (small $|a_i - a_j|$) are mapped to nearby representations (small $\|z_i - z_j\|_2$).

The overall optimization objective is:

$$\mathcal{L}(\mathcal{E}) = w_{CI}\mathcal{L}_{CI} + w_{MBR}\mathcal{L}_{MBR} + w_{MAC}\mathcal{L}_{MAC} + \mathcal{R}(\mathcal{E}) \tag{4}$$

where $w_{CI}, w_{MBR}, w_{MAC} > 0$ are loss weights and $\mathcal{R}(\mathcal{E}) = \lambda_{\text{align}} \cdot \mathcal{L}_{\text{align}} + \lambda_{\text{style}} \cdot \mathcal{L}_{\text{style}} + \lambda_{IB} \cdot \mathcal{L}_{IB}$ includes cross-modal alignment loss $\mathcal{L}_{\text{align}}$ ensuring representations of paired modalities are close in the shared space $\mathcal{Z}$, style consistency regularization $\mathcal{L}_{\text{style}}$ preventing nuisance factors from dominating learned representations, and information bottleneck term $\mathcal{L}_{IB}$ controlling representation complexity. Configuration-specific alignment pairs are $(Z_M, Z_{IM})$ in IM and $(Z_{IY}, Z_M)$ in {IY, DUAL}, with constraint $Z_T \perp Z_{IM}$ enforced in IM configuration (details in Appendix D).

## 3.2 DISCOVERY AND TRACING

Let $\bar{Z} = (Z_T, Z_M, Z_Y, Z_{IM}, Z_{IY}, Z_X)$ be the joint random vector of the learned representations with distribution $P_{\bar{Z}}$, and let $\mathcal{G}_Z$ denote the true causal graph over $\bar{Z}$. We define a causal discovery operator $\Psi : \mathcal{P}(\bar{Z}) \to \text{CPDAG}(\bar{Z})$ that generates the corresponding completed partially directed acyclic graph (CPDAG) from the set of conditional independencies in the distribution.

For any pair of variables $i \neq j$ and any feasible conditioning set

$$\mathcal{S}_{ij} \equiv \left\{ S \subseteq \bar{Z} \setminus \{Z_i, Z_j\} : |\{Z_{IM}, Z_{IY}\} \cap S| \leq 1 \right\}, \tag{5}$$

the separation indicator is defined as $\delta(Z_i, Z_j|S) = \mathbf{1}\{Z_i \perp Z_j | Z_S\}$. Under the assumption of multivariate Gaussian and faithfulness, conditional independence $Z_i \perp Z_j | Z_S$ is equivalent to a zero partial canonical correlation, $\rho_{ij|S} = 0$. For the univariate case where $\dim(Z_i) = \dim(Z_j) = 1$, the test statistic is given by Fisher's z-transformation:

$$T_{ij|S} \equiv \sqrt{n - |S| - 3} \, \text{arctanh}(\widehat{\rho}_{ij|S}) \overset{H_0}{\rightsquigarrow} \mathcal{N}(0, 1).$$

In the general multivariate case, we employ the Wilks-Bartlett approximation. This test uses the product of the eigenvalues $\{\widehat{\kappa}_\ell\}$ of the partial canonical correlations to form the statistic $\Lambda_{ij|S} \equiv \prod_\ell (1 - \widehat{\kappa}_\ell^2)$ and tests the null hypothesis $H_0 : \kappa_\ell = 0 \, \forall \ell$. The significance of all tests is determined using the Benjamini-Hochberg procedure at level $q$ to control the false discovery rate.

From these tests, the graph skeleton $E_Z$ is constructed by connecting any pair $\{Z_i, Z_j\}$ not separated by any conditioning set in $\mathcal{S}_{ij}$ as defined by:

$$E_Z \equiv \{\{Z_i, Z_j\} : \nexists S \in \mathcal{S}_{ij} \text{ s.t. } \delta(Z_i, Z_j|S) = 1\} \tag{6}$$

The set of separating conditionals for a pair is denoted by $\text{Sep}(Z_i, Z_j) = \{S \in \mathcal{S}_{ij} : \delta(Z_i, Z_j|S) = 1\}$. The skeleton is then oriented by first identifying all v-structures according to

$$(Z_i \to Z_k \leftarrow Z_j) \iff \{Z_i, Z_k\}, \{Z_j, Z_k\} \in E_Z \text{ and } \forall S \in \text{Sep}(Z_i, Z_j), Z_k \notin S$$

and subsequently applying the Meek rule closure to propagate all remaining edge orientations, yielding the final estimated graph $\widehat{\text{CPDAG}}_Z = \Psi(P_{\bar{Z}})$. To render the latent graph interpretable, we define a variable tracing mechanism using a surjective mapping $\pi : \bar{Z} \to \{T, M, Y^*\}$ and a calibration map $h : \text{range}(Z_Y) \to \text{range}(Y)$. The surjection $\pi$ links each representation to its source variable:

$$\pi(Z_T) = T, \quad \pi(Z_M^{\text{tab}}) = \pi(Z_M^{\text{img}}) = M, \quad \pi(Z_Y) = Y^*. \tag{7}$$

The calibration map $h$ ensures that $h(Z_Y) = Y + \varepsilon_{\text{cal}}$, where the calibration error satisfies $\mathbb{E}[\varepsilon_{\text{cal}}|Z_Y] = 0$. Finally, we use the pushforward operator $\pi_*$ to project the latent graph back to the variable space, defining:

$$\widehat{\text{CPDAG}}_V \equiv \pi_*(\widehat{\text{CPDAG}}_Z). \tag{8}$$

This operator maps each edge $(Z_i \to Z_j)$ in the latent graph to an edge $(\pi(Z_i) \to \pi(Z_j))$ in the variable-level graph, which is then simplified according to CPDAG rules after node merging.

The guarantee of consistency holds under several standard assumptions: (i) multivariate Gaussianity and faithfulness of the data distribution; (ii) the mutual exclusion and anti-bypass conditions enforced during training; (iii) a minimum non-zero partial correlation lower bound $\gamma > 0$; and (iv) a bounded maximum in-degree for the graph ($\Delta < \infty$). Given these conditions, we have

$$\Pr[\widehat{\text{CPDAG}}_Z = \text{CPDAG}(\mathcal{G}_Z)] \to 1 \text{ as } n \to \infty, \tag{9}$$

and $\pi_*(\text{CPDAG}(\mathcal{G}_Z))$ is topologically equivalent to ground-truth variable graph $\text{CPDAG}(\mathcal{G}_V)$

## 3.3 THEORETICAL PROPERTIES

Under the assumptions established in Sections 2.1 and 2.3, the CARL framework satisfies the following guarantees.

**Theorem 1** (CSP Achievability). *Under Assumptions A.5 and spectral norm constraints in A.8.2, consider the loss function*

$$\mathcal{L} = w_{CI} \cdot \mathcal{L}_{CI} + w_{MBR} \cdot \mathcal{L}_{MBR} + w_{MAC} \cdot \mathcal{L}_{MAC}, \tag{10}$$

*where*

$$\mathcal{L}_{CI} = \mathbb{E}[-\log q_\phi(y \mid z_m)] - \mathbb{E}[-\log q_\theta(y \mid z_t, z_m)], \tag{11}$$

$$\mathcal{L}_{MBR} = -InfoNCE(z_m, \psi_Y(y)), \tag{12}$$

$$\mathcal{L}_{MAC} = -\rho_S(soft\_rank(\Delta a), soft\_rank(\Delta z)), \tag{13}$$

*with $q_\theta, q_\phi$ independently parameterized. Then the empirical risk minimization admits approximate minimizing sequences whose limit point $(\mathcal{E}^*, \mathcal{Z}^*)$ satisfies $\varepsilon$-CSP with*

$$\varepsilon = \max\{\zeta^*, O_P(n^{-1/2}), O_P(K^{-1/2}), O_P(n^{-1/3})\}, \tag{14}$$

*where $K$ denotes the number of negative samples, $n$ is the sample size, and $\zeta^*$ is the essential approximation error for the encoder class.*

**Proof Sketch.** Under **A.5** (realizability + regularity with cross-fitting), our three surrogates are risk-consistent: $\mathcal{L}_{CI}$ calibrates $MI(Z_T; Y \mid Z_M)$, $\mathcal{L}_{MBR}$ prevents mediator collapse via an MI lower bound for $I(Z_M; Y)$, and $\mathcal{L}_{MAC}$ enforces monotone alignment through a soft-rank approximation to Spearman's $\rho_S$ (with approximation error $O_P(n^{-1/3})$). Together with **A.8.2** (spectral/Lipschitz control) ensuring compactness and stability, the sampling and approximation errors aggregate to Eq. (14) (dominated by $\zeta^*$, $O_P(n^{-1/2})$, $O_P(K^{-1/2})$, and $O_P(n^{-1/3})$), yielding $\varepsilon$-CSP.

**Remark 1.** *The three loss components target the CSP conditions: $\mathcal{L}_{CI}$ optimizes condition (i), $\mathcal{L}_{MBR}$ ensures condition (ii), and $\mathcal{L}_{MAC}$ guarantees condition (iii). When $K = \Omega(n)$, the dominant error term becomes $O_P(n^{-1/2})$. The weights $(w_{CI}, w_{MBR}, w_{MAC})$ determine which Pareto optimal point is reached.*

**Theorem 2** (Causal Query Preservation). *Suppose a representation system satisfies $\varepsilon$-CSP and the causal query $Q = \mathbb{E}[Y^*(t)]$ is identifiable in the original data space via criterion $\mathcal{C} \in \{backdoor, frontdoor, instrumental\ variable\}$. Let $\tilde{Q}$ be the corresponding query computed in the representation space using the same criterion. Then*

$$|\tilde{Q} - Q| \leq \kappa \cdot \varepsilon + \delta_{cal}, \tag{15}$$

*where $\kappa$ is the sensitivity constant of the identification formula, and*

$$\delta_{cal} = \sup_{(t,x)} |\mathbb{E}[Y^* \mid t, x] - \mathbb{E}[h(Z_Y) \mid t, x]| \tag{16}$$

*is the conditional calibration error satisfying $\delta_{cal} = o_P(1)$ under cross-fitting and realizability.*

**Proof Sketch.** We apply the same identification functional (e.g., backdoor/frontdoor) in the representation space. By **A.8.2**, the map from inputs to representations is Lipschitz, so the structural deviation $\varepsilon$ propagates *at most linearly* to the causal query, giving $|\tilde{Q} - Q| \leq \kappa\varepsilon + \delta_{\text{cal}}$. The calibration term $\delta_{\text{cal}}$ is $O_P(n^{-1/2})$ under **A.5** via cross-fitting. (Full proofs in Appendix C.9)

**Remark 2.** *For backdoor: $\tilde{Q} = \sum_{z_x} \mathbb{E}[h(Z_Y) \mid E_T(t), z_x]P(z_x)$ with $\kappa = 1$. This holds for IY and DUAL configurations where simultaneous conditioning on $Z_{IM}$ and $Z_{IY}$ is avoided in both training and inference.*

**Corollary 3** (Overall Approximation Bound). *Under the conditions of Theorems 1 and 2, the learned encoders $(\hat{\mathcal{E}}, \hat{\mathcal{Z}})$ satisfy $\varepsilon$-CSP with*

$$\varepsilon \leq \zeta^* + O_P(n^{-1/2}) + O_P(K^{-1/2}) + O_P(n^{-1/3}), \tag{17}$$

*where $\zeta^*$ is the essential approximation error for the encoder class. Moreover, the PC algorithm is consistent in the representation space, and if $\pi$ is compatibility-preserving (all members in each fiber share identical adjacencies and orientations to outside nodes), then*

$$\pi_*(\widehat{CPDAG}_Z) \equiv CPDAG(\mathcal{G}_V). \tag{18}$$

*Proof.* The bound follows from Theorem 1. Consistency and topological equivalence follow from Section 3.2 and the compatibility-preserving property of $\pi$. $\qquad\square$

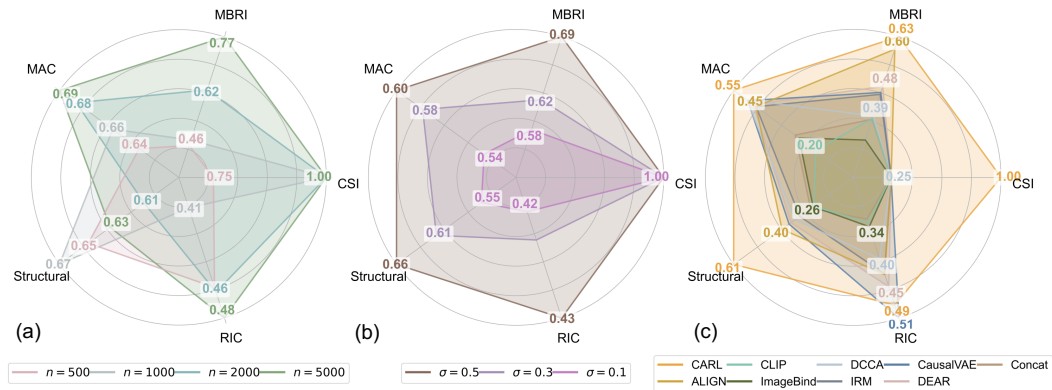

Figure 4: Structure-preservation evaluation. Panels report five metrics (CSI, MBRI, MAC, Structural, RIC) under (a) sample sizes $n \in \{500, 1000, 2000, 5000\}$, (b) noise levels $\sigma \in \{0.1, 0.3, 0.5\}$, and (c) comparision with baselines. Refer to Table 5 for specific values.

## 4 EXPERIMENT RESULTS

We evaluate CARL on two complementary datasets: a synthetic benchmark with known causal ground truth for systematic evaluation, and the real-world HPP dataset for biomedical validation.

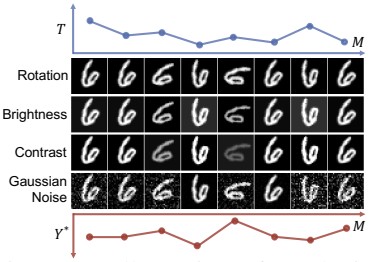

Figure 2: Illustration of synthetic data generation in IM scenario.

**Synthetic dataset.** We manually create a synthetic benchmark dataset to encode a known ground-truth causal relationships between different modalities, using MNIST Lecun et al. (1998). The data generation is grounded in a Structural Causal Model (SCM) that defines a latent causal chain $T \to M \to Y^*$, where $T$ is an exogenous treatment, $M$ is a mediator, and $Y^*$ is an outcome. To create the cross-modal link, the latent causal variables are first mapped to a normalized semantic amplitude $\alpha \in [0, 1]$. This amplitude then deterministically controls a sequence of visual transformations (e.g., rotation, brightness, contrast) applied to a base MNIST digit. This mechanism is used to instantiate three distinct experimental configurations as described in Section 2.2: IM, IY, DUAL. In Figure 2, we illustrate the data synthesis process for the IM scenario, where the mediator $M$ is encoded as the image modality. This design ensures a structured and quantifiable causal dependency that spans both tabular and visual modalities. Full details of the generation process are provided in Appendix I.

**HPP Dataset.** The HPP dataset, introduced in Section 2.4, is used to validate CARL's ability to preserve known causal pathways in a real-world biomedical application. Implementation details are in Appendix H.

**Baselines.** CARL is benchmarked against the most representative cross-modal methods, including CLIP Radford

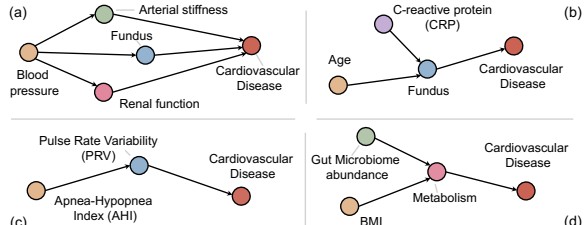

Figure 3: HPP mediation pathways.

et al. (2021), ImageBind Girdhar et al. (2023), DCCA Andrew et al. (2013), CausalVAE Yang et al. (2021), ALIGN Jia et al. (2021), IRM Arjovsky et al. (2019), DEAR Shen et al. (2022) and a feature concatenation baseline.

**Metrics.** We use five core metrics designed to probe causal preservation: Causal Structure Index (CSI), Markov Boundary Retention Index (MBRI), Monotonic Alignment Consistency (MAC), Structural Accuracy (Structural), and Representation Information Content (RIC). Full list of results and experimental setup details are in the Appendix F.

Table 1: HPP results: Summary of total effect (TE), natural direct effect (NDE), natural indirect effect (NIE), and mediation proportion of key pathways in latent space $\mathcal{Z}$.

| Causal Pathways (Latent Space $\mathcal{Z}$) | TE | NDE | NIE | Mediation Proportion (%) |
|---|---|---|---|---|
| $\mathcal{Z}_{\text{Blood pressure}} \rightarrow \mathcal{Z}_{\text{CVD}}$ | 0.486 | 0.271 | 0.215 | 44.24 |
| $\mathcal{Z}_{\text{Blood pressure}} \rightarrow \mathcal{Z}_{\text{Arterial stiffness}} \rightarrow \mathcal{Z}_{\text{CVD}}$ | - | - | 0.097 | 19.96 |
| $\mathcal{Z}_{\text{Blood pressure}} \rightarrow \mathcal{Z}_{\text{Fundus}} \rightarrow \mathcal{Z}_{\text{CVD}}$ | - | - | 0.074 | 15.23 |
| $\mathcal{Z}_{\text{Blood pressure}} \rightarrow \mathcal{Z}_{\text{Renal function}} \rightarrow \mathcal{Z}_{\text{CVD}}$ | - | - | 0.044 | 9.05 |
| $\mathcal{Z}_{\text{Age}} \rightarrow \mathcal{Z}_{\text{Fundus}} \rightarrow \mathcal{Z}_{\text{CVD}}$ | 0.513 | 0.437 | 0.076 | 14.81 |
| $\mathcal{Z}_{\text{CRP}} \rightarrow \mathcal{Z}_{\text{Fundus}} \rightarrow \mathcal{Z}_{\text{CVD}}$ | 0.378 | 0.319 | 0.059 | 15.61 |
| $\mathcal{Z}_{\text{AHI}} \rightarrow \mathcal{Z}_{\text{PRV}} \rightarrow \mathcal{Z}_{\text{CVD}}$ | 0.352 | 0.281 | 0.071 | 20.17 |
| $\mathcal{Z}_{\text{BMI}} \rightarrow \mathcal{Z}_{\text{Metabolism}} \rightarrow \mathcal{Z}_{\text{CVD}}$ | 0.394 | 0.325 | 0.069 | 17.51 |
| $\mathcal{Z}_{\text{Gut Microbiome}} \rightarrow \mathcal{Z}_{\text{Metabolism}} \rightarrow \mathcal{Z}_{\text{CVD}}$ | 0.316 | 0.248 | 0.068 | 21.52 |

## 4.1 CAUSAL PATHWAY DISCOVERY IN HPP DATASET

Based on CARL's structure-preserving learning and causal discovery as shown in Figure 3, we obtained the following stable effect decompositions on HPP data (See Table 1): for "blood pressure → cardiovascular events (CVD)", total effect TE=0.486, NDE=0.271, NIE=0.215, with overall mediation proportion of 44.24%; indirect effects are primarily transmitted through arterial stiffness (19.96%), retinal microvascular changes (15.23%), and renal function (9.05%); we simultaneously identified age → retinal fundus → CVD (14.81%), CRP → retinal fundus → CVD (15.61%), AHI → PRV/HRV → CVD (20.17%), BMI → metabolism → CVD (17.51%), and gut microbiome → metabolism → CVD (21.52%). These pathways are consistent with independent high-level research directions and have interpretable mechanisms: blood pressure reduction shows clear dose-response/risk reduction relationships with major adverse cardiovascular events G.1; arterial stiffness (PWV) can independently predict events and death G.2; retinal microvascular abnormalities predict stroke and coronary heart disease G.3; renal function impairment (decreased eGFR, increased proteinuria) increases CVD and mortality risk G.4; retinal "biological age gap" correlates with all-cause/cardiovascular mortality G.5; systemic inflammation (CRP) shows dose-response with microvascular caliber changes and links to CVD G.6; OSA-related HRV reduction indicates higher CVD risk G.7; obesity mediates considerable proportion of CVD risk through metabolic factors such as blood pressure/lipids/glucose G.8; microbiome-TMAO and other metabolites are closely associated with atherosclerosis and adverse events G.9.

## 4.2 SYNTHETIC DATA VALIDATION

Figure 4 demonstrates CARL's structure-preserving performance across multiple evaluation dimensions. In sample size scaling experiments ($n \in \{500, 1000, 2000, 5000\}$), CARL maintains CSI at 1.0 with structural accuracy stable within 0.61-0.75, validating that the $\varepsilon$-CSP bound tightens at the theoretical $O_P(n^{-1/2})$ rate. Under noise robustness testing ($\sigma \in \{0.1, 0.3, 0.5\}$), CARL's MAC degrades from 0.89 to 0.42 while CSI remains at 1.0, and MBRI decreases from 0.77 to 0.63, indicating that monotonic alignment constraints maintain semantic-geometric correspondence under distributional perturbation. Baseline comparisons reveal CLIP's CSI at 0.25, ImageBind's Structural at 0.33, and similar performance from DCCA, confirming that purely statistical objectives cannot guarantee conditional independence pattern preservation.

## 4.3 ABLATION STUDY

Table 2 quantifies the contribution patterns of each loss function. Removing $\mathcal{L}_{CI}$ causes CSI to drop from 1.0 to 0.25 while MAC increases to 0.83 but structural accuracy falls to 0.40, indicating that semantic alignment and structural preservation decouple when conditional independence constraints are absent. Removing $\mathcal{L}_{MBR}$ reduces MBRI from 0.63 to 0.46 and CSI to 0.75, confirming the necessity of Markov boundary information for complete structural preservation. Removing $\mathcal{L}_{MAC}$ decreases MAC from 0.55 to 0.32 while CSI remains at 1.0, showing that semantic-geometric consistency and conditional independence preservation are mutually independent. The alignment-only

configuration reproduces the fundamental problems of statistical methods: high MAC (0.89) but low CSI (0.25) and structural accuracy (0.32).

Table 3 validates the necessity of key design choices. Shared predictor heads reduce CSI from 1.0 to 0.75 with slight decreases in MBRI and MAC, confirming the importance of independent parameterization for unbiased CMI estimation. Absence of cross-validation causes CSI to plummet to 0.25, indicating severe risks of overfitting spurious conditional dependencies. With $K = 32$ negative samples, all metrics decline and CSI drops to 0.75, validating the impact of InfoNCE estimation quality on structural preservation. Representation dimension $d = 16$ performs comparably to the full configuration, with all metric differences within 0.05, confirming the stability of constraint mechanisms across different representation scales.

Table 2: Loss function ablation: impact on structure preservation performance.

|  | CSI | MBRI | MAC | Structural | RIC |
|---|---|---|---|---|---|
| w/o $\mathcal{L}_{CI}$ | 0.25 | 0.62 | 0.83 | 0.40 | 0.44 |
| w/o $\mathcal{L}_{MBR}$ | 0.75 | 0.46 | 0.54 | 0.52 | 0.37 |
| w/o $\mathcal{L}_{MAC}$ | 1.00 | 0.63 | 0.32 | 0.56 | 0.42 |
| only $\mathcal{L}_{align}$ | 0.25 | 0.66 | 0.89 | 0.32 | 0.49 |
| CARL (Full) | 1.00 | 0.63 | 0.55 | 0.61 | 0.42 |

Table 3: Architecture design ablation study: impact of key design choices on performance.

|  | CSI | MBRI | MAC | Structural | RIC |
|---|---|---|---|---|---|
| Shared Predictor Head | 0.75 | 0.61 | 0.53 | 0.54 | 0.41 |
| w/o Cross Validation | 0.25 | 0.66 | 0.63 | 0.37 | 0.43 |
| $K : 32$ | 0.75 | 0.55 | 0.49 | 0.48 | 0.41 |
| $d : 16$ | 1 | 0.58 | 0.56 | 0.60 | 0.43 |
| CARL (Full) | 1 | 0.63 | 0.55 | 0.61 | 0.42 |

## 5 RELATED WORK

Cross-modal representation learning has achieved great success in aligning heterogeneous data like images and text into a shared semantic space using contrastive objectives Radford et al. (2021); Jia et al. (2021); Girdhar et al. (2023). However, these methods optimize statistical associations and are agnostic to the underlying causal graph, risking the distortion of causal pathways Arjovsky et al. (2019); Geirhos et al. (2020). In parallel, Causal Representation Learning (CRL) aims to learn causally robust representations that support generalization and counterfactual reasoning Schölkopf et al. (2021); Bagi et al. (2023); Zuo et al. (2023). Yet, CRL has predominantly focused on unimodal data and has not systematically addressed the unique challenges of multi-modal settings, such as information asymmetry across modalities or ensuring consistent geometric-semantic alignment Liang et al. (2022); Zhang et al. (2024). While initial exploration into multi-modal causal representation learning exist Sun et al. (2025), the disconnect between the scaling power of modern cross-modal models and the structural requirements for reliable, verifiable causal inference creates a critical gap. Our work aims to bridge this gap by introducing a framework that explicitly preserves key causal structures, including conditional independencies and effect identifiability conditions, during cross-modal representation learning. Please refer to Appendix E for a detailed literature review.

## 6 CONCLUSION

We address the problem of representation-induced structural drift in cross-modal representation learning through the introduction of the Causal Structure Preservation principle and the CARL framework. We identify and formalize three core challenges: Cross-modal Information Bottleneck, Modal Alignment Consistency, and Cross-modal Identifiability Consistency, proposing a joint optimization strategy combining conditional independence preservation, Markov boundary retention, and monotonic alignment consistency. Theoretical analysis establishes achievability guarantees for $\varepsilon$-CSP and proves the preservation of causal query identifiability in representation space. Validation on the HPP dataset demonstrates that CARL preserves complex mediation pathways from blood pressure to cardiovascular disease through arterial stiffness, retinal microvascular changes, and renal function. Synthetic benchmark testing reveals the limitations of purely statistical optimization methods in structural preservation while validating the theoretical necessity and practical effectiveness of our three loss function components. This framework provides verifiable theoretical foundations and methodological support for reliable causal inference in multimodal data environments.

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

# Supplement to "CARL: Preserving Causal Structure in Representation Learning"

LLM USAGE DISCLOSURE

In the preparation of this paper, we made limited use of large language model (LLM) tools such as ChatGPT and Gemini. After drafting the manuscript, we used these tools only for minor language polishing. All research ideas, analyses, and the substantive content of the paper were conceived, developed, and written by the authors.

## A  NOTATION AND ASSUMPTIONS

### A.1  NOTATION

Throughout this paper, we adopt the following notation conventions. Let $T \in \mathcal{T}$ denote the treatment variable, $M \in \mathcal{M}$ the mediator variable, $Y^* \in \mathcal{Y}$ the true but unobserved outcome, $Y \in \mathbb{R}^k$ the observed outcome (reduces to scalar when $k = 1$), and $X \in \mathcal{X}$ the covariate set. Image observations $I^M, I^Y \in \mathbb{R}^{H \times W \times C}$ have height $H$, width $W$, and $C$ channels. The shared representation space is denoted as $\mathcal{Z} \subset \mathbb{R}^d$ where $d$ is the representation dimension.

The encoder family $\mathcal{E}$ comprises the following mappings: treatment encoder $E_T : \mathcal{T} \to \mathcal{Z}$, mediator encoder $E_M : \mathcal{M} \to \mathcal{Z}$, image-mediator encoder $E_{I^M} : I^M \to \mathcal{Z}$, image-outcome encoder $E_{I^Y} : I^Y \to \mathcal{Z}$, and optional outcome encoder $E_Y : \mathbb{R}^k \to \mathcal{Z}$ and covariate encoder $E_X : \mathcal{X} \to \mathcal{Z}$. The corresponding representations are denoted as $Z_T = E_T(T)$ for encoded treatment, $Z_M$ obtained from either $E_M(M)$ or $E_{I^M}(I^M)$, and $Z_Y$ from either $E_Y(Y)$ or $E_{I^Y}(I^Y)$.

Key functions include the parameterized conditional distributions $q_\theta : \mathcal{Z} \times \mathcal{Z} \to \mathcal{P}(\mathbb{R}^k)$ and $q_\phi : \mathcal{Z} \to \mathcal{P}(\mathbb{R}^k)$ modeling $p(y|z_t, z_m)$ and $p(y|z_m)$ respectively, outcome embedding function $\psi_Y : \mathbb{R}^k \to \mathbb{R}^d$, image-to-outcome extraction function $\phi : I^Y \to \mathbb{R}^k$, posterior calibration function $h : \mathcal{Z} \to \mathbb{R}^k$, and differentiable rank approximation $\text{soft\_rank}_{\tau_{\text{rank}}}(\cdot)$ with temperature $\tau_{\text{rank}}$.

Loss function parameters include primary loss weights $w_{CI}, w_{MBR}, w_{MAC}$, regularization coefficients $\lambda_{\text{align}}, \lambda_{\text{style}}, \lambda_{IB}$, temperature parameters $\tau_{\text{align}}$ for InfoNCE alignment, $\tau_{\text{rank}}$ for soft-rank approximation, balance coefficient $\tau_{\text{mbr}}$ for MBR loss weighting, and significance level $\alpha$ for hypothesis testing (default 0.05).

### A.2  IDENTIFICATION CONDITIONS

**Definition A.2.1 (Backdoor Criterion)** Given a DAG $\mathcal{G} = (V, E)$, a set of variables $S \subseteq V \setminus \{T, Y^*\}$ satisfies the backdoor criterion relative to the ordered pair $(T, Y^*)$ if and only if (i) $S \cap \text{de}(T) = \emptyset$ where $\text{de}(T)$ denotes the descendants of $T$, and (ii) $S$ blocks all paths of the form $T \leftarrow \cdots \rightarrow Y^*$.

**Definition A.2.2 (Frontdoor Criterion)** A set of variables $M \subseteq V \setminus \{T, Y^*\}$ satisfies the frontdoor criterion relative to $(T, Y^*)$ if and only if (i) $\forall \pi \in \Pi_{T \to Y^*} : M \cap \pi \neq \emptyset$ where $\Pi_{T \to Y^*}$ denotes all directed paths from $T$ to $Y^*$, (ii) $(T \perp M \mid \emptyset)_{\mathcal{G}_{\overline{T \to M}}}$ holds in the graph with edges $T \to M$ removed, and (iii) $(M \perp Y^* \mid T)_{\mathcal{G}_{\underline{M \to Y^*}}}$ holds in the graph with edges $M \to Y^*$ removed.

**Definition A.2.3 (Instrumental Variable)** A variable $Z$ is an instrumental variable for $(T, Y^*)$ if and only if (i) $\text{Cov}(Z, T) \neq 0$ (relevance), (ii) $(Z \perp Y^*)_{\mathcal{G}_{\overline{T}}}$ holds in the graph with $T$ and its outgoing edges removed (exclusion restriction), and (iii) $Z \perp U$ where $U$ denotes unobserved confounders affecting $(T, Y^*)$.

### A.3  POSITIVITY AND CONSISTENCY

**Definition A.3.1 (Positivity)** The treatment mechanism satisfies the positivity condition if and only if there exist constants $c_1, c_2 \in (0, 1)$ such that $c_1 \leq P(T = t \mid X = x) \leq c_2$ for all $(t, x) \in \text{supp}(T, X)$. For continuous treatments, we require bounded density $\epsilon_1 \leq f_{T|X}(t|x) \leq \epsilon_2$ almost everywhere for some $\epsilon_1, \epsilon_2 > 0$.

**Definition A.3.2 (SUTVA)** Let $Y_i(t)$ denote the potential outcome of unit $i$ under treatment $t$. The Stable Unit Treatment Value Assumption holds if and only if (i) $Y_i(\mathbf{t}) = Y_i(t_i)$ for all treatment

vectors $\mathbf{t} = (t_1, \ldots, t_n)$ (no interference), and (ii) for all $t \in \mathcal{T}$, there exists a unique potential outcome $Y_i(t)$ (treatment uniqueness).

## A.4 CAUSAL MARKOV PROPERTY AND FAITHFULNESS

### A.4.1 FORMAL STATEMENT OF CAUSAL MARKOV ASSUMPTION

**Definition A.1 (Causal Markov Property):** Given a directed acyclic graph $G = (V, E)$, each variable $V_i \in V$ satisfies the conditional independence relation

$$V_i \perp\!\!\!\perp (\text{NonDesc}_G(V_i) \setminus \text{Pa}_G(V_i)) \mid \text{Pa}_G(V_i)$$

*Where* $\text{NonDesc}_G(V_i) \setminus \text{Pa}_G(V_i)$ *denotes the set of non-descendant variables of* $V_i$ *in* $G$ *excluding its parents, and* $\text{Pa}_G(V_i)$ *denotes the parent set of* $V_i$.

In the basic structure $T \to M \to Y^*$ of the CARL framework, the causal Markov property manifests as: for root node $T$, we have $T \perp\!\!\!\perp \text{NonDesc}_G(T)$ (since $\text{Pa}_G(T) = \emptyset$); for $Y^*$, we have $Y^* \perp\!\!\!\perp T \mid \text{Pa}_G(Y^*) = \{M\}$; for $M$, we have $M \perp\!\!\!\perp (\text{NonDesc}_G(M) \setminus \text{Pa}_G(M)) \mid \text{Pa}_G(M) = \{T\}$. When including covariates $X$ and image observations $I^M, I^Y$, the Markov property requires all variables to be conditionally independent of their non-descendant non-parent variables given their graph-structural parents.

### A.4.2 FAITHFULNESS AND ADJACENCY FAITHFULNESS

**Definition A.2 (Faithfulness):** A distribution $P$ is faithful to DAG $G$ if and only if the conditional independence relations in $P$ correspond one-to-one with the $d$-separation relations in $G$:

$$X \perp\!\!\!\perp Y \mid Z \text{ in } P \Leftrightarrow X \perp_d Y \mid Z \text{ in } G$$

For robustness considerations in practical applications, we adopt the adjacency faithfulness condition: if $V_i$ and $V_j$ are adjacent in graph $G$, then for conditioning sets $S$ that exclude the endpoints and their descendants, we *reject* $H_0 : V_i \perp\!\!\!\perp V_j \mid S$ at level $\alpha$ (up to measure-zero exceptions).

Faithfulness fails under parameter cancellation, deterministic relationships, and special parameter configurations. Parameter cancellation occurs when causal effects cancel each other through different paths, and deterministic relationships exist in cases of strict functional dependencies. In continuous parameter spaces, parameter configurations that violate faithfulness constitute measure-zero sets.

### A.4.3 BASIC IDENTIFICATION CONDITIONS

**Positivity Condition:** The treatment variable must have positive probability of being observed under all covariate conditions:

$$0 < P(T = t \mid X = x) < 1 \text{ for all } (t, x) \text{ in support}$$

For mediator variables, we similarly require

$$0 < P(M = m \mid T = t, X = x) < 1$$

For continuous treatments/mediators, we use positive density conditions $f_{T\mid X}(t\mid x) > 0$, $f_{M\mid T,X}(m\mid t, x) > 0$ to replace discrete probabilities.

**Consistency Condition (SUTVA):** This includes the no-interference assumption and treatment version consistency. The no-interference assumption requires that individual $i$'s potential outcomes are not affected by other individuals' treatment status, while treatment version consistency requires that there are no different versions of treatment for the same treatment level.

Positivity violations manifest as treatment probabilities of 0 or 1 under certain covariate combinations, detectable through propensity score distributions. Consistency violations involve treatment spillover effects or ambiguous treatment definitions, requiring domain knowledge for assessment.

## A.5 REALIZABILITY AND REGULARITY FOR ESTIMATOR CONSISTENCY

**Assumption 1 (Causal Structure).** The data generating process follows a directed acyclic graph $G = (V, E)$ with core causal chain $T \to M \to Y^*$. This structure satisfies the Markov property (Definition A.4.1) and faithfulness (Definition A.4.2), with identification conditions detailed in Section A.4.3.

**Assumption 2 (Anti-bypass).** The outcome representation $Z_Y$ satisfies no direct information flow from $(T, M)$ bypassing the outcome pathway. When $Z_Y = E_Y(Y)$, we require $Z_Y \perp (T, M)|Y$. When $Z_Y = E_{I^Y}(I^Y)$ in IY configurations, we require $Z_Y \perp (T, M)|I^Y$ where $I^Y$ captures all outcome-relevant information. These encoders are mutually exclusive and share no feature extraction layers or gradient paths from $(T, M)$, ensuring complete branch isolation in the representation space.

**Assumption 3 (Configuration-Specific Conditions).** In the IM configuration, no spurious path $T \to I^M \to Y^*$ exists, and $I^M$ observes $M$ with independent noise. In the IY configuration, the proxy $Y = \phi(I^Y)$ satisfies monotonic calibratability with $Y^*$, meaning there exists a monotonic function $h$ such that $\mathbb{E}[Y^*|\phi(I^Y)] = h(\phi(I^Y))$. The DUAL configuration satisfies both IM and IY conditions, with $I^M$ and $I^Y$ never simultaneously conditioned in both training loss computation and discovery-phase conditioning sets.

**Assumption 4 (Technical Regularity).** Encoders satisfy Lipschitz continuity with $\|E(x_1) - E(x_2)\| \leq L\|x_1 - x_2\|$ for Lipschitz constant $L > 0$. All representations have bounded second moments $\mathbb{E}[\|Z\|^2] < \infty$. Observed samples are independently and identically distributed or satisfy mixing conditions. Representations maintain non-degeneracy with $\mathrm{Var}(Z) \geq \sigma_0^2 > 0$ preventing representation collapse.

**Assumption 5 (Estimator Consistency).** The following conditions ensure consistency of key estimators. For NLL-based CMI approximation, the predictor $q_\theta$ achieves realizability or bounded calibration error with respect to $p(y|z)$. For InfoNCE lower bounds, in-batch negative samples satisfy approximate independence. Theoretically, $K = \Omega(n)$ achieves optimal rate $O_P(n^{-1/2})$. Practically, $K \asymp \log n$ provides sufficient statistical power with error $O_P((\log n)^{-1/2})$, balancing computational cost and estimation accuracy. For soft-rank consistency, the temperature decays as $\tau_{\mathrm{rank}} = O(n^{-\alpha})$ for $\alpha \in (0, 1/2)$ ensuring both rank consistency and bounded gradients.

## A.6 MEASUREMENT AND PAIRING ASSUMPTIONS

### A.6.1 FORMALIZATION OF SEPARABLE MEASUREMENT ASSUMPTION

**Definition A.3 (Separable Measurement Structure):** Image observations satisfy separable measurement structure

$$I^M = g_M(a_M, b_{\mathrm{style}}; \eta_M), \quad I^Y = g_Y(a_Y, b_{\mathrm{style}}; \eta_Y)$$

where $a_M, a_Y$ represent semantic content variables related to causal variables, $b_{\mathrm{style}}$ denotes the style variable affecting visual appearance but not causal relationships, and $\eta_M, \eta_Y$ are measurement noises independent of all causally relevant variables.

Key independence conditions for separable measurement include noise independence $(\eta_M, \eta_Y) \perp\!\!\!\perp (T, M, Y^*, X)$ and style conditional independence $b_{\mathrm{style}} \perp\!\!\!\perp (T, Y^*) \mid (M, X)$.

### A.6.2 VERIFICATION METHODS

At significance level $\alpha$, we fail to reject $H_0 : \mathrm{Corr}(\hat{\eta}, T) = 0$ (HSIC/permutation-based independence tests; multiple testing controlled via Benjamini–Hochberg FDR). For style independence, we test $H_0 : I(b_{\mathrm{style}}; T, Y^* \mid M, X) = 0$ (conditional mutual information permutation test).

Practical verification strategies include using style disentanglement techniques to verify the separability of style variables, verifying the invariance of causal relationships through multiple style transformations, and cross-validating the consistency of causal effects under different style conditions.

### A.6.3 PAIRED DATA PREREQUISITES

Strong pairing conditions require entity-level exact pairing $(I_i^M, I_i^Y, M_i, Y_i^*, T_i, X_i)$, where the subscript $i$ indicates the same individual or entity. When exact pairing is infeasible, we allow temporal window pairing (observations paired within reasonable time windows), matched pairing (nearest neighbor matching based on key covariates), and distributional alignment (ensuring marginal distribution matching $P(I^M) \approx P_{\text{target}}(I^M)$).

Paired data is used to construct positive sample pairs in InfoNCE loss. When paired data is unavailable, we need to ensure positive sample pairs come from the same semantic category or continuous semantic neighborhood, filtering alternative positive sample pairs through semantic similarity thresholds. Under weak pairing or distributional alignment (e.g., MMD), if the sampling and cross-fitting conditions in B.3 and B.4 are satisfied, **and positive pairs/negatives are sampled independently**, the validity of lower bounds and bias bounds can be maintained.

## A.7 SCENARIO-SPECIFIC CONDITIONS

### A.7.1 FRONT-DOOR CRITERION AND INSTRUMENTAL VARIABLE CONDITIONS

**Front-door Criterion (Three Conditions):** For path $T \to M \to Y^*$, $M$ as a front-door variable must satisfy: $M$ intercepts all causal paths from $T$ to $Y^*$, back-door blocking from $T$ to $M$, and back-door blocking from $M$ to $Y^*$. Specifically, all directed paths from $T$ to $Y^*$ pass through $M$, there are no back-door paths from $T$ to $M$ or all back-door paths are blocked by $X$, and given $T$ and $X$, all back-door paths from $M$ to $Y^*$ are blocked.

**Instrumental Variable (Three Conditions):** Let $Z$ be an instrumental variable for $T$, satisfying Relevance: $\text{Cov}(Z, T) \neq 0$ (excluding weak instruments; first-stage F-statistic exceeds conventional thresholds, e.g., $> 10$); Exclusion restriction: $Z \perp\!\!\!\perp Y^* \mid (T, X)$; Exogeneity: $Z \perp\!\!\!\perp U \mid X$, where $U$ represents unobserved confounders affecting $Y^*$.

Positivity conditions require $0 < P(M = m \mid T = t, X = x) < 1$ in the front-door case and $0 < P(T = t \mid Z = z, X = x) < 1$ in the IV case.

### A.7.2 SCENARIO-SPECIFIC NECESSARY AND SUFFICIENT CONDITIONS

**IM Scenario Conditions:** The necessary condition is the absence of directed path $T \to I^M \to Y^*$. The sufficient condition is satisfied when $I^M = g_M(M, b_{\text{style}}; \eta_M)$ with $\eta_M \perp\!\!\!\perp Y^*$ and $b_{\text{style}} \perp\!\!\!\perp Y^* \mid M$. Violation detection **fails to reject** $H_0 : I(I^M; Y^* \mid M, X) = 0$ at level $\alpha$ (conditional mutual information test); **if $H_0$ is rejected, we deem a violation**.

**IY Scenario Conditions:** Proxy sufficiency requires $Y^* \perp\!\!\!\perp X \mid \phi(I^Y), T$, where $\phi(I^Y)$ contains all information about $Y^*$ given $X$. Strict monotonicity requires the existence of a strictly monotonic function $h$ such that

$$\mathbb{E}[Y^* \mid \phi(I^Y)] = h(\phi(I^Y)), \text{ where } h \text{ is strictly monotonic}$$

Equivalently, monotonicity manifests as $\Pr(Y_1^* > Y_2^*) = \Pr(\phi(I_1^Y) > \phi(I_2^Y))$. The order-consistent equivalent form is used for binary comparisons and does not require linear linkage.

**DUAL Scenario Conditions:** Simultaneously satisfies both IM and IY conditions. Due to the existence of collider structure $I^M \leftarrow M \to Y^* \to I^Y$, simultaneously conditioning on $I^M$ and $I^Y$ introduces spurious associations. Implementation strategy requires avoiding simultaneous use of both image modalities in the same batch or same loss term when estimating conditional mutual information or training discriminators.

## A.8 REGULARITY AND IMPLEMENTATION CONDITIONS

### A.8.1 SAMPLE EXCHANGEABILITY AND MOMENT CONDITIONS

Observed samples $(Z_1, Z_2, \ldots, Z_n)$ satisfy either independent and identically distributed $Z_i \overset{iid}{\sim} P_Z$ or $\beta$-mixing conditions. For time series data, $\beta$-mixing requires $\beta(k) = O(k^{-\alpha})$, where $\alpha > 2$.

Moment conditions include bounded second moments $\mathbb{E}\left[\|Z_i\|_2^2\right] < \infty$ for all $i$, and optional light-tail assumptions where $Z_i$ satisfies sub-Gaussian conditions

$$\mathbb{E}[\exp(tZ_i)] \leq \exp(\sigma^2 t^2/2) \text{ for all } t \in \mathbb{R}$$

Non-degenerate encoder conditions require $\mathrm{Var}(Z_.) \geq \sigma_0^2 > 0$ to prevent representation collapse to constants.

### A.8.2 IMPLEMENTATION METHODS FOR LIPSCHITZ CONSTRAINTS

Encoder functions $E : \mathcal{X} \to \mathcal{Z}$ must satisfy Lipschitz continuity

$$\|E(x_1) - E(x_2)\|_2 \leq L\|x_1 - x_2\|_2$$

where $L > 0$ is the Lipschitz constant.

Practical implementation constrains each layer's weight matrix $W$ through spectral normalization $W_{\mathrm{SN}} = \frac{W}{\sigma(W)}$, where $\sigma(W)$ is the largest singular value of $W$. Gradient clipping limits gradient norms $\nabla_{\mathrm{clip}} = \min\left(1, \frac{C}{\|\nabla\|_2}\right) \nabla$, where $C$ is the clipping threshold. Optional gradient penalty adds $\mathcal{L}_{\mathrm{GP}} = \lambda \mathbb{E}_x\left[\max(0, \|\nabla_x E(x)\|_2 - 1)^2\right]$ to the loss function.

Verification methods numerically estimate the Lipschitz constant $L \approx \max_{x_1,x_2} \frac{\|E(x_1)-E(x_2)\|_2}{\|x_1-x_2\|_2}$, monitor spectral norm changes of encoders during training, and verify the boundedness of representation changes under perturbed inputs. A computable upper bound is $L \leq \prod_\ell \sigma_{\max}(W_\ell)$ (tracked via power iteration).

### A.9 IDENTIFIABILITY CONDITIONS

Causal effect identification requires backdoor, front-door (Section A.7.1), or instrumental variable criteria. Under Assumption A.5 and the structure-preserving losses, these criteria apply in the representation space using $(Z_T, Z_M, Z_X, Z_Y)$ with directional constraints maintaining $T$ as exogenous.

These assumptions provide the foundation for theoretical guarantees in Appendix B and Appendix C, and guide implementation choices in Appendix D.

# B THEORETICAL PROPERTIES AND TECHNICAL CONDITIONS

## B.1 THEORETICAL FOUNDATIONS OF LOSS FUNCTIONS

The alignment loss $\mathcal{L}_{\mathrm{align}}$ provides a lower bound on mutual information via noise contrastive estimation. Let $(z^+, y^+)$ be a positive sample pair and $\{y_k^-\}_{k=1}^K$ be independently sampled negative samples. The InfoNCE loss is defined as:

$$\mathcal{L}_{\mathrm{InfoNCE}} = -\mathbb{E}\left[\log \frac{\exp(f(z^+, y^+)/\tau_{\mathrm{align}})}{\exp(f(z^+, y^+)/\tau_{\mathrm{align}}) + \sum_k \exp(f(z^+, y_k^-)/\tau_{\mathrm{align}})}\right] \quad (19)$$

where $f : \mathcal{Z} \times \mathcal{Y} \to \mathbb{R}$ is the scoring function. By the Donsker-Varadhan variational representation, $I(Z; Y) \geq \log K - \mathcal{L}_{\mathrm{InfoNCE}}$. This lower bound improves with increasing $K$, and its validity relies on the negative sampling independence assumption $Y_k^- \perp (Z^+, Y^+)$ (see Assumption A.5).

Conditional mutual information is estimated via the density ratio method. Define $\Delta\mathrm{NLL} = \mathbb{E}[-\log q_\phi(y|z_m)] - \mathbb{E}[-\log q_\theta(y|z_t, z_m)]$, where $q_\theta$ and $q_\phi$ are two independently parameterized predictors to avoid bias in CMI estimation. Under the realizability assumption (Assumption A.5), i.e., there exists $\theta^*$ such that $q_{\theta^*}(y|z) = p(y|z)$ almost surely, we have $\lim_{n\to\infty} \Delta\mathrm{NLL} = I(Z_T; Y|Z_M)$. Under model misspecification, the bias is $\zeta^* = \inf_\theta \mathrm{KL}(p\|q_\theta)$, and consistency requires $\zeta^* \to 0$.

## B.2 NLL CALIBRATION AND REALIZABILITY

### B.2.1 THEORETICAL FOUNDATION OF NLL SURROGATES

**Theorem B.1 (Calibration of $\Delta\mathrm{NLL}$).** Assume (i) $E_T, E_M$ are Lipschitz and predictors $q_\theta, q_\phi$ are well-specified (realizable), and (ii) the regularity conditions in Appendix A.8 hold. The conditional

independence loss employs two independently parameterized predictors $q_\theta$ and $q_\phi$ to avoid bias in CMI estimation. Specifically:

- $q_\theta : \mathcal{Z}_T \times \mathcal{Z}_M \to \mathcal{P}(\mathcal{Y})$ models $p(y|z_t, z_m)$
- $q_\phi : \mathcal{Z}_M \to \mathcal{P}(\mathcal{Y})$ models $p(y|z_m)$

Then

$$\Delta\mathrm{NLL}_n \xrightarrow{P} I(Z_T; Y \mid Z_M).$$

Without realizability, letting

$$\zeta^\star := \inf_{\theta,\phi} \left\{ \mathbb{E}_{Z_T, Z_M}\big[\mathrm{KL}\big(p(\cdot \mid Z_T, Z_M) \,\|\, q_\theta(\cdot \mid Z_T, Z_M)\big)\big] + \mathbb{E}_{Z_M}\big[\mathrm{KL}\big(p(\cdot \mid Z_M) \,\|\, q_\phi(\cdot \mid Z_M)\big)\big] \right\},$$

we have

$$\Delta\mathrm{NLL}_n = I(Z_T; Y \mid Z_M) + \zeta^\star + \varepsilon_n, \quad \varepsilon_n \xrightarrow{P} 0.$$

**Proof:** Let the true conditional densities be $p(y|z_t, z_m)$ and $p(y|z_m)$, and the parameterized densities be $q_\theta(y|z_t, z_m)$ and $q_\phi(y|z_m)$. Define

$$\mathrm{NLL}_{tzm} = -\frac{1}{n} \sum_{i=1}^{n} \log q_\theta(y_i \mid z_{t,i}, z_{m,i}), \quad \mathrm{NLL}_{zm} = -\frac{1}{n} \sum_{i=1}^{n} \log q_\phi(y_i \mid z_{m,i}).$$

$$\Delta\mathrm{NLL}_n := \mathrm{NLL}_{zm} - \mathrm{NLL}_{tzm} = I(Y; Z_T \mid Z_M) + \zeta^\star + O_P(n^{-1/2}),$$

Under realizability, by the law of large numbers and continuous mapping theorem, convergence to true entropies follows. Under misspecification, the additional fixed bias $\zeta^*$ quantifies the degree of model mismatch.

**Lemma B.1 (Realizability Condition):** If there exist parameters $\theta^*, \phi^*$ such that $q_{\theta^*}(y|z_t, z_m) = p(y|z_t, z_m)$ and $q_{\phi^*}(y|z_m) = p(y|z_m)$ almost surely, then

$$\lim_{n\to\infty} \mathbb{E}[\Delta\mathrm{NLL}_n] = H(Y|Z_M) - H(Y|Z_T, Z_M) = I(Z_T; Y|Z_M).$$

**Lemma B.2 (Approximation Error Bound):** Under model misspecification, let $\theta_n, \phi_n$ be empirical risk minimizers. Then

$$|\Delta\mathrm{NLL}_n - I(Z_T; Y|Z_M) - \zeta^\star| \le \xi_n + \delta_n$$

where $\xi_n = O_P((\log n/n)^{1/2})$ is the finite sample error and $\delta_n = O_P(n^{-1/2})$ is the optimization error.

### B.2.2 PRACTICAL VERIFICATION STRATEGIES

**Corollary B.1 (Calibration Testing):** NLL calibration is verified through the following procedure. First, training convergence is monitored by checking $|\Delta\mathrm{NLL}_{n+k} - \Delta\mathrm{NLL}_n| < \epsilon$ for sufficiently large $k$. Second, cross-validation consistency requires computing $\Delta\mathrm{NLL}_{\mathrm{val}}$ on independent validation sets and ensuring $|\Delta\mathrm{NLL}_{\mathrm{train}} - \Delta\mathrm{NLL}_{\mathrm{val}}| < \delta$. Third, conditional independence testing directly examines the null hypothesis $Z_T \perp\!\!\!\perp Y \mid Z_M$, where rejection at $p < 0.05$ provides statistical evidence for $I(Z_T; Y|Z_M) > 0$.

### B.3 INFONCE LOWER BOUND CONDITIONS

### B.3.1 SUFFICIENT CONDITIONS FOR LOWER BOUND VALIDITY

**Theorem B.2 (InfoNCE Lower Bound Theorem):** Let $(Z^+, Y^+)$ be a positive sample pair and $\{Y_k^-\}_{k=1}^{K}$ be negative samples drawn independently from marginal distribution $P_Y$. Define the InfoNCE loss as

$$\mathcal{L}_{\mathrm{InfoNCE}} = -\mathbb{E}\left[\log \frac{\exp(f(Z^+, Y^+)/\tau)}{\exp(f(Z^+, Y^+)/\tau) + \sum_{k=1}^{K} \exp(f(Z^+, Y_k^-)/\tau)}\right]$$

where $f : \mathcal{Z} \times \mathcal{Y} \to \mathbb{R}$ is a scoring function and $\tau > 0$ is the temperature parameter. Under the following conditions: (i) **Negative independence**: $Y_k^{-} \overset{i.i.d.}{\sim} P_Y$ and $Y_k^{-} \perp\!\!\!\perp (Z^+, Y^+)$; (ii) **Lipschitz scoring**: $|f(z_1, y) - f(z_2, y)| \leq L\|z_1 - z_2\|$; (iii) **Sufficient negatives**: Theoretically $K = \Omega(n)$ minimizes bias to $O_P(n^{-1/2})$. Practically, $K \asymp \log n$ (validation selects constant $C$ such that $K \geq C \log n$) achieves error $\varepsilon_K = O_P(K^{-1/2}) = O_P((\log n)^{-1/2})$, which is sufficient for most applications while maintaining computational tractability. We have the lower bound relation (van den Oord et al., 2019; Poole et al., 2019):

$$I(Z;Y) \geq \log K - \mathcal{L}_{\text{InfoNCE}} - \varepsilon_K$$

where $\varepsilon_K = O_P(K^{-1/2})$.

**Proof:** According to the Donsker-Varadhan variational representation (Donsker & Varadhan, 1975),

$$I(Z;Y) = \sup_{f} \mathbb{E}_{P_{ZY}}[f(Z, Y)] - \log \mathbb{E}_{P_Z \otimes P_Y}[\exp(f(Z, Y))].$$

**Lemma B.3 (Negative Sample Approximation):** Under negative sample independence,

$$\mathbb{E}_{P_Z \otimes P_Y}[\exp(f(Z, Y))] = \frac{1}{K} \sum_{k=1}^{K} \exp(f(Z^+, Y_k^{-})) + O_P(K^{-1/2}).$$

**Lemma B.4 (Temperature Scaling):** For any $\tau > 0$, replacing $f$ by $f_\tau := f/\tau$ yields the same functional family up to a scale; the InfoNCE lower bound retains the form $I(Z;Y) \geq \log K - \mathcal{L}_{\text{InfoNCE}}(f_\tau)$. Hence $\tau$ acts as a numerical conditioning parameter rather than tightening the bound per se; it should be selected via validation.

### B.3.2 TECHNICAL REQUIREMENTS FOR PRACTICAL IMPLEMENTATION

**Corollary B.2 (Negative Sampling Strategy):** To ensure the conditions of Theorem B.2 are satisfied, practical implementation requires the following. Global batch sampling ensures that in distributed training, negative samples are drawn from the global batch rather than local batches. Temperature adjustment selects $\tau \in [0.01, 0.1]$ through validation sets for optimal performance. Negative sample quantity follows $K \asymp \log n$ to ensure statistical power.

**Theorem B.3 (Distributed Consistency):** In distributed settings with $D$ devices and batch size $B$ per device, if negative samples are drawn from the global batch $DB$, InfoNCE lower bound validity is maintained (van den Oord et al., 2019):

$$I(Z;Y) \geq \log(DB - 1) - \mathcal{L}_{\text{InfoNCE}}^{\text{global}} - O_P((DB)^{-1/2}).$$

**Proof:** Extension of Theorem B.2 analysis to multi-device settings, with the key requirement being maintenance of global negative sample independence.

### B.4 CONDITIONAL MUTUAL INFORMATION BIAS CONTROL

### B.4.1 SOURCES OF CMI ESTIMATION BIAS

**Theorem B.4 (CMI Bias Decomposition Theorem):** Let $\hat{I}(Z_T; Y | Z_M)$ be some estimator of CMI. Under the density estimation framework, total bias decomposes as:

$$\mathbb{E}[\hat{I}(Z_T; Y | Z_M)] - I(Z_T; Y | Z_M) = B_{\text{density}} + B_{\text{finite}} + B_{\text{discretization}}$$

where $B_{\text{density}}$ represents conditional density estimation bias, $B_{\text{finite}}$ represents finite sample bias, and $B_{\text{discretization}}$ represents discretization bias when applicable.

**Lemma B.5 (Conditional Density Bias Bound):** Using kernel density estimation with bandwidth $h = O(n^{-1/(d+4)})$ (Silverman's rule or plug-in bandwidth in practice),

$$|B_{\text{density}}| = O(n^{-4/(d+4)})$$

where $d$ is the dimension of the conditioning variable $Z_M$ (Wasserman, 2006).

### B.4.2 SAMPLE SPLITTING AND CROSS-FITTING

**Theorem B.5 (Cross-fitting Bias Control):** When estimating CMI using $K$-fold cross-fitting, let $\mathcal{D} = \cup_{k=1}^{K} \mathcal{D}_k$ be the data partition and $\hat{p}_{-k}$ be density estimators trained on $\mathcal{D} \setminus \mathcal{D}_k$. Define

$$\hat{I}_{\text{CV}} = \frac{1}{K} \sum_{k=1}^{K} \frac{1}{|\mathcal{D}_k|} \sum_{i \in \mathcal{D}_k} \log \frac{\hat{p}_{-k}(y_i | z_{t,i}, z_{m,i})}{\hat{p}_{-k}(y_i | z_{m,i})}.$$

Under regularity conditions (Chernozhukov et al., 2018):

$$|\mathbb{E}[\hat{I}_{\text{CV}}] - I(Z_T; Y | Z_M)| = O(n^{-1/2}).$$

**Proof:** The key advantage of cross-fitting lies in avoiding overfitting bias. For each fold $k$,

$$\mathbb{E}_{i \in \mathcal{D}_k} \left[ \log \frac{\hat{p}_{-k}(Y_i | Z_{T,i}, Z_{M,i})}{\hat{p}_{-k}(Y_i | Z_{M,i})} \right] = I(Z_T; Y | Z_M) + o(1)$$

since $\hat{p}_{-k}$ is independent of $(Y_i, Z_{T,i}, Z_{M,i})$. Summing and averaging yields the desired result.

**Corollary B.3 (Practical Bias Control Strategy):** Effective bias control employs 5-fold or 10-fold cross-fitting. Conditional density estimation utilizes regularized neural networks or Gaussian processes. Monitoring variance across different folds guides regularization strength adjustment when excessive variance is observed.

### B.5 SOFT-RANK TO SPEARMAN CORRELATION CONSISTENCY

### B.5.1 THEORETICAL PROPERTIES OF SOFT-RANK FUNCTIONS

**Definition B.1 (Soft-rank Function):** Given vector $x = (x_1, \ldots, x_n)$ and temperature parameter $\tau > 0$, the soft-rank function is defined as:

$$\text{SoftRank}_\tau(x_i) = \sum_{j=1}^{n} \sigma \left( \frac{x_i - x_j}{\tau} \right)$$

where $\sigma(z) = (1 + \exp(-z))^{-1}$ is the sigmoid function.

**Theorem B.6 (Soft-rank Consistency Theorem):** Let $(X, Y)$ be a bivariate random vector and $\text{Rank}(x_i)$ be the true rank of $x_i$ in the sample. As $\tau \to 0$,

$$\lim_{\tau \to 0} \text{SoftRank}_\tau(x_i) = \text{Rank}(x_i) - 1$$

with uniform convergence over compact domains (Blondel et al., 2020).

**Proof:** As $\tau \to 0$, $\sigma((x_i - x_j)/\tau) \to \mathbf{1}_{x_i > x_j}$, hence

$$\lim_{\tau \to 0} \text{SoftRank}_\tau(x_i) = \sum_{j=1}^{n} \mathbf{1}_{x_i > x_j} = \text{Rank}(x_i) - 1.$$

Uniformity follows from the monotonicity of the sigmoid function.

**Theorem B.7 (Spearman Correlation Convergence Theorem):** Let $\rho_S(X, Y)$ be the true Spearman correlation coefficient and $\hat{\rho}_\tau$ be the soft-rank based estimator:

$$\hat{\rho}_\tau = \text{Corr}(\text{SoftRank}_\tau(X), \text{SoftRank}_\tau(Y)).$$

Under conditions: (i) $(X_i, Y_i)$ independent and identically distributed for $i = 1, \ldots, n$; (ii) joint distribution has continuous marginals; (iii) $\tau = O(n^{-\alpha})$ for some $\alpha \in (0, 1/2)$, we have:

$$|\hat{\rho}_\tau - \rho_S(X, Y)| = O_P(n^{-1/2} + \tau \log n).$$

**Proof:** Decompose error into bias and variance components:

$$\hat{\rho}_\tau - \rho_S(X, Y) = [\mathbb{E}[\hat{\rho}_\tau] - \rho_S(X, Y)] + [\hat{\rho}_\tau - \mathbb{E}[\hat{\rho}_\tau]].$$

**Lemma B.6 (Bias Term Analysis):** $|\mathbb{E}[\hat{\rho}_\tau] - \rho_S(X, Y)| = O(\tau \log n)$.

**Lemma B.7 (Variance Term Analysis):** $\text{Var}[\hat{\rho}_\tau] = O(n^{-1})$.

### B.5.2 Gradient Boundedness and Optimization Stability

**Theorem B.8 (Soft-rank Gradient Bound):** The MAC loss $\mathcal{L}_{\text{MAC}} = -\hat{\rho}_\tau$ has gradient with respect to representation $z$ satisfying:

$$\left\| \frac{\partial \mathcal{L}_{\text{MAC}}}{\partial z} \right\| \leq \frac{C}{\tau} \cdot \frac{1}{n}$$

where $C$ is a constant depending only on the data range (Blondel et al., 2020).

**Proof:**

$$\frac{\partial \text{SoftRank}_\tau(x_i)}{\partial x_i} = \frac{1}{\tau} \sum_{j \neq i} \sigma'((x_i - x_j)/\tau) \leq \frac{1}{\tau} \cdot \frac{n-1}{4}$$

using $\sigma'(z) \leq 1/4$. The result follows by chain rule and correlation coefficient gradient formulas.

**Corollary B.4 (Training Stability):** Choosing $\tau = O(n^{-1/3})$ balances bias and gradient stability: Bias: $O(n^{-1/3} \log n)$; Gradient norm: $O(n^{-2/3})$; Overall rate: $O_P(n^{-1/3})$.

### B.6 Structure Preservation in Representation Space

The conditional data processing inequality guarantees information monotonicity. For deterministic mappings $E$, we have $I(E(X); E(Y)|E(Z)) \leq I(X; Y|Z)$ (Cover & Thomas, 2006). Under standard regularity conditions in Assumption A.5 (Lipschitz continuity and bounded second moments), standard estimators achieve $O_P(n^{-1/2})$ convergence rates.

The anti-bypass condition (Assumption A.5) ensures correct causal pathways. Defining the Markov chain $T \rightarrow (M, Y) \rightarrow Z_Y$, the conditional data processing inequality yields $I(T; Z_Y|M) \leq I(T; Y|M)$. Combined with the original conditional independence $T \perp Y^*|M$ and the IY bridging assumption $\mathbb{E}[Y^*|\phi(I^Y)] = h(\phi(I^Y))$ (Assumption A.5), the mutual information $I(T; Z_Y|M)$ is controlled by the calibration error $\varepsilon_h = \|\mathbb{E}[Y^*|\phi(I^Y)] - h(\phi(I^Y))\|^2$. Under bounded support or sub-Gaussian conditions, standard $f$-divergence inequalities (e.g., Pinsker's inequality) convert the mean squared error $\varepsilon_h$ to a mutual information upper bound (Fedotov et al., 2003).

### B.7 PC Algorithm Consistency Guarantees

Partial correlation testing achieves asymptotic normality via Fisher transformation (Fisher, 1921; 1924). Given conditioning set $\mathcal{S}$, the Fisher-transformed sample partial correlation $T = \sqrt{n - |\mathcal{S}| - 3} \text{arctanh}(\hat{\rho}_{XY \cdot \mathcal{S}})$ is asymptotically $\mathcal{N}(0, 1)$ under $H_0 : \rho_{XY \cdot \mathcal{S}} = 0$.

Multiple testing correction employs the Benjamini-Hochberg procedure. For $m$ hypotheses with ordered p-values $p_{(1)} \leq \ldots \leq p_{(m)}$, define $k^* = \max\{i : p_{(i)} \leq i\alpha/m\}$ and reject $\{H_{(1)}, \ldots, H_{(k^*)}\}$. Under independence or positive regression dependence on subsets (PRDS), this procedure controls the false discovery rate at FDR $\leq \alpha$.

Under the faithfulness assumption (Assumption A.5), consistent conditional independence testing, and the existence of a minimum effect size (minimum non-zero partial correlation $\gamma > 0$) with bounded degree conditions, the PC algorithm recovers the true Markov equivalence class with probability approaching 1. The error probability decays exponentially or polynomially in $n$, with the specific rate depending on $\gamma$, maximum degree, and test power.

## C Main Theoretical Proofs

### C.1 Proof Strategy and Roadmap

To improve readability and make dependencies explicit, we organize the proofs so that the reader can proceed linearly: from the statistical consistency of each surrogate loss (Appendix B), to structural preservation of the joint objective (Theorem 1), and finally to query-level identifiability guarantees (Theorem 2).

1. **Step 1: Statistical Foundations (Appendix B).** We establish risk-consistency of our variational/contrastive surrogates under **Assumption A.5** (Realizability and Regularity with cross-fitting):

   - **Lemmas B.1–B.2:** the calibrated surrogate for conditional mutual information makes $\mathcal{L}_{\mathrm{CI}}$ estimate $\mathrm{MI}(Z_T; Y \mid Z_M)$ with error $O_P(n^{-1/2})$, and the InfoNCE term in $\mathcal{L}_{\mathrm{MBR}}$ lower-bounds $\mathrm{MI}(Z_M; Y)$ with finite-$K$ deviation $O_P(K^{-1/2})$.
   - **Lemmas B.6–B.8:** the soft-rank operator used in $\mathcal{L}_{\mathrm{MAC}}$ approximates Spearman's $\rho_S$ with error $O_P(n^{-1/3})$ while keeping gradients bounded.

2. **Step 2: Structural Preservation (Appendix C.2– C.3).** Using Step 1, we prove **Theorem 1** (CSP Achievability): minimizing the joint loss $\mathcal{L}_{\mathrm{total}}$ enforces the $\varepsilon$-CSP conditions (see equation 14). Here **Assumption A.8.2** (Lipschitz/spectral control) provides compactness and stability, yielding uniform convergence of ERM and controlling the generalization gap at rate $O_P(n^{-1/2})$.

3. **Step 3: Identifiability Preservation (Appendix C.4). Theorem 2** translates $\varepsilon$-CSP into a query-level bound. By the Lipschitz stability in **A.8.2**, the representation error $\varepsilon$ propagates *at most linearly* to the causal query: $|\tilde{Q}-Q| \leq \kappa\,\varepsilon+\delta_{\mathrm{cal}}$, where the calibration term satisfies $\delta_{\mathrm{cal}} = O_P(n^{-1/2})$ under **A.5** (see equation 16).

## C.2 Conditional Independence Preservation Theorem

**Theorem C.1 (Conditional Independence Preservation):** Consider CARL framework's three scenarios: IM/DUAL scenarios with $Y = Y^*$, and IY scenario with $Y = \phi(I^Y)$ as proxy variable. Under the assumptions in Appendix A.5, suppose the original variables satisfy $T \perp\!\!\!\perp Y^* \mid M$. Representations trained by CARL and converged to approximate empirical risk minimizers satisfy:

$$I(Z_T; Z_Y \mid Z_M) \leq \zeta^{\star}(E_T, E_M) + o_P(1)$$

with high probability, where $\zeta^{\star}(E_T, E_M)$ is the model misspecification bias for given learned encoder classes (see Appendix B.2).

**Premise Conditions:** (1) Representation $Z_Y = E_Y(Y)$ depends solely on $Y$ without additional dependence on $(Z_T, Z_M)$ or information bypassing $Y$ (anti-bypass, see Appendix A.5); (2) Markov boundary retention loss $L_{MBR}$ is enabled with weight $w_{mbr} > 0$; (3) Encoders are deterministic functions during evaluation or contain randomness exogenous to $(T, M, Y)$; (4) **IY Bridging**: For IY scenarios, there exists measurement model $Y = h(Y^*, \nu)$ where $\nu \perp\!\!\!\perp (T, M, X)$, such that $T \perp\!\!\!\perp Y^* \mid M, \nu \perp\!\!\!\perp (T, M, X) \Rightarrow T \perp\!\!\!\perp Y \mid M$.

**Proof:** Under IY scenarios, the bridging assumption ensures that original conditional independence $T \perp\!\!\!\perp Y^* \mid M$ implies conditional independence on proxy variables $T \perp\!\!\!\perp Y \mid M$.

Let $(\hat{E}_T, \hat{E}_M, \hat{E}_Y)$ be approximate empirical risk minimizers of the conditional independence loss $L_{CI} = \mathbb{E}[\mathrm{NLL}(Z_T, Z_M \to Y)] - \mathbb{E}[\mathrm{NLL}(\mathrm{detach}(Z_M) \to Y)]$. According to Theorem B.1 in Appendix B.2, upon training convergence we have $L_{CI} = I(Z_T; Y \mid Z_M) + \zeta^{\star}(E_T, E_M) + o_P(1)$, where $\zeta^{\star}(E_T, E_M)$ depends on the learned encoder classes.

The key observation is that since $Z_Y = E_Y(Y)$ depends solely on $Y$ (premise condition 1), there exists a Markov chain $Z_T \to (Y, Z_M) \to Z_Y$. Applying the conditional data processing inequality yields $I(Z_T; Z_Y \mid Z_M) \leq I(Z_T; Y \mid Z_M)$. Notably, even when evaluation involves exogenous noise (random encoding Markov kernels), conditional DPI remains valid since the Markov chain property is preserved.

The Markov boundary retention loss $L_{MBR}$ ensures $Z_M$ does not collapse to zero information by maximizing an InfoNCE lower bound of $I(Z_M; Y)$. Specifically, there exists a constant $c > 0$ determined by temperature parameters and batch size such that $I(Z_M; Y) \geq c - \varepsilon_K$, where $\varepsilon_K = O_P(K^{-1/2})$ (see Appendix B.3). This excludes trivial solutions that achieve conditional independence through $Z_M$ degradation.

When original variables satisfy $T \perp\!\!\!\perp Y \mid M$, theoretically the minimum value of $I(Z_T; Y \mid Z_M)$ is $\zeta^{\star}(E_T, E_M)$ (limited only by model misspecification for given encoder classes). Minimization of $L_{CI}$ drives the system toward this theoretical lower bound. Combining conditional data processing

inequality and NLL calibration results, we obtain:
$$I(Z_T; Z_Y \mid Z_M) \leq I(Z_T; Y \mid Z_M) \leq \zeta^\star(E_T, E_M) + o_P(1).$$

Here $\zeta^\star(E_T, E_M)$ depends on the learned encoder classes; under Lipschitz constraints and sufficient capacity, $\zeta^\star$ can be controlled to be small (see Appendix B.2).

**Remark:** In all scenarios, we employ $Z_Y = E_Y(Y)$ ensuring it depends solely on $Y$ (see Appendix A.5), and enable $L_{MBR}$ to prevent $Z_M$ degradation. The IY scenario bridging assumption ensures conditional independence transfer from $Y^*$ to proxy $Y$, making the conclusion applicable to the original target variable $Y^*$ as well.

## C.3    Markov Boundary Retention Theorem

**Theorem C.2 (Markov Boundary Retention):** Under the assumptions in Appendix A.5 and the premises of Theorem C.1, with $Z_M = g(M)$ as a deterministic encoder function and $Z_Y$ depending solely on $Y$ (anti-bypass condition), representations trained by CARL preserve Markov boundary properties:
$$I(Z_M; Z_Y) \geq I(M; Y^*) - \delta_{MB}$$
$$I(Z_T; Z_Y \mid Z_M) \leq \min\left\{I(T; Y^* \mid M) + H(M \mid Z_M), \zeta^\star(E_T, E_M) + o_P(1)\right\}$$
where $\delta_{MB} = H(M \mid Z_M) + H(Y^* \mid Z_Y)$ represents encoder information loss (interpreted as conditional cross-entropy w.r.t. a common base measure or using discretization approximation; both provide upper bounds for continuous variables), and $\zeta^\star(E_T, E_M)$ is the model misspecification bias from Theorem C.1.

**Proof:** For the first inequality, since $Z_M = g(M)$ is deterministic, $I(Z_M; Y^* \mid M) = 0$, hence the chain rule gives $I(M; Y^*) = I(Z_M; Y^*) + I(M; Y^* \mid Z_M)$. Since $I(M; Y^* \mid Z_M) \leq H(M \mid Z_M)$, we obtain:
$$I(Z_M; Y^*) \geq I(M; Y^*) - H(M \mid Z_M).$$

For the information-theoretic inequality $I(Z_M; Z_Y) \geq I(Z_M; Y^*) - H(Y^* \mid Z_Y)$, we use the standard derivation:
$$I(Z_M; Z_Y) - I(Z_M; Y^*) = H(Z_M \mid Y^*) - H(Z_M \mid Z_Y) \geq -H(Y^* \mid Z_Y)$$
where the inequality follows from:
$$H(Z_M \mid Z_Y) \leq H(Z_M, Y^* \mid Z_Y) = H(Z_M \mid Y^*, Z_Y) + H(Y^* \mid Z_Y) \leq H(Z_M \mid Y^*) + H(Y^* \mid Z_Y).$$

Combining yields:
$$I(Z_M; Z_Y) \geq I(M; Y^*) - H(M \mid Z_M) - H(Y^* \mid Z_Y).$$

Defining $\delta_{MB} = H(M \mid Z_M) + H(Y^* \mid Z_Y)$ yields the first inequality.

For the second inequality, we derive two independent upper bounds. First, by data processing inequality with $T \rightarrow Z_T$ and $Y^* \rightarrow Y \rightarrow Z_Y$:
$$I(Z_T; Z_Y \mid Z_M) \leq I(Z_T; Y \mid Z_M) \leq I(T; Y \mid Z_M).$$

Since conditioning coarsening gives $I(T; Y \mid Z_M) \leq I(T; Y \mid M) + H(M \mid Z_M)$, and using IY bridging $I(T; Y \mid M) \leq I(T; Y^* \mid M)$:
$$I(Z_T; Z_Y \mid Z_M) \leq I(T; Y^* \mid M) + H(M \mid Z_M).$$

Second, Theorem C.1 provides an independent training error bound:
$$I(Z_T; Z_Y \mid Z_M) \leq \zeta^\star(E_T, E_M) + o_P(1).$$

Taking the minimum of these two independent bounds yields the second inequality.

**Estimable Certificate Version:** In IY scenarios, using observable proxy $Y$:
$$I(Z_M; Z_Y) \geq I(M; Y) - H(M \mid Z_M) - H(Y \mid Z_Y)$$
where $H(M \mid Z_M)$ and $H(Y \mid Z_Y)$ are estimated through validation set cross-entropy. Since $Y^* \rightarrow Y$ is post-processing, $I(M; Y) \leq I(M; Y^*)$. The CARL framework optimizes a lower bound of $I(Z_M; Y)$ through the InfoNCE term in $L_{MBR}$, with practical estimation requiring consideration of statistical error $\varepsilon_K = O_P(K^{-1/2})$. The estimation version incorporates cross-fitting bias control from Appendix B.4.

## C.4 Monotonic Alignment Consistency Theorem

**Theorem C.3 (Monotonic Alignment Consistency Preservation):** Under Assumptions A.2 (separable measurement & measurable encoders), A.4.2 (Lipschitz constraints), and B.4 (soft-rank consistency), let $\Delta a_{ij} = |a_i - a_j|$ and $\Delta z_{ij} = \|z_i - z_j\|_2$. If the additional **order-preserving margin assumption** holds: there exists $\gamma > 0$ such that

$$\Pr[(\Delta a_{ij} - \Delta a_{k\ell})(\Delta z_{ij} - \Delta z_{k\ell}) > 0] \geq \frac{1}{2} + \gamma$$

Under copula density upper and lower bound regularity conditions, there exists $\kappa \in (0, 1]$ such that $\rho_S(\Delta a, \Delta z) \geq \kappa \tau_K(\Delta a, \Delta z)$. Since the order-preserving margin assumption gives $\tau_K \geq 2\gamma$, we have $c(\gamma) = \kappa \cdot 2\gamma$.

Then there exists a constant $c = c(\gamma) \in (0, 1]$ such that upon convergence of $\mathcal{L}_{\text{MAC}}$ training,

$$\rho_S(\Delta a, \Delta z) \geq c - \delta_{\text{MAC}} - \varepsilon_{\text{gen}}$$

where

$$\delta_{\text{MAC}} = O_P(\tau \log N + n_{\text{eff}}^{-1/2})$$

$n_{\text{eff}}$ is the number of subsampled pairwise distances, $\varepsilon_{\text{gen}} = O_P(n^{-1/2})$ with constants depending on the Rademacher complexity of the representation class and IB/Lipschitz regularization; details in Appendix B.5. In practice, choosing $\tau \asymp n_{\text{eff}}^{-1/3}$ achieves $\delta_{\text{MAC}} = O_P(n_{\text{eff}}^{-1/3})$ while avoiding gradient vanishing.

**Proof:** By Theorem B.6, we have $\sup_x |\text{SoftRank}_\tau(x) - \text{Rank}(x)| \leq C\tau \log N$, where $N$ is the number of elements participating in ranking, equal to $n_{\text{eff}}$.

By Theorem B.7, we have $|\hat{\rho}_\tau - \rho_S(\Delta a, \Delta z)| = O_P(n_{\text{eff}}^{-1/2} + \tau \log N)$. Here the pairwise distances $\{\Delta a_{ij}, \Delta z_{ij}\}$ are controlled to have size $n_{\text{eff}}$ through subsampling strategies, avoiding excessive computational complexity while maintaining statistical effectiveness.

The order-preserving margin assumption combined with Lipschitz stability provides a positive lower bound for the Spearman correlation through Kendall's $\tau$. This assumption can be verified through pairwise comparison tests on validation sets, equivalent to permutation tests for Kendall's $\tau$.

ERM convergence maximizes the training objective to a neighborhood of the population target, producing generalization error $\varepsilon_{\text{gen}} = O_P(n^{-1/2})$.

Regarding the choice of $\tau$, theoretically without gradient lower bound constraints, the optimal choice would be $\tau^* \asymp (\log N)^{-1} n_{\text{eff}}^{-1/2}$, giving $\delta_{\text{MAC}} = O_P(n_{\text{eff}}^{-1/2})$. However, in practice to avoid soft-rank gradient vanishing, we take $\tau \asymp n_{\text{eff}}^{-1/3}$ to balance approximation bias and optimization stability, achieving the practical convergence rate $\delta_{\text{MAC}} = O_P(n_{\text{eff}}^{-1/3})$.

## C.5 Identifiability Consistency Theorem

**Theorem C.4 (Cross-modal Identifiability Preservation):** Under Assumptions A.4 (causal structure and faithfulness), A.7.1 (identification conditions), A.7 (scenario-specific requirements), and the outcome calibration assumption C.6, suppose the original causal query $\mathbb{E}[Y^*(t)] = Q(t; P(T, M, Y^*, X))$ is identifiable via backdoor adjustment, front-door criterion, or instrumental variables. Let $(Z_T, Z_M, Z_Y, Z_X)$ be the learned representations from CARL training satisfying Theorems C.1-C.3.

Define the corresponding query in representation space $\mathbb{E}[\tilde{Y}(t)] = Q(t; P(Z_T, Z_M, \tilde{Y}, Z_X))$, where $\tilde{Y} := \hat{h}(Z_Y)$. Then

$$\left|\mathbb{E}[\tilde{Y}(t)] - \mathbb{E}[Y^*(t)]\right| \leq \delta_{MB} + \mathbf{1}_{IY} \cdot \delta_{MAC} + \delta_{cal} + \varepsilon_{ident}$$

where $\delta_{MB}$ is from Theorem C.2, $\delta_{MAC}$ is from Theorem C.3 (activated in IY scenarios, otherwise 0), $\delta_{cal} := |\mathbb{E}[Y^*] - \mathbb{E}[\hat{h}(Z_Y)]|$ is the outcome calibration error, and $\varepsilon_{ident}$ represents identification-specific statistical error ($\varepsilon_{ident} = O_P(n^{-1/2})$ under parametric/calibratable settings; $O_P(n^{-\beta})$ under nonparametric settings depending on smoothness and dimensionality).

**Proof:** We establish identifiability preservation for each identification strategy.

**Backdoor Adjustment Case:** By Theorem C.1's conditional independence preservation and A.4.2's Lipschitz conditions, the $d$-separation of $X$ is maintained in $(Z_T, Z_X, Z_Y)$ with $o_P(1)$ error, yielding

$$\mathbb{E}[\tilde{Y}(t)] = \mathbb{E}_{Z_X}\left[\mathbb{E}[\tilde{Y}|Z_T = z_t, Z_X]\right] + o_P(1)$$

which is equivalent to the original backdoor formula, with discrepancies dominated by $\delta_{cal}$ and $\varepsilon_{ident}$.

**Front-door Criterion Case:** Theorem C.2 ensures Markov boundary information retention of $Z_M$ for $Y^*$, combined with C.1's CI preservation, yielding the representation space front-door formula

$$\mathbb{E}[\tilde{Y}(t)] = \mathbb{E}_{Z_M}\left[\mathbb{E}[\tilde{Y}|Z_M] \cdot \mathbb{E}_{Z_X}[P(Z_M|Z_T = z_t, Z_X)]\right] + \delta_{MB}.$$

Due to the deterministic mapping $Z_M = E_M(M)$ and Lipschitz property of $\hat{h}$, the mediation pathway identification formula remains valid in the calibrated representation space, with errors containing $\delta_{MB}$ and $\delta_{cal}$.

**Instrumental Variable Case:** In representation space, relevance $\mathrm{Cov}(Z_Z, Z_T) \neq 0$ and exclusion/exogeneity $Z_Z \perp Z_Y|(Z_T, Z_X)$ approximately hold (by C.1 and A.4.2), yielding IV estimation based on $\hat{h}(Z_Y)$ equivalent to the original formula, with error $\delta_{cal} + \varepsilon_{ident}$.

**Scenario-unified Treatment:** IM scenarios directly apply the above results with $\delta_{MAC} = 0$ since no image-to-outcome monotonic alignment is involved. IY scenarios introduce the $\delta_{MAC}$ term through IY bridging and Theorem C.3's monotonic consistency, ensuring order consistency from $\phi(I^Y)$ to $Y^*$. DUAL scenarios avoid simultaneous conditioning on $I^M$ and $I^Y$ A.7, applying the corresponding identification strategies by pathway.

**Error Decomposition:** The total error comprises four components. $\delta_{MB}$ reflects Markov boundary information loss during encoding, primarily from $H(M|Z_M) + H(Y^*|Z_Y)$. $\delta_{MAC}$ is activated only in IY scenarios, arising from soft-rank approximation and semantic-representation distance monotonic alignment errors. $\delta_{cal}$ captures calibration error from representation space to outcome space, empirically assessable through $|\hat{\mathbb{E}}[Y^*] - \hat{\mathbb{E}}[\hat{h}(Z_Y)]|$ or appropriate calibration losses on validation sets. $\varepsilon_{ident}$ encompasses statistical fluctuations of identification formulas under finite samples and generalization errors from representation learning.

Under regularity conditions A.8 and technical conditions B.2-B.5, all error terms converge at controllable rates, ensuring asymptotic preservation of cross-modal identifiability.

## C.6 OUTCOME CALIBRATION ASSUMPTION

Let $h^*(z) = \mathbb{E}[Y^*|Z_Y = z]$ be the true conditional expectation function. There exists a learned estimator $\hat{h} : Z_Y \to \mathbb{R}$ that is Lipschitz continuous and approximates $h^*$. The calibration error is defined as $\delta_{cal} := |\mathbb{E}[Y^*] - \mathbb{E}[\hat{h}(Z_Y)]|$, which can be empirically assessed through validation set performance of the calibration head. In practice, $\hat{h}$ is implemented through regression heads or calibration layers, inheriting Lipschitz properties from the overall network constraints A.8.2.

## C.7 METRIC CONSISTENCY PROOF

This section establishes formal connections between the four evaluation metrics and the main theoretical theorems, proving that these metrics can effectively detect the theoretical guarantees.

**Theorem (Metric Consistency):** Suppose the representations from converged CARL training satisfy the conditions of Theorems C.1-C.4. Then the four evaluation metrics have the following consistency relationships with the corresponding theoretical results, where $\mathbf{1}_{IY}$ indicates the IY scenario:

**CIP and Conditional Independence Preservation (Theorem C.1):** Under Gaussian conditions, by the data processing inequality $I(\tilde{Y}; Z_T|Z_M) \leq I(Z_Y; Z_T|Z_M) \leq \zeta^*$, we have

$$I(\tilde{Y}; Z_T|Z_M) = -\frac{1}{2}\log(1-\rho_{pc}^2) \Rightarrow |\rho_{pc}| \leq g(\zeta^*) := \sqrt{1-e^{-2\zeta^*}}$$

Let $\beta_n(\rho, \alpha)$ be the power function of the partial correlation test (significance level $\alpha$, sample size $n$). Then under the local alternative $|\rho_{pc}| \leq g(\zeta^*)$,

$$\Pr(\text{CIP}=1) = 1 - \Pr(\text{reject } H_0) \geq 1 - \beta_n(g(\zeta^*), \alpha)$$

**CSI and Identifiability Preservation (Theorem C.4):** Let $S_M := I(M; Y^*)$, $\tilde{S}_M := S_M - \delta_{MB}$ (Markov boundary signal retention). Under linear-Gaussian approximation and calibration models, there exist constants $\kappa_1, \kappa_2 > 0$ such that

$$R_{\text{correct}}^2 \geq \kappa_2 \tilde{S}_M, \quad R_{\text{direct}}^2 \leq \kappa_1 g(\zeta^*)^2$$

Using the definition $\text{CSI} = (R_{\text{correct}}^2 - R_{\text{direct}}^2)/(R_{\text{correct}}^2 + \varepsilon)$, we obtain

$$\text{CSI} \geq 1 - \frac{\kappa_1 g(\zeta^*)^2}{\kappa_2 \tilde{S}_M + \varepsilon} - \xi_n$$

**MBRI and Markov Boundary Retention (Theorem C.2):** Define

$$\widehat{\text{MBRI}} := \frac{\widehat{I}(M; \tilde{Y})}{\widehat{I}(M; \tilde{Y}) + \widehat{I}(T; \tilde{Y}) + \varepsilon}$$

Let $\xi_{\text{MI}}(n, B)$ denote the histogram MI estimation bias with $B$ bins. Then by C.2 and C.1,

$$\mathbb{E}[\widehat{\text{MBRI}}] \geq \frac{I(M; Y^*) - \delta_{MB} - \xi_{\text{MI}}}{I(M; Y^*) - \delta_{MB} - \xi_{\text{MI}} + \zeta^* + \xi_{\text{MI}} + \varepsilon} - O_P(n^{-1/2})$$

**MAC and Monotonic Alignment Consistency (Theorem C.3):**

$$\mathbb{E}[\widehat{\text{MAC}}] \geq c(\gamma) - \delta_{\text{MAC}} - \varepsilon_{\text{gen}} - \xi_{\text{rank}}(n_{\text{eff}})$$

where $\xi_{\text{rank}}(n_{\text{eff}}) = O_P(n_{\text{eff}}^{-1/2})$ comes from the U-statistic convergence of sample Spearman correlation, and $n_{\text{eff}}$ is the non-overlapping pair subsampling size.

## C.8  METRIC CONSISTENCY

**Theorem C.6 (Metric-Theory Correspondence)** Suppose the encoder family $\mathcal{E}$ satisfies Assumptions 1-4 and $(Z_T, Z_M, Z_Y)$ are the converged representations. Then the evaluation metrics have the following correspondence with theoretical guarantees.

**Proposition C.6.1 (CIP Metric)** If $I(Z_T; Z_Y \mid Z_M) \leq \zeta^*$, then the partial correlation coefficient satisfies $|\rho_{pc}(Z_T, Z_Y \mid Z_M)| \leq \sqrt{1 - \exp(-2\zeta^*)}$.

*Proof:* Under Gaussian assumptions, the relationship between conditional mutual information and partial correlation is $I(Z_T; Z_Y \mid Z_M) = -\frac{1}{2}\log(1-\rho_{pc}^2)$. From $I(Z_T; Z_Y \mid Z_M) \leq \zeta^*$, we obtain $\rho_{pc}^2 \leq 1 - \exp(-2\zeta^*)$.

**Proposition C.6.2 (CSI Metric)** Let $\mathcal{L} : T \to M \to Y^*$ be the correct causal path and $\mathcal{L}' : T \to Y^*$ the direct path. If $I(M; Y^*) - \delta_{MB} > 0$, then $\frac{R^2(\mathcal{L}) - R^2(\mathcal{L}')}{R^2(\mathcal{L}) + \epsilon} \geq 1 - \frac{O(\zeta^{*2})}{I(M;Y^*) - \delta_{MB}}$.

*Proof:* By Markov boundary retention (Theorem C.2), $R^2(\mathcal{L}) \propto I(Z_M; Z_Y) \geq I(M; Y^*) - \delta_{MB}$. By conditional independence preservation, $R^2(\mathcal{L}') \propto I(Z_T; Z_Y) \leq \zeta^*$ under linear approximation.

**Proposition C.6.3 (MBRI Metric)** Define $\text{MBRI} = \widehat{I}(M;Y)/[\widehat{I}(M;Y) + \widehat{I}(T;Y) + \epsilon]$. Then $\mathbb{E}[\text{MBRI}] \geq \frac{I(M;Y^*) - \delta_{MB}}{I(M;Y^*) - \delta_{MB} + \zeta^* + \epsilon} + O(n^{-1/2})$.

*Proof:* By Theorem C.2, the numerator $\hat{I}(M;Y) \xrightarrow{P} I(Z_M;Z_Y) \geq I(M;Y^*) - \delta_{MB}$. By Theorem C.1, in the denominator $\hat{I}(T;Y) \xrightarrow{P} I(Z_T;Z_Y) \leq \zeta^*$. The result follows from the continuous mapping theorem.

**Proposition C.6.4 (MAC Metric)** Let $\Delta_a = \{|a_i - a_j|\}_{i,j}$ be semantic distances and $\Delta_z = \{\|z_i - z_j\|_2\}_{i,j}$ be representation distances. Then $\rho_S(\Delta_a, \Delta_z) \geq c(\gamma) - O(\tau \log n + n^{-1/2})$ where $c(\gamma)$ is determined by the order-preserving margin $\gamma$ and $\tau$ is the soft-rank temperature.

*Proof:* By Theorems C.3 and B.7, soft-rank converges to true ranks at rate $O(\tau \log n)$ and sample Spearman correlation converges at rate $O(n^{-1/2})$.

## C.9  PROOFS OF CAUSAL STRUCTURE PRESERVATION THEOREMS

**Lemma C.8.1 (Conditional Independence Transfer).** Under the anti-bypass assumption ($Z_Y$ depends only on $Y$ or $I^Y$), there exists a Markov chain $Z_T \to (Y, Z_M) \to Z_Y$. By the conditional data processing inequality, $I(Z_T;Z_Y \mid Z_M) \leq I(Z_T;Y \mid Z_M)$. Combined with Theorem B.2.1's NLL calibration consistency, when $q$ is realizable or under cross-fitting, $I(Z_T;Y \mid Z_M) = \mathcal{L}_{CI} - \zeta^* - o_P(1)$, where $\zeta^*$ is the model misspecification bias.

**Lemma C.8.2 (Markov Boundary Necessity).** Let $Z_M = g(M)$ be a deterministic encoding. By mutual information decomposition, $I(Z_M;Z_Y) = I(Z_M;Y^*) + I(Z_M;Z_Y \mid Y^*) - I(Z_M;Y^* \mid Z_Y)$. Since $I(Z_M;Z_Y \mid Y^*) = 0$ (Markov property) and $I(Z_M;Y^* \mid Z_Y) \leq H(Y^* \mid Z_Y)$, we have $I(Z_M;Z_Y) \geq I(Z_M;Y^*) - H(Y^* \mid Z_Y)$. Furthermore, by chain decomposition $I(M;Y^*) = I(Z_M;Y^*) + I(M;Y^* \mid Z_M)$ and $I(M;Y^* \mid Z_M) \leq H(M \mid Z_M)$, we obtain $I(Z_M;Y^*) \geq I(M;Y^*) - H(M \mid Z_M)$. Combining yields $I(Z_M;Z_Y) \geq I(M;Y^*) - H(M \mid Z_M) - H(Y^* \mid Z_Y)$. Defining information loss $\delta_{MB} = H(M \mid Z_M) + H(Y^* \mid Z_Y)$, we have $I(Z_M;Z_Y) \geq I(M;Y^*) - \delta_{MB}$. InfoNCE provides a consistent lower bound for $I(Z_M;Z_Y)$ as $K \to \infty$ with estimation error $O_P(K^{-1/2})$.

**Proof of Theorem 1.** Under the parameter space $\Theta = \{\theta : \|W_\ell(\theta)\|_2 \leq c_\ell, \|\text{CNN}_k(\theta)\|_{2 \to 2} \leq \kappa_k\}$ with spectral norm constraints, the loss function $\mathcal{L}$ is lower semicontinuous on compact sets. By variational analysis, the empirical risk minimization problem $\min_{\theta \in \Theta} \mathcal{L}(\theta; P_n)$ admits solution sequences.

For condition (i), by Lemma C.9 and Theorem B.2.1, $\mathcal{L}_{CI}$ converges to a consistent estimate of $I(Z_T;Y|Z_M)$ with bias $|\mathcal{L}_{CI} - I(Z_T;Y|Z_M)| \leq \zeta^*(\mathcal{E}) + O_P(n^{-1/2})$. For condition (ii), $\mathcal{L}_{MBR} = -\text{InfoNCE}(z_m, z_y)$ directly optimizes $I(Z_M;Z_Y)$. By Theorem B.3.2, InfoNCE provides a lower bound with bias $O_P(K^{-1/2})$, and combined with Lemma C.9, $I(Z_M;Z_Y) \geq I(M;Y^*) - \delta_{MB} - O_P(K^{-1/2})$. For condition (iii), by Theorem B.5.1, choosing temperature $\tau = O(n^{-1/3})$ yields $|\hat{\rho}_\tau - \rho_S| = O_P(n^{-1/3})$, and combined with the order-preserving margin assumption, $\rho_S \geq c_0 - O_P(n^{-1/3})$.

Combining estimation errors from all three conditions, there exists $\varepsilon = \max\{\zeta^*(E_T), \zeta^*(E_M), \zeta^*(E_Y), O_P(n^{-1/2}), O_P(K^{-1/2}), O_P(n^{-1/3})\}$ such that the limit point satisfies $\varepsilon$-CSP. When $K \geq cn$, $K^{-1/2}$ and $n^{-1/2}$ are of the same order; if $K = \omega(n)$ (e.g., $K = n \log n$), then $K^{-1/2} = o(n^{-1/2})$. Therefore, the dominant order is $O_P(n^{-1/2})$. Different choices of weights $(w_{CI}, w_{MBR}, w_{MAC})$ correspond to different points on the Pareto frontier.

**Proof of Theorem 2.** We prove for three identification criteria.

*Backdoor criterion*: The original query is $Q = \sum_x \mathbb{E}[Y^*|t, x]P(x)$. In representation space, $\tilde{Q} = \sum_{z_x} \mathbb{E}[h(Z_Y)|E_T(t), E_X(x)]P(z_x)$. Define intermediate quantity $\hat{Q} = \sum_x \mathbb{E}[h(Z_Y)|t, x]P(x)$. Error decomposes as $|\tilde{Q} - Q| \leq |\tilde{Q} - \hat{Q}| + |\hat{Q} - Q|$. The first term is controlled by Lipschitz properties: $|\mathbb{E}[h(Z_Y)|E_T(t), E_X(x)] - \mathbb{E}[h(Z_Y)|t, x]| \leq \varepsilon \cdot \text{Lip}(h)$. The second term $\delta_{cal} = \sup_{(t,x)} |\mathbb{E}[Y^*|t, x] - \mathbb{E}[h(Z_Y)|t, x]|$ is the conditional calibration error. Thus $|\tilde{Q} - Q| \leq \varepsilon + \delta_{cal}$, i.e., $\kappa = 1$.

*Frontdoor criterion*: The original query is $Q = \sum_{m,x} \mathbb{E}[Y^*|m, x]P(m|t)P(x)$. In representation space, $\tilde{Q} = \sum_{z_m, z_x} \mathbb{E}[h(Z_Y)|z_m, z_x]P(z_m|E_T(t))P(z_x)$. Error decomposes into three terms: (a)

mediator distribution error $|P(z_m|E_T(t)) - P(m|t)| \le \varepsilon \cdot \text{Lip}(P_{M|T})$; (b) outcome model error $|\mathbb{E}[h(Z_Y)|z_m, z_x] - \mathbb{E}[Y^*|m, x]| \le \varepsilon \cdot \text{Lip}(h) + \delta_{cal}$; (c) marginal distribution error $|P(z_x) - P(x)| \le \varepsilon \cdot \text{Lip}(P_X)$. Since the frontdoor formula involves nested expectations, total error $|\tilde{Q} - Q| \le (\text{Lip}(P_{M|T}) + \text{Lip}(h) + \text{Lip}(P_X)) \cdot \varepsilon + \delta_{cal}$. Define $\kappa = \max\{\text{Lip}(P_{M|T}), \text{Lip}(h), \text{Lip}(P_X)\}$.

*Instrumental variable criterion*: Let $Z$ be an instrument. The original query via two-stage least squares is $Q = \text{Cov}(Y^*, Z)/\text{Cov}(T, Z)$. In representation space, $\tilde{Q} = \text{Cov}(h(Z_Y), Z_Z)/\text{Cov}(Z_T, Z_Z)$ where $Z_Z = E_Z(Z)$. Numerator error $|\text{Cov}(h(Z_Y), Z_Z) - \text{Cov}(Y^*, Z)| \le \varepsilon \cdot \text{Lip}(h) + \delta_{cal} \cdot \|Z\|_2$; denominator error $|\text{Cov}(Z_T, Z_Z) - \text{Cov}(T, Z)| \le \varepsilon \cdot (\text{Lip}(E_T) + \text{Lip}(E_Z))$. By Lipschitz properties of ratios, when $\text{Cov}(T, Z) \ge c_0 > 0$ (relevance lower bound), total error $|\tilde{Q} - Q| \le c_0^{-2} \cdot [(\text{Lip}(h) + \|Z\|_2) \cdot \varepsilon + \delta_{cal}]$. Thus $\kappa = c_0^{-2} \cdot \max\{\text{Lip}(h), \|Z\|_2, \text{Lip}(E_T), \text{Lip}(E_Z)\}$.

Under realizability assumption $\mathbb{E}[Y^*|Z_Y] = h^*(Z_Y)$ and cross-fitting, $\delta_{cal} = O_P(n^{-1/2})$ by Theorem B.3.1.

# D  IMPLEMENTATION DETAILS

## D.1  ENCODER PARAMETERIZATION

For representation mappings $E : \mathcal{X} \to \mathcal{Z} \subset \mathbb{R}^d$, we specify the functional forms. Throughout this section, $\|\cdot\|_2$ denotes the operator spectral norm. Tabular encoders $E_T$ and $E_M$ employ multilayer perceptron architectures defined as the composition $E = \phi_L \circ \ldots \circ \phi_1$, where each layer $\phi_\ell : \mathbb{R}^{d_{\ell-1}} \to \mathbb{R}^{d_\ell}$ has the form $\phi_\ell(x) = \sigma(W_\ell x + b_\ell)$ with $\sigma$ being a 1-Lipschitz activation function. To satisfy the Lipschitz constraint (Assumption A.5), weight matrices $W_\ell$ satisfy spectral norm constraints $\|W_\ell\|_2 \le c_\ell$, yielding total Lipschitz constant $L \le \prod_\ell c_\ell$.

Image encoders $E_{IM}$ and $E_{IY}$ adopt convolutional architectures $F = \psi_K \circ \text{CNN}_K \circ \ldots \circ \text{CNN}_1$. Each convolutional operator $\text{CNN}_k$ satisfies $\|\text{CNN}_k\|_{2\to 2} \le \kappa_k$ (estimated via spectral normalization). The final projection $\psi_K : \mathbb{R}^{h \times w \times c} \to \mathbb{R}^d$ is a linear mapping satisfying $\|\psi_K\|_2 \le 1$.

## D.2  BATCH CONSTRUCTION FOR LOSS FUNCTIONS

The InfoNCE loss constructs positive and negative samples following this mathematical procedure. Given batch $\mathcal{B} = \{(x_i, y_i)\}_{i=1}^N$, positive pairs are defined as $\mathcal{P} = \{(z_i, \psi_Y(y_i)) : i \in [N]\}$. The negative sample set $\mathcal{N}_i = \{\psi_Y(y_j) : j \ne i\}$ serves as approximately independent samples (see Assumption A.5).

The scoring function $f : \mathcal{Z} \times \mathcal{Y} \to \mathbb{R}$ is a bounded Lipschitz function. Theoretical results do not depend on the specific form; in practice, normalized inner products or other bilinear forms may be employed.

The two terms in conditional mutual information estimation utilize the same predictor family $q_\theta$ to ensure consistency (theoretical justification in Appendix B.1).

## D.3  PREDICTOR ARCHITECTURE FOR CMI ESTIMATION

We adopt a two-tower design to avoid information leakage:

$$\text{Enc}_T : X_T \to Z_T, \qquad \text{Enc}_M : X_M \to Z_M.$$

The joint predictor and the marginal predictor are parameterized separately as

$$q_\theta : [Z_T, Z_M] \to \text{MLP}_\theta \to y, \qquad q_\phi : Z_M \to \text{MLP}_\phi \to y.$$

No layer in $q_\phi$ receives any function of $Z_T$. We compute

$$L_{CI} = \mathbb{E}[-\log q_\phi(y \mid Z_M)] - \mathbb{E}[-\log q_\theta(y \mid Z_T, Z_M)]$$

on held-out mini-batches (or via $K$-fold cross-fitting) to reduce overfitting bias.

### D.4 PC Algorithm Formalization

Conditional independence testing employs hypothesis testing based on partial correlation coefficients. For node pair $(Z_i, Z_j)$ and conditioning set $\mathcal{S}$, the test statistic is $T_{ij|\mathcal{S}} = \sqrt{n - |\mathcal{S}| - 3} \cdot \text{arctanh}(r_{ij|\mathcal{S}})$, where $r_{ij|\mathcal{S}}$ is the sample partial correlation coefficient. Actual testing uses Fisher transformation (see Appendix B.7). The recursive relation

$$r_{ij|\mathcal{S} \cup \{k\}} = \frac{r_{ij|\mathcal{S}} - r_{ik|\mathcal{S}} r_{jk|\mathcal{S}}}{\sqrt{(1 - r_{ik|\mathcal{S}}^2)(1 - r_{jk|\mathcal{S}}^2)}} \tag{20}$$

serves to define partial correlations.

Edge orientation follows the PC algorithm skeleton with Meek orientation rules (Spirtes-Glymour-Scheines): v-structures $Z_i \to Z_k \leftarrow Z_j$ are identified if and only if $Z_i - Z_k - Z_j$ forms a triple and $Z_k \notin \text{Sep}(Z_i, Z_j)$.

Variable tracing is achieved through deterministic mapping $\pi : V_{\mathcal{Z}} \to V_{\text{orig}}$, arising from the single-valuedness of encoders.

### D.5 Optimization Algorithm

CARL employs first-order optimization of the joint objective:

$$\min_{E \in \mathcal{E}} \mathcal{L}(E; \mathcal{D}) = w_{CI} \mathcal{L}_{CI} + w_{MBR} \mathcal{L}_{MBR} + w_{MAC} \mathcal{L}_{MAC} + \lambda \mathcal{R}(E) \tag{21}$$

where the regularization term $\mathcal{R}(E)$ incorporates weight norm and spectral norm penalties to control capacity and maintain Lipschitz bounds.

Parameter updates are implemented via projected gradient descent:

$$\theta^{t+1} = \Pi_{\Theta}[\theta^t - \eta_t \nabla_\theta \mathcal{L}(\theta^t)] \tag{22}$$

where $\Pi_{\Theta}$ is the projection operator onto the feasible parameter space:

$$\Theta = \{\theta : \|W_\ell(\theta)\|_2 \leq c_\ell, \forall \ell; \|\text{CNN}_k(\theta)\|_{2 \to 2} \leq \kappa_k, \forall k\} \tag{23}$$

encompassing spectral norm constraints for multilayer perceptrons and convolutional operators.

## E Related Work

**Cross-Modal Representation Learning.** Cross-modal representation learning has become a cornerstone of modern AI Sun et al. (2025), enabling machines to understand and integrate information from disparate sources, such as vision, language, and tabular data Mao et al. (2022). The primary goal is to learn a shared semantic embedding space where data from different modalities are aligned and comparable. Early pioneering work, such as Deep Canonical Correlation Analysis (DCCA) Andrew et al. (2013), focused on maximizing the statistical correlation between modality-specific representations. In recent years, contrastive learning has emerged as the dominant paradigm, fueling the success of large-scale vision-language models like CLIP Radford et al. (2021) and ALIGN Jia et al. (2021), and more recent architectures that unify even more modalities, such as ImageBind Girdhar et al. (2023) and models from the PaLI family Chen et al. (2023). These models have achieved remarkable zero-shot performance on various tasks by aligning vast unpaired datasets.

Despite their empirical power, the training objectives of these methods are fundamentally statistical and remain agnostic to the underlying causal mechanisms that generate the data Pearl (2009); Arjovsky et al. (2019). By optimizing for reconstruction accuracy, statistical correlation, or contrastive similarity, these powerful nonlinear mappings can inadvertently distort the data's intrinsic causal graph, a critical failure we identify as representation-induced structural drift. This distortion can manifest as the introduction of spurious dependencies not present in the original system or, conversely, the elimination of critical mediators essential for correct causal reasoning. This phenomenon is closely related to the problem of shortcut learning, where models exploit statistical artifacts rather than the intended causal features Geirhos et al. (2020). Consequently, while current cross-modal models excel at pattern recognition and statistical association, they lack the structural guarantees necessary for robust causal inference, motivating the urgent need for a new class of causally-aware representation learning frameworks.

**Causal Representation Learning.** In response to the limitations of traditional methods, the field of Causal Representation Learning (CRL) has emerged with the ambitious goal of learning representations that encode not just statistical patterns, but the underlying causal structure of the data-generating process Schölkopf et al. (2021). Such causally-informed representations are believed to be more robust, generalize better to out-of-distribution scenarios Liu et al. (2021a; 2022); Bagi et al. (2023), and provide a principled basis for fair and explainable models Kusner et al. (2017). A significant body of work in CRL has focused on learning disentangled representations, where latent factors correspond to independent causal mechanisms, often by leveraging assumptions of invariance across different environments or interventions Lu et al. (2022); Brehmer et al. (2022). Other lines of research have focused on enabling counterfactual reasoning within the learned latent space, which is critical for tasks like algorithmic fairness and personalized decision-making Zuo et al. (2023).

However, the vast majority of existing CRL research has been developed and validated on unimodal, often single-domain, datasets (e.g., images). Consequently, these methods are often ill-equipped to handle the unique challenges posed by multi-modal data, such as information asymmetry between modalities or the need to ensure consistent geometric and semantic alignment, which we identified earlier. Furthermore, while some recent studies have begun to explore causal learning from multi-modal biomedical observations Sun et al. (2025), a general framework that explicitly guarantees the preservation of causal effect identifiability conditions, such as the backdoor, frontdoor, and instrumental variable criteria Pearl (2009), in a shared representation space remains an open challenge. This gap is particularly critical in high-stakes domains like healthcare, where preserving the integrity of causal pathways for reliable decision-making is paramount.

**Preserving Causal Structure in Multi-Modal Settings.** A natural and critical frontier is therefore to bridge the gap between these two lines of research: to enable the powerful alignment capabilities of cross-modal models with the robust causal principles from CRL. However, this synthesis is profoundly challenging, as the unique characteristics of multi-modal data introduce specific obstacles that are not addressed by conventional unimodal CRL methods Schölkopf et al. (2021). One major obstacle is the Cross-modal Information Bottleneck (CIB), where information-dense modalities (e.g., images or text) can dominate the learning objective, potentially masking or discarding causally salient variables from sparser modalities (e.g., tabular biomarkers) Zhang et al. (2024); Wei et al. (2024); Xu et al. (2025). Another is ensuring Modal Alignment Consistency (MAC); standard contrastive objectives guarantee coarse-grained alignment but do not enforce a monotonic relationship between semantic similarity and latent space distance, a property crucial for preserving ordered relationships and the relative strength of causal effects Liang et al. (2022).

Most importantly, the complex, nonlinear transformations inherent in deep encoders offer no guarantee of Cross-modal Identifiability Consistency (CIC). The very conditions required for causal effect identification, such as the backdoor, frontdoor, and instrumental variable criteria, can be invalidated during the encoding process, rendering downstream causal queries unreliable Pearl (2009). While initial and important steps have been taken to explore causal learning from multi-modal biomedical data Sun et al. (2025), these pioneering efforts often focus on specific applications or do not provide a general mechanism for verifiably preserving these identifiability conditions. As of yet, a general framework designed to explicitly tackle CIB, MAC, and CIC simultaneously, thereby ensuring that learned cross-modal representations are "causally sufficient" for downstream inference, remains an open and pressing challenge.

# F EXPERIMENT DETAILS

## F.1 EXPERIMENTAL SETUP

Network architectures and optimization protocols are fixed a priori and selected by cross-validated grid search to ensure comparability across methods. Tabular encoders adopt a three-hidden-layer MLP with 128 units per layer, each block ordered as Linear→BatchNorm→ReLU. Image encoders use a ResNet-18 trained from scratch with the same data partitions as the tabular streams. The predictors $q_\theta$ and $q_\phi$ used in the conditional-independence objective are two parameter-independent MLPs with two hidden layers of 64 units and ReLU activations. All models are optimized with Adam at learning rate $1 \times 10^{-4}$, $\beta_1 = 0.9$, $\beta_2 = 0.999$, and $L_2$ regularization $1 \times 10^{-5}$. Mini-batches contain 256 samples. Training runs for at most 200 epochs with early stopping monitored on

the validation objective and a patience of 20 epochs. CARL employs fixed loss weights $w_{CI} = 1.0$, $w_{MBR} = 0.5$, $w_{MAC} = 0.1$, representation dimension $d = 64$, the number of InfoNCE negatives $K_{\text{neg}} = 128$, and soft-rank temperature $\tau_{\text{rank}} = 0.1$. Model selection uses five-fold cross-validation; hyperparameters are chosen by inner validation on the training folds, and all baselines follow the same protocol, budgets, and representation dimension.

### F.2 EVALUATION METRICS

All metric values lie in $[0, 1]$. Let $N$ denote the batch size. The number of InfoNCE negatives is $K_{\text{neg}} = 128$ and the total number of candidates in the denominator is $K_{\text{tot}} = K_{\text{neg}} + 1$. The significance level is $\alpha = 0.05$. The kernel-ridge regularization parameter is $\lambda = 10^{-3}$. Numerical stability uses $\epsilon = 10^{-8}$. The centering matrix is $H = I - \frac{1}{N}\mathbf{1}\mathbf{1}^\top$. Cosine similarity is $s(\mathbf{u}, \mathbf{v}) = \frac{\mathbf{u}^\top \mathbf{v}}{\|\mathbf{u}\|_2 \|\mathbf{v}\|_2}$.

**Causal Structure Index (CSI).**

$$\text{CSI} = \tfrac{1}{4} \sum_{k=1}^{4} \mathbb{I}\big(p^{(k)} > \alpha\big)$$

where the four null hypotheses are $Z_T \perp Z_Y \mid Z_M$, $T \perp Z_Y \mid Z_M$, $Z_T \perp Y \mid Z_M$, and $T \perp Y \mid M$. Each $p$-value is computed by a kernel-based conditional-independence test with Gaussian kernels. Bandwidths follow the median heuristic. Residual kernels are obtained by kernel ridge regression with $\lambda$ and the statistic

$$T_{\text{KCI}}(U, V \mid W) = \frac{1}{N^2} \text{tr}\big(H K_{U \perp W} H K_{V \perp W}\big)$$

is calibrated by wild bootstrap with $B = 500$ resamples.

**Markov Boundary Retention Index (MBRI).**

$$\widehat{I}_{\text{NCE}}(U; V) = \mathbb{E}\Big[ \log \frac{\exp s(\mathbf{u}, \mathbf{v})}{\sum_{\mathbf{v}' \in \mathcal{N}(\mathbf{v})} \exp s(\mathbf{u}, \mathbf{v}')} \Big] + \log K_{\text{tot}}.$$

$$\widetilde{I}(U; V) = \min\{ \widehat{I}_{\text{NCE}}(U; V) / \log K_{\text{tot}}, \, 1 \}.$$

The final score is

$$\text{MBRI} = \frac{\widetilde{I}(Z_M; Z_Y)}{\widetilde{I}(Z_M; Z_Y) + \widetilde{I}(Z_T; Z_Y) + \epsilon}.$$

**Monotonic Alignment Consistency (MAC).** From semantic amplitudes $\{a_i\}$ and representations $\{z_i\}$, a uniformly subsampled set of pairs $P$ of size $|P| = \min\{50{,}000, \frac{N(N-1)}{2}\}$ forms the vectors $\Delta a = \{|a_i - a_j|\}_{(i,j) \in P}$ and $\Delta z = \{\|z_i - z_j\|_2\}_{(i,j) \in P}$. Spearman's rank correlation is mapped to $[0, 1]$ by

$$\text{MAC} = \tfrac{1}{2}\big(1 + \rho_S(\Delta a, \Delta z)\big).$$

**Structural Accuracy.** Let $A_{\text{pred}}^{\text{skel}}$ and $A_{\text{true}}^{\text{skel}}$ denote the predicted and true skeleton adjacency matrices on the evaluation node set. The skeleton Structural Hamming Distance is $\text{SHD}_{\text{skel}} = \|\text{Upper}(A_{\text{pred}}^{\text{skel}} - A_{\text{true}}^{\text{skel}})\|_0$. Let $m_\cup = \|\text{Upper}(A_{\text{pred}}^{\text{skel}} \vee A_{\text{true}}^{\text{skel}})\|_0$ with guard $m_\cup \geq 1$. The score is

$$\text{Structural} = 1 - \frac{\text{SHD}_{\text{skel}}}{m_\cup}.$$

**Representation Information Content (RIC-avg).**

$$\text{RIC-avg} = \tfrac{1}{3}\Big( \widetilde{I}(Z_T; T) + \widetilde{I}(Z_M; M) + \widetilde{I}(Z_Y; Y) \Big)$$

with $\widetilde{I}$ identical to the definition used in MBRI and the same $K_{\text{neg}}$, $K_{\text{tot}}$, and similarity function.

Table 4: Parameter configurations for the synthetic datasets (full-factorial grid).

| $n$ | $\sigma = 0.1$ | | | $\sigma = 0.3$ | | | $\sigma = 0.5$ | | |
|---|---|---|---|---|---|---|---|---|---|
| | linear | quadratic | neural | linear | quadratic | neural | linear | quadratic | neural |
| 500 | ✓ | ✓ | ✓ | ✓ | ✓ | ✓ | ✓ | ✓ | ✓ |
| 1000 | ✓ | ✓ | ✓ | ✓ | ✓ | ✓ | ✓ | ✓ | ✓ |
| 2000 | ✓ | ✓ | ✓ | ✓ | ✓ | ✓ | ✓ | ✓ | ✓ |
| 5000 | ✓ | ✓ | ✓ | ✓ | ✓ | ✓ | ✓ | ✓ | ✓ |

### F.3 PARAMETER CONFIGURATIONS

The synthetic datasets follow a full-factorial design over sample size $n \in \{500, 1000, 2000, 5000\}$, noise level $\sigma \in \{0.1, 0.3, 0.5\}$, and nonlinearity type $\{\text{linear}, \text{quadratic}, \text{neural}\}$. The direct effect is fixed to $\delta = 0$, implying the ground-truth condition $T \perp Y^* \mid M$. For each $(n, \sigma, \text{nonlinearity})$ combination, instances are generated under the IM, IY, and DUAL scenarios using identical random seed and train–validation–test split.

## G SUPPORTING EVIDENCE FOR CAUSAL PATHWAY DISCOVERY

### G.1 BLOOD PRESSURE → CVD EVIDENCE

The Prospective Studies Collaboration meta-analysis of individual data for one million adults demonstrates age-specific relevance of usual blood pressure to vascular mortality, showing robust relationships across age groups Lewington et al. (2002). The Blood Pressure Lowering Treatment Trialists' Collaboration individual participant-level data meta-analysis shows pharmacological blood pressure lowering reduces cardiovascular disease across different blood pressure levels, with each reduction associated with proportional risk reduction in major adverse cardiovascular events Blood Pressure Lowering Treatment Trialists' Collaboration (2021).

### G.2 ARTERIAL STIFFNESS → CVD EVIDENCE

Systematic review and meta-analysis demonstrates that arterial stiffness predicts cardiovascular events and all-cause mortality Vlachopoulos et al. (2010). Individual participant meta-analysis of 17,635 subjects shows aortic pulse wave velocity improves cardiovascular event prediction Ben-Shlomo et al. (2014).

### G.3 RETINAL MICROVASCULAR → CVD EVIDENCE

The Atherosclerosis Risk in Communities Study demonstrates retinal microvascular abnormalities predict incident stroke Wong et al. (2001). The same cohort study shows retinal arteriolar narrowing increases risk of coronary heart disease in men and women Wong et al. (2002).

### G.4 RENAL FUNCTION → CVD EVIDENCE

Collaborative meta-analysis shows association of estimated glomerular filtration rate and albuminuria with all-cause and cardiovascular mortality in general population cohorts Chronic Kidney Disease Prognosis Consortium et al. (2010). Chronic kidney disease independently associates with risks of death, cardiovascular events, and hospitalization Go et al. (2004).

### G.5 AGE → RETINAL FUNDUS → CVD EVIDENCE

Retinal age gap serves as a predictive biomarker for mortality risk, with biological age differences between chronological and retinal-predicted age associating with increased mortality Zhu et al. (2023).

### G.6 Inflammation → Microvascular → CVD Evidence

C-reactive protein associates with retinal microvascular caliber in multiethnic populations, demonstrating inflammation-microvascular connections Cheung et al. (2010). Systematic review and meta-analysis confirms associations between markers of inflammation and retinal microvascular parameters Liu et al. (2021b).

### G.7 Sleep Apnea → HRV → CVD Evidence

Heart rate variability analysis shows differences in obstructive sleep apnea patients with and without excessive daytime sleepiness Ucak et al. (2024). Meta-analysis demonstrates heart rate variability predicts first cardiovascular event in populations without known cardiovascular disease Hillebrand et al. (2013).

### G.8 BMI → Metabolism → CVD Evidence

Pooled analysis of 97 prospective cohorts with 1.8 million participants identifies metabolic mediators of body-mass index effects on coronary heart disease and stroke Global Burden of Metabolic Risk Factors for Chronic Diseases Collaboration (BMI Mediated Effects) et al. (2014).

### G.9 Gut Microbiome → Metabolism → CVD Evidence

Landmark studies demonstrate gut bacterial metabolism of choline and phosphatidylcholine promotes cardiovascular disease through production of proatherogenic metabolite TMAO Wang et al. (2011). NEJM research confirms intestinal microbial metabolism of phosphatidylcholine associates with cardiovascular risk in humans Tang et al. (2013). Nature Medicine studies reveal intestinal microbiota metabolism of L-carnitine, abundant in red meat, also produces TMAO and promotes atherosclerosis, establishing the "microbiome-metabolism-atherosclerosis" pathway Koeth et al. (2013).

## H Real data

### H.1 Description of Human Phenotype Project Dataset

Comprehensive biomedical datasets that integrate multiple modalities are essential for understanding the complex interplay between different physiological systems. The Human Phenotype Project (HPP) provides such a resource, representing a large-scale, longitudinal collection of deep phenotypic profiles. Our study utilizes data from 6,366 adult participants (3,043 male and 3,323 female) within this cohort, with a mean age of 52.4±7.7 years and a mean BMI of 26.1±4.1kg/m$^2$. The project collected a wide array of clinical, physiological, behavioral, and multi-omic profiling data, categorized into 17 major body systems Kohn et al. (2025).

A key strength of the HPP cohort is its rich, multi-modal structure, encompassing tabular data, time-series signals, and medical imaging. For instance, the sleep data alone consists of 448 characteristics collected over 16,812 nights of home sleep apnea testing, providing high-resolution time-series information. This is complemented by other modalities such as tabular data from anthropometrics and body composition (120 features), blood tests for inflammation and immune system markers (12 features), and medical imaging from retinal fundus scans, which is part of the cardiovascular assessment (162 features). In this work, we focus on data spanning six key systems: Anthropometrics, Sleep, Cardiovascular, Fundus, Metabolic Pathway, and Inflammation. The integration of these distinct yet causally related modalities, each with a high-dimensional feature set, provides a unique opportunity to apply cross-modality causal inference methods, aiming to uncover the underlying mechanisms connecting these physiological domains.

### H.2 Body System-Derived Features

**Sleep Monitoring** The sleep monitoring data is a high-resolution, time-series dataset designed to capture the complex dynamics of sleep. Each participant underwent at least one series of home sleep

Table 5: Complete experimental results across varying sample sizes, noise levels, and nonlinearity functions. Performance metrics include Causal Structure Index (CSI), Markov Boundary Retention Index (MBRI), Monotonic Alignment Consistency (MAC), Structural Accuracy, and averaged Representation Information Content (RIC-avg).

| Sample Size | Noise | Nonlinearity | CSI | MBRI | MAC | Structural | RIC-avg |
|---|---|---|---|---|---|---|---|
| 500 | 0.1 | linear | 1.0000 | 0.5284 | 0.5312 | 0.4506 | 0.4483 |
| 500 | 0.1 | quadratic | 1.0000 | 0.5152 | 0.5484 | 0.4994 | 0.4539 |
| 500 | 0.1 | neural | 1.0000 | 0.4877 | 0.5292 | 0.6013 | 0.4388 |
| 500 | 0.3 | linear | 1.0000 | 0.5350 | 0.5419 | 0.4741 | 0.4534 |
| 500 | 0.3 | quadratic | 1.0000 | 0.5380 | 0.5696 | 0.5514 | 0.4526 |
| 500 | 0.3 | neural | 1.0000 | 0.5089 | 0.5358 | 0.6001 | 0.4313 |
| 500 | 0.5 | linear | 1.0000 | 0.5159 | 0.5243 | 0.4526 | 0.4293 |
| 500 | 0.5 | quadratic | 1.0000 | 0.5267 | 0.5490 | 0.5242 | 0.4273 |
| 500 | 0.5 | neural | 1.0000 | 0.4840 | 0.5327 | 0.5821 | 0.4323 |
| 1000 | 0.1 | linear | 1.0000 | 0.5088 | 0.5847 | 0.6693 | 0.4408 |
| 1000 | 0.1 | quadratic | 1.0000 | 0.5405 | 0.5894 | 0.6584 | 0.4388 |
| 1000 | 0.1 | neural | 1.0000 | 0.4692 | 0.5665 | 0.7075 | 0.4518 |
| 1000 | 0.3 | linear | 1.0000 | 0.5149 | 0.5953 | 0.6762 | 0.4419 |
| 1000 | 0.3 | quadratic | 1.0000 | 0.5463 | 0.6068 | 0.6884 | 0.4404 |
| 1000 | 0.3 | neural | 1.0000 | 0.4980 | 0.5702 | 0.7102 | 0.4545 |
| 1000 | 0.5 | linear | 1.0000 | 0.5097 | 0.5804 | 0.6565 | 0.4284 |
| 1000 | 0.5 | quadratic | 1.0000 | 0.5337 | 0.5938 | 0.6651 | 0.4196 |
| 1000 | 0.5 | neural | 1.0000 | 0.4819 | 0.5658 | 0.6954 | 0.4364 |
| 2000 | 0.1 | linear | 1.0000 | 0.5660 | 0.6301 | 0.6893 | 0.4176 |
| 2000 | 0.1 | quadratic | 1.0000 | 0.6027 | 0.6484 | 0.6784 | 0.4212 |
| 2000 | 0.1 | neural | 1.0000 | 0.5102 | 0.6285 | 0.7275 | 0.4298 |
| 2000 | 0.3 | linear | 1.0000 | 0.5893 | 0.6352 | 0.6962 | 0.4248 |
| 2000 | 0.3 | quadratic | 1.0000 | 0.6500 | 0.6536 | 0.7084 | 0.4346 |
| 2000 | 0.3 | neural | 1.0000 | 0.5354 | 0.6320 | 0.7300 | 0.4308 |
| 2000 | 0.5 | linear | 1.0000 | 0.5808 | 0.6174 | 0.6765 | 0.4133 |
| 2000 | 0.5 | quadratic | 1.0000 | 0.6421 | 0.6362 | 0.6851 | 0.4257 |
| 2000 | 0.5 | neural | 1.0000 | 0.5013 | 0.6128 | 0.7154 | 0.4256 |
| 5000 | 0.1 | linear | 1.0000 | 0.6141 | 0.6432 | 0.7093 | 0.3948 |
| 5000 | 0.1 | quadratic | 1.0000 | 0.6344 | 0.6507 | 0.6984 | 0.3954 |
| 5000 | 0.1 | neural | 1.0000 | 0.6193 | 0.6384 | 0.7300 | 0.4231 |
| 5000 | 0.3 | linear | 1.0000 | 0.6437 | 0.6515 | 0.7162 | 0.4024 |
| 5000 | 0.3 | quadratic | 1.0000 | 0.6600 | 0.6609 | 0.7284 | 0.4070 |
| 5000 | 0.3 | neural | 1.0000 | 0.6209 | 0.6457 | 0.7300 | 0.4240 |
| 5000 | 0.5 | linear | 1.0000 | 0.6238 | 0.6397 | 0.6965 | 0.3949 |
| 5000 | 0.5 | quadratic | 1.0000 | 0.6500 | 0.6441 | 0.7051 | 0.4112 |
| 5000 | 0.5 | neural | 1.0000 | 0.6062 | 0.6252 | 0.7300 | 0.4221 |

monitoring tests, with each series consisting of three nights of continuous recording within a two-week timeframe. This multi-night protocol allows for a more reliable assessment by accounting for night-to-night variability. The collected data encompasses a wide range of metrics, including sleep architecture (sleep stages), body position, respiratory events, pulse rate, peripheral blood oxygen saturation ($SpO_2$), and snoring. Figure 5 provides an example of raw time-series records on selected sleep channels, illustrating the continuous nature of the data collected, such as respiratory movement, $SpO_2$, and body position. A primary focus of this dataset is the study of Obstructive Sleep Apnea (OSA), a prevalent sleep disorder characterized by repeated interruptions in breathing due to upper airway obstruction. These interruptions often lead to significant physiological responses, such as loud snoring, reductions in blood oxygen levels, awakenings, and fragmented sleep.

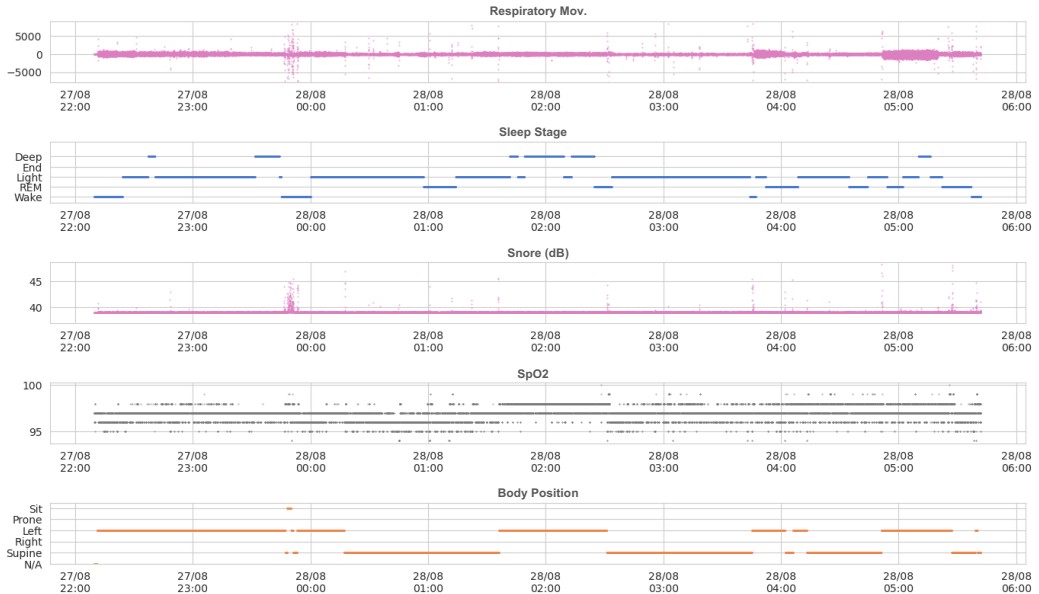

Figure 5: Example of Raw Time-Series Records on Selected Sleep Channels. The figure displays 8 hours of continuous data from a single participant, showing five different channels: Respiratory Movement (top), Sleep Stage, Snore (dB), Peripheral Oxygen Saturation (SpO$_2$), and Body Position (bottom). This illustrates the high-resolution, multi-channel nature of the raw sleep monitoring data.

The raw signals collected by devices are processed by clinically-validated algorithms to extract a rich set of 448 features for each participant. These features include key diagnostic indices for OSA, such as the peripheral Apnea-Hypopnea Index (pAHI), the Respiratory Disturbance Index (RDI), and the Oxygen Desaturation Index (ODI). Figure 7a shows the distribution of pAHI for male and female participants at the baseline visit. The histograms and empirical cumulative distribution functions (ECDF) clearly indicate that the male population exhibits a tendency towards higher pAHI values compared to the female population. In addition to respiratory events, the algorithms provide detailed statistics on sleep architecture, including the percentage of time spent in different sleep stages and overall sleep efficiency. Figure 6 further illustrates the derived events during sleep, such as sleep stages, various respiratory events, and pulse rate changes, highlighting the granular insights extracted from the raw channels. Furthermore, to capture autonomic nervous system activity, 348 Pulse Rate Variability (PRV) features are computed from the PAT channel, spanning time-domain, frequency-domain, and non-linear metrics. Table 6 provides a statistical summary of the key sleep-derived measurements for the cohort at their first visit (N=6,336).

**Fundus imaging**   The fundus data modality provides a non-invasive window into both ocular and systemic health through high-resolution imaging of the interior surface of the eye. In the HPP cohort, center-view 45° retinal images were collected for both eyes of each participant without pupil dilation, using the iCare DRSplus confocal fundus imaging system. This technique allows for the detailed visualization of critical structures within the fundus, including the retina, optic disc, macula, and the retinal microvasculature. The resulting images are essential for diagnosing and monitoring a variety of eye conditions, as the appearance of these structures, particularly the network of small blood vessels, offers valuable information about the health of the eye and the central nervous system.

To transform the raw images into a quantitative dataset suitable for analysis, the HPP employs an open-source deep learning pipeline known as AutoMorph Zhou et al. (2022). This automated pipeline processes the fundus images through several stages, including pre-processing, image quality grading, and anatomical segmentation of the optic disc and retinal vessels into arteries and veins. Following segmentation, a comprehensive set of morphological features is extracted, such as vessel caliber, tortuosity, density, and fractal dimension. Figure 7b provides an example of one such extracted feature, showing the distribution of left eye vessel density for male and female participants at the baseline visit. By quantifying the intricate geometry of the retinal microvasculature, this data

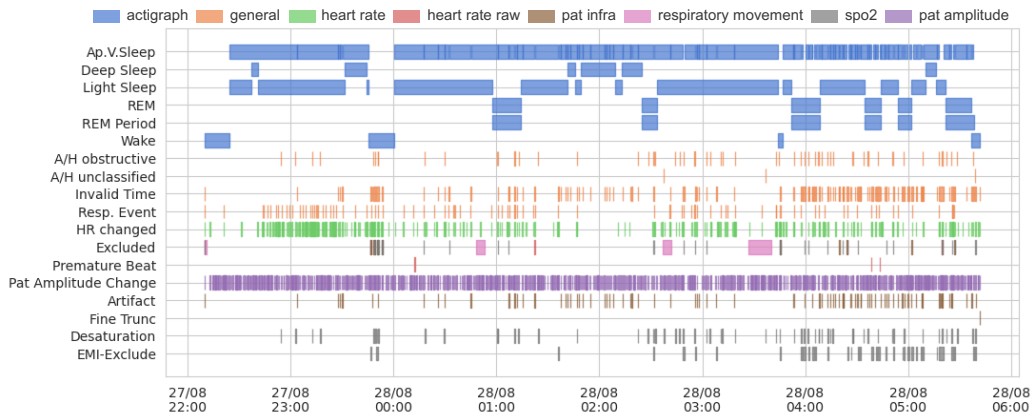

Figure 6: Derived Events During Sleep from Raw Channels. This figure presents a timeline of clinically relevant events and metrics derived from the raw sleep monitoring data. It includes sleep stages (Ap.V.Sleep, Deep Sleep, Light Sleep, REM Period, Wake), various respiratory events (e.g., A/H obstructive, Resp. Event), and other physiological events (e.g., HR changed, Desaturation), providing a summary of key sleep patterns and disturbances.

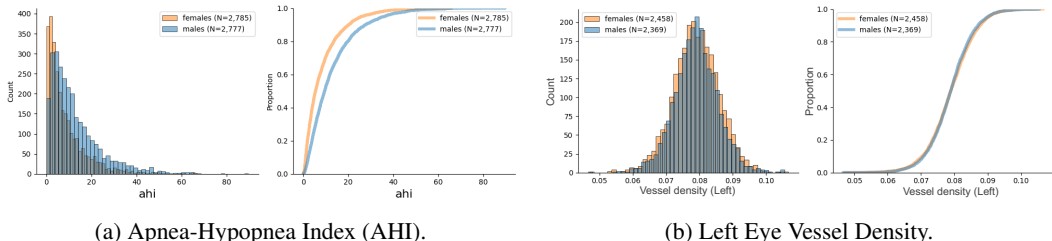

(a) Apnea-Hypopnea Index (AHI).          (b) Left Eye Vessel Density.

Figure 7: (a) Distribution of AHI for male (N=2,777) and female (N=2,785) participants at Baseline Visit. The left panel is a histogram, and the right panel is an ECDF. The data indicate that the AHI distribution for males is generally shifted towards higher values compared to females. (b) Distribution of Left Eye Vessel Density for male (N=2,369) and female (N=2,458) participants at Baseline Visit. The left panel is a histogram, and the right panel is an ECDF.

modality provides a rich set of features for investigating the links between microvascular health and broader systemic disease processes. Figure 8 displays the colored fundus images and corresponding segmentation results in HPP cohort.

**Anthropometrics**    The systematic measurement of the size, weight, and proportions of the human body, serves as a fundamental modality in the HPP cohort for assessing health and nutritional status. This dataset is primarily composed of tabular data collected through standardized protocols. To ensure precision and consistency, these measurements were taken using specific equipment, including a *Shekel Stadiometer* for height and weight. This set of measurements provides essential baseline characteristics for evaluating body composition and identifying potential health risks like obesity.

From these basic anthropometric measurements, important indices such as the Body Mass Index (BMI) can be derived. BMI is a key covariate used in a wide range of phenome-wide association studies. Figure 9a illustrates the distribution of BMI across the cohort at the baseline visit, stratified by sex. The histograms and ECDF clearly show that male participants (N=4,943) in this cohort tend to have a higher BMI compared to female participants (N=5,439). This quantitative, tabular data provides a foundational layer for our cross-modality causal analysis, allowing us to control and investigate the effects of body composition on other physiological systems.

**Cardiovascular System**    The cardiovascular system assessment within the HPP is a comprehensive and multi-modal evaluation, designed to provide a holistic view of cardiac and vascular health. This modality integrates data from several distinct, non-invasive measurement techniques, including

Table 6: Summary of Key Sleep-Derived Measurements.

| Parameters | First visit (*n*=6,336) | | | Second visit (*n*=574) | | |
|---|---|---|---|---|---|---|
| | Min / Max | Mean (s.d.) | P10 / P90 | Min / Max | Meand (s.d.) | P10 / P90 |
| Snore events above 40 dB (h) | 0.01 / 10.77 | 1.63 (1.54) | 0.22 / 3.93 | 0.02 / 8.33 | 1.80 (1.59) | 0.27 / 4.16 |
| Total sleep time (h) | 2.00 / 11.50 | 6.07 (1.21) | 4.54 / 7.51 | 2.00 / 11.58 | 6.08 (1.21) | 4.51 / 7.52 |
| Sleep efficiency (%) | 43.0 / 97.9 | 88.4 (5.6) | 81.1 / 94.3 | 45.1 / 98.0 | 88.5 (5.4) | 81.3 / 94.2 |
| Number of wakes (events) | 1 / 39 | 6.5 (3.9) | 2 / 12 | 1 / 34 | 6.5 (3.9) | 2 / 12 |
| Percentage of deep sleep time (%) | 0 / 38.8 | 17.6 (5.3) | 10.8 / 24.3 | 0 / 37.7 | 17.3 (5.4) | 10.5 / 24.2 |
| Percentage of REM sleep time (%) | 1.8 / 45.2 | 23.9 (7.3) | 14.2 / 33.2 | 2.4 / 44.8 | 23.8 (7.4) | 13.8 / 33 |
| Apnea-Hypopnea Index (events per h) | 0 / 90.8 | 10.8 (10.5) | 1.5 / 24.4 | 0 / 91.5 | 10.5 (10.4) | 1.5 / 23.7 |
| Oxygen Desaturation Index (events per h) | 0 / 86.6 | 4.5 (6.4) | 0.4 / 11.3 | 0 / 96.3 | 4.6 (7.1) | 0.3 / 11.2 |
| Mean heart rate during sleep (BPM) | 35 / 98 | 59.7 (8.0) | 50 / 70 | 35 / 93 | 60.2 (8.1) | 50 / 70 |
| Maximum heart rate during sleep (BPM) | 55 / 180 | 95.3 (11.7) | 82 / 109 | 55 / 180 | 94.8 (11.6) | 81 / 108 |
| Minimum heart rate during sleep (BPM) | 30 / 79 | 46.3 (6.8) | 38 / 55 | 30 / 79 | 46.9 (7.1) | 38 / 56 |
| Mean heart rate during NREM sleep (BPM) | 33.5 / 97.8 | 59.8 (8.0) | 49.8 / 70.2 | 31.9 / 92.8 | 60.3 (8.1) | 50.1 / 70.5 |
| Mean heart rate during REM sleep (BPM) | 33.1 / 107.9 | 61.4 (8.2) | 51.2 / 72.1 | 33.8 / 95.4 | 61.9 (8.2) | 51.5 / 72.6 |
| Mean heart rate during wakefulness (BPM) | 31.9 / 97.6 | 63.7 (8.3) | 53.4 / 74.3 | 34.3 / 100.0 | 64.0 (8.3) | 53.6 / 74.8 |
| Mean nadir of desaturations (%) | 67 / 98 | 92.1 (1.8) | 90 / 94 | 69 / 97 | 91.7 (1.8) | 89.6 / 93.7 |

Summary of key sleep-derived measurements for the first visit (n=6,336) and the second visit (n=574). Data are presented as the minimum and maximum values (Min / Max), the mean and standard deviation (Mean (s.d.)), and the 10th and 90th percentiles (P10 / P90). For example, for the 'Total sleep time' during the first visit, the value '6.07 (1.21)' represents a mean sleep time of 6.07 hours with a standard deviation (s.d.) of 1.21 hours. Similarly, '4.54 / 7.51' in the same row indicates that the 10th percentile (P10) for sleep time was 4.54 hours and the 90th percentile (P90) was 7.51 hours for the same group. AHI, Apnea-Hypopnea Index; BPM, beats per minute.

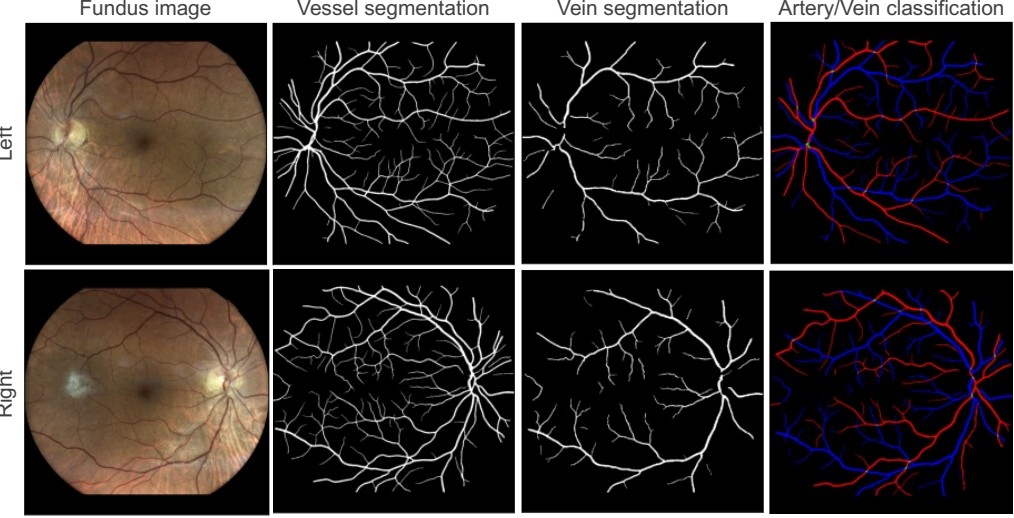

Figure 8: Example of fundus images in HPP. From left to right: (1) The original color fundus image; (2) Vessel segmentation; (3) Vein segmentation; and (4) Artery/Vein classification, distinguishing vessels into arteries (red) and veins (blue).

assessments of peripheral vascular health, multi-positional blood pressure readings, carotid ultrasound imaging, and electrocardiography (ECG). The combination of these methods yields a rich dataset that includes tabular data (e.g., blood pressure, PWV/ABI values, derived ECG features), medical imaging (e.g., carotid ultrasound), and raw time-series signals (e.g., ECG waveforms), making it an ideal modality for cross-modal analysis.

Vascular health is quantified through multiple techniques. Arterial stiffness, a key predictor of future cardiovascular events, is measured via Pulse Wave Velocity (PWV), specifically the leg PWV (faPWV). Peripheral Arterial Disease (PAD) is assessed using the Ankle-Brachial Index (ABI), which is the ratio of ankle to arm systolic blood pressure. Figure 9b shows the distribution of the right Ankle-Brachial Index (r_abi) across different sexes, where the values are concentrated around 1.1, consistent with the normal range for a healthy population. Both PWV and ABI are measured

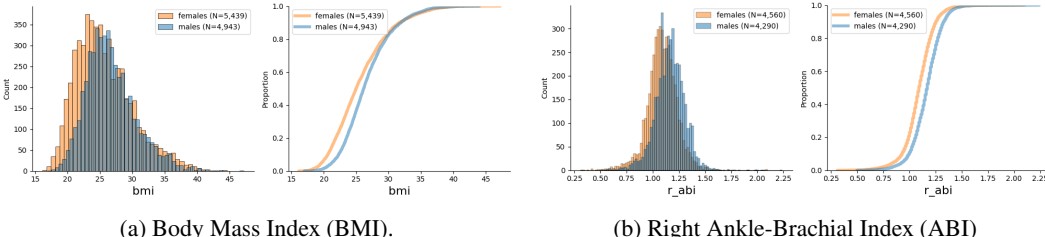

(a) Body Mass Index (BMI).     (b) Right Ankle-Brachial Index (ABI)

Figure 9: (a) Distribution of BMI for male (N=4,943) and female (N=5,439) at Baseline Visit. The left panel is a histogram, and the right panel is an ECDF. The data indicate that the BMI distribution for males is generally shifted towards higher values compared to females. (b) Distribution of Right ABI for male (N=4,290) and female (N=4,560) participants at Baseline Visit. The left panel is a histogram, and the right panel is an ECDF.

using a Falcon (Viasonix) device. Table 7 details the descriptive statistics for these key peripheral vascular health measurements. Furthermore, early signs of atherosclerosis are evaluated by measuring the Carotid Intima-Media Thickness (CIMT) from ultrasound images of the carotid arteries, performed with a Supersonic Aixplorer MACH 30 system. Figure 10a provides a distribution example of the left Carotid Intima-Media Thickness (`imt_left`), showing the specific numerical range of this feature within the cohort, with males showing a slight tendency towards higher values. Blood pressure is also systematically recorded with multiple readings taken while the participant is seated, lying down, and standing to capture a comprehensive hemodynamic profile. Table 8 summarizes the detailed statistical data of these key blood pressure measurements across different positions.

Table 7: Summary of key peripheral vascular health measurements

| Parameters | First visit (*n*=7,632) | | | Second visit (*n*=1,878) | | |
|---|---|---|---|---|---|---|
| | Min / Max | Mean (s.d.) | P10 / P90 | Min / Max | Meand (s.d.) | P10 / P90 |
| Right brachial pressure (mmHg) | 25 / 240 | 114.8 (19.4) | 92 / 139 | 61 / 220 | 114.5 (19.5) | 91 / 139 |
| Left brachial pressure (mmHg) | 24 / 291 | 115.6 (19.4) | 94 / 139 | 28 / 217 | 116 (18.7) | 94 / 139 |
| Right ABI (ratio) | 0.3 / 2.2 | 1.1 (0.1) | 1.0 / 1.3 | 0.45 / 2.09 | 1.14 (0.14) | 0.97 / 1.30 |
| Left ABI (ratio) | 0.46 / 2.19 | 1.13 (0.14) | 0.96 / 1.29 | 0.49 / 2.23 | 1.14 (0.14) | 0.98 / 1.30 |
| PWV from right thigh to right ankle (m/s) | 3.3 / 18.3 | 8.1 (1.8) | 5.9 / 10.5 | 3.2 / 18.5 | 7.4 (1.6) | 5.5 / 9.5 |
| PWV from left thigh to left ankle (m/s) | 3.3 / 17.2 | 8.1 (1.8) | 5.9 / 10.5 | 3.6 / 16.3 | 7.5 (1.6) | 5.6 / 9.5 |

Summary of key peripheral vascular health measurements for the first visit (n=7,632) and the second visit (n=1,878). Data are presented as the minimum and maximum values (Min / Max), the mean and standard deviation (Mean (s.d.)), and the 10th and 90th percentiles (P10 / P90). For example, for the 'Right brachial pressure' during the first visit, the value '114.8 (19.4)' represents a mean pressure of 114.8 mmHg with a standard deviation (s.d.) of 19.4 mmHg. Similarly, '92 / 139' in the same row indicates that the 10th percentile (P10) was 92 mmHg and the 90th percentile (P90) was 139 mmHg for the same group. ABI, Ankle-Brachial Index; PWV, Pulse Wave Velocity.

The electrical activity of the heart is captured using a resting 12-lead electrocardiogram (ECG), as shown in Figure 11. A significant feature of the HPP's ECG data is its dual format. For each participant, the dataset includes not only a set of automatically extracted tabular features (e.g., P-wave duration, QT interval) but also the complete raw 10-second waveform signal from all 12 leads, recorded at a high sampling rate of 1000Hz. For example, Figure 10b shows the distributional difference of one of the key derived features, P wave duration (`p_ms`), between male and female participants, highlighting sex-specific variations in cardiac electrical activity. This dual output provides an exceptionally detailed characterization of cardiac function, suitable for both traditional feature-based analysis and advanced signal processing techniques.

**Inflammation**   The inflammation modality in the HPP is characterized through a panel of hematological and biochemical measurements. The data for this modality primarily originates from blood tests conducted during participants' regular medical care at their respective Health Maintenance Organizations (HMOs), which are then uploaded by the participants themselves. This dataset is entirely tabular, providing a quantitative snapshot of systemic immune and inflammatory status.

The core of this modality consists of the complete blood count (CBC), which includes the total white blood cell count and a detailed differential of its major subtypes: the absolute counts and relative

Table 8: Summary of key blood pressure measurements

| Parameters | First visit (*n*=8,090) | | | Second visit (*n*=2,020) | | |
|---|---|---|---|---|---|---|
| | Min / Max | Mean (s.d.) | P10 / P90 | Min / Max | Meand (s.d.) | P10 / P90 |
| Lying diastolic blood pressure (mmHg) | 61 / 94 | 76.5 (7.6) | 67 / 87 | 61 / 94 | 76.7 (7.4) | 67 / 87 |
| Lying blood pressure pulse rate (BMP) | 35 / 110 | 62.4 (9.7) | 51 / 75 | 35 / 105 | 61.9 (9.6) | 50 / 74 |
| Lying systolic blood pressure (mmHg) | 74 / 199 | 119.2 (14.9) | 101 / 138 | 72 / 189 | 119.5 (15.0) | 101 / 138 |
| Sitting diastolic blood pressure (mmHg) | 47 / 140 | 78.4 (10.1) | 66 / 91 | 41 / 119 | 77.6 (9.5) | 66 / 90 |
| Sitting blood pressure pulse rate (BMP) | 35 / 113 | 66.6 (10.6) | 54 / 80 | 36 / 116 | 66.3 (10.4) | 53 / 80 |
| Sitting systolic blood pressure (mmHg) | 73 / 215 | 120.1 (16.6) | 100 / 142 | 62 / 203 | 119.8 (16.6) | 100 / 142 |
| Sitting second diastolic blood pressure (mmHg) | 47 / 117 | 77.7 (10.0) | 65 / 90 | 47 / 104 | 77.2 (9.4) | 66 / 89 |
| Sitting second blood pressure pulse rate (BMP) | 40 / 108 | 66.3 (10.6) | 54 / 80 | 37 / 110 | 65.2 (10.2) | 53 / 78 |
| Sitting second systolic blood pressure (mmHg) | 77 / 219 | 116.6 (15.8) | 98 / 137 | 76 / 171 | 116.5 (15.0) | 98.7 / 136 |
| Standing one minute diastolic blood pressure (mmHg) | 38 / 138 | 82.1 (9.6) | 70 / 94 | 55 / 127 | 82.2 (9.3) | 71 / 94 |
| Standing one minute blood pressure  pulse rate (BMP) | 36 / 122 | 73.6 (12.0) | 59 / 89 | 35 / 119 | 73.4 (12.0) | 58 / 88 |
| Standing one minute systolic blood pressure (mmHg) | 66 / 214 | 116.9 (16.0) | 98 / 137 | 79 / 197 | 116.8 (16.1) | 97 / 138 |
| Standing three minute diastolic blood pressure | 66 / 101 | 82.4 (7.9) | 72 / 93 | 66 / 101 | 82.5 (7.8) | 72 / 93 |
| Standing three minute blood pressure pulse rate (BMP) | 37 / 122 | 72.9 (11.4) | 59 / 88 | 35 / 127 | 72.8 (11.2) | 59 / 87 |
| Standing three minute systolic blood pressure (mmHg) | 69 / 214 | 117.7 (15.5) | 99 / 137 | 76 / 217 | 118.1 (15.8) | 99 / 139 |

Summary statistics of key blood pressure measurements for the first and second visits (n=8,090). Data are presented as the minimum and maximum values (Min / Max), the mean and standard deviation (Mean (s.d.)), and the 10th and 90th percentiles (P10 / P90). For example, for the 'Lying diastolic blood pressure' during the first visit, the value '76.5 (7.6)' represents a mean of 76.5 mmHg with a standard deviation (s.d.) of 7.6 mmHg. Similarly, '67 / 87' in the same row indicates that the 10th percentile (P10) value was 67 mmHg and the 90th percentile (P90) value was 87 mmHg for the same group.

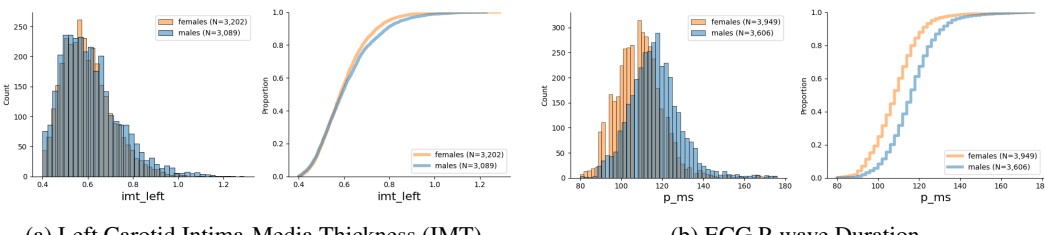

(a) Left Carotid Intima-Media Thickness (IMT).          (b) ECG P-wave Duration

Figure 10: (a) Distribution of Left Carotid IMT for male (N=3,089) and female (N=3,202) participants at Baseline Visit. The left panel is a histogram, and the right panel is an ECDF. (b)Distribution of ECG P-wave Duration (p_ms) for male (N=3,606) and female (N=3,949) participants at Baseline Visit. The left panel is a histogram, and the right panel is an ECDF, with the data showing that males generally have a longer P-wave duration than females.

percentages of neutrophils, lymphocytes, monocytes, eosinophils, and basophils. Furthermore, this modality is enriched with other key inflammatory markers, such as C-reactive protein (CRP) and ferritin, which are widely used clinical indicators for assessing systemic inflammation levels. This comprehensive panel of tests provides a solid foundation for studying baseline inflammatory states and their associations with other physiological systems.

**Metabolic Pathways**   The metabolic pathway data offers a functional perspective on the gut microbiome by characterizing its metabolic potential. This modality is derived from metagenomic sequencing of stool samples collected from the participants. Specifically, the data consists of microbial pathway abundances which are functionally profiled from the metagenomic data using the HUMAnN3 Beghini et al. (2021) software package. This approach allows for the quantification of the relative abundance of various metabolic pathways present in an individual's gut microbial community. The resulting dataset enables the investigation of associations between the collective metabolic functions of the gut microbiota and other physiological systems, providing insights into how microbial activities, such as amino acid biosynthesis or energy metabolism, may influence overall health and disease states.

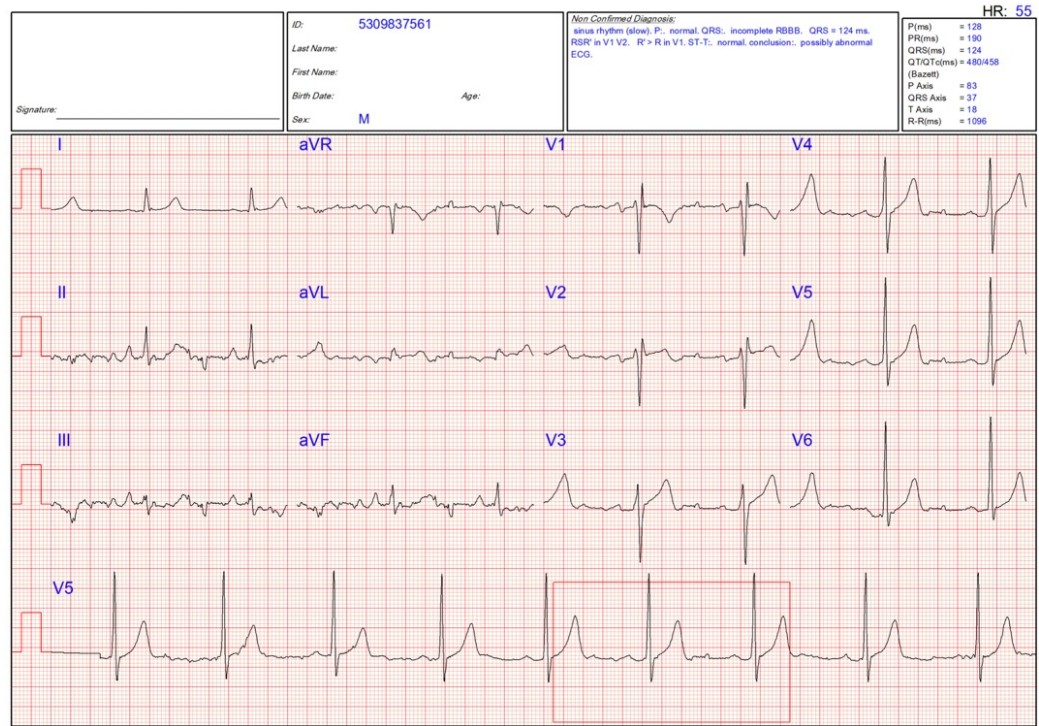

Figure 11: Processed ECG time series

Table 9: Summary of physiological systems, features and medical explanations

| System | Features | Medical Explanation |
|---|---|---|
| **Body Fat Composition** | body fat percentage | Quantifies total fat mass relative to body weight; elevated levels indicate obesity, metabolic dysregulation, and increased cardiovascular risk. |
| | AG fat ratio | Represents android-to-gynoid fat distribution; a higher ratio signals central obesity and greater risk for insulin resistance and vascular disease. |
| | Visceral fat volume | Measures fat accumulation around internal organs; strongly associated with inflammation, impaired glucose metabolism, and higher cardiometabolic risk. |
| | Appendicular lean mass | Reflects muscle mass in the limbs; reduced levels indicate sarcopenia, frailty, and diminished functional capacity. |
| **OSA-Related Phenotypes** | OSA severity | Indicates the degree of obstructive sleep apnea; higher values correlate with elevated cardiometabolic burden and daytime functional impairment. |
| | Snoring severity | Captures snoring intensity and frequency; a proxy marker for airway obstruction and OSA risk. |
| | Snoring ratio | Proportion of sleep time snoring; higher ratios reflect disrupted sleep and elevated OSA probability. |
| **Sleep Structure Quality** | Sleep duration | Objective total sleep time; both short and fragmented sleep are linked to metabolic dysfunction, cognitive decline, and cardiovascular morbidity. |
| | Sleep efficiency | Ratio of actual sleep to time in bed; reduced efficiency indicates impaired sleep maintenance and higher risk of fatigue and poor recovery. |
| | Sleep continuity | Measures the stability of sleep across the night; lower continuity reflects fragmented sleep, often associated with inflammation and stress. |
| | Restorative sleep ratio | Fraction of deep and REM stages relative to total sleep; lower ratios suggest reduced recovery and neurocognitive restoration. |
| **Sleep HRV** | Night mean HR | Average nocturnal heart rate; elevated values suggest reduced autonomic balance and higher stress or metabolic load. |
| | HRV index | Global measure of heart rate variability; lower scores are linked to autonomic dysfunction, systemic inflammation, and metabolic risk. |
| | HRV stage variation | Assesses variability of HRV across sleep stages; reduced variation indicates impaired autonomic adaptation and poorer sleep architecture. |
| | HRV wake delta | Difference in HRV between wake and sleep states; blunted differences may signal impaired recovery and autonomic dysregulation. |
| **Nocturnal Hypoxia Burden** | Oxygen desaturation | Degree of blood oxygen drop during sleep; frequent or severe drops indicate hypoxia-related cardiovascular and neurocognitive stress. |
| | Oxygen burden | Cumulative measure of oxygen deficit across the night; reflects the physiological load of intermittent hypoxia and links to systemic risk. |
| **Blood Pressure - Lying** | Lying diastolic pressure | Diastolic blood pressure while lying; helps assess baseline vascular tone and resting blood pressure. |
| | Lying pulse rate | Pulse rate measured while lying; serves as a marker of basal autonomic function. |
| | Lying systolic pressure | Systolic blood pressure measured while lying; provides baseline arterial load evaluation. |
| **Blood Pressure - Sitting** | Sitting diastolic pressure | Diastolic pressure in the sitting position; useful for detecting positional blood pressure changes. |
| | Sitting pulse rate | Pulse rate while sitting; reflects autonomic balance in a resting seated state. |
| | Sitting systolic pressure | Systolic pressure while sitting; indicates systemic pressure load in a neutral posture. |
| **Blood Pressure - Standing** | Standing 1min diastolic pressure | Diastolic pressure after standing for 1 minute; detects early orthostatic changes. |
| | Standing 1min pulse rate | Pulse rate after standing for 1 minute; useful for assessing autonomic responses to posture change. |
| | Standing 1min systolic pressure | Systolic pressure after standing for 1 minute; evaluates rapid hemodynamic adjustment. |
| | Standing 3min diastolic pressure | Diastolic pressure after 3 minutes of standing; detects delayed orthostatic hypotension. |
| | Standing 3min pulse rate | Pulse rate after 3 minutes standing; prolonged elevation may suggest autonomic dysfunction. |
| | Standing 3min systolic pressure | Systolic pressure after 3 minutes standing; indicates sustained hemodynamic adaptation. |
| **Blood Pressur - Resting** | Resting systolic pressure | Baseline systolic blood pressure; high values indicate elevated cardiovascular risk. |
| | Resting diastolic pressure | Baseline diastolic pressure; reflects resting vascular resistance. |
| | Resting pulse rate | Resting heart rate; higher rates may indicate poor cardiovascular fitness or increased stress. |
| | Orthostatic SBP drop 1min | Systolic pressure drop within 1 minute standing; shows autonomic or volume regulation impairment. |

# I  SYNTHETIC DATASET GENERATION

To facilitate a principled and reproducible evaluation of cross-modal causal inference methods, we developed a comprehensive data generation pipeline to create a synthetic dataset with a fully specified, ground-truth causal structure. Our procedure is grounded in a Structural Causal Model (SCM) that defines the latent causal relationships between variables. The numerical outputs of this model are then mapped to a set of semantic and style parameters, which in turn deterministically guide a sequence of transformations applied to base images from the MNIST Lecun et al. (1998) dataset. This multi-step process allows for precise control over the underlying causal graph, the manifestation of variables across tabular and visual modalities, and the introduction of realistic complexities such as nonlinearities and nuisance variables. The technical details are outlined in Algorithm 1.

## I.1  CAUSAL GROUNDING VIA STRUCTURAL CAUSAL MODEL

To enable rigorous evaluation of cross-modal causal inference methods, we construct a synthetic dataset grounded in a fully specified SCM. This approach ensures that all causal relationships are known by design, providing clear ground truth for method validation. Our SCM implements a three-variable mediation chain with the structure $T \to M \to Y^*$, where $T$ represents the treatment variable, $M$ the mediator, and $Y^*$ the continuous outcome. The data generating process is defined through the following system of structural equations:

$$
\begin{aligned}
T &= \epsilon_T \\
M &= \alpha_1 T + \rho h_1(T) + \epsilon_M \\
Y^* &= \alpha_2 M + \rho h_2(M) + \delta T + \epsilon_Y
\end{aligned}
\tag{24}
$$

where $\epsilon_T, \epsilon_M, \epsilon_Y \sim \mathcal{N}(0, \sigma^2)$ are independent Gaussian noise terms with configurable variances $\sigma_T^2, \sigma_M^2, \sigma_Y^2$. The treatment $T$ is exogenous, while both the mediator $M$ and outcome $Y^*$ incorporate linear and nonlinear components to model realistic causal complexity.

The linear effects are governed by coefficients $\alpha_1$ and $\alpha_2$, representing the primary causal pathways $T \to M$ and $M \to Y^*$ respectively. Nonlinear relationships are introduced through functions $h_1(\cdot)$ and $h_2(\cdot) \in \{\texttt{square}, \texttt{sin}, \texttt{tanh}\}$, scaled by parameter $\rho \in [0, 1]$. Crucially, the direct effect from treatment to outcome is controlled by parameter $\delta$. When $\delta = 0$, all causal influence from $T$ to $Y^*$ flows through the mediator $M$, ensuring that the conditional independence relationship $T \perp\!\!\!\perp Y^* \mid M$ holds by construction. This parameter allows systematic evaluation of both full mediation scenarios ($\delta = 0$) and partial mediation with confounding ($\delta > 0$).

## I.2  NUMERICAL INSTANTIATION AND AUXILIARY VARIABLES

Following the SCM specification, we numerically instantiate the model by sampling $N = 5000$ independent observations. This process yields a concrete realization of the core causal variables $(T, M, Y^*)$ for each sample. To enhance the dataset's complexity and support the subsequent cross-modal synthesis, these core variables are augmented with two distinct types of auxiliary variables. First, we introduce a set of $q = 30$ high-dimensional nuisance variables, denoted by the vector $\mathbf{W} \in \mathbb{R}^{N \times q}$ for each sample $i$. These are generated by drawing each component independently from a standard normal distribution, such that $\mathbf{W}_i \sim \mathcal{N}(0, \mathbf{I}_q)$. By design, these variables are causally independent of the primary $T \to M \to Y^*$ chain, serving as realistic distractors that a robust inference method must learn to ignore. Second, we sample a scalar style variable $S_{\text{style}} \sim \mathcal{N}(0, 1)$, which is causally orthogonal to the main SCM structure. This is specifically designated to control stylistic variations in the image generation phase, thereby ensuring a clean separation between the semantic content driven by the causal variables, and the visual style driven by $S_{\text{style}}$.

## I.3  SEMANTIC PARAMETER MAPPING

To bridge the numerical latent space of the SCM with the perceptual domain of the images, we conducted the semantic parameter mapping. This stage transforms the raw, unbounded variables generated by the SCM into normalized and interpretable parameters that directly govern the visual attributes during image synthesis. This transformation is achieved using the standard normal Cumulative Distribution Function (CDF), denoted as $\Phi(\cdot)$, which maps input to the interval $[0, 1]$.

For the core causal variables, the mediator vector $\mathbf{M}$ and outcome vector $\mathbf{Y}^*$ are converted into semantic amplitude vectors, $\mathbf{a}_M$ and $\mathbf{a}_Y$, respectively. This is accomplished by first standardizing the variables using their empirical mean ($\mu_M, \mu_{Y^*}, \mu_S$) and standard deviation ($\sigma_M, \sigma_Y, \sigma_S$) calculated across all $N$ samples, and then applying the CDF. The specific mappings are defined as:

$$\mathbf{a}_M = \Phi\left(\frac{\mathbf{M} - \mu_M}{\sigma_M}\right), \quad \mathbf{a}_Y = \Phi\left(\frac{\mathbf{Y}^* - \mu_{Y^*}}{\sigma_{Y^*}}\right) \tag{25}$$

An analogous procedure is used to derive the style parameter vector $\mathbf{b}_{\text{style}}$ from the independent style variable vector $\mathbf{S}_{\text{style}}$:

$$\mathbf{b}_{\text{style}} = \Phi\left(\frac{\mathbf{S}_{\text{style}} - \mu_S}{\sigma_S}\right) \tag{26}$$

This normalization ensures the resulting parameters provide stable and consistent control over the subsequent image transformations. Critically, this design enforces a clean separation of generative factors: the semantic parameters $\mathbf{a}_M$ and $\mathbf{a}_Y$ are exclusively derived from their corresponding causal variables, while the style parameter $\mathbf{b}_{\text{style}}$ originates from the causally orthogonal variable $\mathbf{S}_{\text{style}}$.

## I.4 CROSS-MODAL IMAGE GENERATION

With the semantic and style parameters established, we synthesize the cross-modal data by applying a sequence of deterministic transformations to a set of base images. The base images, denoted as a collection of matrices $\{\mathbf{I}_{\text{base},i} \in [0,1]^{28 \times 28}\}_{i=1}^N$, are sourced from the MNIST dataset. To prevent the introduction of spurious correlations, these images are sampled using a class-uniform distribution, ensuring a balanced representation across all digit classes, $c_{\text{class}} \in \{0, \ldots, 9\}^N$. This sampling is performed independently of the SCM variables, thereby guaranteeing statistical independence between the semantic amplitudes and the class labels by construction ($a_M \perp c_{\text{class}}$ and $a_Y \perp c_{\text{class}}$).

For each of the $N$ samples, a fixed sequence of spatial transformations is applied. The transformations are parameterized by the corresponding semantic amplitude $a_i \in \{a_{M,i}, a_{Y,i}\}$ and the style parameter $b_{\text{style},i}$. The steps are as follows:

**Rotation** An angular transformation is applied to orient the base image. The rotation angle $\theta$ is linearly determined by the semantic amplitude $a_i$ according to the formula:

$$\theta = \theta_{\text{deg}} \cdot (2a_i - 1), \tag{27}$$

This function maps the full range of semantic amplitudes $[0,1]$ to a continuous spectrum of rotation angles spanning from $[-\theta_{\text{deg}}, +\theta_{\text{deg}}]$.

**Brightness Adjustment** The overall brightness of the image is modified through an additive shift $\Delta$. This shift is calculated as:

$$\Delta = \beta \cdot (a_i - 0.5), \tag{28}$$

where the hyperparameter $\beta$ controls the strength of the effect. This ensures that the image becomes progressively brighter as the semantic amplitude $a_i$ increases from 0 to 1.

**Contrast Enhancement** The contrast of image is adjusted by scaling pixel values around the mean brightness $\mu_i$. The transformation is controlled by a multiplicative gain factor:

$$g = 1 + \gamma(2a_i - 1), \tag{29}$$

where $\gamma$ is a hyperparameter for the effect strength. A value of $a_i > 0.5$ results in increased contrast, while $a_i < 0.5$ reduces it.

**Style-Dependent Noise** To introduce stylistic variation that is independent of the causal semantics, we add Gaussian noise to the image. The standard deviation of noise $\sigma$ is controlled by the style parameter $b_{\text{style},i}$ via the relation $\sigma = \sigma_{\text{pix}} \cdot b_{\text{style},i}$. where $\sigma_{\text{pix}}$ is the configurable base noise level.

Each step in this transformation pipeline is followed by a clipping operation to maintain all pixel values within the valid range of $[0,1]$. This complete sequence ensures that the final visual features of the generated images systematically and monotonically vary with their corresponding semantic or style parameters.

## I.5 DUAL-SCENARIO FRAMEWORK

Our data generation framework is designed to produce two distinct yet causally equivalent cross-modal scenarios from a single underlying SCM. This unique design allows for a comprehensive and controlled evaluation of causal inference methods by systematically altering which variable in the causal chain $T \rightarrow M \rightarrow Y^*$ is manifested in the visual modality. This approach tests a model's ability to reason flexibly across different modal configurations while the ground-truth causal graph remains fixed. The two scenarios are detailed below:

$I^M$ **Scenario (Image as Mediator).** In this scenario, the mediator is represented by image data, resulting in a `Tabular→Image→Tabular` causal chain. The mediator images, denoted as $I^M$, are synthesized using the semantic amplitudes $\mathbf{a}_M$, which are themselves a direct function of the latent tabular mediator variable $M$. This configuration is designed to test a model's ability to correctly identify a mediated causal pathway ($T \rightarrow M \rightarrow Y^*$) where the intermediate step is only observable through high-dimensional visual data. To succeed, a model must effectively learn to extract the causally salient information encoded in the visual features of $I^M$ and use it to reason about the relationship between the tabular treatment $T$ and outcome $Y^*$. This setup mimics real-world scientific and industrial problems, such as medical diagnostics where a treatment's effect on a patient's outcome is mediated by changes observable only through medical imaging.

$I^Y$ **Scenario (Image as Outcome).** This scenario creates a `Tabular→Tabular→Image` causal chain. The outcome images $I^Y$ are generated using the semantic amplitudes $\mathbf{a}_Y$, which are derived from the latent tabular outcome variable $Y^*$. This configuration evaluates a model's capacity to discover causal relationships where a low-dimensional tabular mediator ($M$) directly influences a high-dimensional, complex visual outcome ($I^Y$). This scenario presents a different but equally important challenge, requiring the model to understand how changes in an abstract tabular variable manifest as structured visual changes. It reflects problems such as predicting the visual outcomes of a manufacturing process based on machine settings or determining the effect of drug treatments on cellular morphology.

## I.6 GROUND TRUTH SPECIFICATION AND ROBUSTNESS TESTING

To facilitate a systematic and rigorous evaluation of inference methods, we explicitly define a set of testable ground truth hypotheses, denoted as $\mathcal{H}$, for each scenario. These hypotheses serve as the benchmark against which model performance is measured. For the $I^M$ scenario, the corresponding set of hypotheses, $\mathcal{H}^M$, requires a model to correctly identify: (i) the causal dependence between treatment and mediator ($T \not\perp M$); (ii) the dependence between the mediator and the final outcome ($M \not\perp Y^*$); and (iii) the crucial conditional independence of the treatment and outcome given the mediator ($T \perp\!\!\!\perp Y^* \mid M$) when the direct causal path is absent ($\delta = 0$).

Beyond the core causal chain, we specify two critical sanity checks. The first is **semantic-class independence** ($a_M \perp c_{\text{class}}$), which verifies that no spurious correlations exist between the semantic amplitude and the underlying MNIST digit class, a condition guaranteed by our independent sampling procedure. The second is a test of **Monotonic Amplitude-Correspondence (MAC)**:

$$\text{MAC}(a_M, \phi(I^M)) > \tau. \tag{30}$$

This metric quantifies the Spearman correlation between the vector of semantic amplitudes and the feature distances of their corresponding images, where $\phi(\cdot)$ is a predefined feature extraction function. A high MAC score confirms that the semantic information is faithfully encoded in the visual modality, a fundamental prerequisite for cross-modal reasoning. An analogous set of hypotheses, $\mathcal{H}^Y$, is defined for the $I^Y$ scenario with the appropriate variable substitutions (e.g., testing $M \not\perp \phi(I^Y)$).

To assess model robustness against corrupted data, we introduce an optional **permutation** step. By randomly shuffling a fraction of the generated images, controlled by permutation ratios $\rho_M$ and $\rho_Y$, we deliberately break the semantic-visual correspondence for a subset of the data. This serves as a critical negative control, enabling the evaluation of both a model's sensitivity (ability to discover true causal links in unpermuted data) and its specificity (ability to avoid discovering spurious relationships in permuted data).

The final generated dataset $\mathcal{D} = \{\mathcal{D}_{\text{tab}}, \mathcal{D}_{\text{img}}, \mathcal{S}, \{\mathcal{H}^M, \mathcal{H}^Y\}\}$, is a comprehensive package. It includes the complete tabular data $\mathcal{D}_{\text{tab}}$, the cross-modal images $\mathcal{D}_{\text{img}}$, stratified train/validation/test splits $\mathcal{S}$, and the full ground truth specification $\mathcal{H}$. This principled construction provides a robust and transparent benchmark for the evaluation of cross-modal causal inference methods.

---

**Algorithm 1** Cross-Modal Causal Synthetic Dataset Generation

---

**Require:** Sample size $N$, SCM parameters $\{\alpha_1, \alpha_2, \rho, \delta\}$, noise variances $\{\sigma_T^2, \sigma_M^2, \sigma_Y^2\}$, nonlinear functions $\{h_1, h_2\}$, imaging parameters $\{\theta_{\text{deg}}, \beta, \gamma, \sigma_{\text{pix}}\}$, cross-modal flags $\{\text{use}_{IM}, \text{use}_{IY}\}$, permutation ratios $\{\rho_M, \rho_Y\}$, nuisance dimension $q$

**Ensure:** Synthetic dataset $\mathcal{D}$ with tabular data, images, ground-truth causal relationships, and evaluation splits

    **// Step 1: Structural Causal Model Sampling**
1: **for** $i = 1$ to $N$ **do**
2:     $T_i \leftarrow \epsilon_{T,i}; \epsilon_{T,i} \sim \mathcal{N}(0, \sigma_T^2)$                ▷ $T$ is exogenous treatment variable
3:     $M_i \leftarrow \alpha_1 T_i + \rho h_1(T_i) + \epsilon_{M,i}; \epsilon_{M,i} \sim \mathcal{N}(0, \sigma_M^2)$   ▷ Mediator with linear + nonlinear effects
4:     $Y_i^* \leftarrow \alpha_2 M_i + \rho h_2(M_i) + \delta T_i + \epsilon_{Y,i}; \epsilon_{Y,i} \sim \mathcal{N}(0, \sigma_Y^2)$   ▷ Outcome with mediation + direct effect
5:     $\mathbf{W}_i \sim \mathcal{N}(\mathbf{0}, \mathbf{I}_q)$                     ▷ $q$-dimensional nuisance variables
6:     $S_{\text{style},i} \sim \mathcal{N}(0, 1)$                  ▷ Independent style variable for imaging
7: **end for**
    **// Step 2: Semantic Parameter Mapping**
8: Let $\Phi$ denote the standard normal cumulative distribution function
9: $\mathbf{a}_M \leftarrow \Phi((\mathbf{M} - \mu_M)/\sigma_M); \mu_M \leftarrow \text{mean}(\mathbf{M}), \sigma_M \leftarrow \text{std}(\mathbf{M})$   ▷ Semantic amplitudes for mediator
10: $\mathbf{a}_Y \leftarrow \Phi((\mathbf{Y}^* - \mu_{Y^*})/\sigma_{Y^*}); \mu_{Y^*} \leftarrow \text{mean}(\mathbf{Y}^*), \sigma_{Y^*} \leftarrow \text{std}(\mathbf{Y}^*)$ ▷ Semantic amplitudes for outcome
11: $\mathbf{b}_{\text{style}} \leftarrow \Phi((\mathbf{S}_{\text{style}} - \mu_S)/\sigma_S); \mu_S \leftarrow \text{mean}(\mathbf{S}_{\text{style}}), \sigma_S \leftarrow \text{std}(\mathbf{S}_{\text{style}})$   ▷ Style parameters in $[0,1]$
    **// Step 3: Base Image Sampling with Class Independence**
12: $\{\mathbf{I}_{\text{base}}, \mathbf{c}_{\text{class}}, \mathbf{id}_{\text{base}}\} \leftarrow$ Sample N MNIST images with class-uniform distribution
    **// Step 4: Cross-Modal Generation**
13: **for** scenario $\in \{I^M, I^Y\}$ **do**                   ▷ Image as mediator or outcome
14:     Select semantic amplitudes: $\mathbf{a} \leftarrow \mathbf{a}_M$ if $I^M$, else $\mathbf{a}_Y$
15:     **for** $i = 1$ to $N$ **do**
16:         $\mathbf{I}'_i \leftarrow \text{Rotate}(\mathbf{I}_{\text{base},i}, \theta_{\text{deg}} \cdot (2a_i - 1))$            ▷ Rotation
17:         $\mathbf{I}''_i \leftarrow \text{clip}_{[0,1]}(\mathbf{I}'_i + \beta \cdot (a_i - 0.5))$           ▷ Brightness
18:         $\mathbf{I}'''_i \leftarrow \text{clip}_{[0,1]}(\mu_i + (1 + \gamma(2a_i - 1))(\mathbf{I}''_i - \mu_i))$       ▷ Contrast
19:         $\mathbf{I}_i^{\text{scenario}} \leftarrow \text{clip}_{[0,1]}(\mathbf{I}'''_i + \mathcal{N}(0, (\sigma_{\text{pix}} b_{\text{style},i})^2))$       ▷ Noise
20:     **end for**
21: **end for**
        Optional: Apply random permutation to fraction $\{\rho_M, \rho_Y\}$ of images
    **// Step 5: Data Assembly and Stratified Split Generation**
22: $\mathcal{D}_{\text{tab}} \leftarrow \{\mathbf{T}, \mathbf{M}, \mathbf{Y}^*, \mathbf{W}, \mathbf{a}_M, \mathbf{a}_Y, \mathbf{b}_{\text{style}}, \mathbf{c}_{\text{class}}\}$
23: $\mathcal{D}_{\text{img}} \leftarrow \{\mathbf{I}^M, \mathbf{I}^Y\}$
24: Create stratified train/validation/test splits using $\mathbf{c}_{\text{class}}$: $\mathcal{S} = \{\mathcal{I}_{\text{train}}, \mathcal{I}_{\text{val}}, \mathcal{I}_{\text{test}}\}$
    **// Step 6: Ground Truth Specification**
25: **For** $I^M$ scenario (image as mediator):
26:     $\mathcal{H}_1^M: T \not\perp M$                    ▷ Treatment causally affects mediator
27:     $\mathcal{H}_2^M: M \not\perp Y^*$                  ▷ Mediator causally affects outcome
28:     $\mathcal{H}_3^M: T \perp Y^* \mid M \Leftrightarrow \delta = 0$          ▷ Conditional independence test
29:     $\mathcal{H}_4^M: a_M \perp c_{\text{class}}$               ▷ Semantic-class independence
30:     $\mathcal{H}_5^M: \text{MAC}(a_M, \phi(\mathbf{I}^M)) > \tau$        ▷ Monotonic correspondence
31: **For** $I^Y$ scenario (image as outcome):
32:     $\mathcal{H}_1^Y: T \not\perp M$                    ▷ Treatment causally affects mediator
33:     $\mathcal{H}_2^Y: M \not\perp \phi(\mathbf{I}^Y)$            ▷ Mediator causally affects image features
34:     $\mathcal{H}_3^Y: T \perp \phi(\mathbf{I}^Y) \mid M \Leftrightarrow \delta = 0$      ▷ Conditional independence test
35:     $\mathcal{H}_4^Y: a_Y \perp c_{\text{class}}$              ▷ Semantic-class independence
36:     $\mathcal{H}_5^Y: \text{MAC}(a_Y, \phi(\mathbf{I}^Y)) > \tau$        ▷ Monotonic correspondence
37: **return** $\mathcal{D} = \{\mathcal{D}_{\text{tab}}, \mathcal{D}_{\text{img}}, \mathcal{S}, \{\mathcal{H}^M, \mathcal{H}^Y\}\}$

---

## I.7 SYNTHETIC DATA SAMPLE

To provide an intuitive understanding of the generated data's properties, we present a series of visualizations. Figure 12 illustrates the statistical relationships within the Image as Mediator ($I^M$) scenario, which corresponds to the Tabular→Image→Tabular causal chain. The plots show

the relationships between the latent treatment variable ($T$), the final outcome ($Y^*$), and four distinct visual features extracted from the synthesized mediator images ($I^M$). These visualizations serve as an empirical confirmation of the dependencies and conditional independencies specified by our ground-truth SCM.

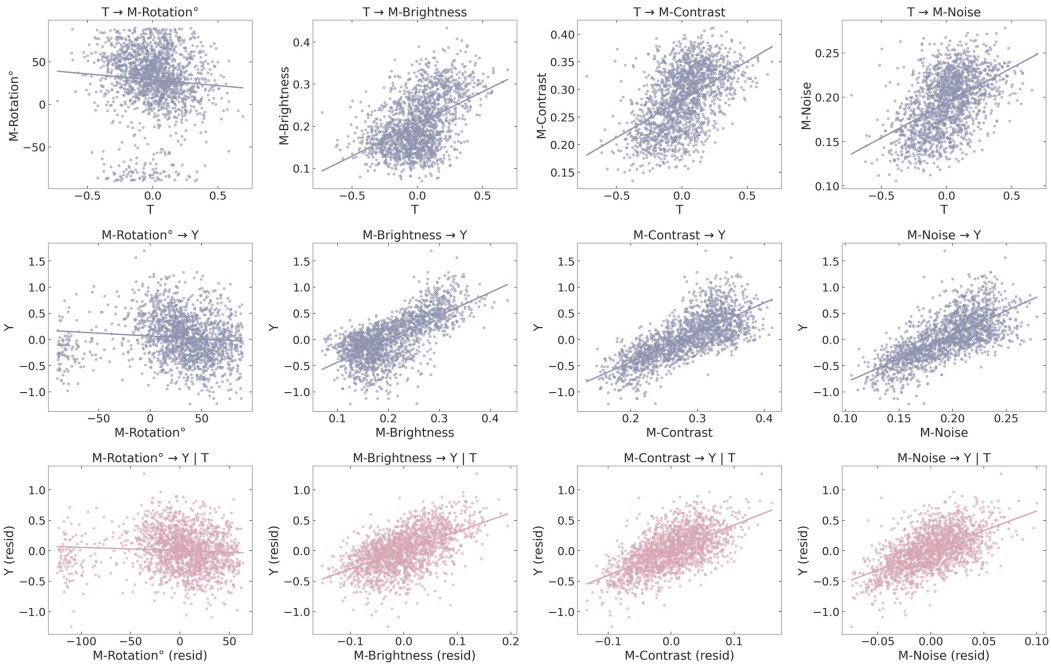

Figure 12: Visualization of the causal relationships in the Image as Mediator ($I^M$) scenario. Top: The causal link between the treatment $T$ and the visual features of the mediator image. Middle: The marginal correlation between the mediator's visual features and the outcome $Y^*$. Bottom: The conditional relationship between the mediator's features and the outcome, controlling for $T$. "Resid" denotes the residuals of each variable after being regressed on $T$, confirming the $M \rightarrow Y^*$ link.

Figure 13 provides an empirical analysis of the Image as Outcome (IY scenario), which instantiates a Tabular→Tabular→Image causal chain. The visualizations confirm that the relationships defined in our SCM are successfully propagated through to the final generated images. The scatter plots illustrate the foundational causal links, starting with the correlation between the tabular treatment $T$ and mediator $M$. They then show how $M$ directly influences various visual features (e.g., brightness, contrast) extracted from the outcome images, $I^Y$. The residual plots in the middle rows confirm that this influence from $M$ to the visual features persists even after controlling for $T$. The image panels at the bottom provide compelling qualitative evidence by showing a clear, monotonic progression in visual characteristics as the value of the mediator $M$ increases across its quantiles, visually demonstrating that the causal information is successfully encoded in the image domain.

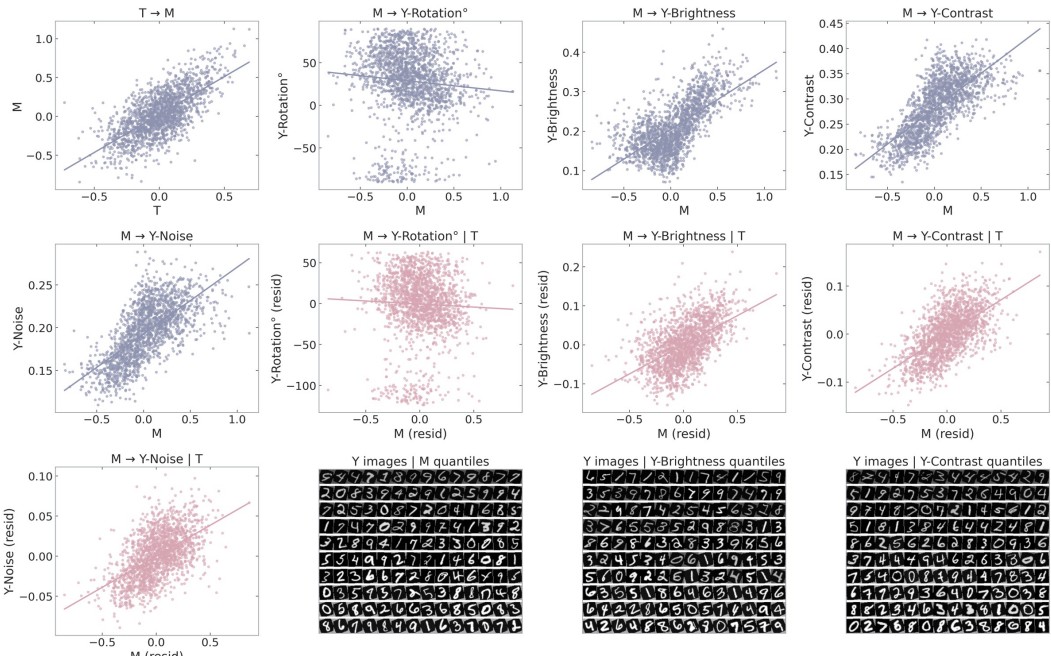

Figure 13: Visualization of the causal relationships in the Image as Outcome ($I^Y$) scenario. Top & Middle Rows: Scatter plots showing the tabular causal link $T \rightarrow M$, the marginal relationships between the mediator $M$ and visual features extracted from the outcome images $I^Y$, and the conditional relationships ($\perp T$) between $M$ and the image features after controlling for $T$. Bottom Row: Example images from the dataset, sorted into columns by quantiles of the mediator $M$ (left) and by quantiles of their extracted brightness and contrast. The visible progression in rotation, brightness, and contrast across the $M$ quantiles provides a qualitative confirmation of the $M \rightarrow I^Y$ causal link.

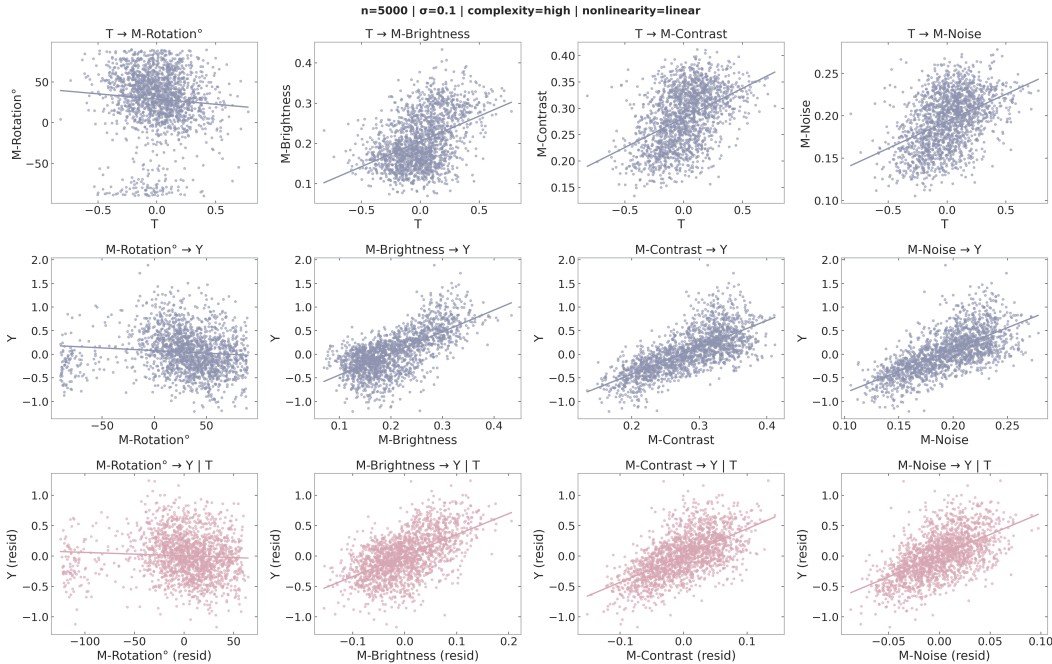

Figure 14: Synthetic data sample in setting 1 ($n = 5000$, $\sigma = 0.1$ and high linear), $T \rightarrow M \rightarrow Y^*$.

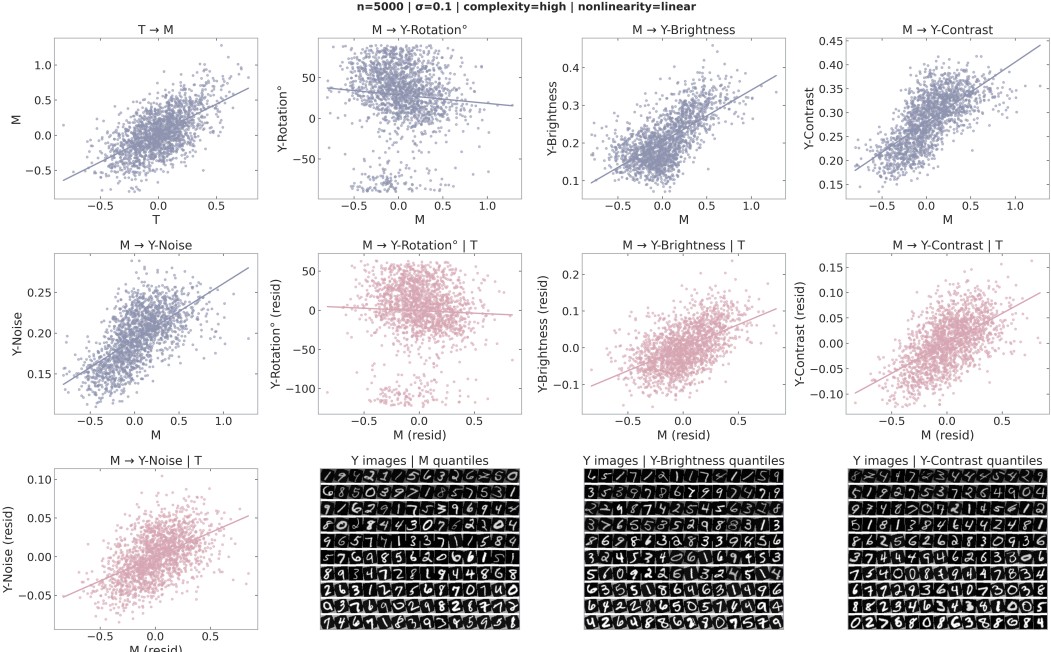

Figure 15: Synthetic data sample in setting 2 ($n = 5000$, $\sigma = 0.1$ and high linear), $T \rightarrow M^* \rightarrow Y$.

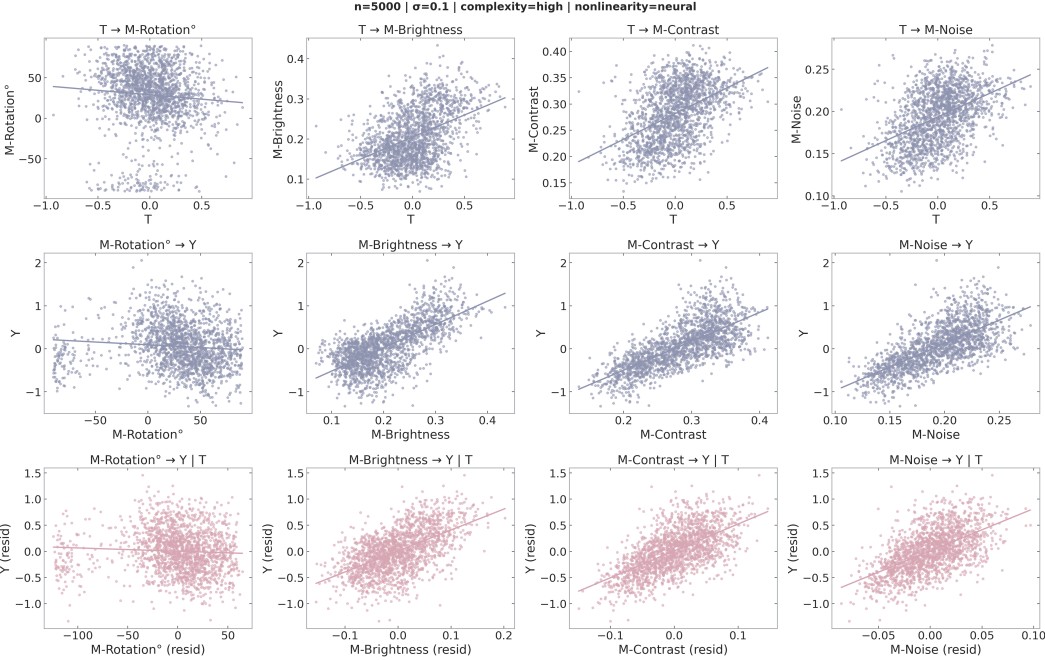

Figure 16: Synthetic data sample in setting 3 ($n = 5000$, $\sigma = 0.1$ and high neural), $T \rightarrow M \rightarrow Y^*$.

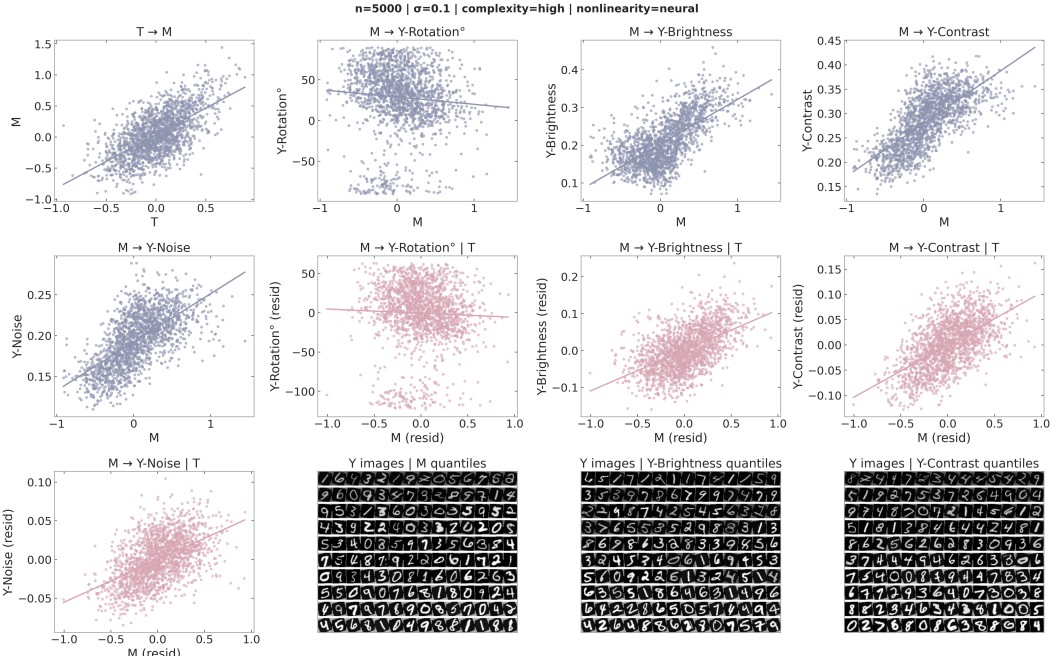

Figure 17: Synthetic data sample in setting 4 ($n = 5000$, $\sigma = 0.1$ and high neural), $T \to M^* \to Y$.

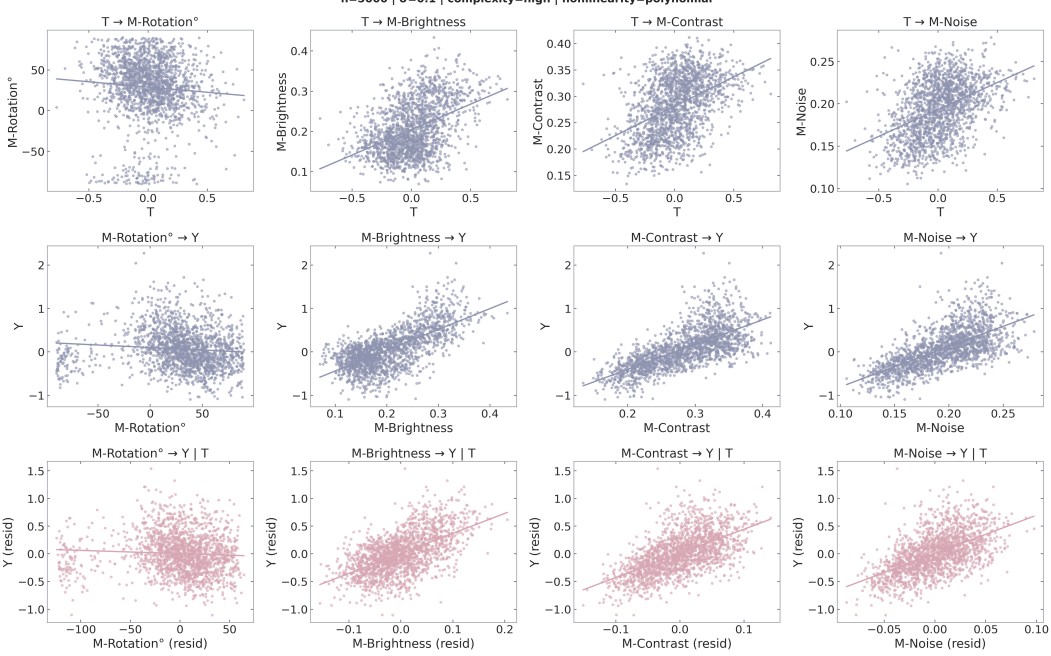

Figure 18: Synthetic data sample in setting 5 ($n = 5000$, $\sigma = 0.1$ and high polynomial), $T \to M \to Y^*$.

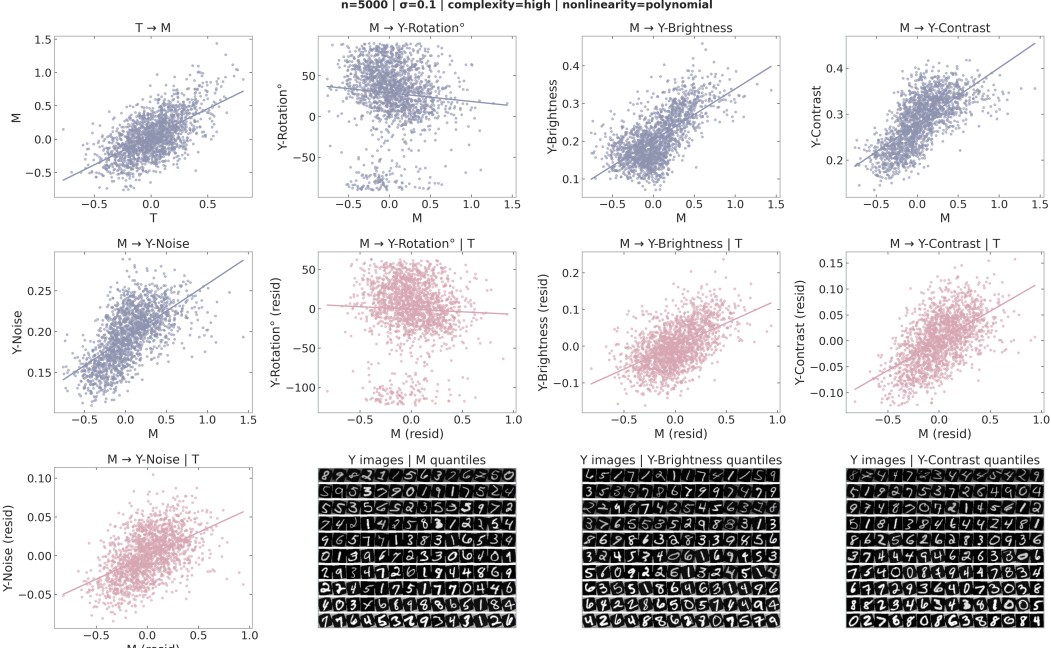

Figure 19: Synthetic data sample in setting 6 ($n = 5000$, $\sigma = 0.1$ and high polynomial), $T \rightarrow M^* \rightarrow Y$.

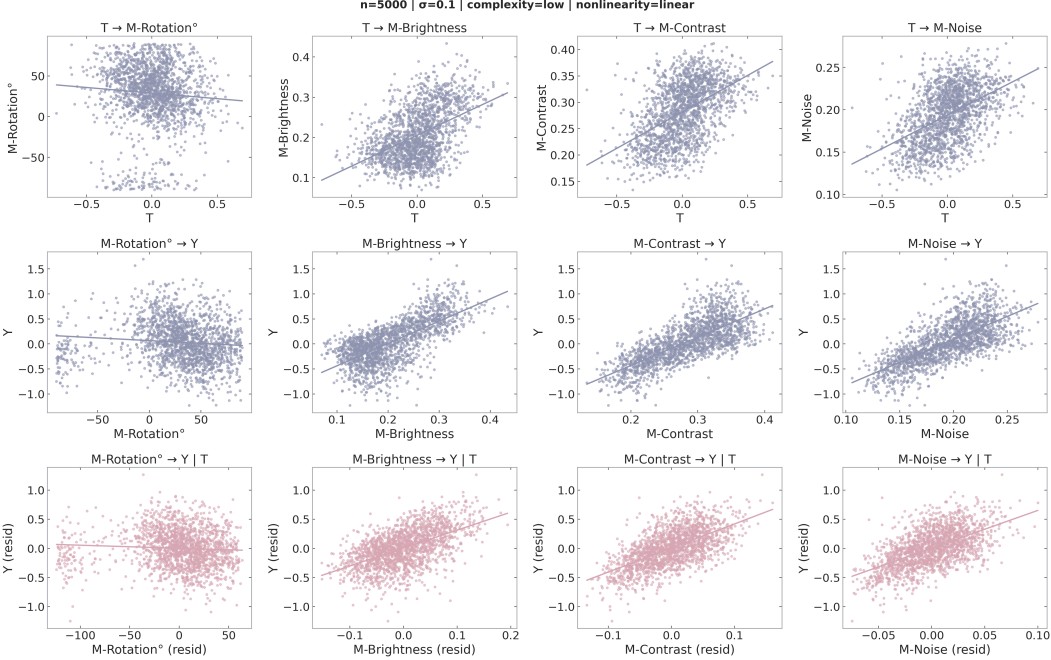

Figure 20: Synthetic data sample in setting 7 ($n = 5000$, $\sigma = 0.1$ and low linear), $T \rightarrow M \rightarrow Y^*$.

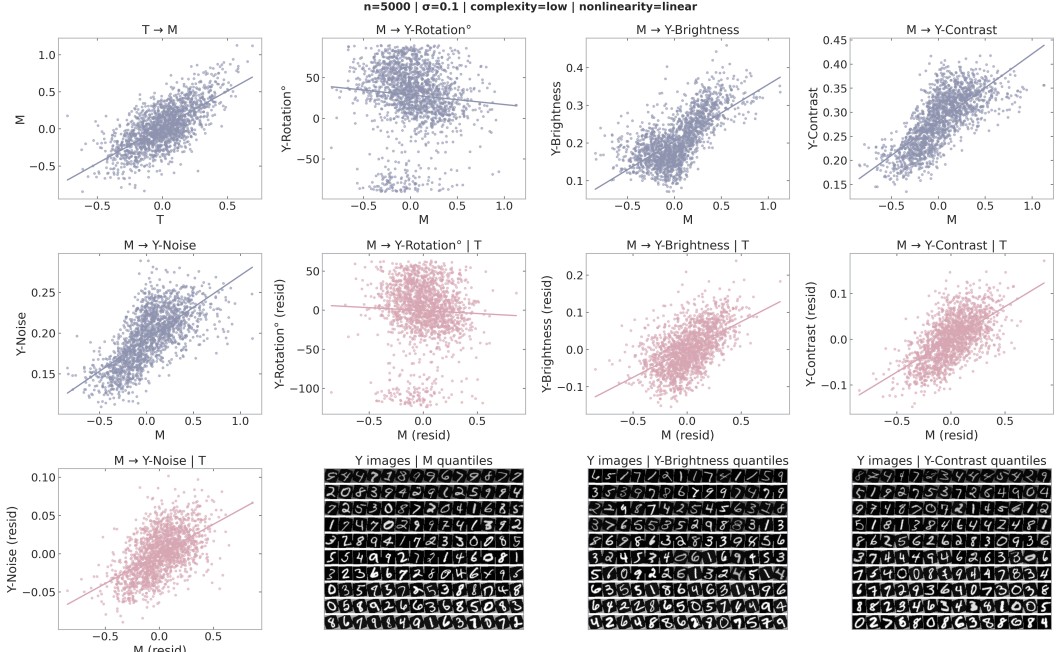

Figure 21: Synthetic data sample in setting 8 ($n = 5000$, $\sigma = 0.1$ and low linear), $T \to M^* \to Y$.

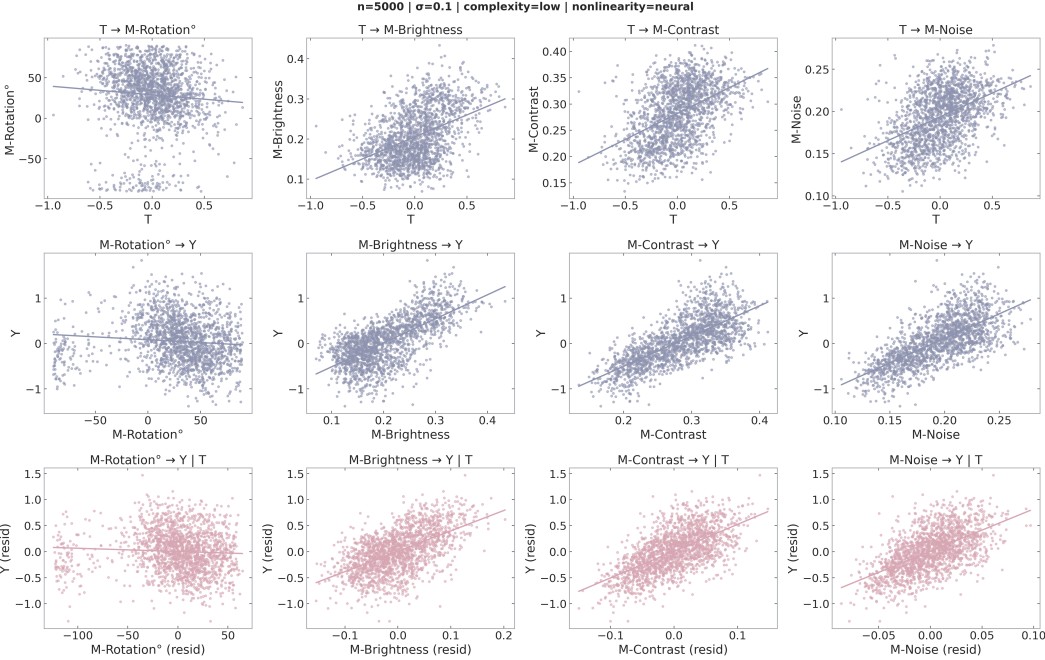

Figure 22: Synthetic data sample in setting 9 ($n = 5000$, $\sigma = 0.1$ and low neural), $T \to M \to Y^*$.

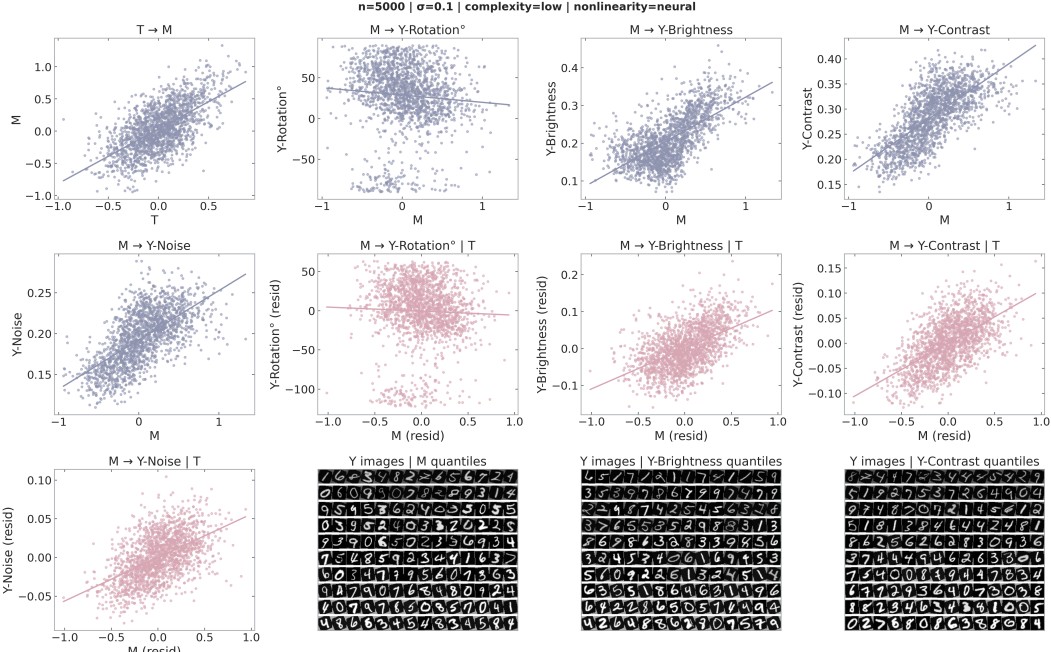

Figure 23: Synthetic data sample in setting 10 ($n = 5000$, $\sigma = 0.1$ and low neural), $T \rightarrow M^* \rightarrow Y$.

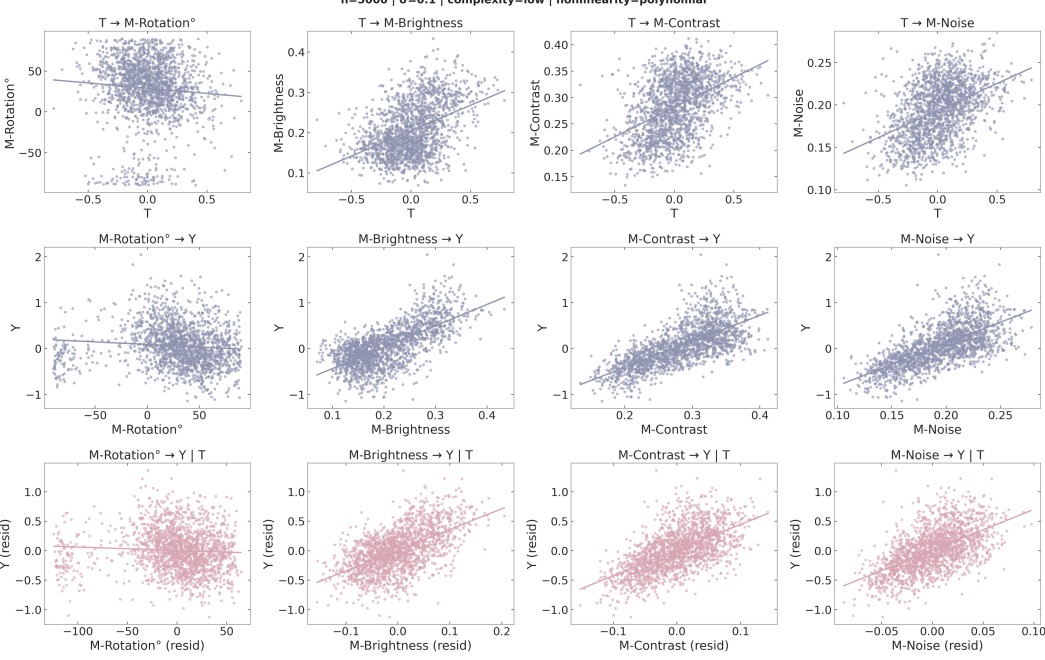

Figure 24: Synthetic data sample in setting 11 ($n = 5000$, $\sigma = 0.1$ and low polynomial), $T \rightarrow M \rightarrow Y^*$.

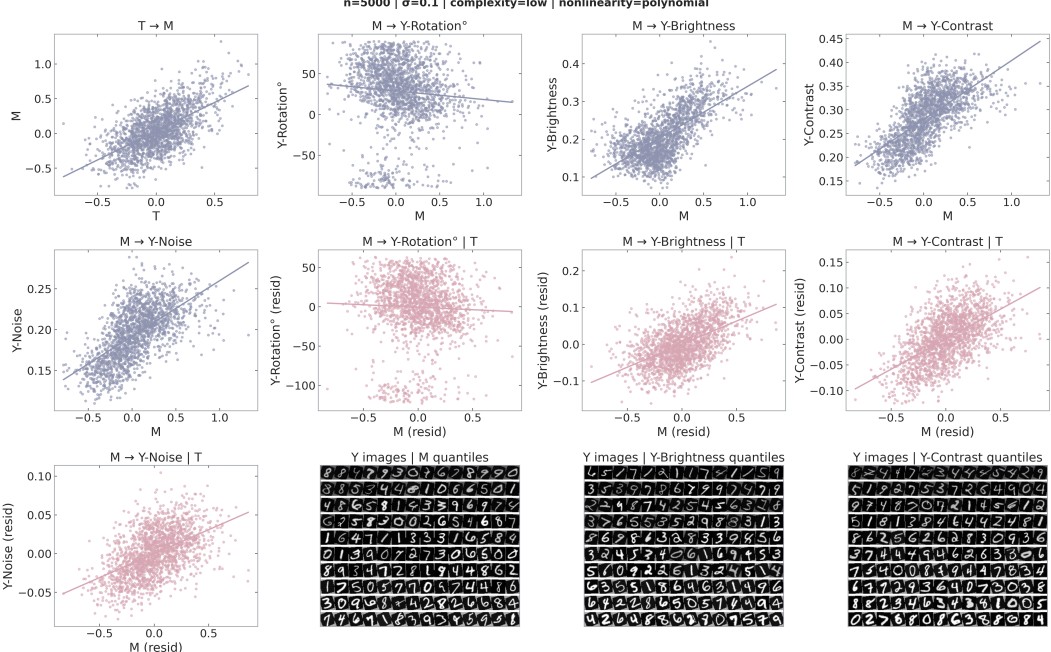

Figure 25: Synthetic data sample in setting 12 ($n = 5000$, $\sigma = 0.1$ and low polynomial), $T \rightarrow M^* \rightarrow Y$.

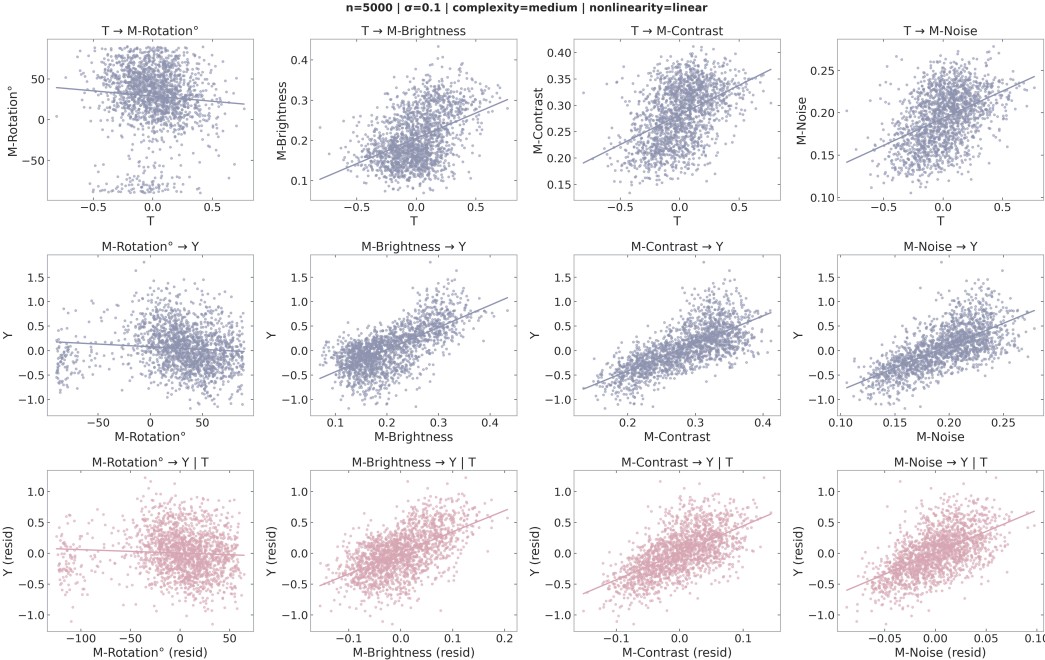

Figure 26: Synthetic data sample in setting 13 ($n = 5000$, $\sigma = 0.1$ and medium linear), $T \rightarrow M \rightarrow Y^*$.

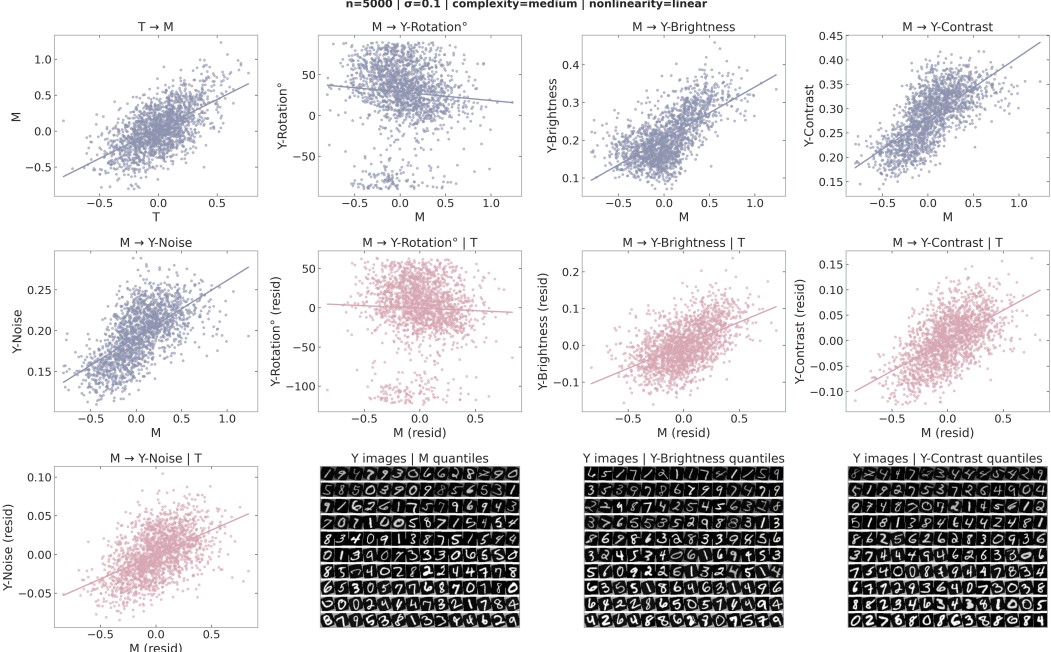

Figure 27: Synthetic data sample in setting 14 ($n = 5000$, $\sigma = 0.1$ and medium linear), $T \rightarrow M^* \rightarrow Y$.

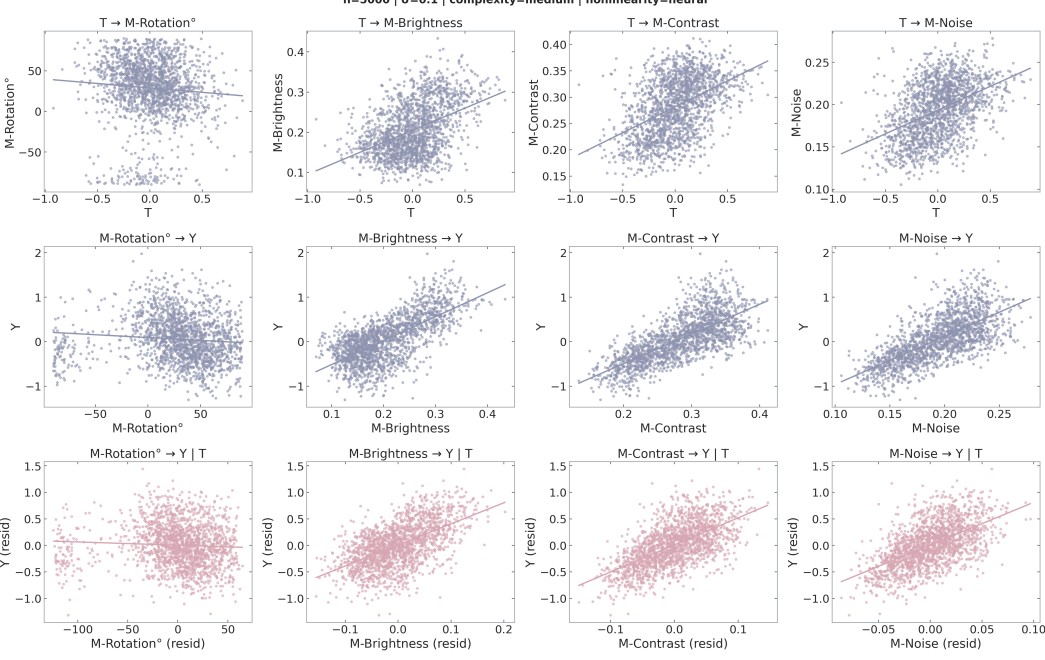

Figure 28: Synthetic data sample in setting 15 ($n = 5000$, $\sigma = 0.1$ and medium neural), $T \rightarrow M \rightarrow Y^*$.

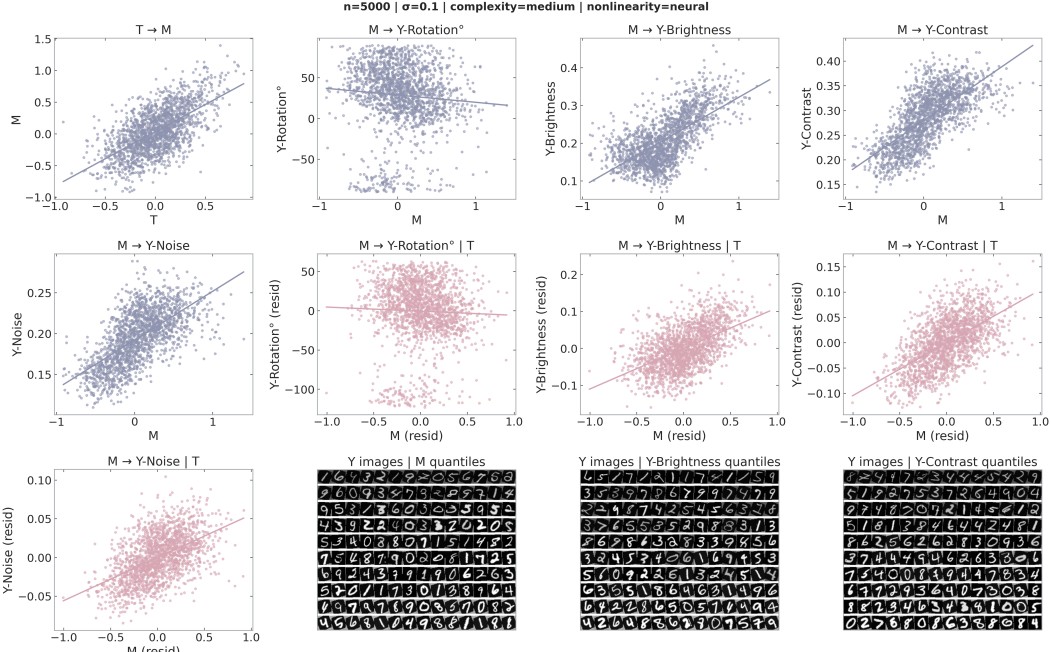

Figure 29: Synthetic data sample in setting 16 ($n = 5000$, $\sigma = 0.1$ and medium neural), $T \rightarrow M^* \rightarrow Y$.

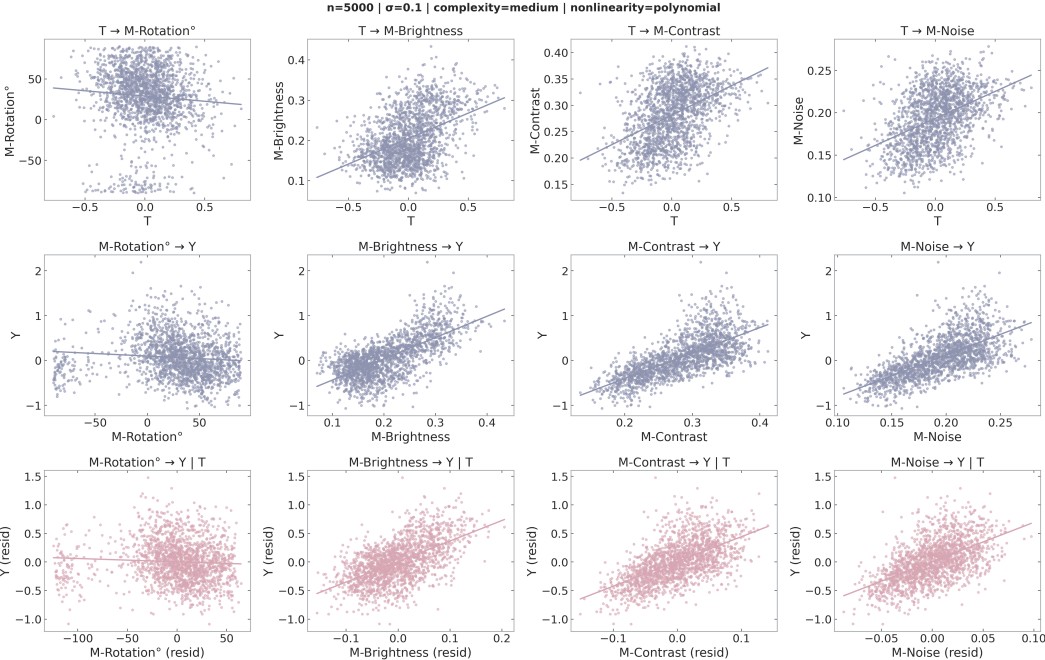

Figure 30: Synthetic data sample in setting 17 ($n = 5000$, $\sigma = 0.1$ and medium polynomial), $T \rightarrow M \rightarrow Y^*$.

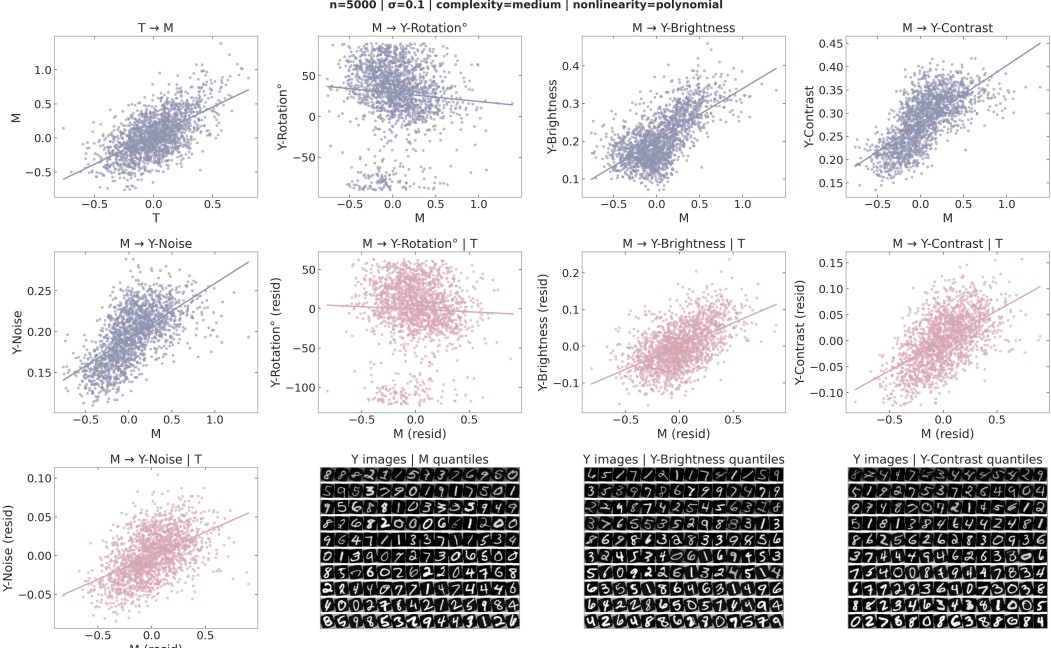

Figure 31: Synthetic data sample in setting 18 ($n = 5000$, $\sigma = 0.1$ and medium polynomial), $T \to M^* \to Y$.

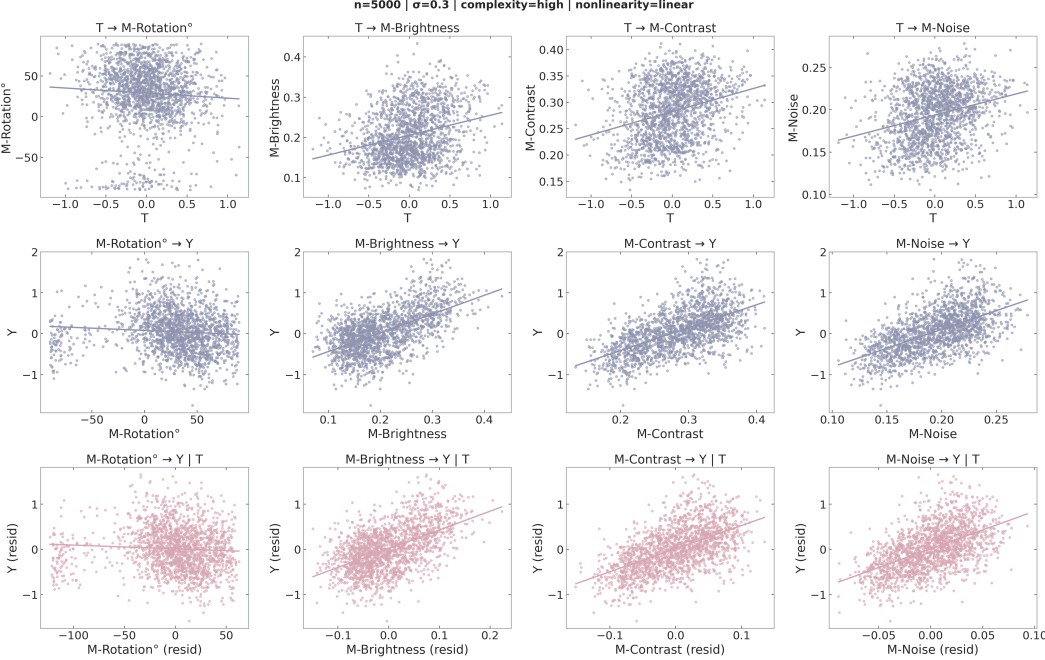

Figure 32: Synthetic data sample in setting 19 ($n = 5000$, $\sigma = 0.3$ and high linear), $T \to M \to Y^*$.

