# CARL: PRESERVING CAUSAL STRUCTURE IN REPRESENTATION LEARNING

# 1 DATASET DESCRIPTION

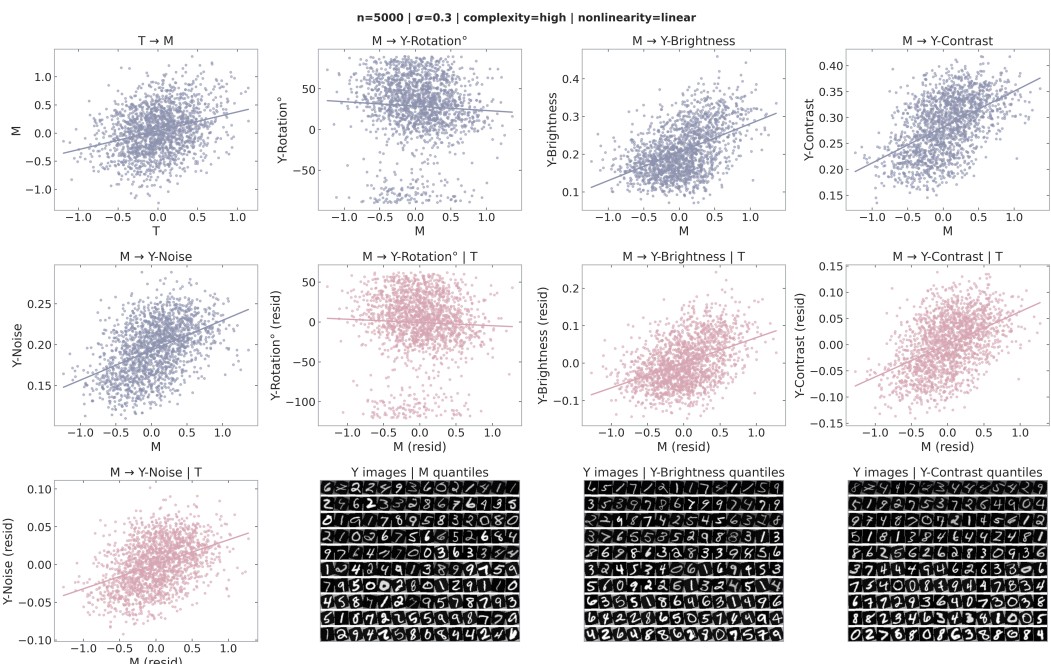

Figure 1: Synthetic data sample in setting 20 ($n = 5000$, $\sigma = 0.3$ and high linear), $T \rightarrow M^* \rightarrow Y$.

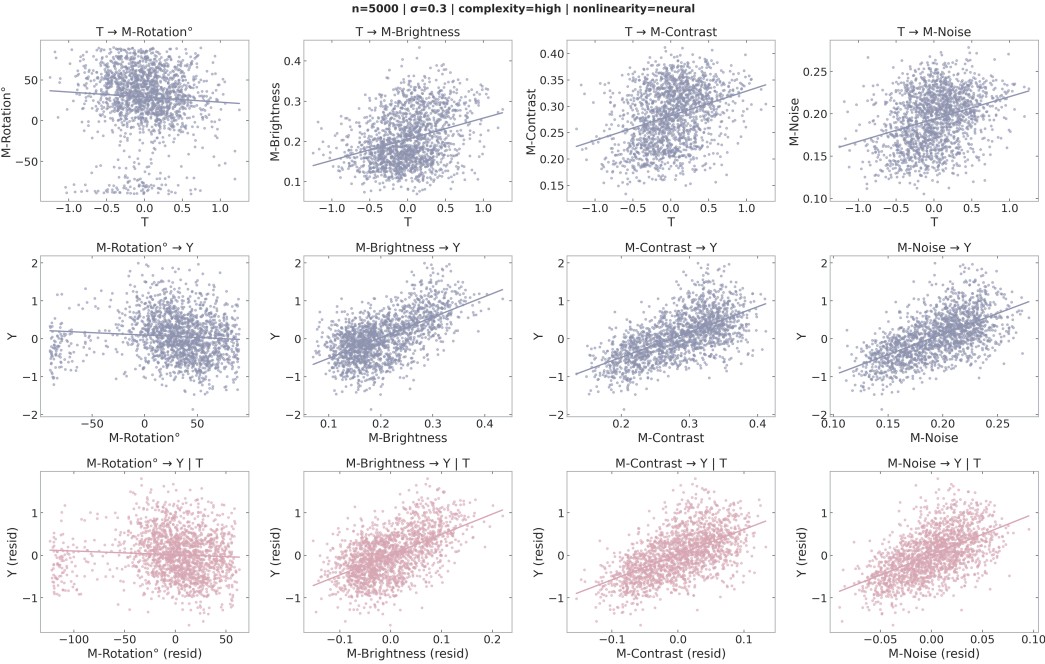

Figure 2: Synthetic data sample in setting 21 ($n = 5000$, $\sigma = 0.3$ and high neural), $T \rightarrow M \rightarrow Y^*$.

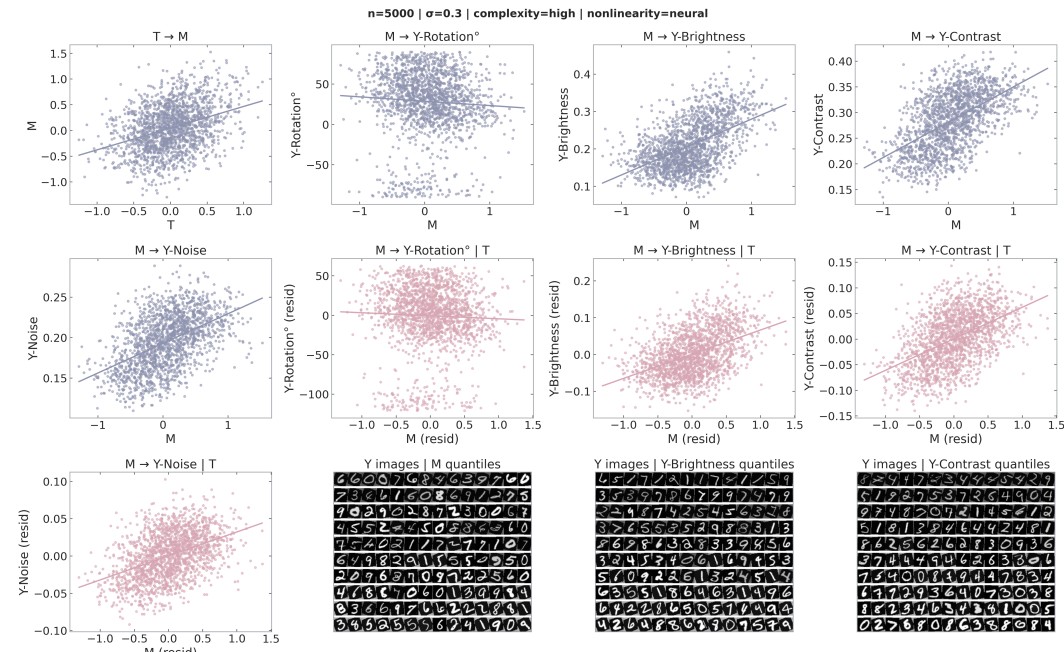

Figure 3: Synthetic data sample in setting 22 ($n = 5000$, $\sigma = 0.3$ and high neural), $T \rightarrow M^* \rightarrow Y$.

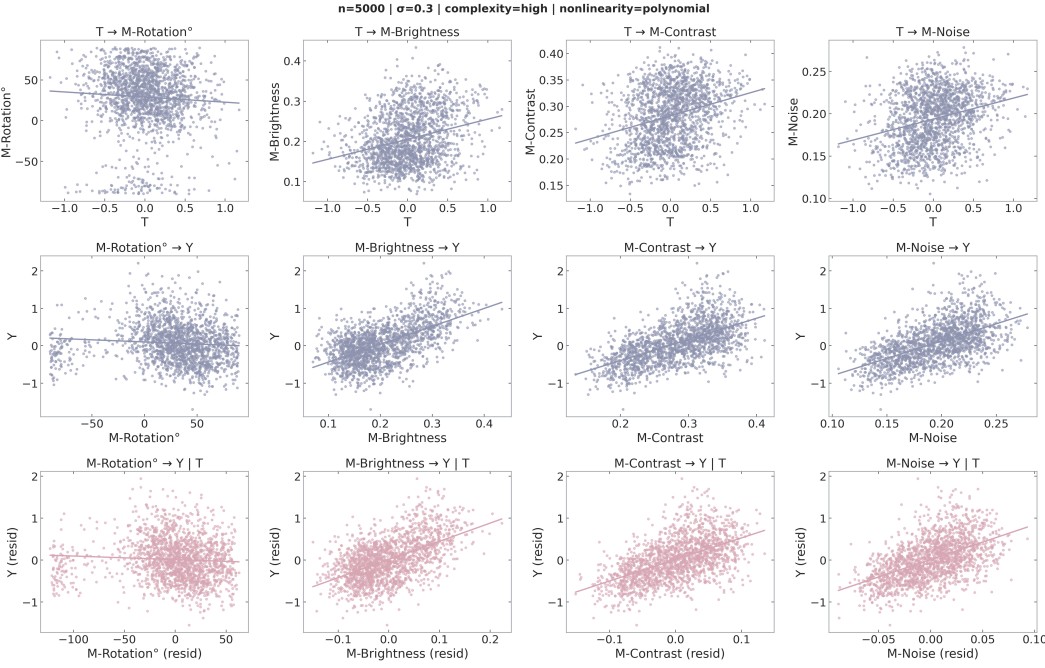

Figure 4: Synthetic data sample in setting 23 ($n = 5000$, $\sigma = 0.3$ and high polynomial), $T \rightarrow M \rightarrow Y^*$.

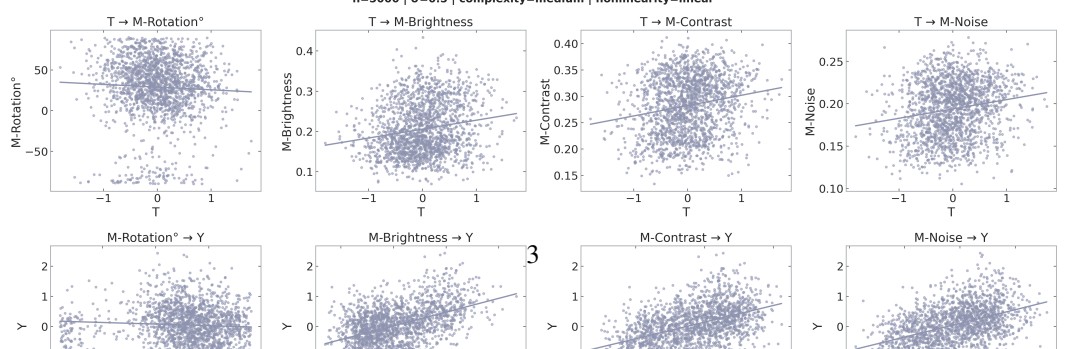

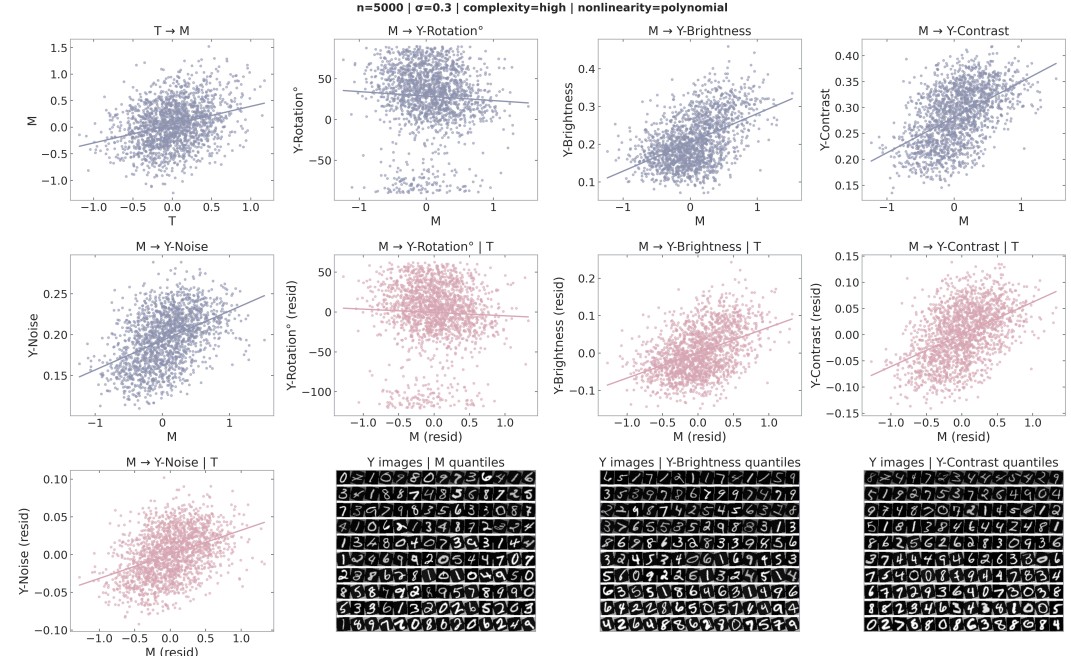

Figure 5: Synthetic data sample in setting 24 ($n = 5000$, $\sigma = 0.3$ and high polynomial), $T \rightarrow M^* \rightarrow Y$.

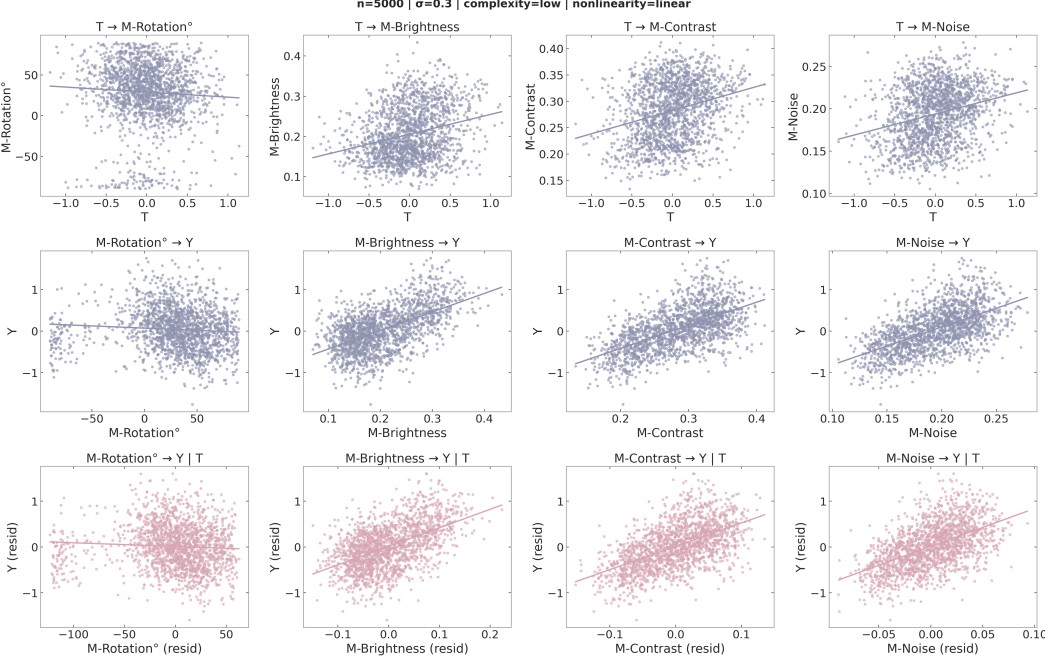

Figure 6: Synthetic data sample in setting 25 ($n = 5000$, $\sigma = 0.3$ and low linear), $T \rightarrow M \rightarrow Y^*$.

# REFERENCES

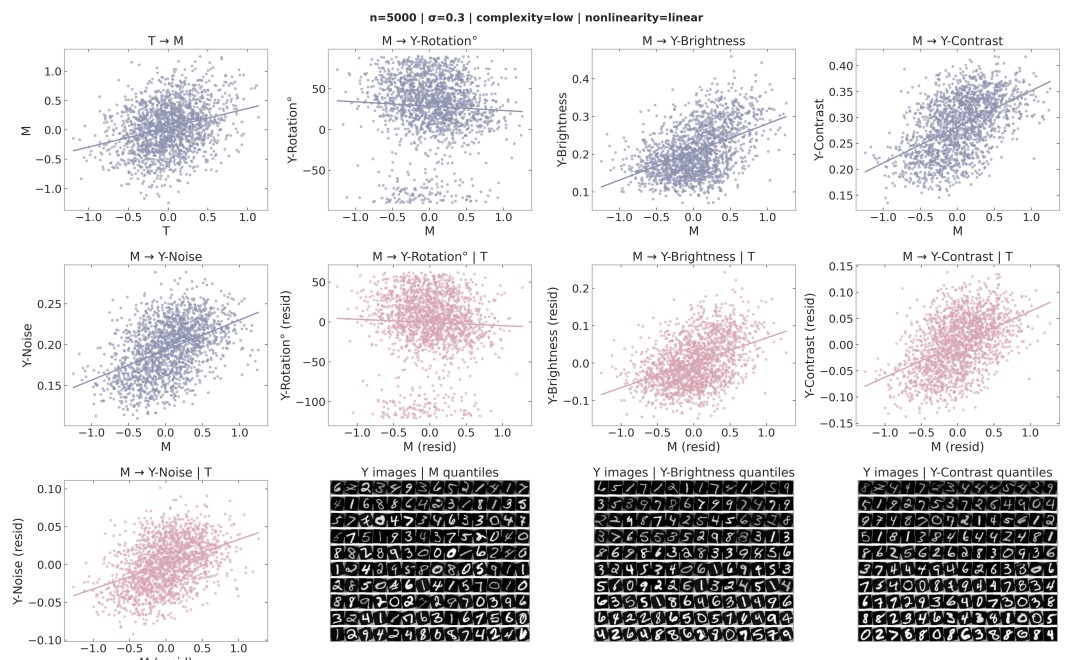

Figure 7: Synthetic data sample in setting 26 ($n = 5000$, $\sigma = 0.3$ and low linear), $T \to M^* \to Y$.

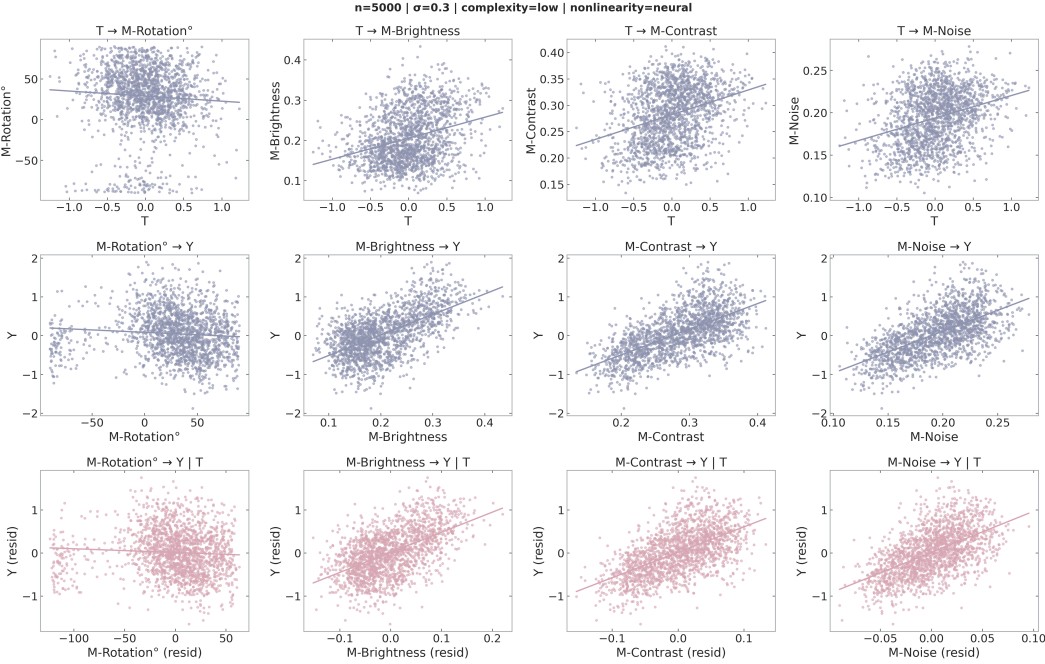

Figure 8: Synthetic data sample in setting 27 ($n = 5000$, $\sigma = 0.3$ and low neural), $T \to M \to Y^*$.

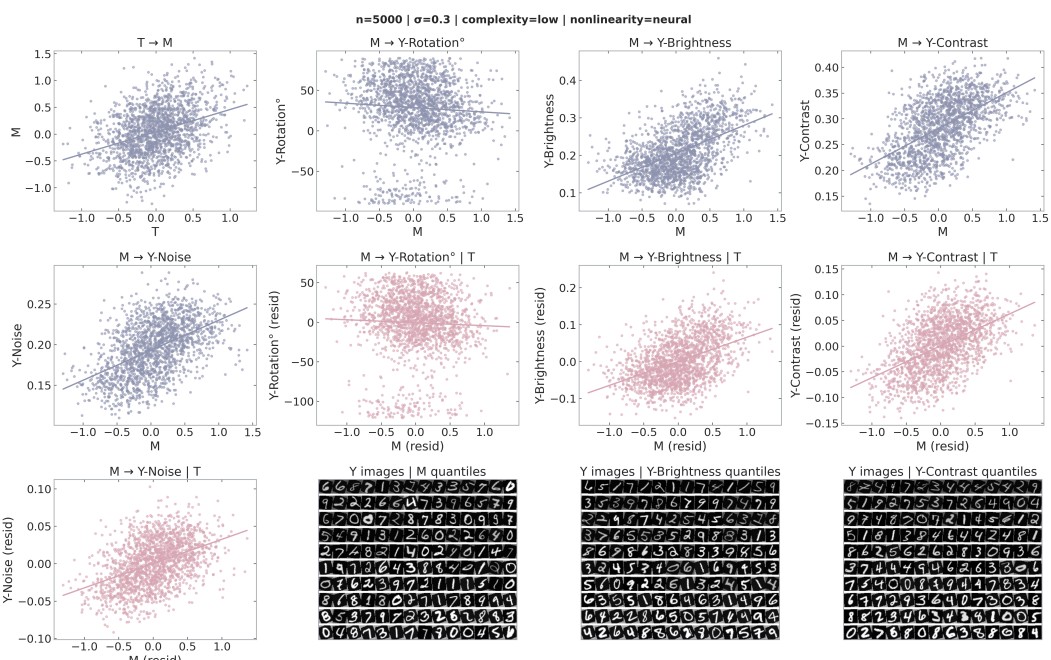

Figure 9: Synthetic data sample in setting 28 ($n = 5000$, $\sigma = 0.3$ and low neural), $T \rightarrow M^* \rightarrow Y$.

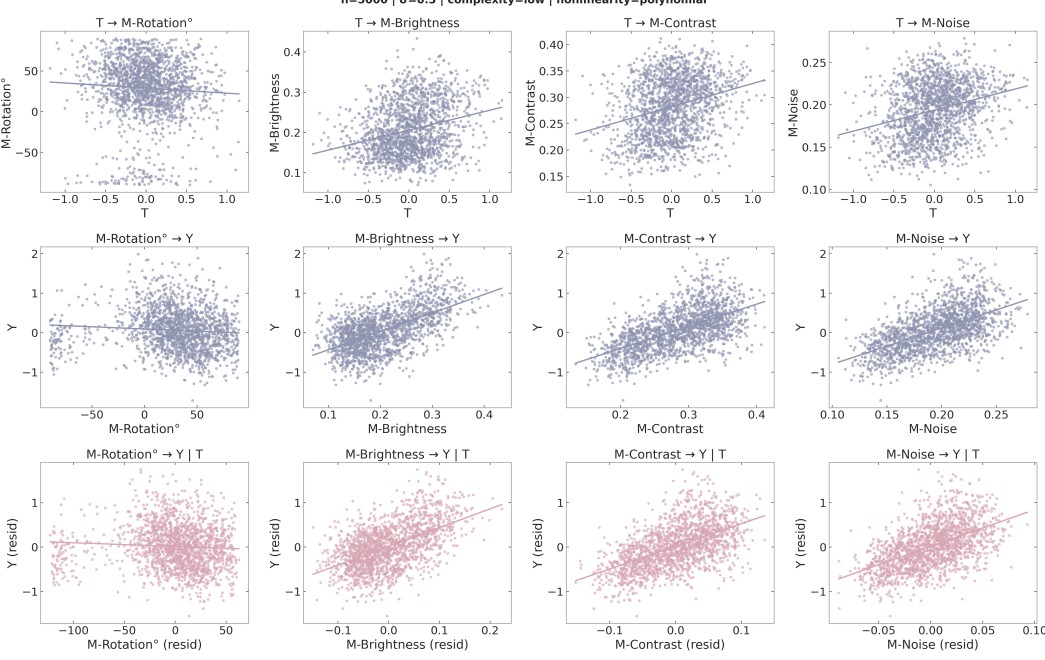

Figure 10: Synthetic data sample in setting 29 ($n = 5000$, $\sigma = 0.3$ and low polynomial), $T \rightarrow M \rightarrow Y^*$.

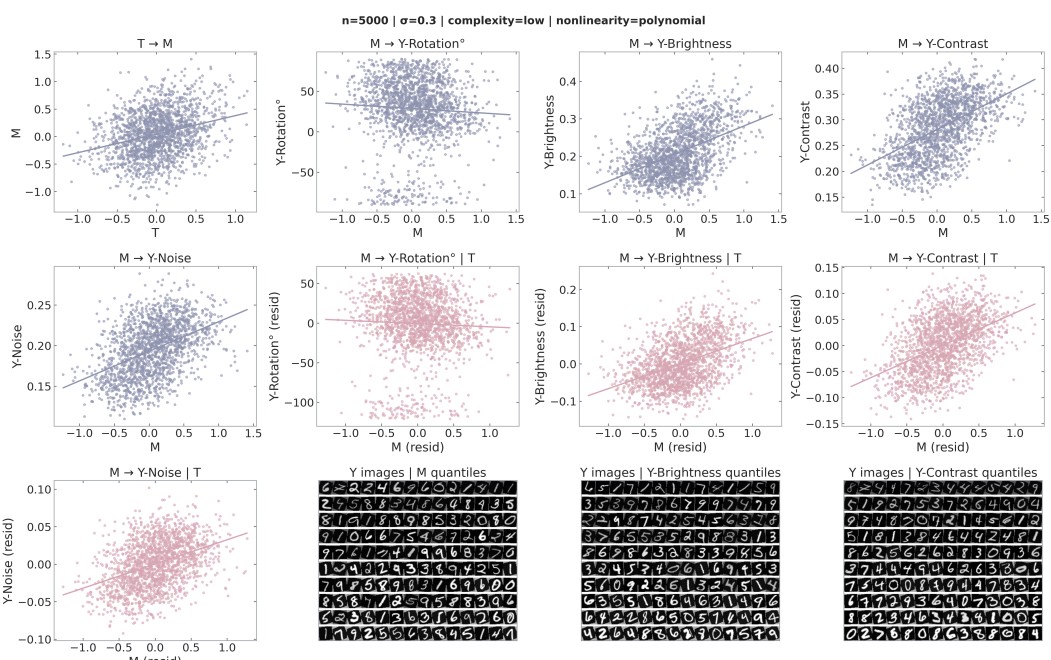

Figure 11: Synthetic data sample in setting 30 ($n = 5000$, $\sigma = 0.3$ and low polynomial), $T \rightarrow M^* \rightarrow Y$.

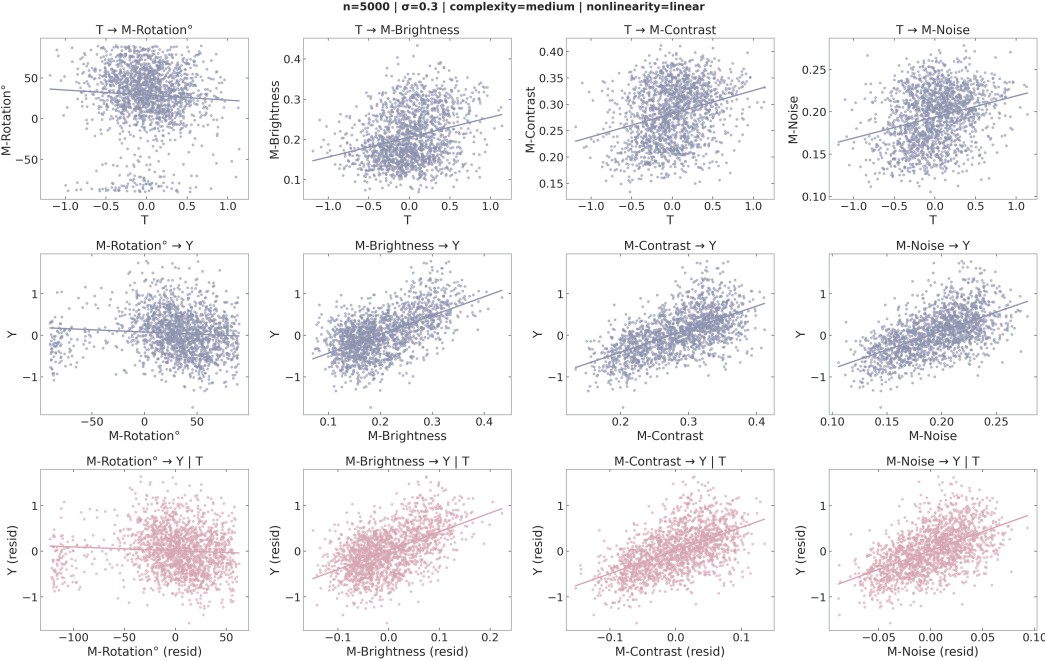

Figure 12: Synthetic data sample in setting 31 ($n = 5000$, $\sigma = 0.3$ and medium linear), $T \rightarrow M \rightarrow Y^*$.

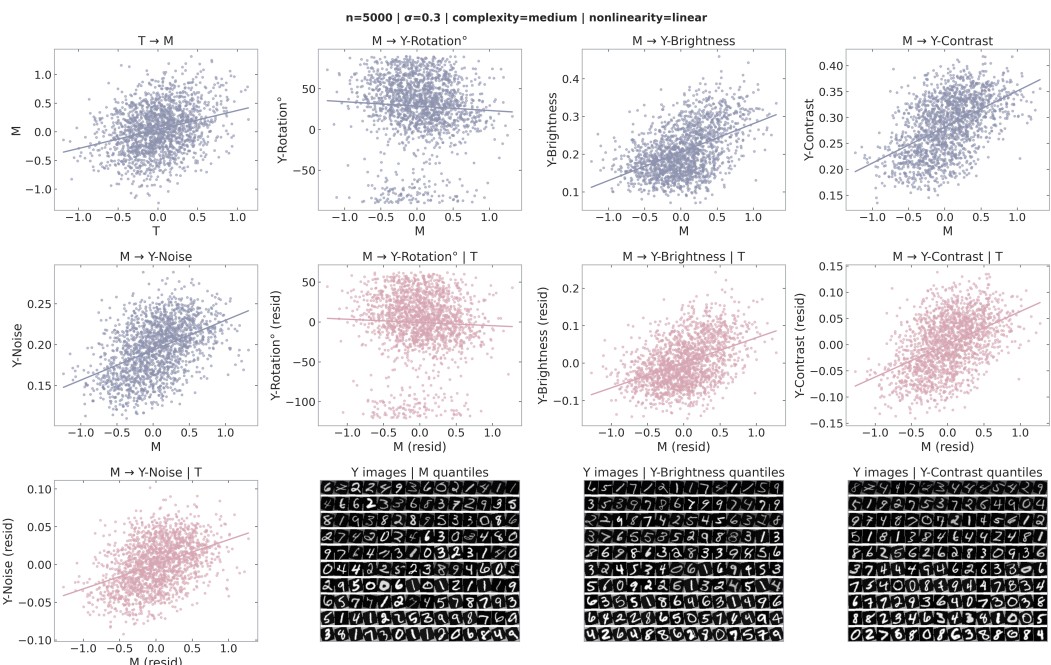

Figure 13: Synthetic data sample in setting 32 ($n = 5000$, $\sigma = 0.3$ and medium linear), $T \rightarrow M^* \rightarrow Y$.

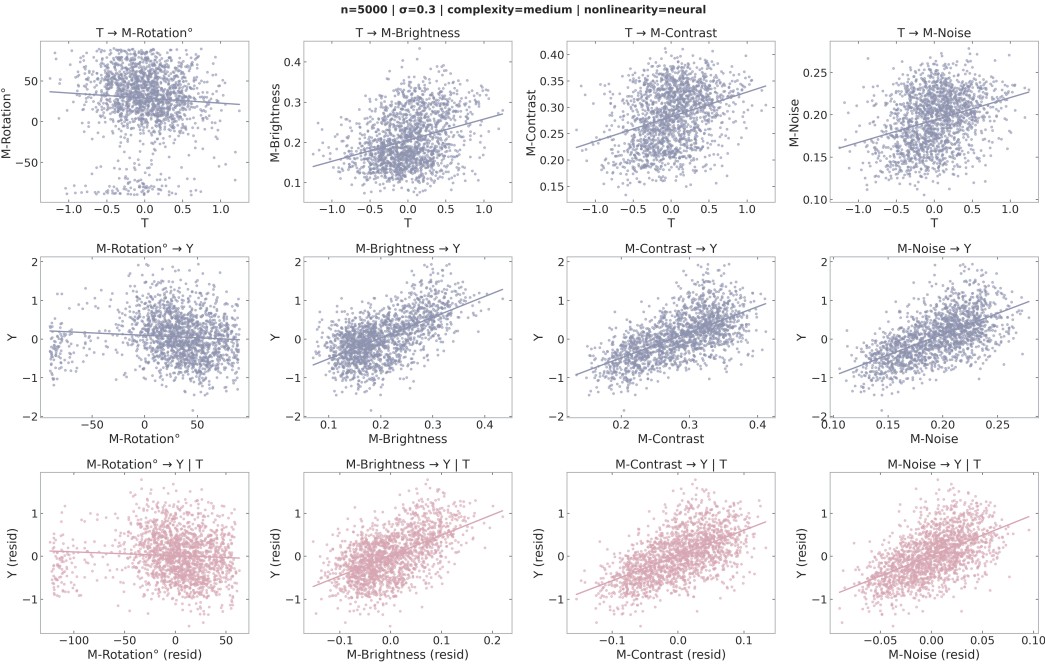

Figure 14: Synthetic data sample in setting 33 ($n = 5000$, $\sigma = 0.3$ and medium neural), $T \rightarrow M \rightarrow Y^*$.

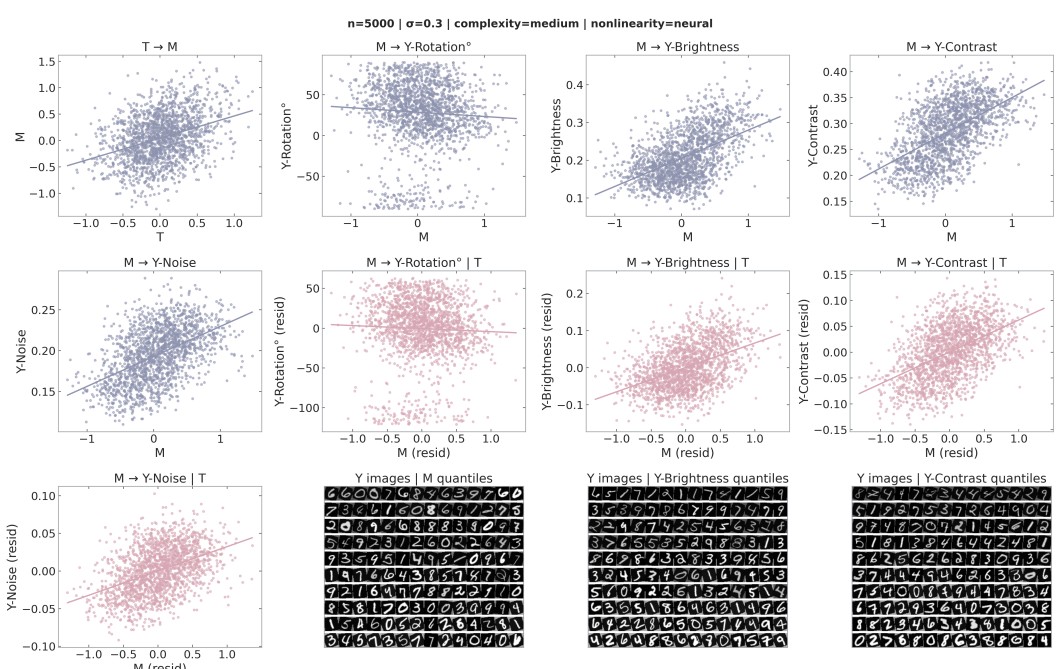

Figure 15: Synthetic data sample in setting 34 ($n = 5000$, $\sigma = 0.3$ and medium neural), $T \rightarrow M^* \rightarrow Y$.

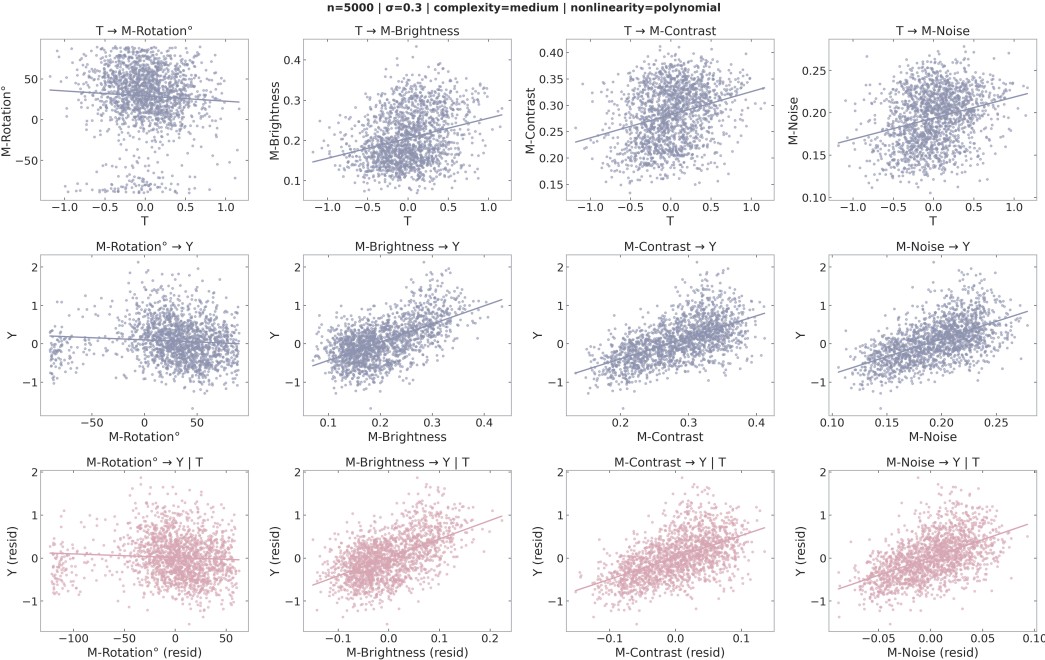

Figure 16: Synthetic data sample in setting 35 ($n = 5000$, $\sigma = 0.3$ and medium polynomial), $T \rightarrow M \rightarrow Y^*$.

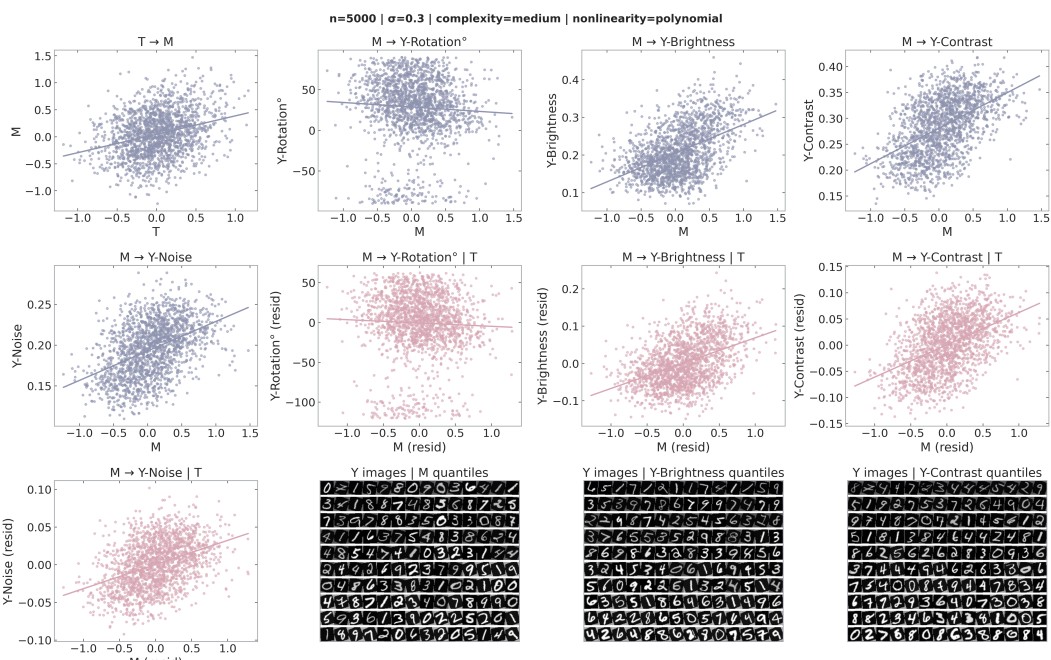

Figure 17: Synthetic data sample in setting 36 ($n = 5000$, $\sigma = 0.3$ and medium polynomial), $T \rightarrow M^* \rightarrow Y$.

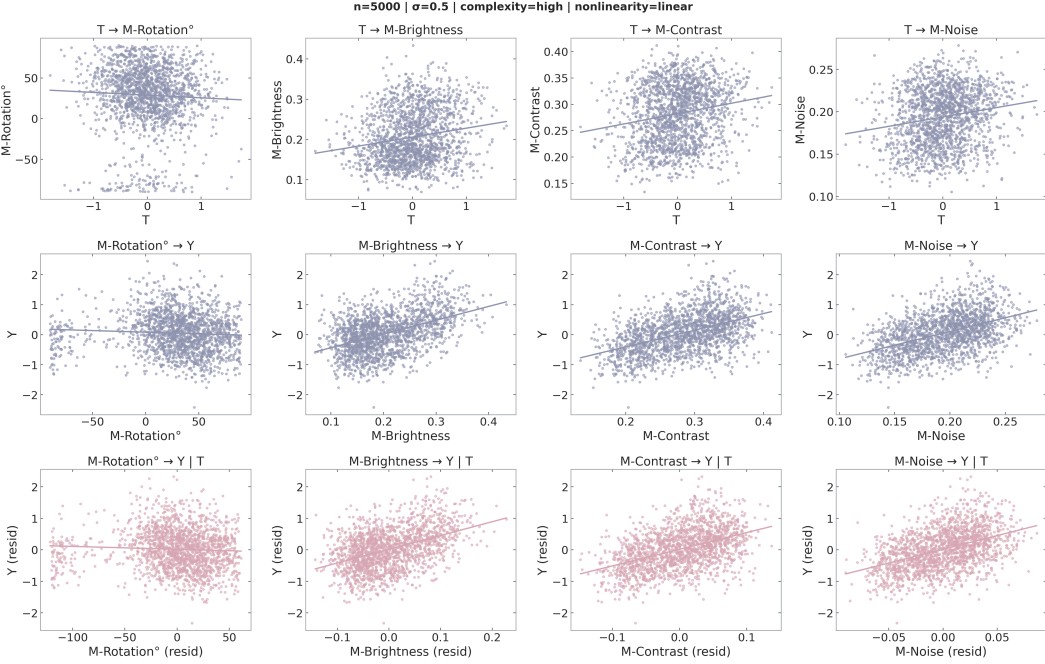

Figure 18: Synthetic data sample in setting 37 ($n = 5000$, $\sigma = 0.5$ and high linear), $T \rightarrow M \rightarrow Y^*$.

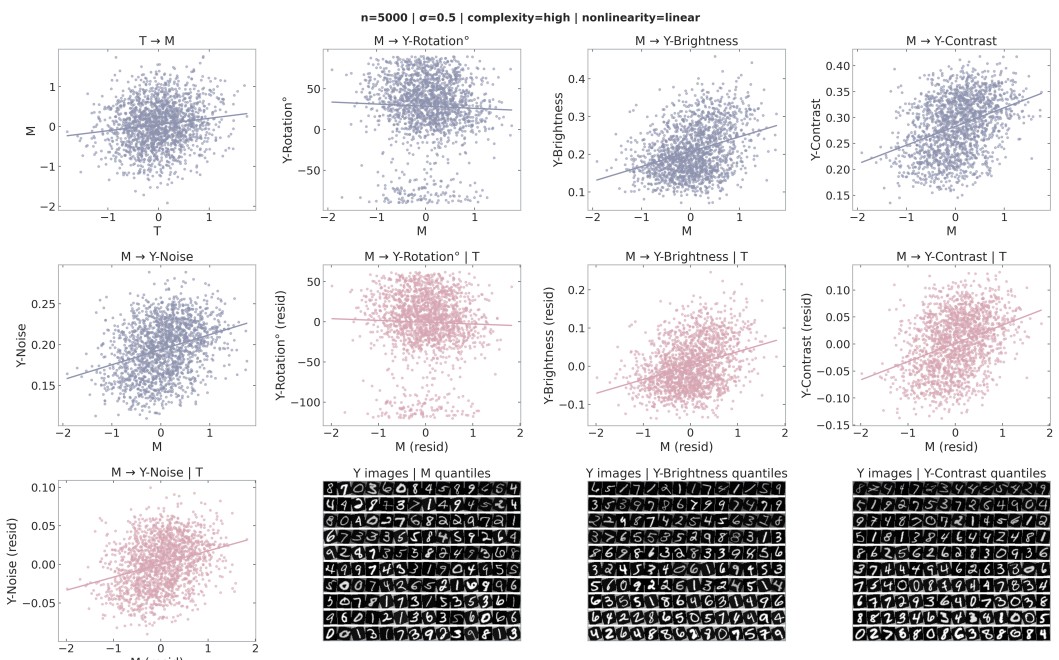

Figure 19: Synthetic data sample in setting 38 ($n = 5000$, $\sigma = 0.5$ and high linear), $T \to M^* \to Y$.

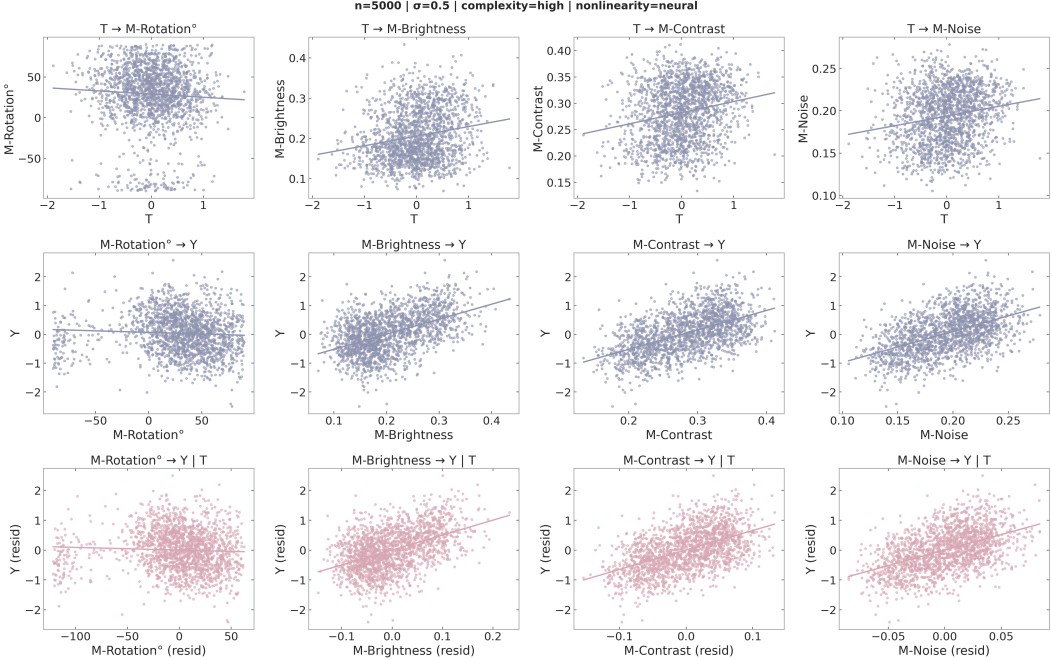

Figure 20: Synthetic data sample in setting 39 ($n = 5000$, $\sigma = 0.5$ and high neural), $T \to M \to Y^*$.

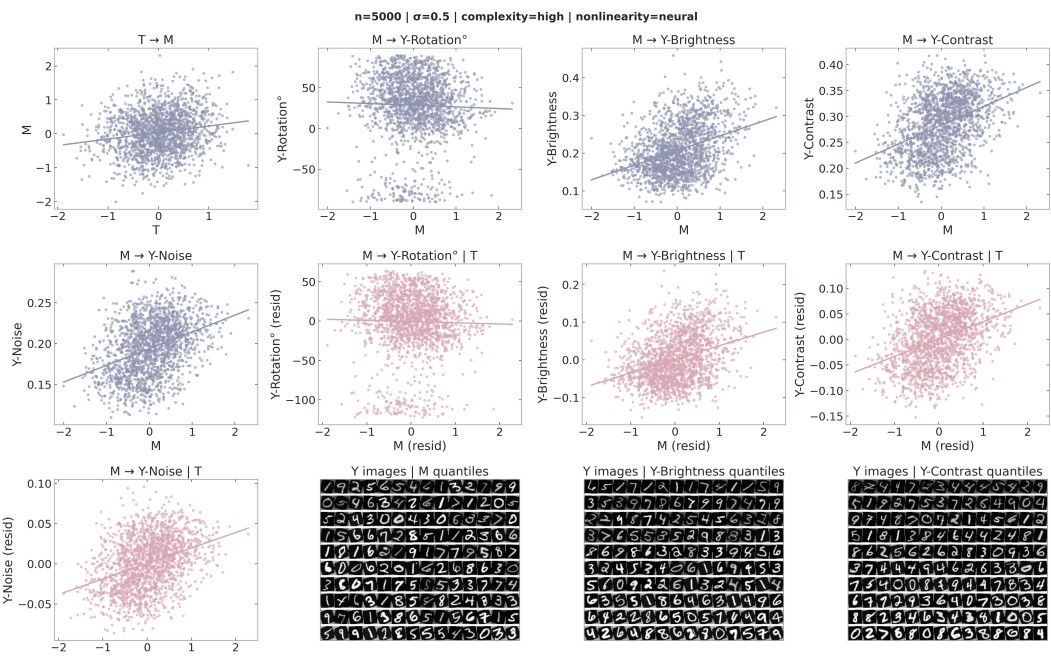

Figure 21: Synthetic data sample in setting 40 ($n = 5000$, $\sigma = 0.5$ and high neural), $T \to M^* \to Y$.

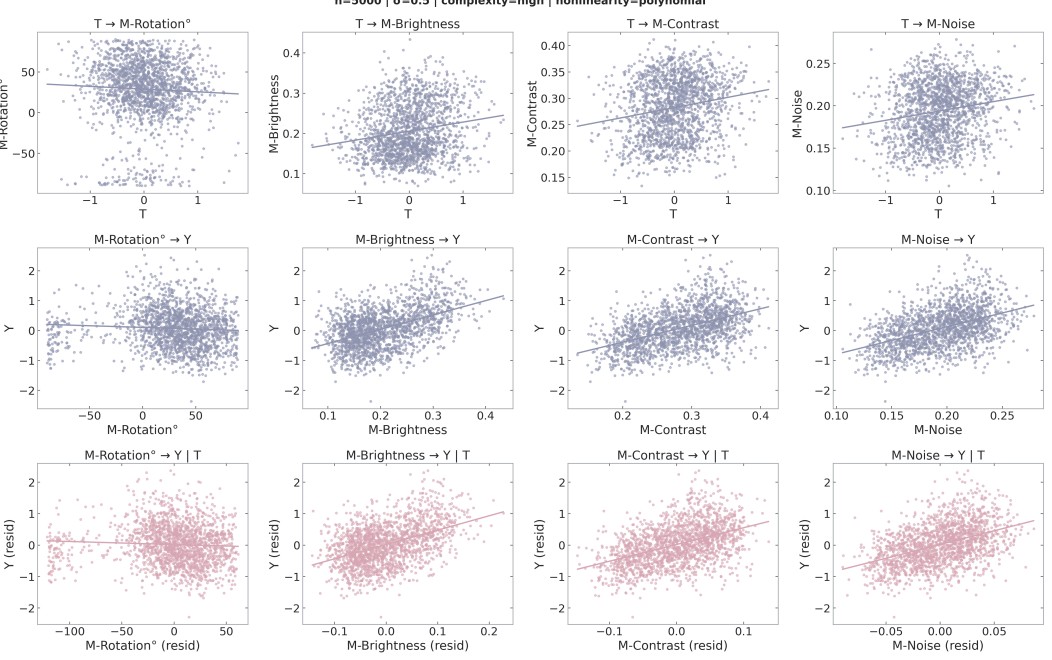

Figure 22: Synthetic data sample in setting 41 ($n = 5000$, $\sigma = 0.5$ and high polynomial), $T \to M \to Y^*$.

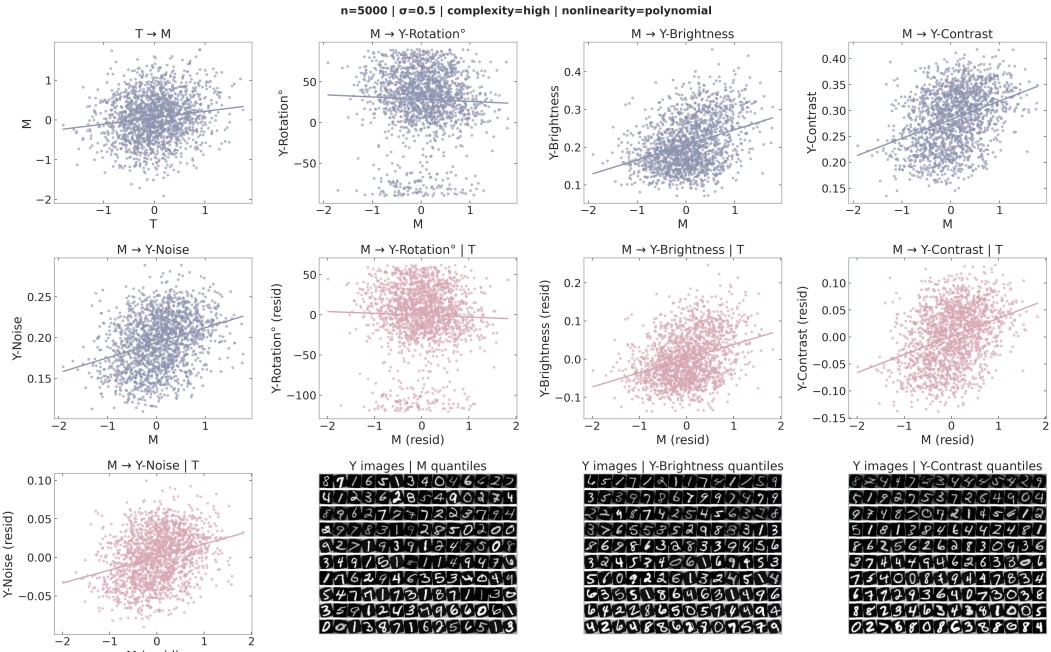

Figure 23: Synthetic data sample in setting 42 ($n = 5000$, $\sigma = 0.5$ and high polynomial), $T \to M^* \to Y$.

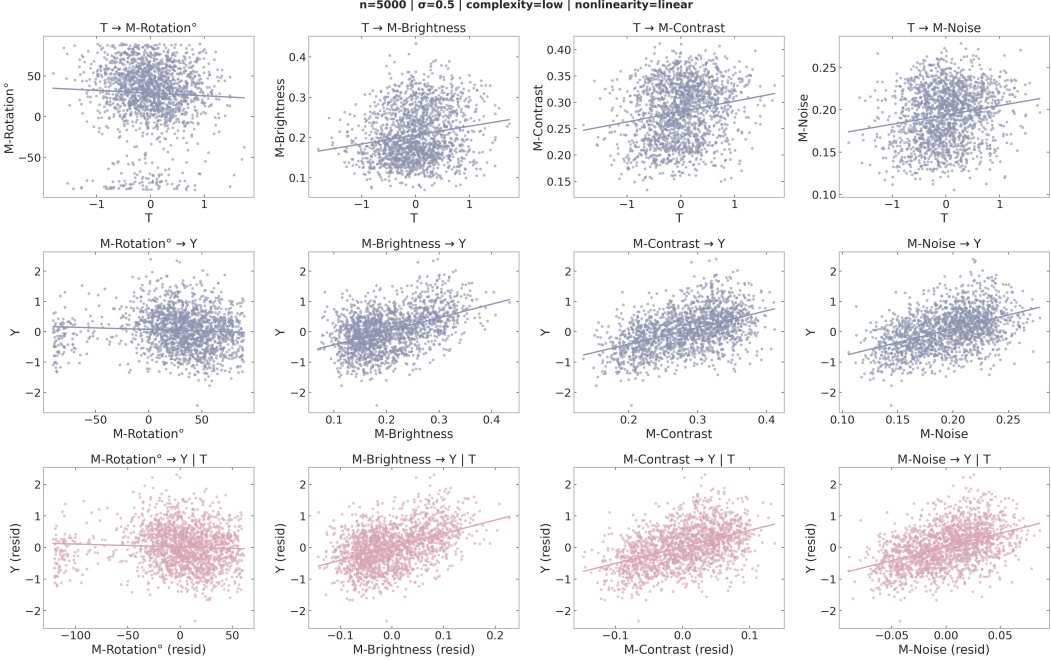

Figure 24: Synthetic data sample in setting 43 ($n = 5000$, $\sigma = 0.5$ and low linear), $T \to M \to Y^*$.

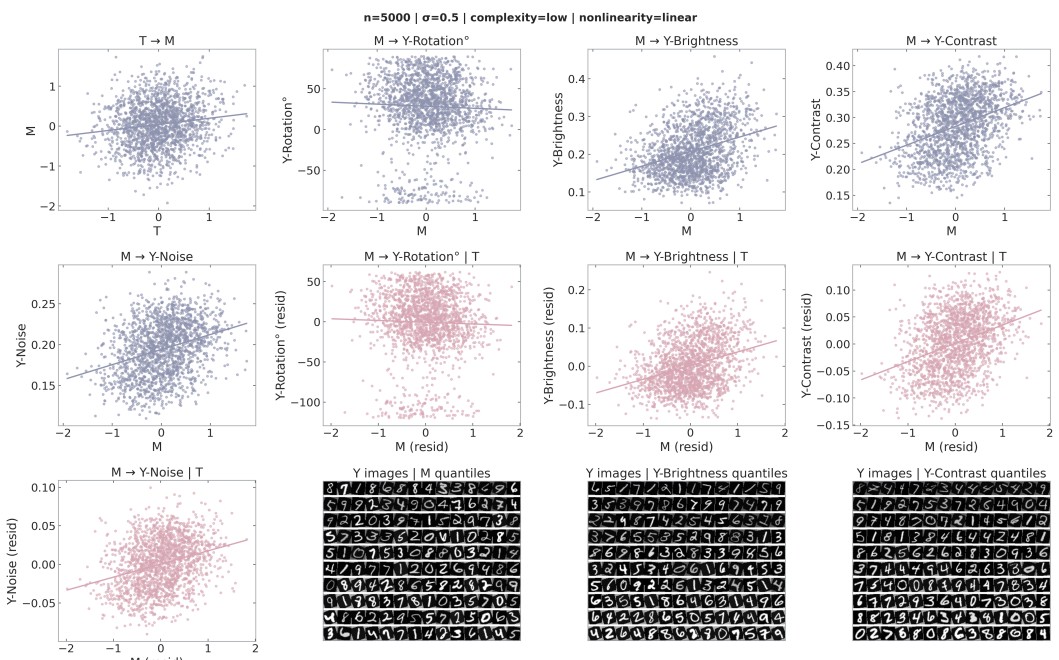

Figure 25: Synthetic data sample in setting 44 ($n = 5000$, $\sigma = 0.5$ and low linear), $T \to M^* \to Y$.

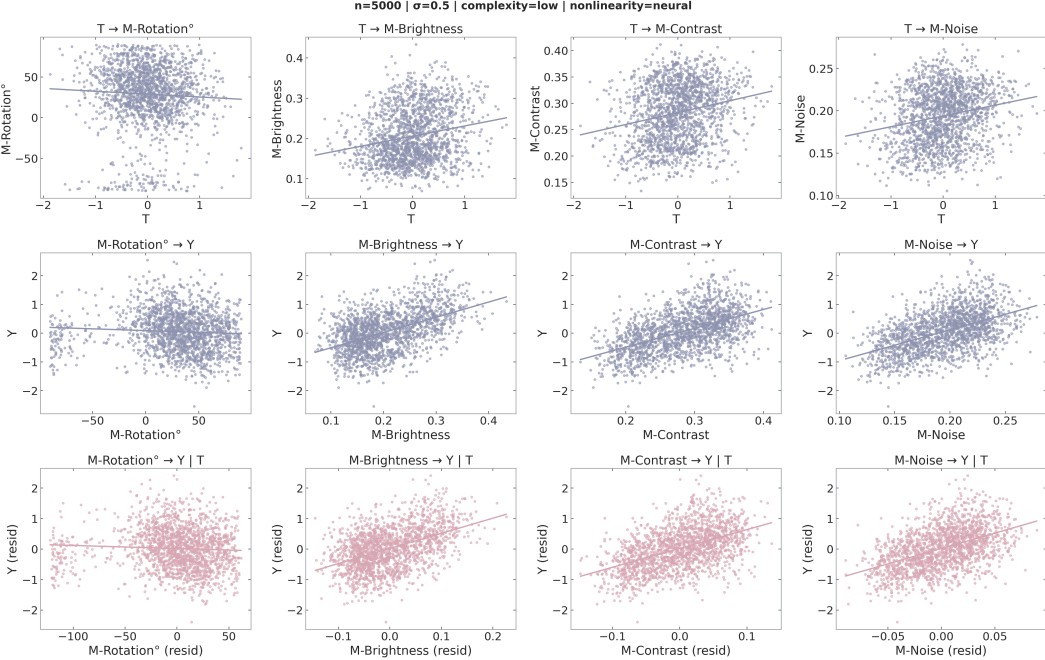

Figure 26: Synthetic data sample in setting 45 ($n = 5000$, $\sigma = 0.5$ and low neural), $T \to M \to Y^*$.

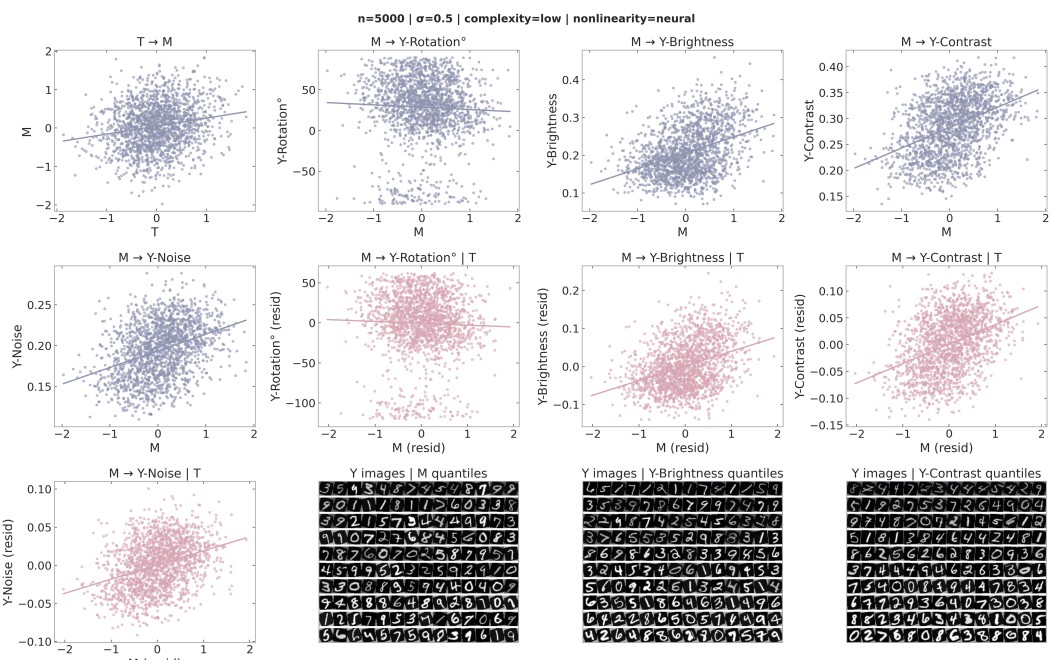

Figure 27: Synthetic data sample in setting 46 ($n = 5000$, $\sigma = 0.5$ and low neural), $T \to M^* \to Y$.

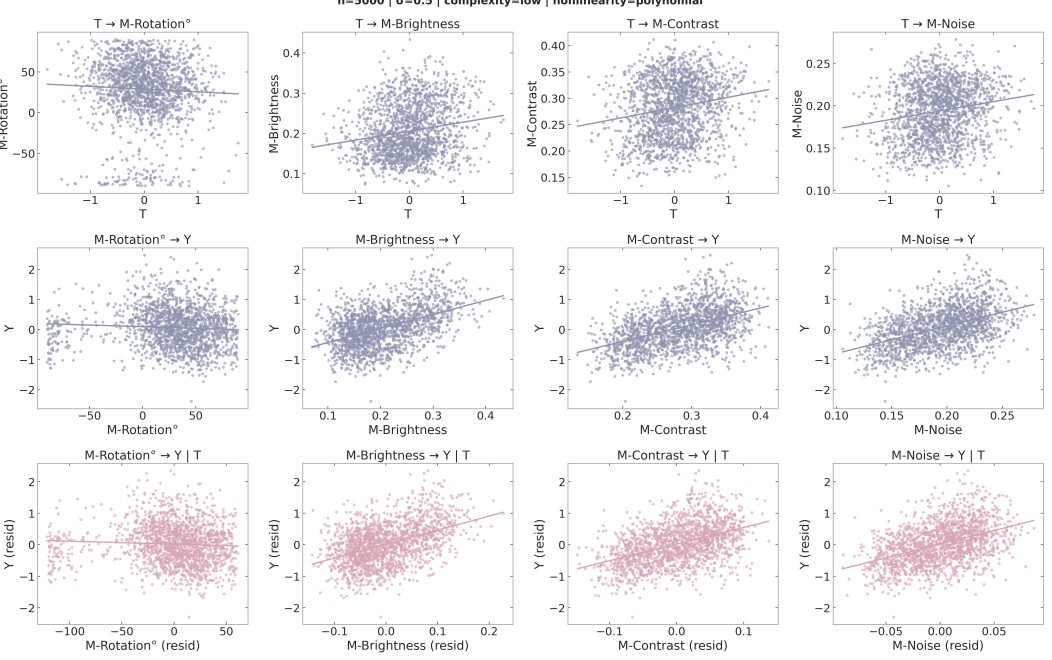

Figure 28: Synthetic data sample in setting 47 ($n = 5000$, $\sigma = 0.5$ and low polynomial), $T \to M \to Y^*$.

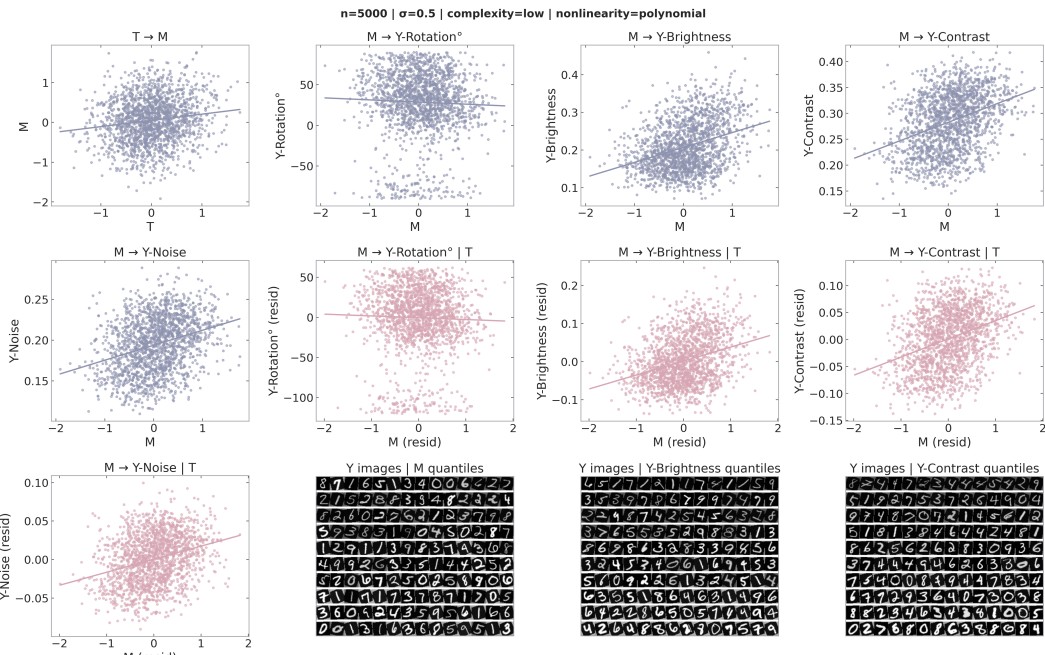

Figure 29: Synthetic data sample in setting 48 ($n = 5000$, $\sigma = 0.5$ and low polynomial), $T \to M^* \to Y$.

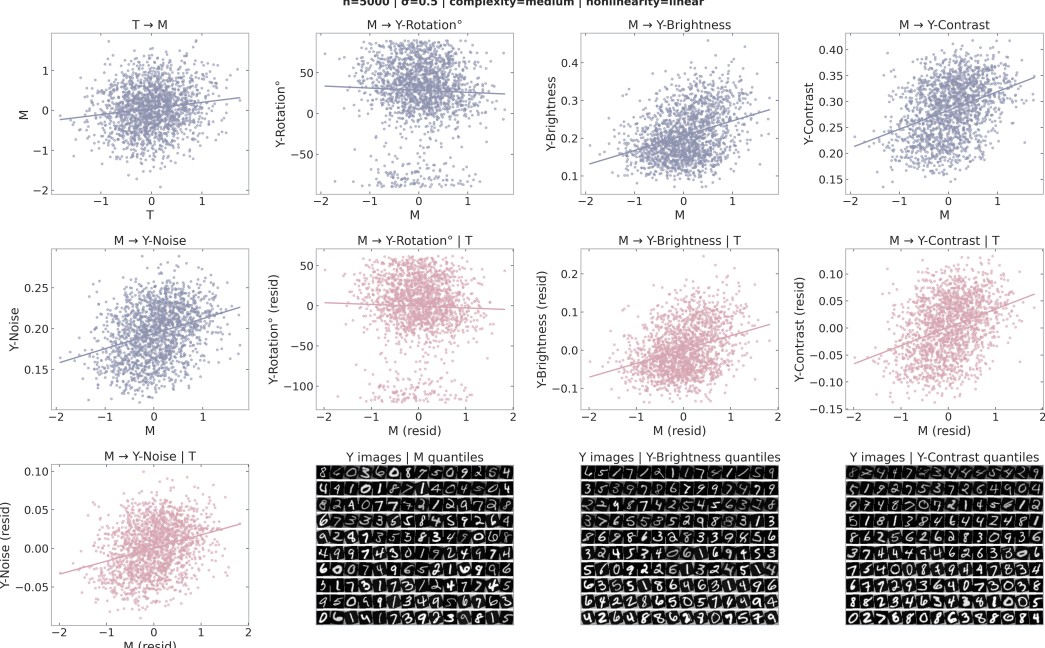

Figure 31: Synthetic data sample in setting 50 ($n = 5000$, $\sigma = 0.5$ and medium linear), $T \to M^* \to Y$.

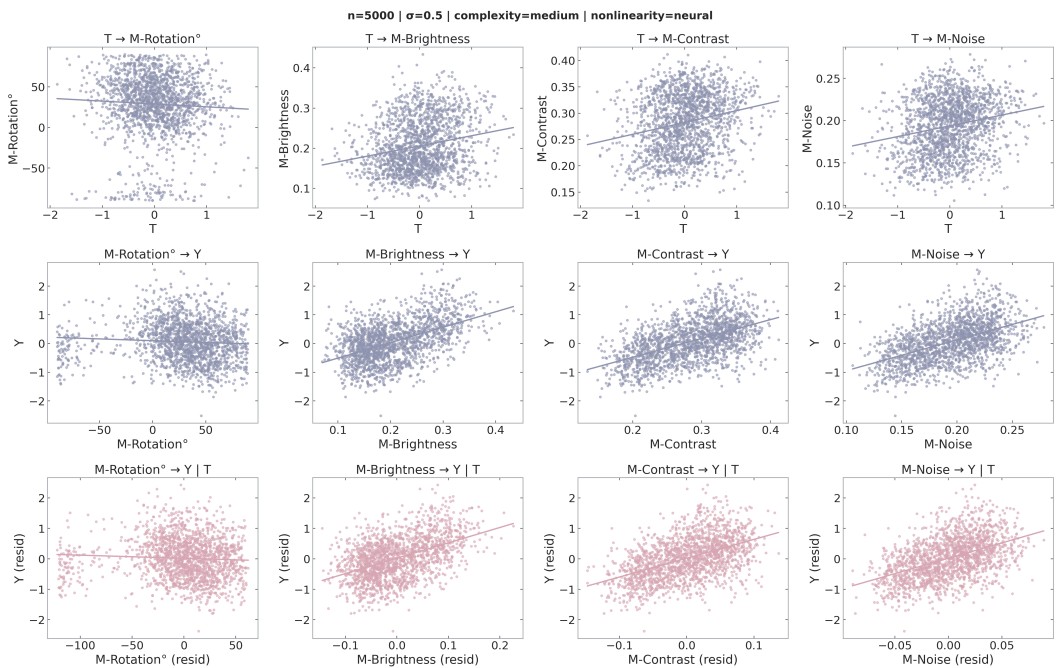

Figure 32: Synthetic data sample in setting 51 ($n = 5000$, $\sigma = 0.5$ and medium neural), $T \rightarrow M \rightarrow Y^*$.

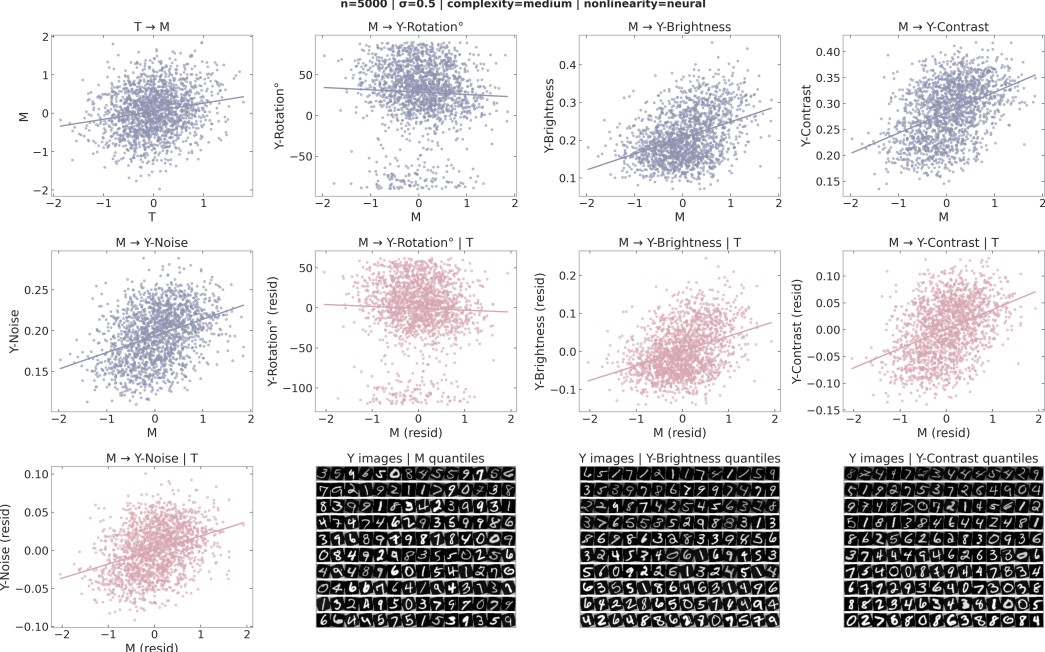

Figure 33: Synthetic data sample in setting 52 ($n = 5000$, $\sigma = 0.5$ and medium neural), $T \rightarrow M^* \rightarrow Y$.

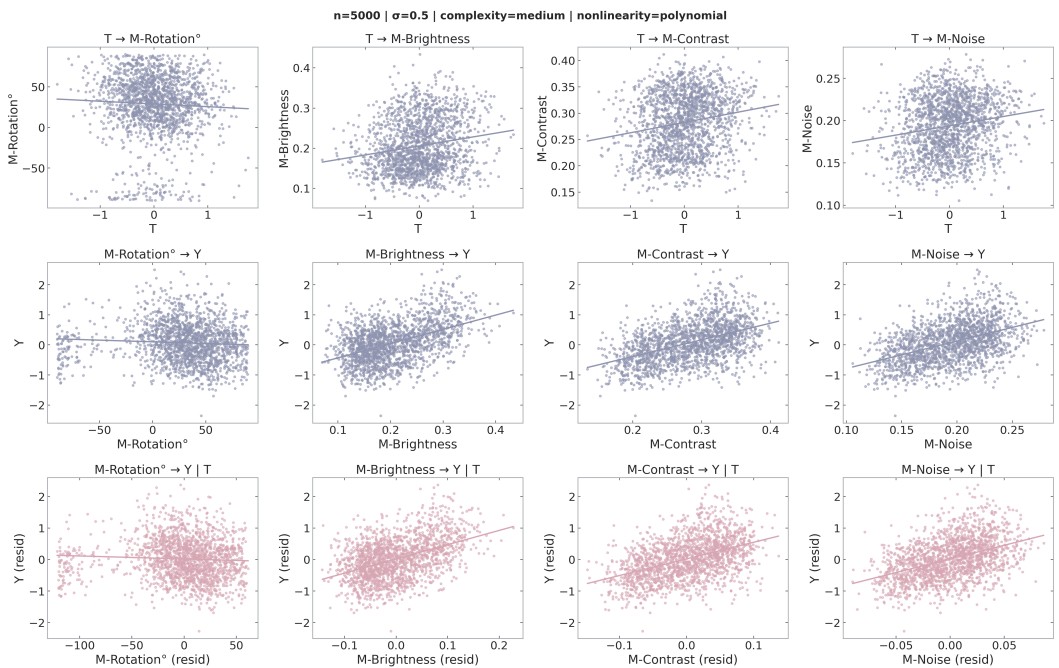

Figure 34: Synthetic data sample in setting 53 ($n = 5000$, $\sigma = 0.5$ and medium polynomial), $T \to M \to Y^*$.

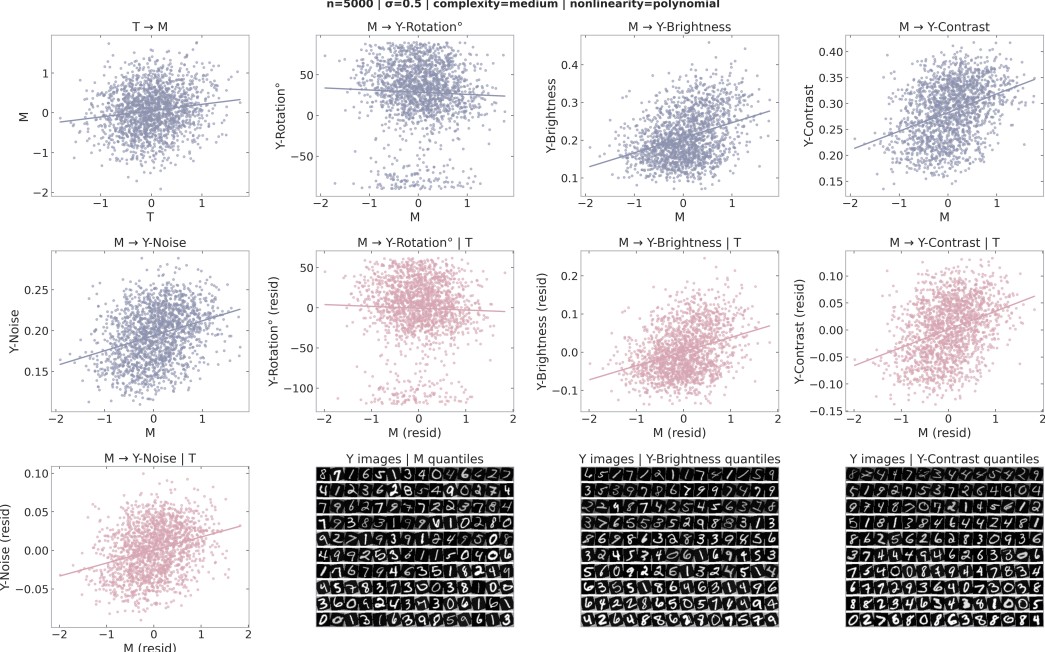

Figure 35: Synthetic data sample in setting 54 ($n = 5000$, $\sigma = 0.5$ and medium polynomial), $T \to M^* \to Y$.