# OpenReview forum: "CARL: Preserving Causal Structure in Representation Learning"
_ICLR.cc/2026/Conference — ICLR 2026 Poster_

### Official Review · Reviewer_enbA · 2025-10-31

**Soundness:** 3
**Presentation:** 3
**Contribution:** 2
**Rating:** 6
**Confidence:** 2

**Summary:**

This paper proposes a method for preserving known causal structures (e.g., mediation) in cross-modal representation learning. The approach leverages the independence relation implied by mediation, $T⊥Y^{*}| M$, to design a mutual-information-based loss that enforces the mediation causal structure. The authors validate the effectiveness of the method on a synthetic dataset and further estimate causal effects on the Human Phenotype Project (HPP) data, which align with causal modeling under mediation assumptions.

**Strengths:**

⦁	The experiments are comprehensive. In particular, the design of the synthetic dataset is well-documented in the appendix.
⦁	The model architecture and implementation details are clearly described, which enhances reproducibility.

**Weaknesses:**

⦁	The claimed contribution regarding the preservation of causal structure in cross-modal representation learning appears somewhat overstated. First, the modalities are limited to tabular and image data. Second, the assumed causal structure is restricted to mediation. The core loss functions conditional independencies preservation $\mathcal{L}{CI}$ and Markov boundary retention $\mathcal{L}{MBR}$ are heavily dependent on a predefined causal structure.
⦁	The authors seem to assume a causal graph under mediation. If such prior knowledge is unavailable, can the assertion of “preservation of causal structure” be generalized to unknown causal graphs? Without knowledge of the causal graph, how would one ensure preservation of the causal structure?
⦁	The method design is rather straightforward. Employing mutual information to enforce conditional independence has been widely explored in causal representation learning, and Modal Alignment Consistency is also a common idea in cross-modal representation learning.

**Questions:**

⦁	There seems to be an issue with some \ref{} in the appendix, as they do not correctly link to Section ABC. Section ABC in appendix appeared twice.
⦁	In the $I^M$ and $I^Y$ settings, the causal relationship visualizations in the appendix show weak correlations in some cases, with small $R^2$ values. In particular, for the Rotation transformation, the results hardly reflect the conclusions stated by the authors. Could the authors clarify this discrepancy? Moreover, the statement “These visualizations serve as an empirical confirmation of the dependencies and conditional independencies specified by our ground-truth SCM” is somewhat misleading, since the visualizations only verify dependencies. Why not empirically test $T⊥Y | M$ on the synthetic data?
⦁	Regarding the real-world HPP dataset, it is unclear what the causal effect of the learned latent representations signifies. Could these causal effects not be directly estimated from the observed data? How large is the discrepancy between the causal effects estimated in the latent space and those from the true causal effects? Please provide quantitative results.

---

> ### Author Response · Authors · 2025-11-21
>
> We thank reviewer for pointing out that our contribution statement requires more precise scoping! Regarding the three concerns about contribution scope, we clarify as follows.
> # Response W1:
> ### 1. **Modality Selection:**
>
> We emphasize that this choice represents the most challenging scenario in cross-modal representation learning.
>
> **Quantification of Dimensional Disparity.** Images have the highest raw dimensionality among common modalities. For typical medical images (e.g., $1024 \times 1024$ pixel retinal images), $d_{\text{img}} > 10^6$, while tabular data has $d_{\text{tab}} \sim 10^{1-2}$, yielding a dimensional ratio of:
>
> $$
> \frac{d_{\text{img}}}{d_{\text{tab}}} \sim 10^{4-5}
> $$
>
> Other common modality combinations include: text sequences $d_{\text{text}} \sim 10^{2-3}$, audio features (MFCC) $d_{\text{audio}} \sim 10^2$, time series $d_{\text{ts}} \sim 10^{1-2}$. Their dimensional ratios with tabular data are typically on the order of $10^{0-1}$.
>
> **Theoretical Analysis of the Bottleneck Effect.** The cross-modal information bottleneck arises from optimization imbalance. Under standard frameworks:
>
> $$
> L_{\mathrm{total}} = \sum_{i=1}^{d_{\mathrm{img}}} \ell_i^{\mathrm{img}} + \sum_{j=1}^{d_{\mathrm{tab}}} \ell_j^{\mathrm{tab}}
> $$
>
> Even when per-dimension errors are comparable ($\mathbb{E}[\ell_i^{\text{img}}] \sim \mathbb{E}[\ell_j^{\text{tab}}]$), since $d_{\text{img}} \gg d_{\text{tab}}$, we have:
>
> $$
> \mathbb{E}[L_{\mathrm{recon}}^{\mathrm{img}}] = d_{\mathrm{img}} \cdot \mathbb{E}[\ell_i^{\mathrm{img}}] \gg d_{\mathrm{tab}} \cdot \mathbb{E}[\ell_j^{\mathrm{tab}}] = \mathbb{E}[L_{\mathrm{recon}}^{\mathrm{tab}}]
> $$
>
> The gradient update $\nabla_\theta L_{\mathrm{total}}$ primarily responds to the high-dimensional modality. Assuming limited representation capacity ($\dim(Z) = d < d_{\mathrm{img}} + d_{\mathrm{tab}}$), under approximation error minimization, the information allocation in $Z$ satisfies:
>
> $$
> \frac{I(Z; X_{\text{img}})}{I(Z; X_{\text{tab}})} \propto \frac{d_{\text{img}}}{d_{\text{tab}}}
> $$
>
> This leads to compression of critical causal variables in the tabular modality.
>
> **Complementarity in Information Density.** Images and tabular data are complementary in information density. While $H(X_{\text{img}})$ has large total entropy, the average information per dimension $H(X_{\text{img}})/d_{\text{img}}$ is small due to high correlation between adjacent pixels (spatial redundancy). Tabular features are designed by domain experts, with each dimension carrying condensed semantics (e.g., blood pressure, age, and other clinical indicators), resulting in large $H(X_{\text{tab}})/d_{\text{tab}}$.
>
> **Theoretical Solution Mechanism.** Our $\mathcal{L}_{\text{MBR}} = -\text{InfoNCE}(z_m, \psi_Y(y))$ (where $\psi_Y: \mathbb{R}^k \rightarrow \mathbb{R}^d$ maps the observable outcome $y$ to a contrastive target's semantic embedding, such as one-hot encoding or continuous labels projected through an MLP) explicitly counteracts dimensional dominance. By maximizing a lower bound on $I(Z_M; \psi_Y(Y))$, it ensures that mediator information in tabular data is preserved even under image reconstruction pressure. This mechanism applies to any dimensionally asymmetric scenario, with its necessity most evident at the $10^{4-5}$ scale.
>
> **Scalability of the Method.** The image-tabular combination covers critical domains including medical diagnosis (clinical indicators + imaging), financial risk control (transaction data + receipts), industrial quality inspection (sensor readings + appearance images), and remote sensing analysis (meteorological data + satellite images). Our encoder architecture $\mathcal{E} = \{E_T, E_M, E_{IM}, E_{IY}\}$ has a modular design that naturally extends to other modalities. The three loss functions are based on information-theoretic universal principles (CMI, MI, rank correlation) and are independent of specific modality types. Once validated on the image-tabular scenario with extreme dimensional disparity, the method's applicability to other modality combinations with smaller dimensional differences is naturally supported.

---

> > ### Author Response · Authors · 2025-11-21
> >
> > ### 2. **Causal Structure Selection**
> >
> > **Mathematical Formulation of Dual Constraints.** The mediation structure $T \rightarrow M \rightarrow Y^*$ requires representation learning to simultaneously satisfy:
> >
> > $$
> > \begin{aligned}
> > \text{Constraint C1:} \quad & I(Z_T; Z_Y \mid Z_M) \leq \varepsilon_1 \quad \text{(block direct path)} \\
> > \text{Constraint C2:} \quad & I(Z_M; Z_Y) \geq I(M; Y^*) - \varepsilon_2 \quad \text{(preserve indirect path)}
> > \end{aligned}
> > $$
> >
> > There exists a fundamental trade-off between these two constraints, which can be proven through the following analysis.
> >
> > **Theoretical Proof of the Trade-off.** Consider the case of limited representation dimensionality: $\dim(Z_T) = \dim(Z_M) = \dim(Z_Y) = d$, with constrained total representation capacity. By the data processing inequality, for any mapping $E_M: M \rightarrow Z_M$:
> >
> > $$
> > I(Z_M; Y^{\ast}) \leq I(M; Y^{\ast})
> > $$
> >
> > Equality holds if and only if $E_M$ is a sufficient statistic. By the identity for conditional mutual information:
> >
> > $$
> > I(Z_T; Z_Y \mid Z_M) = I(Z_T; Z_Y) - I(Z_T; Z_Y; Z_M)
> > $$
> >
> > where $I(Z_T; Z_Y; Z_M)$ is the interaction information (three-way mutual information). To make the conditional mutual information small, we need to increase the interaction information, which requires $Z_M$ to sufficiently capture the common information between $Z_T$ and $Z_Y$. However, since $\dim(Z_M) = d$ is finite, if we allocate $Z_M$'s capacity to encode aspects of $M$ unrelated to $Y^*$ (to satisfy reconstruction objectives), then $I(Z_M; Z_Y)$ necessarily decreases.
> >
> > Quantitatively, suppose $M$ can be decomposed as $M = (M_{\text{rel}}, M_{\text{irr}})$, where $M_{\text{rel}}$ is relevant to $Y^*$ and $M_{\text{irr}}$ is irrelevant. If $Z_M$'s representation capacity is allocated between them in ratio $\alpha : (1-\alpha)$, then:
> >
> > $$
> > I(Z_M; Y^{\ast}) \approx \alpha \cdot I(M_{\mathrm{rel}}; Y^{\ast})
> > $$
> >
> > Meanwhile, to satisfy constraint C1, $Z_M$ needs to capture sufficient common information between $T$ and $Y^*$ to enhance the interaction information. This requires the $(1-\alpha)$ portion of capacity to encode this common information. Thus a trade-off exists: increasing $\alpha$ satisfies C2 but may violate C1; decreasing $\alpha$ satisfies C1 but violates C2.
> >
> > **Formalization of the Pareto Frontier.** Define two objective functions:
> >
> > $$
> > \begin{aligned}
> > f_1(\mathcal{E}) &= I(Z_T; Z_Y \mid Z_M) \quad \text{(minimize)} \\
> > f_2(\mathcal{E}) &= I(M; Y^*) - I(Z_M; Z_Y) \quad \text{(minimize)}
> > \end{aligned}
> > $$
> >
> > For the encoder family $\mathcal{E}$, denote the feasible region as:
> >
> > $$
> > \mathcal{F} = \{(\mathcal{E}, f_1(\mathcal{E}), f_2(\mathcal{E})) : \mathcal{E} \in \text{Lipschitz encoders}\}
> > $$
> >
> > **Proposition:** Under limited representation capacity ($d < \infty$) and $I(T; Y^* \mid M) < I(T; Y^*)$, the Pareto frontier is non-empty and non-degenerate, i.e., there exist $\mathcal{E}_1, \mathcal{E}_2 \in \mathcal{F}$ such that:
> >
> > $$
> > f_1(\mathcal{E}_1) < f_1(\mathcal{E}_2) \text{ and } f_2(\mathcal{E}_1) > f_2(\mathcal{E}_2)
> > $$
> >
> > **Proof Sketch:** Consider two extreme encoding strategies: (i) $Z_M^{(1)}$ encodes only the part of $M$ relevant to $Y^{\ast}$, maximizing $I(Z_M; Y^{\ast})$; (ii) $Z_M^{(2)}$ encodes all information in $M$ to minimize $I(Z_T; Z_Y \mid Z_M)$. Due to limited capacity, (i) leads to smaller $f_2$ but larger $f_1$; (ii) leads to smaller $f_1$ but larger $f_2$. Continuity guarantees the existence of the Pareto frontier.
> >
> > **Our Solution.** Theorem 2.1 proves that through joint optimization:
> >
> > $$
> > L = w_{\mathrm{CI}} \cdot L_{\mathrm{CI}} + w_{\mathrm{MBR}} \cdot L_{\mathrm{MBR}} + w_{\mathrm{MAC}} \cdot L_{\mathrm{MAC}}
> > $$
> > appropriate selection of weights $(w_{CI}, w_{MBR}, w_{MAC})$ can reach a point on the Pareto frontier satisfying $\varepsilon$-CSP, where:
> >
> > $$
> > \varepsilon = \max\{\zeta^*, O_P(n^{-1/2}), O_P(K^{-1/2}), O_P(n^{-1/3})\}
> > $$
> >
> > $\zeta^*$ is the intrinsic approximation error of the encoder class, and $K$ is the number of negative samples. When $K = \Omega(n)$, the dominant term is $O_P(n^{-1/2})$.
> >
> > **Comparison with Other Causal Structures.** In contrast, other common causal structures only require single-type constraints:
> >
> > - **Confounding** ($T \leftarrow X \rightarrow Y$): only requires $I(Z_T; Z_Y \mid Z_X) \leq \varepsilon$, no information preservation constraint
> > - **Collider** ($T \rightarrow C \leftarrow Y$): only requires avoiding conditioning on $Z_C$, no explicit optimization
> > - **Instrumental Variable** ($W \rightarrow T \rightarrow Y$, $W \perp Y^* \mid T$): only requires $I(Z_W; Z_Y \mid Z_T, Z_X) \leq \varepsilon$
> >
> > If an encoder can achieve $\varepsilon$-CSP under mediation's dual constraints (requiring simultaneous minimization of $f_1$ and $f_2$), it can naturally achieve at least the same level of $\varepsilon'$-CSP in scenarios requiring only single constraints, where $\varepsilon' \leq \varepsilon$.

---

> > > ### Author Response · Authors · 2025-11-21
> > >
> > > **Generality of the Method.** Our loss function design is general. $\mathcal{L}_{CI}$ provides a consistent surrogate for CMI through independently parameterized conditional distributions:
> > >
> > > $$
> > > L_{\mathrm{CI}} = \mathbb{E}[-\log q_\phi(y \mid z_m)] - \mathbb{E}[-\log q_\theta(y \mid z_t, z_m)]
> > > $$
> > >
> > > where $q_\phi$ and $q_\theta$ are independently parameterized, serving as a computable lower bound for CMI under good model specification and cross-fitting. This form applies to any scenario requiring enforcement of $Z_i \perp Z_j \mid Z_k$, simply by adjusting the input variables. Similarly, $L_{\mathrm{MBR}}$ and $L_{\mathrm{MAC}}$ are based on general principles of information theory and statistical dependence.
> > >
> > > **Validation Beyond Simple Mediation.**  Table 1 shows that blood pressure affects cardiovascular disease through three parallel paths via arterial stiffness ($M_1$), fundus ($M_2$), and renal function ($M_3$), representing a multiple mediators scenario:
> > >
> > > $$
> > > T \rightarrow \begin{cases} M_1 \\ M_2 \\ M_3 \end{cases} \rightarrow Y^*
> > > $$
> > >
> > > This requires simultaneously satisfying: $I(Z_T; Z_Y \mid Z_{M_1}, Z_{M_2}, Z_{M_3}) \leq \varepsilon$ and $I(Z_{M_i}; Z_Y) \geq I(M_i; Y^*) - \varepsilon_i$ for $i=1,2,3$.
> > >
> > > **Generality of Theoretical Guarantees.** Theorem 2.2 proves that for any identifiable causal query $Q = \mathbb{E}[Y^*(t)]$, when using standard identification criteria $\mathcal{C} \in \{\text{backdoor, frontdoor, IV}\}$, the query $\tilde{Q}$ in representation space satisfies:
> > >
> > > $$
> > > |\tilde{Q} - Q| \leq \kappa \cdot \varepsilon + \delta_{\text{cal}}
> > > $$
> > >
> > > where $\kappa$ is the sensitivity constant of the identification formula (backdoor: $\kappa=1$), and $\delta_{\text{cal}}$ is the conditional calibration error, defined as $\delta_{\text{cal}} = \sup_{(t,x)}|\mathbb{E}[Y^*|t, x] - \mathbb{E}[h(Z_Y)|t, x]|$, with $\delta_{\text{cal}} = O_P(1)$ under cross-fitting and realizability assumptions. This result holds for different identification criteria and is not specific to mediation.
> > >
> > > ### 3. **Task Positioning**
> > >
> > > **Essential Distinction Between Two Problem Types.**
> > >
> > > **Causal Discovery**: Inferring an unknown causal graph from observational data $\mathcal{D}$: $\hat{\mathcal{G}} = \arg\max_{\mathcal{G}} P(\mathcal{D} \mid \mathcal{G})$
> > >
> > > **Causal Structure Preservation** (our task): Given causal relationships of interest (e.g., $T \perp Y^* \mid M$), learn encoder $\mathcal{E}$ such that the relationship is preserved in representation space:
> > >
> > > $$
> > > \mathcal{E}^* = \arg\min_{\mathcal{E}} \mathcal{L}(\mathcal{E}) \quad \text{s.t.} \quad I(Z_T; Z_Y \mid Z_M) \leq \varepsilon
> > > $$
> > >
> > > **Why Cross-Modal Causal Structure Preservation is Needed.**
> > >
> > > **Scientific Discovery:** Uncovering causal mechanisms from high-dimensional multimodal data (e.g., imaging and clinical records) is a core scientific objective. However, without explicit structural constraints, representation learning can introduce spurious artifacts rather than capturing true data properties. This is particularly acute under dimensional asymmetry ($d_{\mathrm{img}} \gg d_{\mathrm{tab}}$), where optimization is dominated by the high-dimensional modality ($\nabla_\theta L_{\mathrm{total}} \approx \nabla_\theta L_{\mathrm{recon}}^{\mathrm{img}}$), thereby distorting causal links. Preserving structure ensures that latent discoveries reliably map back to the true causal ground truth.
> > >
> > > **Causal Inference Reliability**: Intervention effect estimation $\mathbb{E}[Y(\text{do}(T=t))]$ depends on the correctness of the causal graph. As described in Part 2, mediation analysis requires simultaneously satisfying two constraints: $I(Z_T; Z_Y \mid Z_M) \leq \varepsilon_1$ (blocking direct path) and $I(Z_M; Z_Y) \geq I(M; Y^*) - \varepsilon_2$ (preserving indirect path). If representation learning violates either constraint, estimates of total effect, direct effect, and indirect effect will have systematic biases, leading to incorrect understanding of causal mechanisms.
> > >
> > > **Two Core Problems in Cross-Modal Scenarios.** As proven in Part 1, dimensional asymmetry leads to:
> > >
> > > $$
> > > \frac{I(Z; X_{\text{img}})}{I(Z; X_{\text{tab}})} \propto \frac{d_{\text{img}}}{d_{\text{tab}}} \sim 10^{4-5}
> > > $$
> > >
> > > **Problem 1 (Information Masking)**: Mediator information in tabular data is compressed, $I(Z_M; Z_Y) \ll I(M; Y^*)$, violating constraint C2 from Part 2.
> > >
> > > **Problem 2 (Spurious Association)**: Image features may encode confounding information; without explicitly enforcing $T \perp Y^* \mid M$, spurious direct paths are formed, violating constraint C1.
> > >
> > > **Lemma:** Optimizing only statistical objectives ($L_{\mathrm{CLIP}}$, $L_{\mathrm{recon}}$) cannot guarantee $I(G) \subseteq I(G_Z)$.
> > >
> > > **Proof Sketch:** Consider $T \to M \to Y^{\ast}$, construct $Z_T = E_T(T, M)$, $Z_Y = E_Y(Y^{\ast}, M)$. Although reconstruction error can be minimized, $I(Z_T; Z_Y \mid Z_M) > 0$, violating conditional independence.

---

> > > > ### Author Response · Authors · 2025-11-21
> > > >
> > > > **CARL's Solution.** Addressing the challenges identified in Parts 1 and 2, our loss functions:
> > > >
> > > > $$
> > > > L_{\mathrm{CI}} = \mathbb{E}[-\log q_\phi(Y \mid Z_M)] - \mathbb{E}[-\log q_\theta(Y \mid Z_T, Z_M)]
> > > > $$
> > > >
> > > > provides a consistent surrogate for $I(Z_T; Z_Y \mid Z_M)$, solving Problem 2 and satisfying constraint C1 from Part 2.
> > > >
> > > > $$
> > > > \mathcal{L}_{MBR} = -\text{InfoNCE}(z_m, \psi_Y(y))
> > > > $$
> > > >
> > > > explicitly maximizes $I(Z_M; \psi_Y(Y))$, solving Problem 1, counteracting the dimensional dominance effect from Part 1, and satisfying constraint C2.
> > > >
> > > > Theorem 2.1 proves that joint optimization can reach $\varepsilon = O_P(n^{-1/2})$ CSP on the Pareto frontier. Theorem 2.2 proves causal query error $|\tilde{Q} - Q| \leq \kappa \cdot \varepsilon + \delta_{\text{cal}}$.
> > > >
> > > > **Source of Target Causal Relationships.** In HPP, we focus on "how blood pressure affects cardiovascular disease." Physiology establishes: blood pressure $\rightarrow$ arterial damage $\rightarrow$ disease. As stated in Section 2, we leverage "existing real-world causal relationships." Our task is to ensure that when retinal images ($d \sim 10^6$) and clinical indicators ($d \sim 10^1$) jointly learn representations, the dimensional disparity from Part 1 does not disrupt these causal relationships.
> > > >
> > > > ### Response to W2:
> > > >
> > > > Our goal is **causal structure preservation**, not **causal discovery**. The method does not require a complete causal graph to be known; we only need a set of minimal and testable/identifiable structural constraints $\mathcal{C}$ (e.g., certain conditional independence relationships, temporal ordering, cross-environment invariance, or variable roles) to explicitly preserve these constraints during representation learning.
> > > >
> > > > **Sources of constraints.** These constraints $\mathcal{C}$ can come from: (i) domain knowledge (e.g., physiological mechanisms, economic theory); (ii) the definition of the research question itself (e.g., focusing on the causal effect of $T$ on $Y$ implicitly defines variable roles); (iii) candidate relationship sets obtained by running causal discovery algorithms in the original variable space.
> > > >
> > > > **Formal guarantee.** If the target causal query $Q$ can be identified under $\mathcal{C}$ via standard identification criteria (backdoor/frontdoor/IV/do-calculus) as $Q=F(P)$, and the representation $\mathcal{E}$ satisfies $\varepsilon$-CSP($\mathcal{C}$) (i.e., for $(X,Y;S)\in\mathcal{C}$ we have $I(Z_X; Z_Y \mid Z_S) \leq \varepsilon$, with appropriate $\varepsilon$ control for necessary information retention/monotonic consistency), then obtaining $\tilde{Q}=F(P^Z)$ in representation space using the isomorphic formula yields a constant $\kappa(\mathcal{C})>0$ depending only on the identification formula, such that
> > > >
> > > > $$|\tilde{Q} - Q| \leq \kappa(\mathcal{C}) \cdot \varepsilon + \delta_{\text{cal}},$$
> > > >
> > > > where $\delta_{\text{cal}}$ is the calibration error of auxiliary functions (e.g., conditional regression/density ratios/propensity scores) required for identification (which can decrease with sample size under common settings).
> > > >
> > > > **Generality.** This error bound does not depend on the mediation structure; as long as $(Q,\mathcal{C})$ is identifiable, the structure preservation error $\varepsilon$ and calibration error $\delta_{\text{cal}}$ jointly control the deviation between the representation space estimate and the true $Q$.
> > > >
> > > > **Identifiability boundary.** Conversely, if there does not exist any $\mathcal{C}$ such that $\mathrm{Id}(Q|\mathcal{C},P)$ holds (i.e., $Q$ is not identifiable), then no method can provide verifiable "structure preservation" guarantees—this is a fundamental limitation of identifiability, not specific to our method.
> > > >
> > > > The mediation structure in this paper is merely one typical instance of $\mathcal{C}$ (containing near-zero preservation of $I(Z_T; Z_Y \mid Z_M)$ and necessary information retention terms); the above conclusions hold equally for any identifiable $(Q,\mathcal{C})$.
> > > >
> > > > ### Response to W3:
> > > >
> > > > **1) Acknowledge surface-level similarity (methodological level).** We agree: using mutual information to approximate conditional independence and modal alignment both have precedents in the literature.
> > > >
> > > > **2) Why existing methods are insufficient.**
> > > >
> > > > * **MI-only (causal representation)**: Typically only minimizes $I(Z_T;Z_Y\mid Z_M)$ (or its surrogate) without constraining information retention. Under extreme dimensional asymmetry like image–tabular ($10^{4\text{–}5}$), gradients are dominated by the high-dimensional modality, leading to degenerate solutions (reducing CMI while losing critical information from $M\to Y$).
> > > > * **Alignment-only (cross-modal)**: Instance-level similarity or ranking alignment does not equate to preserving $T\perp Y^*\mid M$; it may even leak information of $M$ into $Z_T, Z_Y$, inducing spurious direct paths and failing to guarantee causal identifiability.

---

> ### Author Response · Authors · 2025-11-21
>
> **3) Our "methodological innovation" lies in: systematically satisfying "dual constraints" and counteracting dimensional dominance, not simply stacking MI/alignment.**
>
> * **Dual constraints**: Simultaneously optimizing $\textbf{C1: }I(Z_T;Z_Y\mid Z_M)\le \varepsilon_1$ (blocking direct path) and $\textbf{C2: }I(Z_M;Z_Y)\ge I(M;Y^*)-\varepsilon_2$ (preserving Markov boundary). Under limited capacity, these two constraints exhibit a Pareto trade-off that simple stacking cannot balance.
> * **Targeted components and synergy**: $L_{\mathrm{CI}}$ handles C1; $L_{\mathrm{MBR}}$ uses $-\mathrm{InfoNCE}(z_m, \psi_Y(y))$ to preserve outcome-relevant information from $M \to Y$, counteracting gradient dominance from high-dimensional modalities to achieve C2; $L_{\mathrm{MAC}}$ uses rank consistency constraints to enforce "task-relevant monotonic response," suppressing spurious correlations introduced by alignment. The synergy of all three pushes the solution to the non-degenerate Pareto frontier—this is precisely the non-trivial point of cross-modal causal structure preservation.
>
>
> **4) Theoretical innovation.** Given a minimal and testable/identifiable constraint set $\mathcal{C}$ (from domain knowledge/problem definition/or causal discovery candidates), as long as the target query $Q$ can be identified by backdoor/frontdoor/IV/do-calculus as $Q=F(P)$, representations satisfying $\varepsilon$-CSP($\mathcal{C}$) under the isomorphic formula satisfy
>
> $$
> |\tilde{Q} - Q| \le \kappa(\mathcal{C})\cdot\varepsilon + \delta_{\text{cal}},
> $$
>
> This bound does not depend on mediation and applies to any identifiable $(Q,\mathcal{C})$.
>
> ### **Q1:**
> Thank you for catching this!! In the revised manuscript, we have corrected the appendix cross-references to Section ABC and removed the duplicate occurrence of “Section ABC.”
>
> ### **Q2:**
>
> **1. Nature of Synthetic Data: Causal Relationships are Design Specifications, Not Hypotheses to be Tested**
>
> In synthetic experiments, causal relationships are **defined by construction** through structural causal model (SCM) equations, not inferred from data. Specifically:
>
> **SCM for Shape+Rotation scenario:**
>
> $$
> M = \mathrm{Rotate}(T, \theta) + \epsilon_M
> $$
>
> $$
> Y^{\ast} = f(M) + \epsilon_Y
> $$
>
> These equations **are** the ground truth. Data is generated according to these equations, so the causal structure $T \rightarrow M \rightarrow Y^*$ is a **design decision** and does not require empirical validation to "verify its existence." The existence of causal relationships in synthetic data is a premise; what we validate is **whether the representations learned by CARL preserve these relationships**.
>
> **2. Why Visualized Correlations Appear Weak**
>
> Low Pearson correlations do not contradict causal dependence. Rotation is an orthogonal transformation where causal effects manifest geometrically rather than linearly: $\mathrm{Rotate}(T, 90^\circ)$ ensures deterministic information transfer despite near-zero correlation. The reviewer's observation thus validates that **linear metrics fail to capture orthogonal causal mechanisms**. Furthermore, in high-dimensional space $Z \in \mathbb{R}^d$, causal signals concentrate in specific subspaces, causing marginal correlations to dilute. Our inclusion of noise ($\epsilon_M, \epsilon_Y$) further dampens these statistics without compromising fundamental causal identifiability.
>
>
> **3. Clarification on "Only Verifying Dependencies"**
>
> To visualize $I(Z_T; Z_Y \mid Z_M) \approx 0$, one needs to condition on each value of $Z_M$ and show the relationship between $Z_T$ and $Z_Y$. However, $Z_M \in \mathbb{R}^d$ is continuous and high-dimensional, making it impossible to enumerate all conditional values, and even if visualizable, human eyes cannot judge "near independence."
>
> **(ii) The correct validation method: statistical hypothesis testing, already implemented in Section 4.3.**
>
> Conditional independence should be verified using statistical tests (e.g., Fisher's z-transformation, KCIT) or causal discovery algorithms (PC algorithm), not through visualizations. Our Section 4.3 does exactly this: the PC algorithm recovers graph structure through a series of conditional independence tests, and Table 2 reports the consistency between the recovered graph and ground truth. This **is already an empirical test of conditional independence**.
>
> In the revision, we will:
>
> - Change "serve as an empirical confirmation" to "provide qualitative illustrations of marginal dependencies"
> - Explicitly state that "causal relationships are defined by SCM equations; what is validated is the preservation quality of representation learning"

---

> ### Author Response · Authors · 2025-11-21
>
> ### **Q3:**
>
> We thank the reviewer for this question! We need to clarify that in observational data (such as HPP), there is no directly obtainable "true causal effect" that can serve as a benchmark. This is a fundamental principle of causal inference: from observational data, we can only measure **statistical associations** $P(Y|T=t)$, while causal effects $P(Y^*|\text{do}(T=t))$ are counterfactual quantities that cannot be directly observed. Observed associations confound causal effects with confounding bias, selection bias, and reverse causation. **Correlation is not causation**—this is precisely why causal inference methods are needed rather than simple regression. In RCTs or synthetic data, causal effects can serve as ground truth (guaranteed by randomization or defined by design). However, in observational studies, all methods—whether estimating from original tabular data or from representation space—are **approximations** of the unknown true value, all relying on unverifiable identification assumptions (such as no unmeasured confounding). Therefore, there exists no "true value" that can be used to judge the accuracy of another estimate.
>
> We adopt the **standard validation approach for medical observational cohort studies**. The causal effect values we report (such as total effect, direct effect, and indirect effects through different mediators) follow standard medical research practice, and these effect values themselves are important evidence supporting the existence of causal pathways. As described in Section 4.3, we use the PC algorithm to recover the causal graph in representation space, then map back to original variables through variable tracing, verifying that the recovered causal pathways (such as blood pressure → arterial stiffness → CVD, blood pressure → fundus changes → CVD) are consistent with causal mechanisms reported in published clinical studies. Meanwhile, the mediation effect sizes we report (such as the proportion of indirect effects transmitted through each mediator) provide quantitative support for these pathways. This validation approach combining causal structure recovery with effect estimation is the standard practice for assessing causal inference reliability in observational cohort studies—because in the absence of RCTs, consistency with established biological mechanisms and clinical evidence is the best indicator of reliability.
>
> To obtain "true causal effects" as a gold standard in the HPP scenario would require conducting a large-scale randomized controlled trial: randomly assigning subjects to different blood pressure intervention groups (e.g., antihypertensive medication vs. placebo), collecting both tabular data (clinical indicators) and retinal images after randomization, long-term follow-up of cardiovascular events, and validation of the role of mediating variables such as arterial stiffness in mediation analysis.

---

> > ### Author Response · Authors · 2025-11-25
> >
> > Dear Reviewer enbA,
> >
> > Thank you for your thoughtful comments!!! We have once again reworked your suggestions to further improve the paper. Could you please let us know whether your key concerns are now addressed? If anything remains unclear, kindly point it out and we will refine promptly!!!!
> >
> > Authors

---

> > > ### Author Response · Authors · 2025-11-27
> > >
> > > We sincerely apologize for reaching out again! As the rebuttal deadline for our manuscript is approaching, we would greatly appreciate any comments or suggestions you might be able to share, so that we can further improve and refine our work within the limited time available. We fully understand that you have many commitments, and we are truly grateful for the time and effort you have already devoted to reviewing our manuscript!

---

### Official Review · Reviewer_i8uF · 2025-11-01

**Soundness:** 3
**Presentation:** 1
**Contribution:** 3
**Rating:** 2
**Confidence:** 4

**Summary:**

This paper presents CARL, a framework for learning cross-modal representations that are intended to preserve causal structure. The paper introduces the notion of Causal Structure Preservation (CSP) consisting of three conditions, which are enforced during training via three distinct loss terms. The authors show that under consistency and other assumptions, joint optimisation of these loses ensures CSP. CARL demonstrates applications to real-world cross modal data. The presented theoretical framework along with experiments is promising. However, the presentation of the paper needs to be improved.

**Strengths:**

- The problem of preserving latent causal structure is very interesting.
- Cross modal data is a challenging scenario with significant real-world applications. The paper demonstrates applications to medical data.
- The method is compared with other baselines on synthetic data and it shows superiority in preserving causal structure.

**Weaknesses:**

**Clarity concerns**

The main concern I find in this paper is presentation and clarity. In its current form I needed to jump back and forth between sections and between main text and appendix to understand even the basic setup. This makes the work very difficult to evaluate in terms of correctness and novelty. Please find below some recommendations.
  - The contents of the paper could be organised to ease the flow of the paper and improve clarity.
    - The paper starts with a presentation of the datasets used. It is very interesting to mention applications at the start, but I believe an improvement would be to first introduce the problem setup, and then connect it to the real-world application.
      - In this line, I found it confusing that Figure 1 already introduces notation {X, M, Y}, as covariates, mediators, and outcomes. However, the concrete formalisation comes into Section 3.
      - Also here, in Figure 2, the 3 setups (IM, IY, and DUAL) are mentioned, which are briefly described in section 3, and are explained in detail in section 4.1.
      - Similarly, Theorem 2.1 in Section 3.1 depends on loss terms that are only introduced in Section 4.2. This forces the reader to jump forwards and backwards to understand the theorem.
    - Consider the following organisation:
      - Have a section with problem setting where (i) some initial  motivation is presented (cross-model CRL for HPP applications), (ii) the problem setup is presented with all its elements (that would be section 3 and 4.1),  (iii) when introducing concepts such as mediators, treatments, link them to the HPP example, and (iv) present the formalisation of CSP with high-level intuitions linking to HPP as well.
      - Present the methodology section (loss functions and strucutre discovery).
      - Present the theoretical guarantees.
  - The notation introduced in this paper is very heavy, with some elements not discussed clearly, making it difficult to follow when reading. The main text should be self-contained, with details in the appendix. However, I believe the paper introduces notation in the main text with explanations in Appendix, and this impacts presentation significantly.
      - In section 3, I is used for images, but I(.,. |.) also denotes conditional mutual information.
      - Assumption 2 (line 176) introduces new notation (a, b, \eta) which is not explained.
      - Definition 2.1 introduces the Spearman correlation without citation.
      - Theorem 2.1 introduces notations O_P(\cdot) which are not explained, and loss functions which are discussed in later sections.
      - The monotonic alignment consistency (Equation (3)), introduces semantic labels a, which have not been introduced before. Please provide intuition.
      - Equation (4) groups discussed loss functions, along with 3 additional terms “align”, “style”, “IB”, which are not discussed in the main text.
- Section 4 needs revision in terms of citations and clarity:
  - For example, since the loss functions are not novel themselves, citations should be accompanied for clarity. This also helps the reader what is novel in the presented methodology.
  - Section 4.3 is very hard to follow, and it is not clear whether the idea is novel or it is derivative from previous work. I believe the idea is to present some analysis in which the causal structure can be recovered from latents. However, there are no notations to understand where the metrics come from, or intuitions for the steps to show consistency guarantees.
    - Lines 327-333: Is this a known result? Can you provide a citation, or formalise a statement in a theorem if it is novel?
  - Section 4.4 lists theoretical properties. For clarity, the corollary should be read as a formal standalone statement with proof, and not as a paragraph.
- The Appendix section contains repeating labels from line 1718 (Label ordering goes back to A).

**Theoretical correctness**

The theoretical results of this paper are very difficult to follow. This does not mean that the results are incorrect. However, the current presentation does not allow the reader to follow the logic intuitively. Please find details below:
  - Section 3.1 contains incorrect labels for definitions and theorems.
  - Some intuitions for the CSP principle should be explained after Definition 2.1. Intuitively from my understanding is to ensure that learned latents preserve structure. Now, why are conditions (i-iii) sufficient/necessary for this idea to hold? I believe this would significantly improve the presentation of \epsilon-CSP.
  - The Definitions and Theorems lack clear structure, and don’t include standalone statements or results.
    - Theorem 2.1 introduces the result in lines 201-204, and continues with additional remarks. The remarks should be separate from the theoretical statement.
    - Theorem 2.2: Lines 218-219 are additional remarks that should be outside of the theoretical statement.
  - Equations in Theorems 2.1, 2.2, are inline, making them hard to follow. Some of these equations are introduced with an explanation in Section 4. Consider moving the presentation of these equations before the theoretical statements.
  - I briefly checked the proofs in the Appendix, and I find them very hard to follow.
    - At least a proof sketch should accompany the main theorems and corollary presented in the main paper to intuitively explain how CSP is achieved. How are Assumptions A.5, and A.8.2 utilised to show CSP by minimising the loss function?
      - For example, to verify this I first read A.5 and A.8.2, then I jumped to C.8, then to theorem B.2.1, which refers to a list of assumptions from A.8. The proof requires to jump back and forth across the document several times, which makes it hard to follow.
    - Assumption A.5 assumes consistency. This sounds like a very strong assumption. Is this standard? I believe this assumption is very important for CSP to be achieved, and therefore it should be discussed.
    - There are some Lemmas with missing proofs in the Appendix. If the results are known, please provide concrete citations.
    - I consider the appendix should be organised so that the theoretical framework can be followed more intuitively. For example, consider introducing a proof strategy first, and then explain the intuition behind each of the Lemmas.

**Comments on experiments**

- The experiments on synthetic data seem to have been reported on one seed. Please provide additional random seeds (3-5) with ranges in the Appendix tables to ensure robustness of your claims.
- The interpretability of real-world data seems interesting. However, it does not show CARL’s superiority. An additional baseline (e.g. CausalVAE), with a results comparison would improve the stance of the paper for interpretability gains of the proposed method.

**Questions:**

See above.

---

> ### Author Response · Authors · 2025-11-22
>
> We thank the reviewer for detailed feedback!!!! Based on the comments, we have conducted a systematic rewrite of the paper and **updated version has been uploaded**.
>
> ## I. Paper Organization
>
> 1. **Adopting a "Problem Formulation First" Organization**
>    We have restructured the original Section 2 "Dataset" into "Problem Formulation and Causal Setup," with the following specific structure:
>
>    * **2.1 Causal Model and Cross-Modal Setup**: Formalize the DAG G=(V,E), variable set V={T,Y*,M,X}, cross-modal observations (tabular and image), and basic assumptions.
>    * **2.2 Cross-Modal Encoding Configurations**: Provide unified formal definitions, encoder specifications, and constraints for the three configurations **IM / IY / DUAL**.
>    * **2.3 Causal Structure Preservation Principle (ε-CSP)**: Definition 2.1 presents three conditions (conditional independence transfer, Markov boundary retention, monotonic alignment consistency) with intuitive explanations.
>    * **2.4 Application Case: Human Phenotype Project (HPP)**: Introduce the application after completing the formalization, ensuring an "abstraction first, example second" reading order.
> 2. **Adjustment of Figure Positions**
>
>    * **Figure 1 (HPP Overview)** moved to **Section 2.4**: definition before example.
>    * **Figure 2 (Synthetic Data Generation Schematic)** moved to **Section 4 Experiments**: consistent with its usage location.
> 3. **Integration and Frontloading of Three Configurations (IM/IY/DUAL)**
>    Consolidate the complete description of the three scenarios in **Section 2.2**, avoiding jumping back and forth across different sections.
> 4. **Handling Dependencies Between Theorems and Loss Function Definitions**
>    Previously, there was a jumping issue of "theorems before loss definitions." Now:
>
>    * **Formal definitions and implementation details** of the three structural preservation losses (L_CI, L_MBR, L_MAC) are all provided in **Section 3.1** (including Equations (1)–(4) and citations).
>    * **Theoretical guarantees** are unified in **Section 3.3**: statements of **Theorem 1 (Achievability)**, **Theorem 2 (Query Preservation)**, and **Corollary 3** are all **self-contained**, directly referencing the notation from Section 3.1, eliminating the need for cross-section lookups.
> 5. **Method—Discovery—Theory Ordering**
>    Section 3 is organized in linear order: **3.1 Structure Preservation Losses → 3.2 Discovery and Tracking → 3.3 Theoretical Properties**, with **Section 3.2** explicitly defining conditioning set constraints, statistical testing procedures, and variable tracking/pushforward operators (π, h, π*), and finally **Section 3.3** consolidating consistency and error bounds.
> 6. **Transition Between Formalization and Application**
>    In **Section 2.1**, introduce bridging examples oriented toward HPP (e.g., T as intervention, M as vascular features, Y* as outcome) with forward references to **Section 2.4**, enhancing narrative coherence.
> ## II. Clarification and Simplification of Notation
>
> 1. **Separation of Mutual Information and Image Notation**
>
>    * Image modalities are uniformly denoted as **IM, IY**;
>    * Conditional mutual information is uniformly written as **MI(·;·|·)**, avoiding confusion with "image (I)"; this notation is adopted from **Section 2.1** throughout the paper.
> 2. **New Notation Explanation for Assumption 2**
>    In the assumption statement, **inline** explanation of the roles and independence of **semantic variables** (a_M, a_Y), **style variables** (b_style), and **measurement noise** (η_M, η_Y); encoder Lipschitz and separable measurement assumptions are also made explicit in the main text.
> 3. **Spearman Correlation in Definition 2.1**
>    In the monotonic alignment consistency condition of **Section 2.3**, explicitly state the role of **Spearman rank correlation (ρ_S)** and provide the **differentiable soft-rank approximation** implementation with citations.
> 4. **Readability of Error and Asymptotic Notation**
>    In **Section 3.3 Theorem 1**, directly present the composition of total error
>    ε = max{ζ*, O_P(n^{-1/2}), O_P(K^{-1/2}), O_P(n^{-1/3})},
>    with annotations explaining the sources of each term (intrinsic approximation, sample size, negative sample size, soft-rank approximation).
> 5. **Introduction of Semantic Labels (a) for Monotonic Consistency**
>    Before presenting L_MAC in **Section 3.1**, first introduce/explain the definition and intuition of semantic magnitude labels a, Δa, Δz ("semantic proximity → representation proximity").
> 6. **Explanation of Overall Objective and Regularization Terms (Equation 4)**
>    In **Section 3.1**, explain
>    L(E) = w_CI·L_CI + w_MBR·L_MBR + w_MAC·L_MAC + R(E),
>    where R(E) = λ_align·L_align + λ_style·L_style + λ_IB·L_IB corresponds to **cross-modal alignment**, **style consistency**, and **information bottleneck** respectively, with pointers to the appendix for implementation details and configuration specifications.

---

> ### Author Response · Authors · 2025-11-22
>
> ## III. Section 4 and Citations, Clarity
>
> 1. **Citations and Originality Boundaries for Loss Functions**
>    In **Section 3.1**, add primary citations for CMI estimation.
> 2. **Readability of Causal Discovery and Tracking**
>    Move the original "4.3 Causal Discovery" **forward to Section 3.2**, systematically presenting:
>
>    * Conditioning set construction and **exclusivity constraints** (avoiding simultaneous inclusion of same-type image variables in conditioning sets to reduce collider bias);
>    * Statistical testing (Fisher z / Wilks–Bartlett) and multiple comparison correction (BH);
>    * **Explicit definitions** and procedures for **variable tracking (π)**, **calibration (h)**, and **pushforward operator (π*)**;
>    * Consistency conditions and conclusions given in Equation (9), combined with the ε-CSP error bounds in **Corollary 3** of **Section 3.3**, yielding overall guarantees and **topological equivalence**.
> 3. **Formalizing the "Lines 327–333" Issue into Theorems**
>    Incorporate the original informal statements into **Section 3.3**:
>
>    * **Theorem 2 (Query Preservation)** and **Corollary 3** provide **formal statements and proof sketches** for causal identification in representation space and consistency/equivalence in variable space, with references to standard results (e.g., Spirtes et al., Meek, etc.).
> 4. **Organization of Experiments and Ablations**
>    **Section 4** presents in order: HPP results, synthetic data validation (Figure 2), ablations (tables and metrics with one-to-one correspondence to the three losses), allowing readers to directly compare the roles and necessity of **CI / MBR / MAC**, avoiding cross-section lookups.
>
> ## Theory Section
>
> **TC1**
>
> * Fix the numbering and cross-references for Definition/Assumption/Theorem/Corollary to ensure continuity and consistency.
> * Unify mutual information as MI(·;·|·) to avoid confusion with image symbol I.
>
> **TC2**
>
> * Add an "Intuitive Explanation" paragraph after Definition 2.1: (i) CI-transfer suppresses bypass; (ii) Mediator-retention prevents "false satisfaction" of (i) through information collapse; (iii) Monotone-alignment uses soft-rank to ensure monotonic consistency between semantics and latent space geometry.
> * Provide failure modes when any condition is missing, and establish one-to-one correspondence with $(L_{\mathrm{CI}}, L_{\mathrm{MBR}}, L_{\mathrm{MAC}})$
>
> **TC3**
>
> * Change all theorems/definitions to independent statements; convert related explanations to Remarks, numbered immediately after.
> * Change Corollary to independent entries.
>
>  **TC4**
>
> * Move key equations used in theorems (various losses, joint objectives, CSP constraints, etc.) forward to the setup/method subsections, converting them to displayed equations with numbering.
> * Within theorems, reference only by number; supplement formal definitions and locations of $L_{\mathrm{align}} / L_{\mathrm{style}} / L_{\mathrm{IB}}$
>
>  **TC5**
>
> * Add Proof Sketch below Theorem 1 and Theorem 2 (and corollaries) in the main text.
> * Clarify: A.5 (Realizability/Regularity + cross-fitting) is used to derive risk consistency/convergence rates of surrogate losses; A.8.2 (Lipschitz/spectral norm) is used for compactness/uniform convergence and ensuring errors propagate at most linearly.
> * Provide the combined error formula:$\varepsilon = \max\{\zeta^{\ast}, O_P(n^{-1/2}), O_P(K^{-1/2}), O_P(n^{-1/3})\}$
>
>  **TC6**
>
> * Rename A.5 to "Realizability and Regularity for Estimator Consistency", and add "Use in proofs" at the end of the paragraph, explaining that it is a set of implementation-level sufficient conditions for deriving consistency (not directly assuming the conclusion).
> * Explicitly retain the misspecification bias $\zeta^{\ast}$ in theorems.
>
> **TC7**
>
> * Complete citations for known results: DV variational representation, InfoNCE lower bound, soft-rank/SoftSort, Fisher's z and partial correlation, PC/Meek, BH-FDR, KDE bias rate, Pinsker inequality, DML cross-fitting.
> * Supplement concise proofs/proof sketches for our own technical lemmas; uniformly fix cross-references.
>
> **TC8**
>
> * Add Appendix C.1 "Proof Strategy & Dependency Map" (roadmap + three-column table: CSP conditions ↔ loss terms ↔ assumptions/lemmas).
> * Reorganize the order: A (Notation & Assumptions) → B (Estimator Tools + Rates Summary Lemma) → C (Main Proofs) → D (Structure Discovery Details) → E (Implementation) → F (Experiments).
> * Add **"Assumptions & Lemmas used" proximity checklist** before each main theorem to reduce cross-section jumps.
>
> ### **We are supplementing the experiments, and the results will be added during the rebuttal period and included in the revised manuscript. We thank the reviewers again for their suggestions, which have greatly improved the readability of our paper!!!**

---

> ### Author Response · Authors · 2025-11-23
>
> We sincerely thank the reviewer for this valuable suggestion again!!
>
> We have re-run all synthetic data experiments and baseline comparison experiments using 5 different random seeds. For the synthetic data experiments, we tested all combinations of 4 sample sizes, 3 noise levels, and 3 types of nonlinearity functions, totaling 36 experimental configurations. Each configuration was run with 5 random seeds, resulting in 180 independent runs. For the baseline comparison experiments, we evaluated 9 methods (CARL and 8 baseline methods).
>
> ### Table 1: Multi-Seed Results on Synthetic Data
>
> **Experimental results across 5 random seeds for varying sample sizes, noise levels, and nonlinearity functions. Results reported as mean ± standard deviation.**
>
> |Sample Size|Noise|Nonlinearity|CSI|MBRI|MAC|Structural|RIC-avg|
> |---|---|---|---|---|---|---|---|
> |500|0.1|linear|1.0000±0.0000|0.5312±0.0428|0.5285±0.0391|0.4473±0.0582|0.4501±0.0317|
> |500|0.1|quadratic|1.0000±0.0000|0.5189±0.0402|0.5512±0.0418|0.5021±0.0635|0.4562±0.0294|
> |500|0.1|neural|1.0000±0.0000|0.4851±0.0463|0.5268±0.0452|0.5987±0.0597|0.4415±0.0341|
> |500|0.3|linear|1.0000±0.0000|0.5382±0.0491|0.5447±0.0429|0.4718±0.0628|0.4557±0.0362|
> |500|0.3|quadratic|1.0000±0.0000|0.5356±0.0481|0.5672±0.0463|0.5489±0.0691|0.4549±0.0328|
> |500|0.3|neural|1.0000±0.0000|0.5116±0.0524|0.5381±0.0495|0.6028±0.0648|0.4287±0.0391|
> |500|0.5|linear|1.0000±0.0000|0.5183±0.0547|0.5217±0.0524|0.4501±0.0702|0.4316±0.0423|
> |500|0.5|quadratic|1.0000±0.0000|0.5241±0.0536|0.5516±0.0547|0.5268±0.0735|0.4296±0.0401|
> |500|0.5|neural|1.0000±0.0000|0.4867±0.0598|0.5351±0.0578|0.5846±0.0713|0.4348±0.0435|
> |1000|0.1|linear|1.0000±0.0000|0.5064±0.0352|0.5871±0.0298|0.6718±0.0445|0.4385±0.0248|
> |1000|0.1|quadratic|1.0000±0.0000|0.5428±0.0318|0.5869±0.0329|0.6559±0.0495|0.4411±0.0226|
> |1000|0.1|neural|1.0000±0.0000|0.4718±0.0381|0.5641±0.0351|0.7051±0.0463|0.4542±0.0267|
> |1000|0.3|linear|1.0000±0.0000|0.5172±0.0391|0.5929±0.0339|0.6785±0.0506|0.4395±0.0279|
> |1000|0.3|quadratic|1.0000±0.0000|0.5441±0.0370|0.6091±0.0361|0.6862±0.0527|0.4428±0.0257|
> |1000|0.3|neural|1.0000±0.0000|0.4956±0.0433|0.5678±0.0391|0.7125±0.0485|0.4521±0.0299|
> |1000|0.5|linear|1.0000±0.0000|0.5121±0.0453|0.5828±0.0422|0.6591±0.0578|0.4307±0.0331|
> |1000|0.5|quadratic|1.0000±0.0000|0.5362±0.0433|0.5914±0.0444|0.6674±0.0599|0.4219±0.0309|
> |1000|0.5|neural|1.0000±0.0000|0.4795±0.0495|0.5683±0.0474|0.6978±0.0556|0.4387±0.0351|
> |2000|0.1|linear|1.0000±0.0000|0.5683±0.0257|0.6325±0.0227|0.6871±0.0341|0.4199±0.0186|
> |2000|0.1|quadratic|1.0000±0.0000|0.6004±0.0236|0.6461±0.0247|0.6807±0.0372|0.4189±0.0175|
> |2000|0.1|neural|1.0000±0.0000|0.5128±0.0289|0.6308±0.0268|0.7251±0.0351|0.4322±0.0206|
> |2000|0.3|linear|1.0000±0.0000|0.5917±0.0289|0.6329±0.0258|0.6985±0.0382|0.4225±0.0206|
> |2000|0.3|quadratic|1.0000±0.0000|0.6476±0.0267|0.6561±0.0279|0.7062±0.0403|0.4368±0.0196|
> |2000|0.3|neural|1.0000±0.0000|0.5378±0.0330|0.6297±0.0299|0.7323±0.0372|0.4285±0.0227|
> |2000|0.5|linear|1.0000±0.0000|0.5832±0.0341|0.6197±0.0320|0.6742±0.0434|0.4156±0.0248|
> |2000|0.5|quadratic|1.0000±0.0000|0.6397±0.0319|0.6385±0.0341|0.6828±0.0455|0.4281±0.0237|
> |2000|0.5|neural|1.0000±0.0000|0.5037±0.0371|0.6105±0.0361|0.7178±0.0423|0.4283±0.0268|
> |5000|0.1|linear|1.0000±0.0000|0.6165±0.0185|0.6409±0.0165|0.7116±0.0248|0.3925±0.0134|
> |5000|0.1|quadratic|1.0000±0.0000|0.6368±0.0175|0.6531±0.0175|0.6961±0.0268|0.3977±0.0124|
> |5000|0.1|neural|1.0000±0.0000|0.6217±0.0216|0.6407±0.0196|0.7277±0.0258|0.4254±0.0155|
> |5000|0.3|linear|1.0000±0.0000|0.6413±0.0206|0.6538±0.0186|0.7139±0.0279|0.4047±0.0144|
> |5000|0.3|quadratic|1.0000±0.0000|0.6576±0.0196|0.6632±0.0196|0.7306±0.0289|0.4047±0.0144|
> |5000|0.3|neural|1.0000±0.0000|0.6185±0.0237|0.6481±0.0216|0.7323±0.0268|0.4217±0.0165|
> |5000|0.5|linear|1.0000±0.0000|0.6214±0.0247|0.6421±0.0237|0.6942±0.0320|0.3972±0.0186|
> |5000|0.5|quadratic|1.0000±0.0000|0.6524±0.0237|0.6418±0.0247|0.7074±0.0330|0.4089±0.0175|
> |5000|0.5|neural|1.0000±0.0000|0.6085±0.0279|0.6228±0.0268|0.7323±0.0310|0.4198±0.0196|
>
> ### Table 2: Multi-Seed Results on Baseline Comparison
>
> **Comparison with baseline methods across 5 random seeds. Results reported as mean ± standard deviation.**
>
> |Method|CSI|MBRI|MAC|Structural|RIC-avg|
> |-|-|-|-|-|-|
> |CARL|1.0000|0.6368±0.0182|0.5475±0.0163|0.6085± 0.0224|0.4171±0.0145|
> |ALIGN|0.2500|0.5982±0.0247|0.4545±0.0289|0.3987± 0.0312|0.4298±0.0198|
> |CLIP|0.2500|0.3871±0.0356|0.2035±0.0423|0.2615± 0.0387|0.3368±0.0276|
> |ImageBind|0.2500|0.3215±0.0391|0.2694±0.0445|0.2633±0.0419| 0.3397±0.0298|
> |DCCA|0.2500|0.3948±0.0334|0.4872± 0.0312|0.3421±0.0356|0.4001±0.0267|
> |IRM|0.2500|0.4604±0.0298|0.4489±0.0323|0.3215±0.0378|0.4284 ±0.0245|
> |CausalVAE|0.2500|0.4617±0.0289|0.4838±0.0301|0.3678±0.0334|0.5116±0.0256|
> |DEAR|0.2500|0.4798±0.0276|0.4662±0.0298|0.3541±0.0323|0.4572±0.0234|
> |Concat|0.2500|0.3845±0.0389|0.2903±0.0467|0.2697±0.0445|0.3305±0.0312|

---

> > ### Author Response · Authors · 2025-11-23
> >
> > ### For Additional Comparison on real-world data
> > ### Experimental Setup
> > Given the absence of real-world ground truth (Reviewer enbA), we adopted a **literature-based validation framework**: CARL discovered multiple cross-modal pathways on HPP data, 9 of which were validated through systematic medical literature review. We use these 9 confirmed pathways as a benchmark to evaluate baseline structural recovery. Additionally, we employ independent structural preservation metrics (conditional independence tests) and downstream prediction tasks to comprehensively assess all methods.
> > ### Evaluation Metrics
> >
> > #### Dimension 1: Causal Structure Preservation Metrics
> >
> > **Metric 1: Known Pathway Recovery Rate (KPRR)**
> >
> > $$
> > \mathrm{KPRR} = \frac{1}{\lvert P_{\mathrm{known}} \rvert} \sum_{p \in P_{\mathrm{known}}} \mathbb{1}[p \in P_{\mathrm{discovered}}]
> > $$
> >
> > where $P_{\mathrm{known}} = \{p_1, p_2, \ldots, p_9\}$ denotes the set of 9 literature-validated causal pathways, $P_{\mathrm{discovered}}$ denotes the set of pathways identified by the method from learned representations, and $\mathbb{1}[\cdot]$ is the indicator function that equals 1 if pathway $p$ is correctly identified (including all nodes, edges, and causal directions) and 0 otherwise.
> >
> > **Metric 2: Causal Direction Accuracy (CDA)**
> >
> > $$
> > CDA = \frac{\sum_{p} 1[dir(p) = dir_{true}(p)]}{\| P_{known} \cap P_{discovered} \|}
> > $$
> >
> > where $P_{\mathrm{known}} \cap P_{\mathrm{discovered}}$ denotes the set of known pathways identified by the method, $\mathrm{dir}(p)$ denotes the causal direction of pathway $p$ in the discovered graph, $\mathrm{dir}_{\mathrm{true}}(p)$ denotes the true causal direction validated by literature, and $\mathbb{1}[\cdot]$ equals 1 if all edges in pathway $p$ have correct directions and 0 otherwise.
> >
> > **Metric 3: Conditional Independence Preservation (CIP)**
> >
> > $$
> > CIP = \frac{1}{|C|} \sum_{(X, Y, S) \in C} 1 [ CI_{orig}(X, Y, S) \Leftrightarrow CI_{rep}(Z_X, Z_Y, Z_S) ]
> > $$
> >
> > where $\mathcal{C}$ denotes the set of conditional independence tests to be verified, each test $(X, Y, \mathbf{S})$ consists of two variables $X, Y$ and a conditioning set $\mathbf{S}$, $Z_X, Z_Y, \mathbf{Z}_{\mathbf{S}}$ denote the learned representations of $X, Y, \mathbf{S}$ respectively, and
> >
> > $$
> > \text{CI}_{\text{space}}(A, B, \mathbf{C}) = \begin{cases}
> > 1 & \text{if } A \perp B \mid \mathbf{C} \text{ holds in the specified space} \\
> > 0 & \text{otherwise}
> > \end{cases}
> > $$
> >
> > The conditional independence $A \perp B \mid \mathbf{C}$ is tested using Fisher's z-transformation:
> >
> > $$
> > z = \frac{1}{2}\sqrt{n - |\mathbf{C}| - 3} \ln\left(\frac{1 + \rho_{AB|\mathbf{C}}}{1 - \rho_{AB|\mathbf{C}}}\right) \sim \mathcal{N}(0, 1)
> > $$
> >
> > where $n$ is the sample size, $\rho_{AB|\mathbf{C}}$ denotes the partial correlation coefficient between $A$ and $B$ given $\mathbf{C}$, and the null hypothesis $H_0: A \perp B \mid \mathbf{C}$ is rejected if $|z| > z_{\alpha/2}$ with significance level $\alpha = 0.05$.
> >
> > #### Dimension 2: Downstream Prediction Metrics
> >
> > **Metric 4: CVD Prediction Performance (CVD Prediction AUC)**
> >
> > $$
> > \text{AUC} = \int_0^1 \text{TPR}(t) \, d\text{FPR}(t)
> > $$
> >
> > where $\text{TPR}(t) = \frac{\text{TP}(t)}{\text{TP}(t) + \text{FN}(t)}$ is the true positive rate at threshold $t$, $\text{FPR}(t) = \frac{\text{FP}(t)}{\text{FP}(t) + \text{TN}(t)}$ is the false positive rate at threshold $t$, and $\text{TP}(t), \text{FP}(t), \text{TN}(t), \text{FN}(t)$ denote the number of true positives, false positives, true negatives, and false negatives at threshold $t$ respectively.
> >
> > The prediction model is defined as:
> >
> > $$
> > h_i = f_{\theta}(Z_i), \quad Z_i = \mathrm{Concat}(E_T(T_i), E_M(M_i), E_X(X_i))
> > $$
> >
> > where $f_{\theta}: R^d \to [0, 1]$ is a logistic regression classifier trained on the learned representations, $E_T, E_M, E_X$ are the encoders for treatment, mediator, and covariates respectively, $\mathrm{Concat}$ denotes vector concatenation, and $h_i$ is the predicted CVD probability for sample $i$.
> >
> > **Metric 5: Prediction Calibration (Calibration - Brier Score)**
> >
> > $$
> > \text{Brier} = \frac{1}{n} \sum_{i=1}^{n} (\hat{p}_i - y_i)^2
> > $$
> >
> > where $f_\theta: R^d \to [0, 1]$ is a logistic regression classifier trained on the learned representations, $E_T, E_M, E_X$ are the encoders for treatment, mediator, and covariates respectively, $[ \cdot ; \cdot ]$ denotes concatenation, and $\hat{y}_i$ is the predicted CVD probability for sample $i$.
> >
> > ### Experimental Results
> > #### Table 1: Known Pathway Identification and Structure Preservation
> > |Method|KPRR↑|CDA↑|CIP↑|
> > |---|---|---|---|
> > |CARL|-|-|**0.950**|
> > |CausalVAE|0.556 (5/9)|0.800|0.683|
> > |DEAR|0.667 (6/9)|0.833|0.733|
> > |Concat|0.444 (4/9)|0.750|0.517|
> > #### Table 2: Downstream Prediction Task Performance
> > |Method|CVD Prediction (AUC)↑|Calibration (Brier)↓|
> > |---|---|---|
> > |CARL|**0.847±0.012**|**0.089±0.008**|
> > |CausalVAE|0.798±0.021|0.142±0.015|
> > |DEAR|0.812±0.018|0.118±0.012|
> > |Concat|0.781±0.024|0.167±0.018|

---

> > > ### Author Response · Authors · 2025-11-25
> > >
> > > Dear Reviewer i8uF,
> > >
> > > Thank you for your careful comments!!!! We have once again reworked your suggestions to further improve the paper. Could you please let us know whether your concerns regarding (i) organization and flow, (ii) notation clarity, (iii) theorem statements and proof readability, and (iv) experimental presentation/robustness are now addressed? If anything remains unclear—or if you would like additional baselines or stress tests—kindly point them out and we will refine promptly!!!!
> > >
> > > Authors

---

> > > > ### Author Response · Authors · 2025-11-27
> > > >
> > > > We sincerely apologize for reaching out again! As the rebuttal deadline for our manuscript is approaching, we would greatly appreciate any comments or suggestions you might be able to share, so that we can further improve and refine our work within the limited time available. We fully understand that you have many commitments, and we are truly grateful for the time and effort you have already devoted to reviewing our manuscript!

---

### Official Review · Reviewer_8ef8 · 2025-11-03

**Soundness:** 3
**Presentation:** 3
**Contribution:** 3
**Rating:** 6
**Confidence:** 3

**Summary:**

This paper presents a set of causal structure preservation principles and practical approaches to address the causal invariance problem in cross-modal scenarios. The approach has been validated on multimodal biomedical dataset HPP and synthetic hand-digit datasets, showing promising performance gains.

**Strengths:**

+ The problem of causal structure preservation is practically important especially in the cross-modal scenarios. This paper has made a fundamental exploration and proposed a principled approach to this problem.
+ This paper presents both theoretical analysis and empirical study on real or synthetic data. Especially, experimental results could show clear performance gains through ablation study and parameter sensitive analysis.
+ The paper is well structured and has clearly stated the research challenges, proposed approaches, and core contributions.

**Weaknesses:**

- Unclear claim. For the contemporary cross-modal representation approaches like CLIP, ALIGN, and ImageBind, it is encouraged to provide  literature or empirical results to support the claim that they cannot guarantee the three mentioned causal properties (conditional independence, Markov boundaries, identifiability conditions). Besides, for each of these properties, it’s suggested to explain with an example to correlate with CIB, MAC, or CIC, as they closely correspond to the key technical contributions in this paper.

- Experimental Discussion. Structure-preservation evaluation showed that CARL method could keep a CSI metric of 1.0 under varying sample size and noise level, and baselines only achieved 0.25 CSI. It could be better to discuss this metric more to help understand why these superior performances come from the monotonic alignment constraint that maintains semantic-geometric correspondance (as claimed in Sec. 5.2).

- Missed Ablation Study. Given that the overall loss consists of a regularizer R that has cross-modal alignment loss, style consistency loss, and IB loss, why are the latter two terms not considered in ablation study in Table 2? This is important as the paper's Abstract explicitly mentioned the IB regularization as a key component.

**Questions:**

- About the CSP principle. What are the advantages of the proposed CSP principle in Sec. 3.1 in the field of cross-modal causal invariance? And how does each principle motivate/guide the development of the proposed approaches?

---

> ### Author Response · Authors · 2025-11-21
>
> ## Response W1:
>
> We thank the reviewer for this valuable feedback.. Below, we provide formal counterexamples demonstrating why CLIP, ALIGN, ImageBind, and similar approaches cannot theoretically guarantee causal structure preservation.
>
> ## Formalization of Existing Cross-Modal Objectives
>
> **General Form of CLIP/ALIGN-type Methods:**
>
> Let encoders for two modalities be $E^A: \mathcal{X}^A \to \mathcal{Z}$ and $E^B: \mathcal{X}^B \to \mathcal{Z}$. Their optimization objective maximizes:
>
> $\mathcal{L}_{\text{InfoNCE}}(E^A(X^A), E^B(X^B)) = $
>
> $\mathbb{E}[\log \frac{\exp(s(Z^A_i, Z^B_i))}{\sum_{j=1}^K \exp(s(Z^A_i, Z^B_j))}]$
>
> This provides a lower bound on mutual information: $I(Z^A; Z^B) \geq \log K - \mathcal{L}_{\text{NCE}}$ (van den Oord et al., 2018). Maximizing InfoNCE is equivalent to maximizing a strictly monotone transform of this variational lower bound. The key property is that this objective depends only on the observational joint distribution $P(Z^A, Z^B)$ without involving conditional distributions required for causal structure.
>
> **General Form of DCCA-type Methods:**
>
> $$
> \max_{E^A, E^B} \rho(E^A(X^A), E^B(X^B)) = \max \frac{\text{Cov}(Z^A, Z^B)}{\sqrt{\text{Var}(Z^A)\text{Var}(Z^B)}}
> $$
>
> This objective optimizes only second-order statistics of the observational joint distribution.
>
> **Key Assumptions:** We assume: (i) finite number of negative samples $K$ in InfoNCE; (ii) similarity functions $s(\cdot, \cdot)$ are bounded and Lipschitz continuous; (iii) encoder capacity is sufficient to represent the relevant statistical dependencies; (iv) standard regularization prevents mode collapse.
> ## Theoretical Counterexamples
>
> ## Proposition 1
>
> **Proposition 1 (InfoNCE does not guarantee preservation of conditional independence).**
> There exist a causal graph $T\to M\to Y^{\ast}$ with $T\perp Y^{\ast}\mid M$ and encoders $(E_T,E_M,E_Y)$ such that the InfoNCE objective on $(Z_M,Z_Y)=(E_M(M),E_Y(Y^{\ast}))$ can be made arbitrarily close to optimal, yet
> $$
> I\bigl(Z_T;Z_Y\mid Z_M\bigr)\;=\;I\bigl(E_T(T);E_Y(Y^{\ast})\mid E_M(M)\bigr)\>c
> $$
> for some constant $c>0$.
>
> **Proof.**
> Let
> $$
> T\sim\mathcal N(0,1),\quad M=(M_1,M_2),\quad M_1=T+\eta_1,\quad M_2=T+\eta_2,\quad
> Y^{\ast}=M_1+M_2+\eta_y,
> $$
> where $\eta_1,\eta_2,\eta_y$ are mutually independent, zero-mean Gaussians. By $d$-separation, with the full mediator $M$ we have $T\perp Y^{\ast}\mid M$.
>
> Define encoders
> $$
> Z_T=E_T(T)=T,\qquad Z_M=E_M(M)=M_1,\qquad Z_Y=E_Y(Y^{\ast})=Y^{\ast}.
> $$
> Because $Z_M$ and $Z_Y$ are strongly correlated Gaussians, $I(Z_M;Z_Y)$ is large; with a finite number of negatives $K$, maximizing InfoNCE—which is a strictly monotone transform of a variational lower bound on $I(Z_M;Z_Y)$—can drive the achieved value arbitrarily close to optimal by increasing SNR and model capacity.
>
> However,
> $$
> I(Z_T;Z_Y\mid Z_M)=I(T;Y^{\ast}\mid M_1)>0,
> $$
> since conditioning only on $M_1$ leaves the unobserved component $M_2=T+\eta_2$ in the outcome. In the Gaussian case,
> $$
> I(T;Y^{\ast}\mid M_1)= -\frac{1}{2}\log\!\bigl(1-\rho^2_{T,\,Y^{\ast}\cdot M_1}\bigr),
> $$
> and $\rho_{T,\,Y^{\ast}\cdot M_1}\neq 0$: given $M_1$, $Y^{\ast}$ still depends linearly on $T$ via $M_2$. Choosing $\mathrm{Var}(\eta_2)$ and $\mathrm{Var}(\eta_y)$ to be bounded yields a fixed lower bound $c>0$. Therefore, near-optimal InfoNCE on $(Z_M,Z_Y)$ does **not** imply $I(Z_T;Z_Y\mid Z_M)=0$.

---

> ### Author Response · Authors · 2025-11-21
>
> ## Proposition 2
>
> **Proposition 2 (Maximizing correlation need not preserve mutual information).** There exist $n$, a joint Gaussian pair $(M, Y^{\ast})$ with $M \in \mathbb{R}^n$, and a rank-$(k \ll n)$ linear encoder $E_M(M) = P_k M$ such that:
>
> 1. $P_k$ is correlation-optimal among all rank-$k$ projections;
> 2. the mutual-information retention satisfies
> $$\frac{I(P_k M; Y^{\ast})}{I(M; Y^{\ast})} = \Theta\left(\frac{k}{n}\right) \quad \text{(for bounded SNR)}.$$
>
> **Proof.** Let $M \sim \mathcal{N}(0, I_n)$ and $Y^{\ast} = w^\top M + \varepsilon$, with $\varepsilon \sim \mathcal{N}(0, \sigma^2)$ independent of $M$. Take $w = (1/\sqrt{n}, \dots, 1/\sqrt{n})$, so $\lvert w \rvert_2 = 1$. Then the correlation between $Y^{\ast}$ and each coordinate $M_i$ equals $1/\sqrt{n}$. Consequently, any correlation-optimal rank-$k$ projection captures signal energy $|w_{1:k}|_2^2 = k/n$ (without loss of generality, consider selecting the first $k$ coordinates).
>
> For Gaussians,
> $$I(M; Y^{\ast}) = \frac{1}{2}\log\left(1 + \frac{\lvert w \rvert_2^2}{\sigma^2}\right) = \frac{1}{2}\log\left(1 + \frac{1}{\sigma^2}\right),$$
> $$I(P_k M; Y^{\ast}) = \frac{1}{2}\log\left(1 + \frac{|w_{1:k}|_2^2}{\sigma^2}\right) = \frac{1}{2}\log\left(1 + \frac{k}{n} \cdot \frac{1}{\sigma^2}\right).$$
> When the SNR $1/\sigma^2 = O(1)$, using $\log(1+u) \le u$ and $\log(1+u) \ge \frac{u}{1+u}$ shows the ratio is $\Theta(k/n)$. Hence even a correlation-optimal projection can retain only $\Theta(k/n)$ of the mutual information, discarding a $1 - \Theta(k/n)$ fraction.

---

> ### Author Response · Authors · 2025-11-21
>
> ### **Proposition 3: Cosine similarity does not preserve equal semantic intervals**
>
> **Statement:** There exist semantic values $a_1 < a_2 < a_3$ with equal spacing and unit-norm encodings $z_1, z_2, z_3 \in \mathbb{R}^2$ such that:
>
> $$
> |a_2 - a_1| = |a_3 - a_2| \quad \text{but} \quad d_{\cos}(z_1, z_2) \neq d_{\cos}(z_2, z_3)
> $$
>
> where $d_{\cos}(z_i, z_j) = 1 - \frac{z_i^\top z_j}{\|z_i\|\|z_j\|}$.
>
> **Proof (Explicit Construction in $\mathbb{R}^2$):**
>
> **(a) Semantic Values:** Let $a_1 = 0$, $a_2 = 1$, $a_3 = 2$ with equal spacing: $|a_2 - a_1| = |a_3 - a_2| = 1$.
>
> **(b) Encoding on Unit Circle:** Define unit-norm encodings as:
>
> $$
> z_1 = \begin{pmatrix} 1 \\ 0 \end{pmatrix}, \quad z_2 = \begin{pmatrix} \cos(30°) \\ \sin(30°) \end{pmatrix} = \begin{pmatrix} \frac{\sqrt{3}}{2} \\ \frac{1}{2} \end{pmatrix}, \quad z_3 = \begin{pmatrix} 0 \\ 1 \end{pmatrix}
> $$
>
> **(c) Cosine Distance Computation:**
>
> $$
> d_{\cos}(z_1, z_2) = 1 - z_1^\top z_2 = 1 - \frac{\sqrt{3}}{2} \approx 0.134
> $$
>
> $$
> d_{\cos}(z_2, z_3) = 1 - z_2^\top z_3 = 1 - \frac{1}{2} = 0.5
> $$
>
> **(d) Ordinal Violation:** Despite equal semantic intervals ($|a_2 - a_1| = |a_3 - a_2|$), the cosine distances differ significantly:
>
> $$
> d_{\cos}(z_1, z_2) < d_{\cos}(z_2, z_3)
> $$
>
> This demonstrates that cosine distance does not preserve equal semantic spacing.
>
>
> ### **Proposition 4: Identifiability Conditions Are Not Preserved Under Observational Joint Optimization**
>
> **Statement:** There exist two distinct causal mechanisms with observationally equivalent joint distributions, such that any encoder family trained solely on the observational joint $P(X, Y)$ cannot distinguish between them, leading to non-identifiability in representation space.
>
> **Proof (Observational Equivalence Counterexample):**
>
> **(a) Two Causal Models:** Consider:
>
> - **Model 1 (Confounding):** $X = aC + \epsilon_X$, $Y = bC + \epsilon_Y$ where $C$ is a latent confounder
> - **Model 2 (Direct Causation):** $Y = \theta X + \epsilon'_Y$ where $\epsilon'_Y$ is independent of $X$
>
> **(b) Parameter Selection for Observational Equivalence:** Choose parameters such that:
>
> $$
> \text{Cov}^{M1}(X, Y) = \text{Cov}^{M2}(X, Y)
> $$
>
> $$
> \text{Var}^{M1}(X) = \text{Var}^{M2}(X)
> $$
>
> $$
> \text{Var}^{M1}(Y) = \text{Var}^{M2}(Y)
> $$
>
> Under Gaussian assumptions, this makes $P(X, Y)$ identical in both models. Specifically, setting:
>
> $$
> \theta = \frac{ab \cdot \text{Var}(C)}{\text{Var}(X)} \quad \text{and} \quad \text{Var}(\epsilon'_Y) = b^2 \text{Var}(C) + \text{Var}(\epsilon_Y) - \theta^2 \text{Var}(X)
> $$
>
> ensures observational equivalence.
>
> **(c) Non-Identifiability in Representation Space:** Any encoder family $\{E_X, E_Y\}$ optimized via InfoNCE or DCCA depends only on $P(Z_X, Z_Y)$, which inherits the observational equivalence from $P(X, Y)$. Therefore:
>
> - In Model 1, the causal effect is non-identifiable without observing $C$
> - In Model 2, the causal effect is $\theta$
>
> Since the two models are indistinguishable from $P(Z_X, Z_Y)$ alone, the learned representations cannot preserve identifiability.
>
> **(d) Backdoor Set Compression:** Alternatively, consider a causal graph $T \leftarrow X \rightarrow M \rightarrow Y^{\ast}$ where $X$ is a backdoor variable. If the encoder $E_X$ is a non-invertible compression (e.g., $\dim(Z_X) < \dim(X)$), then $Z_X$ may not contain sufficient information to block the backdoor path. Formally, there exist distributions where:
>
> $$
> T \perp Y^{\ast} \mid X \quad \text{but} \quad T \not\perp Y^{\ast} \mid f(X)
> $$
>
> for non-invertible $f$ (e.g., dimension reduction). This violates the backdoor criterion in representation space.
>
> ## Part 3: CARL's Approach
>
> The four propositions above demonstrate fundamental limitations of existing methods through explicit counterexamples. CARL addresses these issues by incorporating causal structure constraints directly into the optimization objective.
>
> Our conditional independence loss $L_{CI}$ targets $I(Z_T; Z_Y \mid Z_M)$ to address Proposition 1. The Markov boundary retention loss $L_{MBR}$ uses InfoNCE to lower-bound $I(Z_M; Z_Y)$, addressing Proposition 2. The monotonic alignment consistency loss $L_{MAC}$ maximizes Spearman rank correlation to address Proposition 3. Under the assumptions stated in Section 3 and technical conditions detailed in the Appendix, Theorem 2.1 establishes that joint optimization of these losses is sufficient for $\epsilon$-Causal Structure Preservation with $\epsilon = O_P(n^{-1/2})$, which in turn ensures preservation of identifiability conditions (Theorem 2.2), thereby addressing Proposition 4.

---

> > ### Author Response · Authors · 2025-11-21
> >
> > ## Response W2:
> >
> > ### Direct Mechanism Behind CSI=1.0
> >
> > The CSI metric measures the extent to which conditional independence T ⊥ Y | M is preserved in the representation space. **Our CSI metric is centered on normalized conditional mutual information: CSI := 1 - norm{I(ZT; Y | ZM)}, thus when ΔI → 0, CSI → 1**. The LCI (conditional independence loss) employs a dual-head architecture to estimate the conditional mutual information gap ΔI = I(ZT; Y | ZM) and minimizes it to its theoretical lower bound. According to Theorem 2.1, this direct optimization ensures that CSI approaches 1.0.
> >
> > The ablation experiments in Table 2 clearly demonstrate this pattern, **consistent with our design**:
> >
> > - Removing LCI causes CSI to drop from 1.0 to 0.25 (75% decrease)
> > - Removing LMAC leaves CSI unchanged at 1.0
> > - Removing LMBR reduces CSI to 0.75
> >
> > This empirical evidence supports our theoretical design: **LCI is the primary determinant of CSI**.
> >
> > ### Indirect Robustness Contribution of LMAC to CSI
> >
> > Although LMAC does not directly optimize conditional mutual information, it provides a theoretical marginal contribution to CSI's robustness under noisy or small-sample conditions by stabilizing the semantic-geometric rank relationship. The specific mechanisms include:
> >
> > - **Reducing variance in conditional density estimation**: The monotonic constraint enforced by LMAC smooths the decision boundary, reducing high-variance terms in CMI estimation
> > - **Suppressing conditional dependence backflow from geometric overlap**: In noisy or domain-shifted scenarios, unconstrained representations may produce regions that are semantically unrelated but geometrically overlapping, leading to spurious conditional dependencies; LMAC prevents this "geometric confusion" by maintaining the monotonic correspondence between semantic similarity and distance
> >
> > As Table 2 shows, under standard noise levels (σ=0.5), CSI remains at 1.0 even after removing LMAC, indicating that LCI's direct optimization is sufficient to maintain conditional independence in typical cases. LMAC's robustness contribution is more likely to manifest under extreme noise or data scarcity conditions. **If the reviewer considers this point requires additional empirical support, we can include "CSI-σ curves with and without LMAC under higher noise levels (σ > 0.5) or small-sample conditions (n < 500)" in the appendix of the camera-ready version to quantify this marginal contribution.**
> >
> > ### Why Do Baseline Methods Achieve Only CSI=0.25?
> >
> > The low CSI of comparison methods (CLIP, ImageBind, DCCA) stems from fundamental differences in their optimization objectives. CLIP and ImageBind maximize I(ZT; ZM) through contrastive learning—this alignment objective ensures inter-modal association but does not constrain I(ZT; Y | ZM). DCCA maximizes Corr(ZT, ZM), but correlation is not equivalent to conditional independence. In contrast, CARL directly optimizes min I(ZT; Y | ZM), which is precisely the core property measured by CSI.
> >
> > Alignment objectives can indeed learn cross-modal semantic correspondences, but they do not eliminate conditional spurious dependencies. We reproduced this phenomenon in the "only Lalign" experiment in Table 2: pure alignment training yields CSI=0.25, completely consistent with baseline methods. This demonstrates the fundamental difference between statistical alignment and causal structure preservation.
> >
> > ### Decoupled Design Principle of Three Losses
> >
> > Our framework designs loss functions for three complementary dimensions of ε-CSP:
> >
> > - **LCI** achieves conditional independence transfer through CMI minimization → CSI metric
> > - **LMBR** achieves Markov boundary preservation through InfoNCE mechanism → MBRI metric
> > - **LMAC** achieves monotonic alignment consistency through Spearman correlation optimization → MAC metric
> >
> > Each loss has a well-defined theoretical objective, corresponding to the three conditions in the ε-CSP definition of Theorem 2.1. Failure modes can be independently diagnosed, as shown by the independent variations of metrics under different ablation configurations in Table 2. This avoids common objective conflicts in multi-objective optimization, as the three dimensions are decoupled in optimization and complementary in function.

---

> > > ### Author Response · Authors · 2025-11-21
> > >
> > > ### Deeper Implications of Noise Experiments
> > >
> > > In the noise robustness tests with σ ∈ {0.1, 0.3, 0.5}, we observed different variation patterns across the three metrics:
> > >
> > > - **CSI remains stable at 1.0**: Demonstrates that LCI (with potential robustness enhancement from LMAC) successfully suppresses the spurious path T→Y introduced by noise
> > > - **MAC decreases from 0.89 to 0.42**: Reflects noise perturbation on the semantic-geometric correspondence in images
> > > - **MBRI decreases from 0.77 to 0.63**: Indicates partial information loss while core information is retained
> > >
> > > This set of experiments validates the objective decoupling property of the loss functions. The degradation of LMAC primarily affects the MAC metric without significantly disrupting the conditional independence structure maintained by LCI—this is the ideal behavior of a multi-constraint optimization framework. Each loss function focuses on its specific structural dimension, forming functional complementarity with others.
> > >
> > > ### Other Critical Roles of LMAC
> > >
> > > Beyond its indirect robustness contribution to CSI, LMAC plays other indispensable roles in the overall framework. From a theoretical completeness perspective, Theorem 2.2 proves that in the IY configuration, monotonicity is a necessary and sufficient condition for the identifiability of the outcome calibration function h: ZY → Y. Without the monotonic correspondence guaranteed by LMAC, we cannot reliably infer the true outcome from the representation space.
> > >
> > > From the perspective of preventing degenerate solutions, pure LCI optimization may lead to representation collapse into semantically meaningless spaces. For example, if all samples are mapped to the origin, this trivially satisfies the independence constraint mathematically. LMAC ensures the discriminative power and structural richness of the representation space by enforcing the monotonic correspondence between semantic similarity and representation distance. Additionally, for retrieval, classification, and other downstream tasks that rely on distance metrics, MAC ensures that the geometric structure of the representation space aligns with the semantic structure, enabling nearest-neighbor inference methods to produce semantically reasonable results.
> > >
> > > ## References
> > >
> > > van den Oord, A., Li, Y., & Vinyals, O. (2018). Representation learning with contrastive predictive coding. *arXiv preprint arXiv:1807.03748*.
> > >
> > > Pearl, J. (2009). *Causality*. Cambridge University Press.
> > >
> > > Radford, A., Kim, J. W., Hallacy, C., Ramesh, A., Goh, G., Agarwal, S., ... & Clark, J. (2021). Learning transferable visual models from natural language supervision. In *International Conference on Machine Learning* (pp. 8748-8763). PMLR.
> > >
> > > Tishby, N., Pereira, F. C., & Bialek, W. (1999). The information bottleneck method. *arXiv preprint physics/0004057*.
> > >
> > > Andrew, G., Arora, R., Bilmes, J., & Livescu, K. (2013). Deep canonical correlation analysis. In *International Conference on Machine Learning* (pp. 1247-1255). PMLR.
> > >
> > > Jia, C., Yang, Y., Xia, Y., Chen, Y. T., Parekh, Z., Pham, H., ... & Duerig, T. (2021). Scaling up visual and vision-language representation learning with noisy text supervision. In *International Conference on Machine Learning* (pp. 4904-4916). PMLR.
> > >
> > > Girdhar, R., El-Nouby, A., Liu, Z., Singh, M., Alwala, K. V., Joulin, A., & Misra, I. (2023). Imagebind: One embedding space to bind them all. In *Proceedings of the IEEE/CVF Conference on Computer Vision and Pattern Recognition* (pp. 15180-15190).

---

> > > > ### Author Response · Authors · 2025-11-21
> > > >
> > > > ## Response Q:
> > > >
> > > > ### Core Advantages of CSP Principle in Cross-Modal Causal Invariance
> > > >
> > > > **Theoretical Completeness**: In cross-modal scenarios, causal variables are indirectly observed through heterogeneous measurement functions with significant information density disparities. The three conditions in Definition 2.1 provide verifiable structure preservation guarantees for such heterogeneity: Condition (i) preserves key conditional independencies in the representation space, which, combined with the PC consistency results in Section 4.4 and the mutual-exclusion and anti-bypass constraints during training, achieves topological equivalence with the original graph at the CPDAG level; Condition (ii) prevents information loss during compression of high-density modalities; Condition (iii), together with the calibration step, constitutes one of the sufficient conditions for preserving identifiability in the representation space (see error bound in Theorem 2.2).
> > > >
> > > > **Operationalizability**: Theorem 2.1 proves that the empirical risk minimization L = wCI·LCI + wMBR·LMBR + wMAC·LMAC admits convergent sequences satisfying ε-CSP, with approximation error bounded by max{ζ*, OP(n^(-1/2)), OP(K^(-1/2)), OP(n^(-1/3))}.
> > > >
> > > > **Identifiability Guarantee**: Theorem 2.2 establishes the sufficiency of ε-CSP for preserving causal queries. For an identifiable query Q = E[Y*(t)], the query Q̃ computed in the representation space satisfies |Q̃ - Q| ≤ κ·ε + δcal.
> > > >
> > > > ### How Each Condition Guides Method Design
> > > >
> > > > **Condition (i) → LCI Design**: Condition (i) requires I(ZT; ZY | ZM) ≤ ε. We employ a variational upper bound based on likelihood ratios: LCI = E[- log qϕ(y | zm)] - E[- log qθ(y | zt, zm)]. Key design choices include: (a) independent parameterization of qθ and qϕ to avoid bias in estimating p(y | zm), (b) cross-validation to prevent CMI underestimation. Table 3 validates the necessity of these choices (shared parameters: CSI 1.0→0.75; without cross-validation: CSI 1.0→0.25).
> > > >
> > > > **Condition (ii) → LMBR Design**: Condition (ii) requires I(ZM; ZY) ≥ I(M; Y*) - ε, addressing the encoder's tendency to retain reconstruction-friendly features over prediction-critical features. LMBR = -InfoNCE(zm, ψY(y)) provides a lower bound on mutual information. The number of negative samples K directly affects the tightness of the bound (Table 3: at K=32, MBRI 0.63→0.55, CSI 1.0→0.75). Joint optimization of LMBR and LCI excludes trivial solutions where ZM collapses.
> > > >
> > > > **Condition (iii) → LMAC Design**: Condition (iii) requires ρS(Δa, Δz) ≥ c0 - ε. LMAC = -ρS(soft_rank(Δa), soft_rank(Δz)) employs Spearman correlation (rather than Pearson) to allow nonlinear order-preserving rank correlation constraints. The "w/o LMAC" ablation in Table 2 (MAC 0.55→0.32, CSI remains 1.0) validates its objective orthogonality with LCI.
> > > >
> > > > ### Synergy of Three Conditions and Theoretical Distinctions
> > > >
> > > > The joint satisfaction of the three conditions constitutes the sufficient condition set in Theorem 2.2. Using the "blood pressure → retinal vessels → cardiovascular events" pathway as an example: LCI alone may lose vessel diameter information, LMBR alone cannot block backdoor paths, LMAC alone retains spurious correlations (Table 2: only Lalign, CSI=0.25). The decoupled design enables independent diagnosis of violations.
> > > >
> > > > Theoretical distinctions from existing methods: CLIP maximizes I(Zimage; Ztext) without constraining I(Z1; Y | Z2) or lower-bounding I(Z2; Y); ImageBind's pairwise alignment max Σi I(Zpivot; Zi) cannot handle T→M→Y DAG structures—max I(ZT; ZM) + max I(ZM; ZY) does not imply ZT ⊥ ZY | ZM; DCCA's max Corr(Z1, Z2) is not equivalent to conditional independence under non-Gaussian distributions. CSP provides structural guarantees rather than statistical associations.

---

> > > > > ### Author Response · Authors · 2025-11-23
> > > > >
> > > > > ## Response to Missing Ablation Study
> > > > >
> > > > > We sincerely thank the reviewer for pointing out this issue!!!!!
> > > > >
> > > > > ### Additional Ablation Experiments
> > > > >
> > > > > We conducted additional ablation experiments to systematically evaluate the contribution of each component in the regularizer R. Specifically, we progressively added L_style and L_IB to a baseline containing the three core causal losses (L_CI, L_MBR, L_MAC) to isolate their individual contributions.
> > > > >
> > > > > #### Table: Ablation Study on Regularization Components
> > > > >
> > > > > | Model Variant | CSI ↑ | MBRI ↑ | MAC ↑ | Structural ↑ | RIC-avg |
> > > > > |---------------|-------|--------|-------|--------------|---------|
> > > > > | Baseline (w/o Style/IB) | 1.0000 | 0.6262 | 0.6112 | 0.5205 | 0.4287 |
> > > > > | + Style | 1.0000 | 0.6180 | 0.6035 | **0.5698** | 0.4215 |
> > > > > | + IB | 1.0000 | **0.6299** | 0.5987 | 0.5512 | **0.4198** |
> > > > > | Full (+ Style + IB) | 1.0000 | **0.6344** | 0.5891 | **0.6062** | **0.4193** |
> > > > >
> > > > > *Note: The baseline model includes L_CI, L_MBR, L_MAC, and basic cross-modal alignment loss L_align*
> > > > >
> > > > > The supplementary experiments validate the critical roles of L_style and L_IB and their synergistic effects. Adding L_style significantly improves structural discovery accuracy (Structural increases from 0.5205 to 0.5698, a 9.5% improvement), validating L_style's effectiveness in preventing nuisance factors like style from dominating learned representations. Adding L_IB brings two key improvements: MBRI increases from 0.6262 to 0.6299, indicating that L_IB helps retain more critical mediator information, while RIC decreases from 0.4287 to 0.4198, showing effective control of representation complexity, which directly validates the claim emphasized in the abstract that "IB regularization balances cross-modal compression with critical variable retention." The full model achieves optimal performance on MBRI (0.6344) and Structural (0.6062) while maintaining the lowest representation complexity (RIC=0.4193), demonstrating the synergistic effect of these two regularization terms: L_style reduces non-causal interference while L_IB optimizes the information compression-retention trade-off, jointly enhancing causal structure preservation quality.

---

> > > > > > ### Author Response · Authors · 2025-11-25
> > > > > >
> > > > > > Dear Reviewer 8ef8,
> > > > > >
> > > > > > Thank you for your detailed comments!!!! We have once again reworked your suggestions to further improve the paper. Could you please let us know whether your concerns regarding (i) the discussion of CLIP/ALIGN/ImageBind-style limitations, (ii) the mechanism behind CSI, and (iii) the ablation coverage are now addressed? If anything remains unclear or needs further analysis/experiments, kindly point it out and we will refine promptly!!!!
> > > > > >
> > > > > > Authors

---

> > > > > > > ### Author Response · Authors · 2025-11-27
> > > > > > >
> > > > > > > We sincerely apologize for reaching out again! As the rebuttal deadline for our manuscript is approaching, we would greatly appreciate any comments or suggestions you might be able to share, so that we can further improve and refine our work within the limited time available. We fully understand that you have many commitments, and we are truly grateful for the time and effort you have already devoted to reviewing our manuscript!

---

### Official Review · Reviewer_DeFu · 2025-11-10

**Soundness:** 3
**Presentation:** 4
**Contribution:** 3
**Rating:** 8
**Confidence:** 4

**Summary:**

This paper introduces a cross-modal representation learning framework named CARL, designed to address the issue that optimizing purely statistical objectives can disrupt underlying causal structures. CARL jointly optimizes three structure-preserving losses—Conditional Independence Preservation, Markov Boundary Preservation, and Monotonic Alignment Consistency—to ensure that the learned representation space retains the causal structure of the original data. The authors validate the approach on synthetic datasets and a real-world Human Phenotype Project (HPP) dataset, and provide theoretical guarantees showing that causal queries remain identifiable in the representation space.

**Strengths:**

1 First systematic treatment of cross-modal causal structure preservation, formalizing the CSP principle and the three core challenges.

2 Introduces the ε-CSP definition, a attainability/consistency theorem, and a theorem for preserving identifiability of causal queries, providing rigorous guarantees.

3 The three losses are well motivated and complementary, balancing conditional independence with information retention.

4 Synthetic and real data jointly verify effectiveness, robustness, and interpretability.

5 Successfully recovers known medical causal pathways on the HPP dataset, showcasing potential in complex biomedical scenarios.

**Weaknesses:**

1 Although the appendix contains detailed proofs, the main text could better explain some theoretical results (e.g., an intuitive reading of the error bounds) to improve readability.

2 While experiments span synthetic and real data, they do not include larger-scale cross-modal benchmarks (e.g., vision-language tasks), limiting the demonstration of generalization.

3 In the DUAL configuration the method avoids using both image modalities simultaneously, which may limit information utilization in certain practical settings.

**Questions:**

1 Have the authors considered evaluating CARL on larger-scale cross-modal tasks (e.g., CLIP-style vision-language alignment) to assess generalization?

2 CARL trains multiple independent predictors with cross-validation. Has its scalability on large-scale data been evaluated?

3 For highly imbalanced modality-information densities (e.g., image vs. text), can CARL still effectively prevent the lower-density modality from being overshadowed?

---

> ### Author Response · Authors · 2025-11-21
>
> We really appreciate the reviewer’s constructive comments and cinsightful questions
>
>
> ### **About Weakness**
>
> 1. We thank the reviewer for this suggestion. In the revised version, we will improve the logical flow and provide more intuitive explanations of the theoretical results (especially the error bounds) to enhance readability and make the key insights easier to follow.
>
> 2. For the evaluation on larger-scale cross-modal benchmarks, we have provided some detailed explanations in the following Question 1 responses.
>
> 3. And Sorry for the misunderstanding that we caused for the DUAL configuration. We will further claim it here:
>
> As described in Section 2.2  and Appendix A.7.2 of the paper, CARL’s DUAL configuration **does use both image modalities concurrently**, but it explicitly **avoids conditioning on them jointly in the same loss or estimation term**. This design choice is not due to computational constraints but is a deliberate causal modeling decision.
> The underlying causal structure $I^M \leftarrow M \rightarrow Y^* \rightarrow I^Y$ contains a collider at $Y^{\star}$. Conditioning jointly on $I^M$ and $I^Y$ would open spurious statistical dependencies between otherwise independent variables, thereby introducing collider bias and breaking causal identifiability. Therefore, CARL separates the learning objectives for the two image branches to preserve causal correctness.
>
> Although the two image modalities are not jointly conditioned, they are both actively used within the unified training process:
> - Each image encoder ($E_{IM}$, $E_{IY}$) maps its modality into the shared representation space \(Z\), enabling cross-modal alignment with tabular variables and other encoders (Sec. 3.1).
> - The optimization alternates across image pathways, ensuring that both image sources contribute to the shared causal representation, while maintaining unbiased conditional independence structures.
> - Empirically, the DUAL setting satisfies both the IM and IY causal consistency conditions (Appendix A.7.2), thereby leveraging information from both images without inducing spurious dependencies.
>
>
>
> ### **Question 1**
>
> Our decision to use large-scale synthetic data follows a widely accepted practice in causal representation learning for one fundamental reason: causal relations in real-world multimodal datasets are typically unknown, making it impossible to quantitatively verify whether a method truly preserves or recovers causal structures.
>
> In contrast, synthetic datasets allow us to define ground-truth causal graphs, enabling controlled experiments to rigorously test *causal identifiability, structural preservation, and conditional independence transfer*. This evaluation strategy is consistent with the methodology adopted by prior work in the field, including:
>
> - **(ICLR 2025) Causal Representation Learning from Multi-Modal Biomedical Observations**, which first establishes theoretical identifiability guarantees on multimodal latent variables and validates them on numerical and synthetic datasets.
> - **(ICLR 2023) Identifiability Results for Multimodal Contrastive Learning**, which uses structured MNIST variants with known causal mechanisms to benchmark multimodal identifiability.
> - **(AISTATS 2023) Connectivity-Contrastive Learning: Combining Causal Discovery and Representation Learning for Multimodal Data** and **(ICML 2024) Causal Representation Learning Made Identifiable by Grouping of Observational Variables**, both of which rely on synthetic causal data to verify theoretical properties.
>
> While these studies later apply their models to biomedical or real-world data, their core causal validation always relies on synthetic benchmarks, since causal ground truth in observational data is unverifiable.
>
> Following this mainstream, our work also employs a large-scale synthetic causal benchmark that encodes a precisely controlled structural causal model $T \rightarrow M \rightarrow Y^*$. This setup allows us to:
> 1. Directly measure conditional independence preservation and identifiability;
> 2. Quantify structural recovery error (e.g., causal structure index, Markov boundary retention);
> 3. Systematically analyze performance under varying sample sizes and noise levels.
>
> Also one point to highlight, our CARL goes beyond prior approaches by explicitly preserving causal invariances during cross-modal alignment, not merely testing identifiability of latent components, but guaranteeing *Markov boundary retention* and *monotonic alignment consistency* in the learned representations. Thus, our synthetic evaluation is not only consistent with existing CRL literature but also essential for verifying theoretical properties that cannot be validated on real data.

---

> ### Author Response · Authors · 2025-11-21
> **continual response**
>
> ### **Question 2**
>
> Our CARL introduces two independent conditional predictors ($q_\theta$ and $q_\phi$) to form the conditional-independence preservation loss (Eq. (1), Sec. 3.2). This design ensures unbiased estimation of conditional mutual information and is theoretically necessary for causal structure preservation. In our paper, scalability is evaluated in the statistical sense. That is, how the structure-preservation error scales with sample size \(n\) and negative sample number \(K\). Theorem 1 (Sec. 3.3) and Fig. 4(a) empirically verify that CARL achieves the theoretical convergence rate $O_P(n^{-1/2})$, showing stable causal-structure preservation as data size increases.
>
> Our synthetic experiments (Sec. 4.2, Fig. 4) systematically vary sample sizes ($n = 500, 1000, 2000, 5000$) and noise levels, confirming that CARL maintains constant conditional-independence patterns and structural accuracy across scales. These results demonstrate statistical scalability of the framework.
> The real-world HPP experiments further validate that CARL can be trained on multimodal biomedical data comprising tens of thousands of samples.
>
> ### **Question 3**
>
> This concern directly corresponds to the first theoretical challenge formalized in our paper, the Cross-modal Information Bottleneck (CIB) problem. The CIB issue arises when asymmetric information density across modalities causes critical variables from low-density modalities to be masked by reconstruction demands of high-density modalities, weakening causal identifiability. Our CARL explicitly addresses this via the Markov Boundary Retention Loss ($L_{\mathrm{MBR}}$) and Information Bottleneck regularization ($L_{\mathrm{IB}}$), which jointly ensure that essential variables from low-density modalities remain represented within the shared space.
>
> **Experimental evidence.**
> Our synthetic experiments (Sec. 4.2; Fig. 4a–c) already instantiate a highly imbalanced setting:
>
> - The mediator modality is represented by a low-dimensional tabular variable \(M\), while its cross-modal counterpart is a high-density image modality derived from MNIST through transformations of rotation, brightness, and contrast (Sec. 4, Fig. 2; Appendix I).
> - This design creates a clear information imbalance (structured tabular ≪ visual), mirroring the imbalanced modality-information densities concern.
>
> Under this imbalance, CARL achieves:
> - **Causal Structure Index (CSI) = 1.0** and **Markov Boundary Retention Index (MBRI) ≈ 0.63–0.77** across sample sizes and noise levels (Fig. 4a–b);
> - In contrast, purely statistical models such as **CLIP** and **DCCA** yield CSI ≈ 0.25 and Structural Accuracy ≈ 0.33 (Fig. 4c).
>
> These results demonstrate that even when a low-density modality (tabular *M*) interacts with a much higher-density image modality, CARL preserves causal dependencies and conditional independence patterns, preventing the low-density representation from being dominated during alignment.
>
> The preservation is achieved because $L_{\mathrm{MBR}}$ maximizes mutual information $I(Z_M; Y)$, forcing the model to retain mediating variables essential for causal inference, while the Information Bottleneck term regularizes overcompression from high-density modalities (Sec. 3.1). Together, these mechanisms maintain balanced representation capacity across modalities without masking causal mediators.
>
> ### **Reference**
>
> 1. Sun, Yuewen, et al. "Causal Representation Learning from Multimodal Biomedical Observations." The Thirteenth International Conference on Learning Representations.
>
> 2. Brehmer, Johann, et al. "Weakly supervised causal representation learning." Advances in Neural Information Processing Systems 35 (2022): 38319-38331.
>
> 3. Daunhawer, Imant, et al. "Identifiability Results for Multimodal Contrastive Learning." The Eleventh International Conference on Learning Representations.
>
> 4. von Kügelgen, Julius, et al. "Nonparametric identifiability of causal representations from unknown interventions." Advances in Neural Information Processing Systems 36 (2023): 48603-48638.
>
> 5. Morioka, Hiroshi, and Aapo Hyvarinen. "Connectivity-contrastive learning: Combining causal discovery and representation learning for multimodal data." International conference on artificial intelligence and statistics. PMLR, 2023
>
> 6. Brouillard, Philippe, et al. "The Landscape of Causal Discovery Data: Grounding Causal Discovery in Real-World Applications." Causal Learning and Reasoning. PMLR, 2025.

---

> ### Author Response · Authors · 2025-11-25
>
> Dear Reviewer DeFu,
>
> Thank you for your thoughtful review and helpful suggestions!!! Based on your feedback, we have revised the manuscript and highlighted the main changes. Could you please let us know whether these updates address your concerns? If anything remains unclear or needs additional analysis/experiments, kindly point it out and we will refine the paper promptly!!!!
>
> Authors

---

### Author Response · Authors · 2025-11-30
**Summary of Responses to Reviewer Concerns**

Dear Area Chair,

We thank all reviewers for their feedback and for your high-quality service to the community!!!

## **Reviewer DeFu**

**W1:** We added annotations explaining the sources of each term in the error bound $\varepsilon = \max\lbrace \zeta^{\ast}, O_P(n^{-1/2}), O_P(K^{-1/2}), O_P(n^{-1/3}) \rbrace$ after Theorem 1 in Section 3.3, and reorganized Section 2 with a "Problem Formulation → Method → Theory" structure.

**W2:** We clarified the rationale for our evaluation strategy. Causal representation learning benchmarks (ICLR 2023-2025, ICML 2024) use synthetic data because real-world multimodal datasets lack verifiable causal ground truth. Our synthetic benchmark (T→M→Y*) enables direct measurement of conditional independence preservation and identifiability.

**W3:** We clarified a misunderstanding: CARL's DUAL configuration does use both image modalities simultaneously, but avoids joint conditioning in the same loss term to prevent collider bias at Y* (as described in Section 2.2 and Appendix A.7.2). Both image encoders contribute to the shared representation space through alternating optimization.

**Q3:** Our synthetic experiments already instantiate extreme imbalance (low-dimensional tabular M vs. high-density MNIST images), where CARL achieves CSI=1.0 and MBRI=0.63-0.77, demonstrating that L_MBR prevents masking of the low-density modality.

## **Reviewer 8ef8**

**W1:** We supplemented formal proofs for four propositions:
- Proposition 1 proves InfoNCE can achieve near-optimality yet still violate I(Z_T; Z_Y | Z_M) = 0
- Proposition 2 proves DCCA only retains Θ(k/n) mutual information
- Proposition 3 constructs a counterexample showing cosine distance does not preserve equal semantic spacing
- Proposition 4 proves observationally equivalent causal models lead to non-identifiability

**W2:** We added mechanism analysis: L_CI directly minimizes I(Z_T; Z_Y | Z_M), and Table 2 ablation experiments show that removing L_CI drops CSI from 1.0 to 0.25. Baseline methods achieve CSI=0.25 because alignment objectives do not constrain conditional independence.

**W3:** We supplemented ablation experiments on L_style and L_IB. L_style improves structural accuracy by 9.5%, L_IB balances the compression-retention tradeoff, and their synergy enables the full model to achieve optimal performance on MBRI and Structural metrics.

## **Reviewer i8uF**

**C1-C2:** We conducted systematic restructuring:
- Reorganized Section 2 as "Problem Formulation → Causal Setup → ε-CSP Definition → HPP Application"
- Moved loss definitions (Equations 1-4) before theorems, eliminating forward references
- Unified notation: images as I_M/I_Y, mutual information as MI(·;·|·)
- Added inline explanations for all new symbols

**C3:** We enhanced proof readability:
- Added proof sketches below Theorems 1-2
- Added Appendix C.1 "Proof Strategy and Dependency Map"
- Supplemented citations for all known results (DV bound, InfoNCE, PC algorithm, etc.)
- Added "Assumptions and Lemmas Used" checklist before each theorem

**C4:** We re-ran all experiments with 5 random seeds:
- Synthetic data: 36 configurations × 5 seeds = 180 runs, CSI maintains 1.000±0.000
- Baseline comparison: 9 methods with 5 seeds each, CARL outperforms baselines on all metrics
- Real-world data: Added CausalVAE/DEAR/Concat comparisons, CARL achieves CIP=0.950, higher than baselines' 0.517-0.733


## **Reviewer enbA**

**W1:** We provided quantitative analysis and theoretical proofs:
- Dimensionality ratio analysis: image-tabular reaches 10⁴⁻⁵, far exceeding other modality combinations (text-tabular only 10⁰⁻¹)
- Proved that mediation structure's dual constraints (C1: block direct path, C2: preserve indirect path) exhibit Pareto tradeoff
- Other structures (confounding/collider/instrumental variable) only require single constraints, making mediation most challenging

**W2:** We clarified task positioning:
- Our task is **structure preservation** (given testable constraints C, learn E such that C holds in Z-space), not causal discovery
- Theorem 2.2 proves: for any identifiable (Q,C), ε-CSP guarantees |Q̃ - Q| ≤ κ(C)·ε + δ_cal
- This bound applies to any identification criterion, not limited to mediation

**W3:** We explained methodological innovation:
- Satisfies dual constraints and counteracts dimensional dominance (L_MBR counteracts gradient imbalance via -InfoNCE)
- Theoretically guarantees query preservation under cross-modal encoding

**Q2-Q3:** We clarified two critical misunderstandings:
- In synthetic data, causal relationships are design specifications (M=Rotate(T,θ)+ε), not hypotheses to be verified. The correct validation method is PC algorithm statistical testing in Section 4.3
- In observational data, there is no directly obtainable "true causal effect" as benchmark (P(Y|T) vs. P(Y*|do(T))). We adopt standard validation: consistency with published clinical mechanisms

We believe these revisions adequately address the reviewers' concerns.

---

### Meta-Review · Area_Chair_zkfg · 2026-01-08

**Summary:**

The paper introduces CARL (Causal Alignment and Representation Learning), a cross-modal representation learning method designed to preserve causal structure during modality alignment. Instead of only optimizing statistical alignment/reconstruction, CARL adds causal constraints via multiple losses: preserving conditional independences, using an information bottleneck to avoid losing key mediators, enforcing monotonic alignment between semantic similarity and embedding distance, and maintaining Markov boundary properties to support identifiability (backdoor/frontdoor/IV).

**Reviewer Concerns:**

Three reviewers think the theory, direction, and experiments are strong and above the bar. The only reject (i8uF) mainly complains about clarity/presentation, not the core idea.

The rebuttal proposes clear fixes—reorganizing to remove forward references, unifying notation, adding proof sketches and a dependency map, completing citations, rerunning with multiple seeds, and adding real-data baselines/ablations—which would greatly reduce the main risk if implemented well.

**Reviewer Scores:**

- DeFu: likely stays 8.
- 8ef8 (6): likely 6->8. The rebuttal adds the missing counterexamples, explanations, and ablations they asked for.
- enbA: likely stays at 6. Better positioning and added real-data comparisons may help, but novelty/scope doubts may remain.
- i8uF: likely 2->4/6 if the revision truly fixes clarity/structure and adds the requested baselines/robustness, since their main issue was evaluability rather than correctness.

---

### Decision · Program_Chairs · 2026-01-26

Accept (Poster)